# A POLICY GRADIENT METHOD FOR CONFOUNDED POMDPS

**Mao Hong**
Johns Hopkins University
`mhong26@jhu.edu`

**Zhengling Qi**[*]
George Washington University
`qizhengling@gwu.edu`

**Yanxun Xu**
Johns Hopkins University
`yanxun.xu@jhu.edu`

## ABSTRACT

In this paper, we propose a policy gradient method for confounded *partially observable Markov decision processes* (POMDPs) with continuous state and observation spaces in the offline setting. We first establish a novel identification result to non-parametrically estimate any history-dependent policy gradient under POMDPs using the offline data. The identification enables us to solve a sequence of conditional moment restrictions and adopt the min-max learning procedure with general function approximation for estimating the policy gradient. We then provide a finite-sample non-asymptotic bound for estimating the gradient uniformly over a pre-specified policy class in terms of the sample size, length of horizon, concentratability coefficient and the measure of ill-posedness in solving the conditional moment restrictions. Lastly, by deploying the proposed gradient estimation in the gradient ascent algorithm, we show the global convergence of the proposed algorithm in finding the history-dependent optimal policy under some technical conditions. To the best of our knowledge, this is the first work studying the policy gradient method for POMDPs under the offline setting.

## 1 INTRODUCTION

Policy gradient methods in reinforcement learning (RL) have been extensively studied and utilized across many tasks due to their adaptability and straightforward implementation schemes (Sutton et al., 1999; Kakade, 2001; Silver et al., 2014). Despite its success in practice (Peters & Schaal, 2006; 2008b; Yu et al., 2017; Qiu et al., 2019; Li et al., 2022), most existing policy gradient methods were developed for the fully observable environment with Markovian transition dynamics, which may not known a priori. Little work has been done for the partially observable Markov decision processes (POMDPs), which is a more practical model for sequential decision making in many applications (Sawaki & Ichikawa, 1978; Albright, 1979; Monahan, 1982; Singh et al., 1994; Jaakkola et al., 1994; Cassandra, 1998; Young et al., 2013; Zhang & Bareinboim, 2016; Bravo et al., 2019). In this paper, we study policy gradient methods for finite-horizon and confounded POMDPs with continuous state and observation spaces under the offline setting. Compared to existing literature, we consider a more general setting where a flexible non-parametric model is used for the system dynamics of POMDPs, and the history-dependent policy class is indexed by a finite-dimensional parameter. We establish both statistical and computational convergence guarantees for the proposed policy gradient method in POMDPs under the offline setting.

There are several challenges for studying policy gradient methods in confounded POMDPs under the offline setting. First of all, in the offline data, the unobserved state variables at each decision point are unmeasured confounders that can simultaneously affect the action, the reward and the future transition. Directly implementing standard policy gradient methods can incur bias estimation, leading to suboptimal policies. Therefore identification is required for estimating the gradient of history-dependent policies using the offline data. Second, when the state and observation spaces are continuous, function approximation is inevitable for the gradient estimation. How to non-parametrically and consistently estimate the policy gradient in POMDPs remains unknown. Lastly, since the policy value to be optimized is a non-concave function with respect to the policy parameter, together with unobserved continuous states, a global convergence for finding the optimal policy in POMDPs is

---

[*]Corresponding author.

challenging. For example, the potentially misspecified policy class cannot ensure the improvement of the policy value by the estimated gradient towards optimality.

**Our main contribution:** We propose a policy gradient method in the offline setting for POMDPs under a non-parametric model with both statistical and computational guarantees. Specifically, we first establish a novel identification for directly estimating the gradient of any (history-dependent) policy using the offline data, circumventing the issue of *partial observability*. Based on the identification result, which leads to solving a sequence of conditional moment equations, we adopt the min-max estimating procedure to compute the policy gradient non-parametrically using the offline data. The estimated policy gradient is then incorporated into the gradient ascent algorithm for learning the optimal history-dependent policy. As for theoretical contribution, we investigate the statistical error for estimating the policy gradient at each step of our algorithm. In particular, we provide a non-asymptotic error bound for estimating the gradient uniformly over the policy class in terms of all key parameters. To establish the global (i.e., computational) convergence of our proposed algorithm, we study the landscape of the policy value in POMDPs, and leverage the compatible function approximation for proving the global convergence of the proposed policy learning algorithm. To the best of our knowledge, this is the first work studying policy gradient methods for POMDPs under the offline setting with a complete characterization of both statistical and computational errors.

## 2  RELATED WORK

**Policy gradient in MDPs and POMDPs.** There are mainly two lines of research on policy gradient methods in Markov decision processes (MDPs). The first line focuses on off-policy policy gradient estimation (Williams, 1992; Precup, 2000; Degris et al., 2012; Silver et al., 2014; Hanna & Stone, 2018; Liu et al., 2019; Tosatto et al., 2020; Xu et al., 2020; Kallus & Uehara, 2020; Xu et al., 2021; Tosatto et al., 2021; Ni et al., 2022; Tosatto et al., 2022). The second line focuses on the convergence of gradient ascent methods (Wang et al., 2019; Bhandari & Russo, 2019; Mei et al., 2020; Liu et al., 2020; Zhang et al., 2020; Hambly et al., 2021; Agarwal et al., 2021; Xiao, 2022). Little work has been done on the policy gradient in POMDPs. Azizzadenesheli et al. (2018) was among the first that studied a policy gradient method in POMDPs in the *online* setting, where the unobserved state at each decision point did not directly affect the action, and thus there was no confounding issue in their setting. As a result, the gradient can be estimated by generating trajectories from the underlying environment. In contrast, our work, which considers POMDPs in the offline setting, meets the significant challenge caused by the unobserved state, necessitating a more thorough analysis for identifying the policy gradient using the *offline* data. The presence of unobserved states also poses additional challenges in demonstrating the global convergence of policy gradient methods for learning the optimal history-dependent policy. A substantial analysis is required for controlling the error related to the growing dimension of the policy class in terms of the length of horizon.

**Confounded POMDP.** Employing proxy variables for identification in confounded POMDPs has attracted great attention in recent years. For example, the identification of policy value given a single policy has been investigated (Tennenholtz et al., 2020; Bennett & Kallus, 2021; Nair & Jiang, 2021; Shi et al., 2022; Miao et al., 2022). However, these studies are unsuitable for policy learning because identifying all policy values within a large policy class to find the best one is computationally infeasible. Another line of research focuses on policy learning in confounded POMDPs (Guo et al., 2022; Lu et al., 2022; Wang et al., 2022). Nevertheless, these works either lack computational algorithms (Guo et al., 2022; Lu et al., 2022) or necessitate a restrictive memorylessness assumption imposed on the unmeasured confounder (Wang et al., 2022). In contrast to the aforementioned works, we directly identify the policy gradient rather than the policy value, enabling the construction of an efficient algorithm through gradient ascent update. Furthermore, the proposed policy iteration algorithm circumvents the restrictive memoryless assumption imposed on the unobserved state that is typically necessary for fitted-Q-type algorithms in POMDPs.

## 3  PRELIMINARIES AND NOTATIONS

We consider a finite-horizon and episodic POMDP represented by $\mathcal{M} := (\mathcal{S}, \mathcal{O}, \mathcal{A}, T, \nu_1, \{P_t\}_{t=1}^T, \{\mathcal{E}_t\}_{t=1}^T, \{r_t\}_{t=1}^T)$, where $\mathcal{S}$, $\mathcal{O}$ and $\mathcal{A}$ denote the state space, the observation space, and the action space respectively. In this paper, both $\mathcal{S}$ and $\mathcal{O}$ are considered to be *continuous*, while $\mathcal{A}$ is *finite*.

The integer $T$ is set as the total length of the horizon. We use $\nu_1 \in \Delta(\mathcal{S})$ to denote the distribution of the initial state, where $\Delta(\mathcal{S})$ is a class of all probability distributions over $\mathcal{S}$. Denote $\{P_t\}_{t=1}^T$ to be the collection of state transition kernels over $\mathcal{S} \times \mathcal{A}$ to $\mathcal{S}$, and $\{\mathcal{E}_t\}_{t=1}^T$ to be the collection of observation emission kernels over $\mathcal{S}$ to $\mathcal{O}$. Lastly, $\{r_t\}_{t=1}^T$ denotes the collection of reward functions, i.e., $r_t : \mathcal{S} \times \mathcal{A} \to [-1, 1]$ at each decision point $t$. In a POMDP, given the current (hidden) state $S_t$ at each decision point $t$, an observation $O_t \sim \mathcal{E}_t(\cdot \mid S_t)$ is observed. Then the agent selects an action $A_t$, and receives a reward $R_t$ with $\mathbb{E}[R_t \mid S_t = s, A_t = a] = r_t(s, a)$ for every $(s, a)$. The system then transits to the next state $S_{t+1}$ according to the transition kernel $P_t(\cdot \mid S_t, A_t)$. The corresponding directed acyclic graph (DAG) is depicted in Figure 1. Different from MDP, the state variable $S_t$ cannot be observed in the POMDP.

In this paper, we focus on finding an optimal history-dependent policy for POMDPs. Let $H_t := (O_1, A_1, ..., O_t, A_t) \in \mathcal{H}_t$, where $\mathcal{H}_t := \prod_{j=1}^t \mathcal{O} \times \mathcal{A}$ denotes the space of observable history up to time $t$. Then at each $t$, the history-dependent policy $\pi_t$ is defined as a function mapping from $\mathcal{O} \times \mathcal{H}_{t-1}$ to $\Delta(\mathcal{A})$. Given such a policy $\pi = \{\pi_t\}_{t=1}^T$, the corresponding value is defined as

$$\mathcal{V}(\pi) := \mathbb{E}^\pi \Big[\sum_{t=1}^T R_t \mid S_1 \sim \nu_1\Big],$$

where $\mathbb{E}^\pi$ is taken with respect to the distribution induced by the policy $\pi$. We aim to develop a policy gradient method in the offline setting to find an optimal policy $\pi^*$ defined as,

$$\pi^* \in \arg\max_{\pi \in \Pi} \mathcal{V}(\pi),$$

where $\Pi$ is a pre-specified class of all history-dependent policies. To achieve this goal, for each decision point $t$, we consider a pre-specified class $\Pi_{\Theta_t}$ to model $\pi_t^*$. A generic element of $\Pi_{\Theta_t}$ is denoted by $\pi_{\theta_t}$, where $\theta_t \in \Theta_t \subset \mathbb{R}^{d_{\Theta_t}}$. The overall policy class is then represented by $\Pi_\Theta = \bigotimes_{t=1}^T \Pi_{\Theta_t}$. Let $\theta := vec(\theta_1, \theta_2, ..., \theta_T) \in \Theta \subset \mathbb{R}^{d_\Theta}$ be the concatenation of the policy parameters in each step, where $d_\Theta = \sum_{t=1}^T d_{\Theta_t}$. Similarly, we denote $\pi_\theta = \{\pi_{\theta_t}\}_{t=1}^T \in \Pi_\Theta$.

In the offline setting, an agent cannot interact with the environment but only has access to an offline dataset generated by some behavior policy $\{\pi_t^b\}_{t=1}^T$. We assume that the behavior policy depends on the unobserved state $S_t$, i.e., $\pi_t^b : \mathcal{S} \to \Delta(\mathcal{A})$ for each $t$. We use $\mathbb{P}^{\pi^b}$ to denote the offline data distribution and summarize the data as $\mathcal{D} := (o_t^n, a_t^n, r_t^n)_{t=1:T}^{n=1:N}$, which are $N$ i.i.d. copies from $\mathbb{P}^{\pi^b}$.

To find an optimal policy $\pi^*$ using the offline data, the policy gradient $\nabla_\theta \mathcal{V}(\pi_\theta) = \nabla_\theta \mathbb{E}^{\pi_\theta}[\sum_{t=1}^T R_t \mid S_1 \sim \nu_1]$ needs to be computed/estimated as an ascent direction when one is searching over the policy parameter space $\Theta$. In the vanilla policy gradient method, one can approximate the optimal policy $\pi^*$ via updating the policy parameter $\theta$ iteratively for finitely many times, i.e., at $(k+1)$-th step, we can obtain $\theta^{(k+1)}$ via

$$\theta^{(k+1)} = \theta^{(k)} + \eta_k \nabla_\theta \mathcal{V}(\pi_\theta)|_{\theta=\theta^{(k)}}, \tag{1}$$

where $\eta_k$ is a pre-specified stepsize. To implement the update rule (1), there are two main issues in our problem: (1) estimation of the policy gradient $\nabla_\theta \mathcal{V}(\pi_\theta)$ based on the offline dataset $\mathcal{D}$ with function approximations and (2) the global convergence of the policy gradient method in POMDPs. The challenge of Problem (1) lies in that the state variable $S_t$ is not observed and the policy gradient may not be identified by the offline data. Furthermore, function approximations are needed when both state and observation spaces are continuous. The challenge of Problem (2) originates from the non-concavity of $\mathcal{V}(\pi_\theta)$, which requires an in-depth analysis of the underlying POMDP structure. Additionally, the limited capacity of the policy class complicates the global convergence of gradient ascent methods when state and observation spaces are continuous. In the following sections, we provide a solution to address these two issues and derive a non-asymptotic upper bound on the suboptimality gap of the output policy given by our algorithm.

**Notations**. Throughout this paper, we assume that $\mathbb{E}$ is taken with respect to the offline distribution, and $\mathbb{E}^{\pi_\theta}$ is taken with respect to distributions induced by $\pi_\theta$. Similarly, we use the notation $X \perp\!\!\!\perp Y \mid Z$ when $X$ and $Y$ are conditionally independent given $Z$ under the offline distribution. For any two sequences $\{a_n\}_{n=1}^\infty$, $\{b_n\}_{n=1}^\infty$, $a_n \lesssim b_n$ denotes $a_n \leq Cb_n$ for some $N, C > 0$ and every $n > N$. If $a_n \lesssim b_n$ and $b_n \lesssim a_n$, then $a_n \asymp b_n$. Big $O$ and $O_P$ are used as conventions. For any policy $\pi$ that depends on the observed data, the suboptimality gap is defined as

$$\mathrm{SubOpt}(\pi) := \mathcal{V}(\pi^\star) - \mathcal{V}(\pi).$$

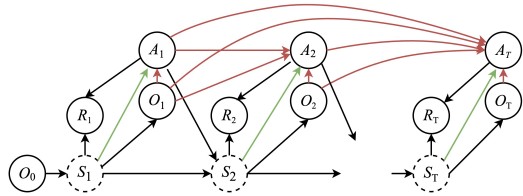

Figure 1: The directed acyclic graph of the data generating process in POMDPs, where states $S_t$ are not observed. Green arrows indicate the generation of actions via the behavior policy, while red arrows indicate the generation through a history-dependent policy.

## 4  POLICY GRADIENT IDENTIFICATION

Since the state variable $S_t$ is unobserved in POMDPs, standard off-policy gradient estimation methods developed in MDPs (Kallus & Uehara, 2020) are no longer applicable. As seen from Figure 1, $S_t$ can be regarded as a time-varying unmeasured confounder as it simultaneously confounds the action $A_t$, the outcome $R_t$ and the future state $S_{t+1}$ in the offline data. Therefore ignoring the effect of $S_t$ will lead to a biased estimation of the policy gradient. To elaborate, we take a partially observable contextual bandit as a simple example. Under some direct calculations, we can show that $\nabla_\theta \mathcal{V}(\pi_\theta) = \mathbb{E}_{S_1 \sim \nu_1}[\sum_a \mathbb{E}[R_1 \nabla_{\theta_1} \pi_{\theta_1}(a \mid O_1) \mid S_1, A_1 = a]]$, which implies that any vanilla estimation procedures based on this equation are not suitable because $S_1$ is not observable - the observable triple $(A_1, O_1, R_1)$ alone is not enough to estimate $\nabla_\theta \mathcal{V}(\pi_\theta)$. One may also think of ignoring $S_1$ and simply applying any standard off-policy gradient estimation method developed in MDPs by treating $O_1$ as the state. However, it can be seen that this *naive* method will incur a non-negligible bias due to $\mathbb{E}[R_1 \mid S_1, O_1, A_1 = a] \neq \mathbb{E}[R_1 \mid O_1, A_1 = a]$ in general. The failure of the naive method is also demonstrated through a simulation study of a toy example (see Appendix K). It can be seen that the naive estimator cannot converge to the true policy gradient no matter how large the sample size would be. In contrast, the proposed estimator (introduced later) is consistent.

In the following, we present a novel identification for addressing the issue of partial observability in POMDPs. Specifically, we develop a non-parametric identification result for estimating $\nabla_\theta \mathcal{V}(\pi_\theta)$ via solving a series of conditional moment equations by using the observed data generated by the behavior policy. Such non-parametric identification results will allow general function approximations for estimating the policy gradient, which is inevitable when state and observation spaces are continuous.

To begin with, we assume the availability of some baseline covariates, represented by $O_0$, that carry some information before the decision-making process. The initial data for all individuals can be recorded as $\{o_0^n\}_{n=1}^N$. To enable the observable trajectory $\{O_t\}_{t=0}^T$ for identifying the policy gradient, we impose Assumption 1 in the following.

**Assumption 1.** *Under the offline distribution $\mathbb{P}^{\pi^b}$, it holds that*

$$O_0 \perp\!\!\!\perp (O_t, O_{t+1}, R_t) \mid S_t, A_t, H_{t-1}, \ \forall t = 1, ..., T. \tag{2}$$

Assumption 1 basically requires $O_0$ is pre-collected before the decision process, which is mild. Next, we rely on the existence of certain *bridge functions* that link the policy gradient and the offline data distribution, which are summarized in the following assumption.

**Assumption 2** (Existence of $V$-bridge and $\nabla V$-bridge functions). *For any $\pi_\theta \in \Pi_\Theta$, there exist real-valued bridge functions $\{b_{V,t}^{\pi_\theta} : \mathcal{A} \times \mathcal{O} \times \mathcal{H}_{t-1} \to \mathbb{R}\}_{t=1}^T$ and vector-valued bridge functions $\{b_{\nabla V,t}^{\pi_\theta} \in \mathcal{A} \times \mathcal{O} \times \mathcal{H}_{t-1} \to \mathbb{R}^{d_\Theta}\}_{t=1}^T$ that satisfy the following conditional moment restrictions:*

$$\mathbb{E}[b_{V,t}^{\pi_\theta}(A_t, O_t, H_{t-1}) \mid A_t, H_{t-1}, O_0] = \mathbb{E}[(R_t \pi_{\theta_t}(A_t \mid O_t, H_{t-1})$$
$$+ \sum_{a' \in \mathcal{A}} b_{V,t+1}^{\pi_\theta}(a', O_{t+1}, H_t) \pi_{\theta_t}(A_t \mid O_t, H_{t-1}) \mid A_t, H_{t-1}, O_0], \tag{3}$$

$$\mathbb{E}[b_{\nabla V,t}^{\pi_\theta}(A_t, O_t, H_{t-1}) \mid A_t, H_{t-1}, O_0] = \mathbb{E}[(R_t + \sum_{a'} b_{V,t+1}^{\pi_\theta}(a', O_{t+1}, H_t))\nabla_\theta$$

$$\pi_{\theta_t}(A_t \mid O_t, H_{t-1}) + \sum_{a'} b_{\nabla V,t+1}^{\pi_\theta}(a', O_{t+1}, H_t)\pi_{\theta_t}(A_t \mid O_t, H_{t-1}) \mid A_t, H_{t-1}, O_0], \tag{4}$$

*where $b_{V,T+1}^{\pi_\theta}$ and $b_{\nabla V,T+1}^{\pi_\theta}$ are set to be 0.*

We refer to $\{b_{V,t}^{\pi_\theta}\}_{t=1}^T$ as value bridge functions, and $\{b_{\nabla V,t}^{\pi_\theta}\}_{t=1}^T$ as gradient bridge functions. Intuitively, the value bridge functions and gradient bridge functions can be understood as bridges that link the value functions as well as their gradients for $\pi_\theta$ with the offline distribution induced by $\pi^b$ in POMDPs. Under some mild regularity conditions stated in Appendix B, one can ensure that Assumption 2 always holds. The usage of such bridge functions for identification was first introduced in the field of proximal causal inference (Miao et al., 2018; Tchetgen et al., 2020), and has been investigated in off-policy evaluation for confounded POMDPs (Bennett & Kallus, 2021; Shi et al., 2022; Miao et al., 2022). Here we generalize the idea along with the following completeness assumption for identifying the policy gradient.

**Assumption 3** (Completeness). *For any measurable function $g_t : \mathcal{S} \times \mathcal{A} \times \mathcal{H}_{t-1} \to \mathbb{R}$, and any $1 \le t \le T$,*
$$\mathbb{E}[g_t(S_t, A_t, H_{t-1}) \mid A_t, H_{t-1}, O_0] = 0$$
*almost surely if and only if $g_t(S_t, A_t, H_{t-1}) = 0$ almost surely.*

Assumption 3 is imposed to identify the policy gradient by using the observable $\{O_0, H_{t-1}\}$ instead of unobservable $S_t$. There are many commonly-used models such as exponential families (Newey & Powell, 2003) and location-scale families (Hu & Shiu, 2018) that can ensure the completeness stated in Assumption 3. The completeness assumption is also widely made in identification problems when there exists unmeasured confounding (Darolles et al., 2011; Miao et al., 2018).

Intuitively, the combination of Assumption 2 and 3 implies that for each action, the confounding effect of the unobservable state $S_t$ on the bridge function matches the confounding effect of the unobservable state $S_t$ on the outcome of interest, i.e., gradients of future cummulative rewards. Hence the bridge function can be used as a good "substitute". In addition, the bridge function can correct the bias because it is conditionally independent of $A_t$ given $S_t$, which allows us to identify the policy value and policy gradient. Finally, under Assumptions 1-3, we obtain the key identification result summarized in Theorem 1.

**Theorem 1** (Policy gradient identification). *Under Assumptions 1-3, for any $\pi_\theta \in \Pi_\Theta$, the policy gradient and policy value for $\pi_\theta$ can be identified as*

$$\nabla_\theta \mathcal{V}(\pi_\theta) = \mathbb{E}[\sum_{a \in \mathcal{A}} b_{\nabla V,1}^{\pi_\theta}(a, O_1)] \text{ and } \mathcal{V}(\pi_\theta) = \mathbb{E}[\sum_{a \in \mathcal{A}} b_{V,1}^{\pi_\theta}(a, O_1)]. \tag{5}$$

According to (5) and conditional moment restrictions (3) and (4), the policy gradient can then be estimated from the offline dataset $\mathcal{D}$. This is due to the fact that both bridge functions and conditional moment restrictions rely solely on the observed data. In Appendix B.6, we use a generic partially observable discrete contextual bandit to illustrate the idea of Theorem 1. This example shows that the policy gradient can be explicitly identified as a form of matrix multiplications using observed variables. In Appendix K, we conduct a simulation study under a partially observable discrete contextual bandit to demonstrate the effectiveness of the proposed identification results.

In the subsequent section, we present a min-max estimation method based on (3-5) utilizing the offline dataset $\mathcal{D}$ when state and observation space is continuous and $T > 1$, and describe a policy gradient ascent algorithm grounded in the estimated policy gradient.

## 5 POLICY GRADIENT ESTIMATION AND OPTIMIZATION

In this section, we estimate $\nabla_\theta \mathcal{V}(\pi_\theta)$ based on offline data $\mathcal{D} = \{o_0^n, (o_t^n, a_t^n, r_t^n)_{t=1}^T\}_{n=1}^N$ and then use a gradient ascent method with the estimated $\nabla_\theta \mathcal{V}(\pi_\theta)$ for policy optimization. For the sake of clarity, we introduce some additional variables used in Sections 5 and 6. Let $\mathcal{Z}_t := \mathcal{O} \times [-1, 1] \times \mathcal{A} \times \mathcal{O} \times \mathcal{H}_{t-1}$, $\mathcal{X}_t := \mathcal{A} \times \mathcal{H}_{t-1} \times \mathcal{O}$ and $\mathcal{W}_t := \mathcal{A} \times \mathcal{O} \times \mathcal{H}_{t-1}$. Then we define three variables

$Z_t \in \mathcal{Z}_t$, $X_t \in \mathcal{X}_t$, $W_t \in \mathcal{W}_t$ as $Z_t := (O_{t+1}, R_t, A_t, O_t, H_{t-1})$, $X_t := (A_t, H_{t-1}, O_0)$ and $W_t := (A_t, O_t, H_{t-1})$.

**Conditional moment restrictions.** By Theorem 1, for each $\pi_\theta \in \Pi_\Theta$, to estimate the policy gradient, it suffices to solve a sequence of conditional moment restrictions:

$$\mathbb{E}[m_V(Z_t; b_{V,t}, b_{V,t+1}, \theta_t) \mid X_t] = 0, \text{ and,} \tag{6}$$

$$\mathbb{E}[m_{\nabla V}(Z_t; b_{\nabla V,t}, b_{V,t+1}, b_{\nabla V,t+1}, \theta_t) \mid X_t] = 0, \forall t = 1, ..., T, \quad \text{where} \tag{7}$$

$$m_V(\cdot) = b_{V,t}(A_t, O_t, H_{t-1}) - (R_t + \sum_{a' \in \mathcal{A}} b_{V,t+1}(a', O_{t+1}, H_t)) \pi_{\theta_t}(A_t \mid O_t, H_{t-1}),$$

$$m_{\nabla V}(\cdot) = b_{\nabla V,t} - (R_t + \sum_{a' \in \mathcal{A}} b_{V,t+1}(a', O_{t+1}, H_t)) \nabla_\theta \pi_{\theta_t} - \sum_{a' \in \mathcal{A}} b_{\nabla V,t+1}(a', O_{t+1}, H_t) \pi_{\theta_t}.$$

Equations (6) and (7) form a sequential nonparametric instrumental variables (NPIV) problem, where $Z_t$ can be regarded as an endogenous variable and $X_t$ as an instrumental variable. Let $b_t := (b_{\nabla V,t}^\top, b_{V,t})^\top : \mathcal{W}_t \to \mathbb{R}^{d_\Theta + 1}$ for each $t$. Define $m(Z_t; b_t, b_{t+1}, \theta_t) := (m_{\nabla V}(Z_t; b_{\nabla V,t}, b_{V,t+1}, b_{\nabla V,t+1}, \theta_t)^\top, m_V(Z_t; b_{V,t}, b_{V,t+1}, \theta_t))^\top : \mathcal{Z}_t \to \mathbb{R}^{d_\Theta + 1}$. Consequently, solving (6) and (7) is equivalent to solving

$$\mathbb{E}[m(Z_t; b_t, b_{t+1}, \theta_t) \mid X_t] = 0, \forall t = 1, ..., T. \tag{8}$$

Therefore, for any measurable $f : \mathcal{X}_t \to \mathbb{R}^{d_\Theta + 1}$, it holds that $\mathbb{E}[m(Z_t; b_t, b_{t+1}, \theta_t)^\top f(X_t)] = 0, \forall t = 1, ..., T$. In this way, we are able to use the unconditional expectations rather than the conditional one (8). Since the moment restriction holds for infinitely many unconstrained $f(\cdot)$'s, a min-max strategy with some pre-specified function classes can be employed to find a solution.

**Min-max estimation procedure.** Motivated by the above discussion, to solve (8), we sequentially adopt the min-max estimator from Dikkala et al. (2020) that permits non-parametric function approximation such as Reproducing Kernel Hilbert Space (RKHS), random forests, and Neural Networks (NN). In particular, let $\widehat{b}_{T+1}^{\pi_\theta} = 0$, we can estimate $b_t^{\pi_\theta}$ sequentially for $t = T, ..., 1$ by solving

$$\widehat{b}_t^{\pi_\theta} = \arg\min_{b \in \mathcal{B}^{(t)}} \sup_{f \in \mathcal{F}^{(t)}} \Psi_{t,N}(b, f, \widehat{b}_{t+1}^{\pi_\theta}, \pi_\theta) - \lambda_N \|f\|_{N,2,2}^2 + \mu_N \|b\|_{\mathcal{B}^{(t)}}^2 - \xi_N \|f\|_{\mathcal{F}^{(t)}}^2, \tag{9}$$

where $\Psi_{t,N}(b, f, \widehat{b}_{t+1}^{\pi_\theta}, \pi_\theta) := \frac{1}{N} \sum_{n=1}^N m(Z_t; b, \widehat{b}_{t+1}^{\pi_\theta}, \theta_t)^\top f(X_t^n)$, the spaces $\mathcal{B}^{(t)} = \{b : \mathcal{W}_t \to \mathbb{R}^{d_\Theta + 1} \mid b_j = 0, \forall 1 \leq j \leq \sum_{i=1}^{t-1} d_{\Theta_i}\}$ are used for modeling the bridge function, and the spaces of the test functions $\mathcal{F}^{(t)} := \{f : \mathcal{X}_t \to \mathbb{R}^{d_\Theta + 1} \mid f_j \in \mathcal{F}_j^{(t)} \text{ with } f_j = 0, \forall 1 \leq j \leq \sum_{i=1}^{t-1} d_{\Theta_i}\}$. In particular, $\mathcal{F}_j^{(t)} = \{f_j : \mathcal{X}_t \to \mathbb{R}\}, j = 1, ..., d_\Theta + 1$ are some user-defined function spaces such as RKHS. $\|f\|_{N,2,2}^2$ is the empirical $\mathcal{L}^2$ norm defined as $\|f\|_{N,2,2} := (\frac{1}{N} \sum_{n=1}^N \|f(x_t^n)\|_{\ell^2}^2)^{1/2}$, and $\|b\|_{\mathcal{B}^{(t)}}^2$, $\|f\|_{\mathcal{F}^{(t)}}^2$ denote the functional norm (see Definition D.3) of $b$, $f$ associated with $\mathcal{B}^{(t)}$, $\mathcal{F}^{(t)}$ respectively. Moreover, $\Psi_{t,N}(b, f, \widehat{b}_{t+1}^{\pi_\theta}, \pi_\theta)$, can be understood as an empirical loss function measuring the violation of (8). Finally, $\lambda_N, \mu_N, \xi_N$ are all tuning parameters. It is worth noting that at each iteration $t$, the first $(t-1)$ blocks of the solution $b_t^{\pi_\theta}$ to (8) are all zero according to the conditional moment restriction (4). Thus this restriction is also necessary to impose when constructing the bridge function class $\mathcal{B}^{(t)}$ and the test function class $\mathcal{F}^{(t)}$ as described above.

**Policy gradient estimation.** After solving (9) for $T$ times, we obtain $\widehat{b}_1^{\pi_\theta}$ and estimate the policy gradient $\nabla_\theta \mathcal{V}(\pi_\theta)$ by a plug-in estimator

$$\widehat{\nabla_\theta \mathcal{V}(\pi_\theta)} = \frac{1}{N} \sum_{n=1}^N [\sum_{a \in \mathcal{A}} \widehat{b}_{1,1:d_\Theta}^{\pi_\theta}(a, o_1^n)], \tag{10}$$

where $\widehat{b}_{1,1:d_\Theta}^{\pi_\theta}$ is formed by the first $d_\Theta$ elements of the vector $\widehat{b}_1^{\pi_\theta}$. The numerical results of an instantiation can be found in Appendix L.

**Policy optimization.** With the estimated policy gradient, we can develop a policy gradient ascent algorithm for learning an optimal policy, i.e., iteratively updating the policy parameter $\theta$ via the rule

$$\theta^{(k+1)} = \theta^{(k)} + \eta_k \widehat{\nabla_{\theta^{(k)}} \mathcal{V}(\pi_{\theta^{(k)}})}.$$

Algorithm 1 summarizes the proposed policy gradient ascent algorithm for POMDP in offline RL. More details can be found in Appendix I.1. Specifically, assuming all function classes are reproducing kernel Hilbert spaces (RKHSs), the min-max optimization problem (9) in step 5 leads to a quadratic concave inner maximization problem and a quadratic convex outer minimization problem. Consequently, a closed-form solution of (9) can be obtained, as demonstrated in Appendix H. Furthermore, we can show that the computational time of Algorithm 1 is of the order $Kd_\Theta N^2 T \max\{T, N\}$, where $K$ is the number of iterations. The details of the derivation can be found in Appendix I.2.

---

**Algorithm 1** Policy gradient ascent for POMDP in offline RL

---

**Input:** Dataset $\mathcal{D}$, step sizes $\{\eta_k\}_{k=0}^{K-1}$, function classes $\{\mathcal{B}^{(t)}\}_{t=1}^T$, $\{\mathcal{F}^{(t)}\}_{t=1}^T$.
1: Initialization: $\theta^{(0)} \in \mathbb{R}^{d_\Theta}$
2: **for** $k$ from 0 to $K-1$ **do**
3:     Initialization: Let $\widehat{b}_{T+1}^{\pi_{\theta^{(k)}}} = 0 \in \mathbb{R}^{d_\Theta+1}$.
4:     **for** $t$ from $T$ to 1 **do**
5:         Solve the optimization problem (9) by plugging in $\widehat{b}_{t+1}^{\pi_{\theta^{(k)}}}$, $\pi_{\theta^{(k)}}$ for $\widehat{b}_t^{\pi_{\theta^{(k)}}}$.
6:     **end for**
7:     Let $\widehat{b}_{1,1:d_\Theta}^{\pi_{\theta^{(k)}}}$ be the vector formed by the first $d_\Theta$ elements of $\widehat{b}_1^{\pi_{\theta^{(k)}}}$.
8:     Compute $\nabla_{\theta^{(k)}} \widehat{\mathcal{V}(\pi_{\theta^{(k)}})} = \frac{1}{N} \sum_{n=1}^N [\sum_{a \in \mathcal{A}} \widehat{b}_{1,1:d_\Theta}^{\pi_{\theta^{(k)}}}(a, o_1^n)]$ and update .
9:     Update $\theta^{(k+1)} = \theta^{(k)} + \eta_k \nabla_{\theta^{(k)}} \widehat{\mathcal{V}(\pi_{\theta^{(k)}})}$.
10: **end for**
**Output:** $\widehat{\pi} \sim \mathrm{Unif}(\{\pi_{\theta_t^{(0)}}\}_{t=1}^T, \{\pi_{\theta_t^{(1)}}\}_{t=1}^T, ..., \{\pi_{\theta_t^{(K-1)}}\}_{t=1}^T)$.

---

## 6 THEORETICAL RESULTS

This section studies theoretical performance of the proposed algorithm in finding $\pi^*$. We show that, given the output $\widehat{\pi}$ from Algorithm 1, under proper assumptions, the suboptimality $\mathrm{SubOpt}(\widehat{\pi}) = \mathcal{V}(\pi^*) - \mathcal{V}(\widehat{\pi})$ converges to 0 as the sample size $N \to \infty$ and the number of iterations $K \to \infty$. In particular, we provide a non-asymptotic upper bound on the suboptimality $\mathrm{SubOpt}(\widehat{\pi})$ that depends on $N$ and $K$. The suboptimality consists of two terms: the statistical error for estimating the policy gradient at each iteration and the optimization error for implementing the gradient ascent algorithm.

### 6.1 STATISTICAL ERROR FOR POLICY GRADIENT

To begin with, we impose the following assumption for analyzing the statistical error.

**Assumption 4.** *The following conditions hold.*
*(a) (Full coverage).* $C_{\pi^b} := \sup_{\theta \in \Theta} \max_{t=1,..,T} (\mathbb{E}^{\pi^b}[(\frac{p_t^{\pi_\theta}(S_t, H_{t-1})}{p_t^{\pi^b}(S_t, H_{t-1})\pi_t^b(A_t|S_t)})^2])^{\frac{1}{2}} < \infty$ *where* $p_t^\pi(\cdot, \cdot)$ *denotes the density function of the marginal distribution of* $(S_t, H_{t-1})$ *following* $\pi$.
*(b) (Richness of* $\mathcal{B} = \{\mathcal{B}^{(t)}\}_{t=1}^T$*).* *For any* $t = 1, ..., T$, $\theta \in \Theta$, *and* $b_{t+1} \in \mathcal{B}^{(t+1)}$, *there exists* $b_t \in \mathcal{B}^{(t)}$ *depending on* $b_{t+1}$ *and* $\theta$ *such that the conditional moment equation (8) is satisfied.*
*(c) (Richness of* $\mathcal{F} = \{\mathcal{F}^{(t)}\}_{t=1}^T$*).* *For any* $t = 1, ..., T$, $\theta \in \Theta$, $b_{t+1} \in \mathcal{B}^{(t+1)}$, $b_t \in \mathcal{B}^{(t)}$, *we have* $\mathbb{E}[m(Z_t; b_t, b_{t+1}, \theta_t) \mid X_t] \in \mathcal{F}^{(t)}$. *For any* $f \in \mathcal{F}^{(t)}$, *we have* $rf \in \mathcal{F}^{(t)}, \forall r \in [-1, 1]$.
*(d) (Uniform boundedness of* $\{\mathcal{B}^{(t)}\}_{t=1}^T$ *and* $\{\mathcal{F}^{(t)}\}_{t=1}^T$*).* *There exist constants* $M_\mathcal{B} > 0$ *and* $M_\mathcal{F} > 0$ *such that for any* $t = 1, ..., T$, $\sup_{o_t, h_{t-1}} \|\sum_a b_t(a, o_t, h_{t-1})\|_{\ell^2} \le M_\mathcal{B}$ *for every* $b_t \in \mathcal{B}^{(t)}$, *and that* $\sup_{x_t} \|f(x_t)\|_{\ell^2} \le M_\mathcal{F}$ *for every* $f \in \mathcal{F}^{(t)}$.
*(e) There exists a constant* $G < \infty$ *such that* $\sup_{t, \theta_t, a_t, o_t, h_{t-1}} \|\nabla_\theta \log \pi_{\theta_t}(a_t \mid o_t, h_{t-1})\|_{\ell^\infty} \le G$.

Assumption 4(a) is a commonly-made full coverage assumption in offline RL (Chen & Jiang, 2019; Xie & Jiang, 2020). It essentially requires that the offline distribution $\mathbb{P}^{\pi^b}$ can calibrate the distribution $\mathbb{P}^{\pi_\theta}$ induced by $\pi_\theta$ for all $\theta$. See Appendix B.4 for more discussions on Assumption 4(a). Assumption 4(b) requires that $\mathcal{B}^{(t)}$ is sufficiently large such that there is no model misspecification error when solving the conditional moment restriction (8). It is often called Bellman closedness in the MDP setting (Xie et al., 2021). Assumption 4(c) requires that the testing function classes

$\{\mathcal{F}^{(t)}\}_{t=1}^{T}$ are large enough to capture the projection of $m(Z_t; b_t, b_{t+1}, \theta_t)$ onto the space $\mathcal{X}_t$. Under this assumption, solving the min-max (9) is equivalent to minimizing $\mathcal{L}^2(\ell^2, \mathbb{P}^{\pi^b})$-norm (see Definition D.4) of $\mathbb{E}[m(Z_t; b_t, \widehat{b}_{t+1}^{\pi_\theta}, \theta_t) \mid X_t]$, which provides the projected residual mean squared error (RMSE) of the min-max estimator $\widehat{b}_t^{\pi_\theta}$ returned by (9). See Dikkala et al. (2020) for more details. Assumptions 4(d)-(e) are two mild technical conditions, which can be easily satisfied.

Next, we present the finite-sample error bound on the statistical error for policy gradient *uniformly* over all $\theta \in \Theta$. All required lemmas and a complete proof are provided in Appendix F.

**Theorem 2** (Policy gradient estimation error). *Under Assumptions 1-4, for some constant $c_1 > 0$, with probability at least $1 - \zeta$ it holds that*

$$\sup_{\theta \in \Theta} \|\nabla_\theta \mathcal{V}(\pi_\theta) - \widehat{\nabla_\theta \mathcal{V}(\pi_\theta)}\|_{\ell^2} \lesssim \tau_{\max} C_{\pi^b} T^{\frac{5}{2}} M_{\mathcal{B}} M_{\mathcal{F}} d_\Theta \sqrt{\frac{\log(c_1 T/\zeta) \gamma(\mathcal{F}, \mathcal{B})}{N}}, \quad (11)$$

*where $\tau_{\max}$ is a measure of ill-posedness (See Definition D.1), and $\gamma(\mathcal{F}, \mathcal{B})$ measures the complexity of user-defined bridge function classes and test function classes and is independent of $T, d_\Theta, N$.*

As seen from Theorem 2, the estimation error achieves an optimal statistical convergence rate in the sense that $\sup_{\theta \in \Theta} \|\nabla_\theta \mathcal{V}(\pi_\theta) - \widehat{\nabla_\theta \mathcal{V}(\pi_\theta)}\|_{\ell^2} = O_P(1/\sqrt{N})$. The ill-posedness measure $\tau_{\max}$ quantifies how hard to obtain RMSE from the projected RMSE of $\widehat{b}_t^{\pi_\theta}$. It is commonly used in the literature of conditional moment restrictions (e.g., Chen & Pouzo (2012)). The term $d_\Theta$ comes from the dimension of the target parameter (i.e., policy gradient) and a need for an upper bound uniformly over $\theta \in \Theta$. The upper bound $\mathcal{M}_\mathcal{B}$ can be understood as the size of gradient that scales with $T^2$ as discussed in Appendix B. The term $\gamma(\mathcal{F}, \mathcal{B})$ can be quantified by VC-dimension or metric entropy of the user-defined function classes. For example, when $\{\mathcal{F}_j^{(t)}\}_{j=1,t=1}^{d_\Theta+1,T}$, $\{\mathcal{B}_j^{(t)}\}_{j=1,t=1}^{d_\Theta+1,T}$ are all linear function classes. i.e. $\mathcal{B}_j^{(t)} = \{\phi_{j,t}(\cdot)^T \omega : \omega \in \mathbb{R}^d\}$ and $\mathcal{F}_j^{(t)} = \{\psi_{j,t}(\cdot)^T \omega : \omega \in \mathbb{R}^d\}$, then $\gamma(\mathcal{B}, \mathcal{F}) \asymp d$. When $\{\mathcal{F}_j^{(t)}\}_{j=1,t=1}^{d_\Theta+1,T}$, $\{\mathcal{B}_j^{(t)}\}_{j=1,t=1}^{d_\Theta+1,T}$ are all general RKHS, then $\gamma(\mathcal{F}, \mathcal{B})$ is quantified by the speed of eigen-decay for RKHS. More results for $\gamma(\mathcal{F}, \mathcal{B})$ are provided in Appendix C.

## 6.2 SUBOPTIMALITY

We then investigate the computational error of the proposed algorithm and establish a non-asymptotic upper bound on the suboptimality of $\widehat{\pi}$. Firstly, we present the following assumption.

**Assumption 5.** *The following conditions hold.*
*(a) ($\beta$-smoothness). For $1 \le t \le T$ and any $a_t \in \mathcal{A}, o_t \in \mathcal{O}, h_{t-1} \in \mathcal{H}_{t-1}$, there exists a constant $\beta > 0$, such that $\|\nabla_{\theta_t} \log \pi_{\theta_t}(a_t \mid o_t, h_{t-1}) - \nabla_{\theta_t} \log \pi_{\theta'_t}(a_t \mid o_t, h_{t-1})\|_{\ell^2} \le \beta \|\theta_t - \theta'_t\|_{\ell^2}$.*
*(b) (Positive definiteness of Fisher information matrix). For each $t = 1, ..., T$ and any $\theta \in \Theta$, let $F_t(\theta) := \mathbb{E}^{\pi_\theta}[\nabla_{\theta_t} \log \pi_{\theta_t}(A_t \mid O_t, H_{t-1}) \nabla_{\theta_t} \log \pi_{\theta_t}(A_t \mid O_t, H_{t-1})^\top]$, then the matrix $F_t(\theta) - \mu \cdot I_{d_{\Theta_t}}$ is positive semidefinite for some positive constant $\mu$.*
*(c) (L-smoothness). $\mathcal{V}(\pi_\theta)$ is L-smooth with respect to $\theta$ as per Definition D.6 in Appendix D.*

Assumption 5(a) is satisfied by a wide range of policy classes. For example, the well-known log-linear policy classes satisfy Assumption 5(a) (see remark 6.7 of Agarwal et al. (2021)). The positive definiteness of $F_t(\theta)$ in Assumption 5(b) is also commonly used in the literature related to the (natural) policy gradient methods. See Appendix B.5 for more discussion. Assumption 5(c) is satisfied if Assumption 4(e) holds in our setting. Indeed, $L$ scales with $T^4$ in the POMDP setting. See Appendix B for further discussion.

Next, we provide a non-asymptotic upper bound for $\mathrm{SubOpt}(\widehat{\pi})$ in terms of all key parameters.

**Theorem 3** (Suboptimality). *Under Assumptions 1-5, with probability at least $1 - \zeta$, for some $c > 0$, we have*

$$\mathcal{V}(\pi_{\theta^*}) - \max_{0 \le k \le K-1} \mathcal{V}(\pi_{\theta^{(k)}}) \lesssim \underbrace{\frac{(1 + \frac{1}{\mu})\sqrt{d_\Theta}}{\sqrt{K}} T^{4.5}}_{\text{optimization error}} + \underbrace{(1 + \frac{1}{\mu}) \tau_{\max} C_{\pi^b} T^{5.5} d_\Theta \sqrt{\frac{\log(cT/\zeta)}{N}}}_{\text{statistical error}}$$

$$+ \underbrace{(1 + \frac{1}{\mu})\sqrt{K}\tau_{\max}^2 C_{\pi^b}^2 T^{9.5} \frac{d_\Theta^{2.5} \log(cT/\zeta)}{N}}_{\text{compound error between statistical and optimization errors}} + \varepsilon_{approx}, \quad (12)$$

*where $\varepsilon_{approx}$ is defined in Definition D.2 in Appendix D, which is used to measure the expressive power of policy class in finding the optimal policy. In particular, when $K \asymp N$, we have*

$$\text{SubOpt}(\widehat{\pi}) = O_P((1 + \frac{1}{\mu})\tau_{\max}^2 C_{\pi^b}^2 T^{9.5} d_\Theta^{2.5} \log(T)\frac{1}{\sqrt{N}} + (1 + \frac{1}{\mu})\frac{\sqrt{d_\Theta}}{\sqrt{K}}T^{4.5}) + \varepsilon_{approx}.$$

According to Theorem 3, we have an upper bound on the suboptimality gap in terms of statistical error, optimization error and an approximation error. As we can see, $\text{SubOpt}(\widehat{\pi}) = O_P(\frac{1}{\sqrt{N}} + \frac{1}{\sqrt{K}}) + \varepsilon_{approx}$, which matches the existing result in MDPs (Theorem 3 of Xu et al. (2021)). In Appendix B.7, we further discuss how finding a history-dependent optimal policy in our confounded POMDP setting affects the suboptimality and computational time of the proposed algorithm compared with that in the standard MDP setting.

## 7 NUMERICAL RESULTS

We evaluate the performance of Algorithm 1 by conducting a simulation study using RKHS endowed with a Gaussian kernel. Details of of the simulation setup can be found in Appendix L. Figure 2 summarizes the performance of three methods: the proposed method, the naive method and behavioral cloning. According to Figure 2, the proposed method can find the in-class optimal policy, since the proposed policy gradient estimation procedure has uniformly good performance over the policy class. Consequently, at each iteration, the proposed policy gradient estimator can be sufficiently close to the true policy gradient and find the correct update direction in the optimization step. In contrast, the naive method cannot achieve the in-class optimal policy no matter how large the number $K$ of iterations is, because there exists an irreducible bias in the policy gradient estimation at each iteration. Furthermore, the performance of behavior cloning is significantly worse than the proposed algorithm, since the behavior cloning only clones the behavior policy that generates data instead of finding the optimal actions. This demonstrates the superior performance of our method in finding an optimal policy in the confounded POMDP.

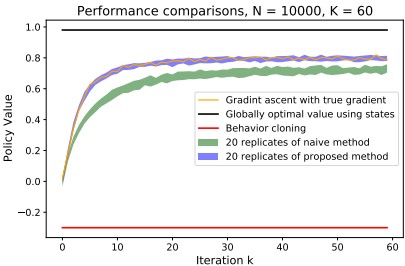

Figure 2: Policy values versus iterations under different methods.

## 8 DISCUSSION AND LIMITATIONS

In this paper, we propose the first policy gradient method for POMDPs in the offline setting. Under some technical conditions, we establish a non-asymptotic upper bound on suboptimality, which is polynomial in all key parameters. There are several promising directions for future research. For example, it will be interesting to study if our suboptimality bound is minimax optimal and explore more efficient algorithms under the current setting. In addition, by leveraging the idea of pessimism, one may develop an algorithm by only requiring the partial coverage, relaxing Assumption 4(a). Lastly, applying the proposed algorithm in practical RL problems with unobserved state variables could be intriguing.

ACKNOWLEDGMENTS

The authors wish to thank anonymous reviewers for their feedback on an early version of this paper. The work of Xu was supported by NSF 1918854, NSF 1940107, and NIH R01MH128085.

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

CONTENTS

# A  LIST OF NOTATIONS

Table 1: List of notations

| Notations | Descriptions |
|---|---|
| $\mathcal{M}$ | an episodic POMDP |
| $T$ | length of horizon |
| $S_t \in \mathcal{S}$ | unobserved state at stage $t$ and state space |
| $O_t \in \mathcal{O}$ | observed variable at stage $t$ and space of observation |
| $A_t \in \mathcal{A}$ | action at stage $t$ and discrete action space |
| $\nu_1$ | distribution of the initial state |
| $\{P_t\}_{t=1}^T$ | collection of state transition kernels over $\mathcal{S} \times \mathcal{A}$ to $\mathcal{S}$ |
| $\{\mathcal{E}_t\}_{t=1}^T$ | collection of observation emission kernels over $\mathcal{S}$ to $\mathcal{O}$ |
| $\{r_t\}_{t=1}^T$ | collection of reward functions, i.e. $r_t : \mathcal{S} \times \mathcal{A} \to [-1, 1]$ |
| $H_t$ | history $(O_1, A_1, ..., O_t, A_t)$ |
| $\pi = \{\pi_t\}_{t=1}^T$ | a history-dependent policy |
| $\pi^b = \{\pi_t^b\}_{t=1}^T$ | the behaviour policy |
| $\pi_\theta = \{\pi_{\theta_t}\}_{t=1}^T$ | a parameterized policy |
| $d_\Theta$ | dimension of policy parameter space |
| $\mathbb{E}^\pi$ | expectation w.r.t. the distribution induced by any policy $\pi$ |
| $\mathbb{E}$ | expectation taken w.r.t. offline distribution |
| $\mathcal{V}(\pi)$ | policy value of $\pi$ defined as $\mathbb{E}^\pi[\sum_{t=1}^T R_t \mid S_1 \sim \nu_1]$ |
| $\mathcal{D}$ | offline data $\{o_0^n, (o_t^n, a_t^n, r_t^n)_{t=1}^T\}_{n=1}^N$ |
| $X \perp\!\!\!\perp Y \mid Z$ | $X$ and $Y$ are conditionally independent given $Z$ |
| $b_{V,t}^{\pi_\theta}, \widehat{b}_{V,t}^{\pi_\theta}$ | true and estimated value-bridge functions at $t$ |
| $b_{\nabla V,t}^{\pi_\theta}, \widehat{b}_{\nabla V,t}^{\pi_\theta}$ | true and estimated gradient-bridge functions at $t$ |
| $b_{V,t}$ | an element in the value-bridge function class |
| $b_{\nabla V,t}$ | an element in the gradient-bridge function class |
| $b_t$ | concatenation of $b_{\nabla V,t}$ and $b_{V,t}$ |
| $f$ | an element in the test function class |
| $m$ | moment function defined in (8) |
| $Z_t$ | argument of $m$, defined as $(O_{t+1}, R_t, A_t, O_t, H_{t-1})$ |
| $W_t$ | argument of $b_t$, defined as $(A_t, O_t, H_{t-1})$ |
| $X_t$ | argument of $f$, defined as $(A_t, H_{t-1}, O_0)$ |
| $\mathcal{B}^{(t)}$ | the bridge function class |
| $\mathcal{F}^{(t)}$ | the test function class |
| $\|f\|_{2,2}$ | population $\mathcal{L}^2$ norm, defined as $\left(\mathbb{E}_{X \sim \mathbb{P}^{\pi^b}} \|h(X)\|_{\ell^2}^2\right)^{1/2}$ |
| $\|f\|_{N,2,2}$ | empirical $\mathcal{L}^2$ norm, defined as $\left(\frac{1}{N}\sum_{n=1}^N \|f(x_n)\|_{\ell^2}^2\right)^{1/2}$ |
| $\|b\|_{\mathcal{B}^{(t)}}$ | a function norm associated with $\mathcal{B}^{(t)}$ |
| $\|f\|_{\mathcal{F}^{(t)}}$ | a function norm associated with $\mathcal{F}^{(t)}$ |
| $\Psi_{t,N}$ | empirical loss function in (9) |
| $\widehat{\nabla_\theta \mathcal{V}(\pi_\theta)}$ | estimated policy gradient of $\theta$ |
| $\theta^{(k)}$ | the policy parameter at the $k-th$ iteration in gradient ascent |
| $\eta_k$ | step size of gradient ascent |
| $\widehat{\pi}$ | the output policy |
| $\lambda_N, \mu_N, \xi_N$ | tunning parameters in (9) |
| $C_{\pi^b}$ | concentratability coefficient defined in Assumption 4 |
| $\tau_{\max}$ | a measure of ill-posedness defined in D.1 |
| $M_\mathcal{B}, M_\mathcal{F}$ | the upper bounds for bridge & test function classes (Assumption 4) |
| $\gamma(\mathcal{F}, \mathcal{B})$ | a complexity measure of bridge & test function classes |
| $\mu$ | the smallest eigenvalue of Fisher information matrix (Assumption 5) |
| $K$ | number of iterations in gradient ascent |
| $\varepsilon_{approx}$ | transferred compatible function approximation error (Definition D.2) |

## B  FURTHER DISCUSSIONS

### B.1  EXISTENCE OF BRIDGE FUNCTIONS

In this section, we discuss sufficient conditions for Assumption 2. Assumption 2 is about the existence of solutions to a sequence of linear integral equations. A rigorous study on this problem is to utilize tools from singular value decomposition in functional analysis (Kress et al., 1989). Specifically, we present the following result from Kress et al. (1989).

**Lemma 1** (Theorem 15.16 in Kress et al. (1989)). *Given Hilbert spaces $H_1$ and $H_2$, a compact operator $K : H_1 \longmapsto H_2$ and its adjoint operator $K^* : H_2 \longmapsto H_1$, there exists a singular system $(\lambda_n, \varphi_n, \psi_n)_{n=1}^{+\infty}$ of $K$ with nonzero singular values $\{\lambda_n\}$ and orthogonal sequences $\{\varphi_n \in H_1\}$ and $\{\psi_n \in H_2\}$ such that*

$$K\varphi_n = \lambda_n \psi_n, \quad K^*\psi_n = \lambda_n \varphi_n.$$

**Lemma 2** (Theorem 15.18 in Kress et al. (1989)). *Given Hilbert spaces $\mathcal{H}_1$ and $\mathcal{H}_2$, a compact operator $K : \mathcal{H}_1 \to \mathcal{H}_2$ and its adjoint operator $K^* : \mathcal{H}_2 \to \mathcal{H}_1$, there exists a singular system $(\lambda_\nu, \phi_\nu, \psi_\nu)_{\nu=1}^\infty$ of $K$, with singular values $\{\lambda_\nu\}$ and orthogonal sequences $\{\phi_\nu\} \subset \mathcal{H}_1$ and $\{\psi_\nu\} \subset \mathcal{H}_2$ such that $K\phi_\nu = \lambda_\nu \psi_\nu$ and $K^*\psi_\nu = \lambda_\nu \phi_\nu$. Given $g \in \mathcal{H}_2$, the Fredholm integral equation of the first kind $Kh = g$ is solvable if and only if*
*(a) $g \in \mathrm{Ker}\,(K^*)^\perp$ and*
*(b) $\sum_{\nu=1}^\infty \lambda_\nu^{-2} |\langle g, \psi_\nu \rangle|^2 < \infty$ where $\mathrm{Ker}\,(K^*) = \{h : K^*h = 0\}$ is the null space of $K^*$, and $\perp$ denotes the orthogonal complement to a set.*

We then present the following sufficient conditions for the existence of bridge functions.

For a probability measure function $\mu$, let $\mathcal{L}^2\{\mu(x)\}$ denote the space of all squared integrable functions of $x$ with respect to measure $\mu(x)$, which is a Hilbert space endowed with the inner product $\langle g_1, g_2 \rangle = \int g_1(x) g_2(x) \mathrm{d}\mu(x)$. For all $a_t, h_{t-1}, t$, define the following operator

$$K_{a_t, h_{t-1}; t} : \mathcal{L}^2 \left\{ \mu_{O_t | A_t, H_{t-1}}(o_t \mid a_t, h_{t-1}) \right\} \to \mathcal{L}^2 \left\{ \mu_{O_0 | A_t, H_{t-1}}(o_0 \mid a_t, h_{t-1}) \right\}$$
$$h \mapsto \mathbb{E} \left\{ h\left(O_t, A_t, H_{t-1}\right) \mid O_0 = o_0, A_t = a_t, H_{t-1} = h_{t-1} \right\},$$

and its adjoint operator

$$K_{a_t, h_{t-1}; t}^* : \mathcal{L}^2 \left\{ \mu_{O_0 | A_t, H_{t-1}}(o_0 \mid a_t, h_{t-1}) \right\} \to \mathcal{L}^2 \left\{ \mu_{O_t | A_t, H_{t-1}}(o_t \mid a_t, h_{t-1}) \right\}$$
$$g \mapsto \mathbb{E} \left\{ g\left(O_0, A_t = a_t, H_{t-1}\right) \mid O_t = o_t, A_t = a_t, H_{t-1} = h_{t-1} \right\}$$

**Assumption 6** (Completeness). . *For any measurable function $g_t : \mathcal{O} \times \mathcal{A} \times \mathcal{H}_{t-1} \to \mathbb{R}$, and any $1 \leq t \leq T$,*

$$\mathbb{E}\left[g_t\left(O_0, A_t, H_{t-1}\right) \mid O_t, A_t, H_{t-1}\right] = 0$$

*almost surely if and only if $g_t\left(O_0, A_t, H_{t-1}\right) = 0$ almost surely.*

**Assumption 7** (Regularities Assumptions). *For any $O_o = o_0, O_t = o_t, A_t = a_t, H_{t-1} = h_{t-1}$ and $1 \leq t \leq T$,*
*(a) $\iint_{\mathcal{O} \times \mathcal{O}} f_{O_t | O_0, A_t, H_{t-1}}(o_t \mid o_0, a_t, h_{t-1}) f_{O_0 | O_t, A_t, H_{t-1}}(o_0 \mid o_t, a_t, h_{t-1}) \mathrm{d}w \, \mathrm{d}z < \infty$, where $f_{O_t | O_0, A_t, H_{t-1}}$ and $f_{O_0 | O_t, A_t, H_{t-1}}$ are conditional density functions.*
*(b) For any uniformly bounded $g_1, g_2, g_3 : \mathcal{O} \times \mathcal{H}_t \to \mathbb{R}$, $\theta \in \Theta$,*

$$\int_{\mathcal{O}} \left[\mathbb{E}\left\{ (R_t + g_1(O_{t+1}, H_t)) \pi_{\theta_t}(A_t \mid O_t, H_{t-1}) \mid O_0 = o_0, A_t = a_t, H_{t-1} = h_{t-1} \right\}\right]^2 \tag{13}$$
$$f_{O_0 | A_t, H_{t-1}}(o_0 \mid a_t, h_{t-1}) \mathrm{d}o_0 < \infty.$$

*and*

$$\int_{\mathcal{O}} [\mathbb{E}\{ (R_t + g_2(O_{t+1}, H_t))(\nabla_\theta \pi_{\theta_t}(A_t \mid O_t, H_{t-1}))_i + g_2(O_{t+1}, H_t) \pi_{\theta_t}(A_t \mid O_t, H_{t-1})$$

$$\mid O_0 = o_0, A_t = a_t, H_{t-1} = h_{t-1} \}]^2 f_{O_0 | A_t, H_{t-1}}(o_0 \mid a_t, h_{t-1}) \mathrm{d}o_0 < \infty, \text{ for each } i = 1, ..., d_\Theta$$

*(c) There exists a singular decomposition $\left(\lambda_{a_t, h_{t-1}; t; \nu}, \phi_{a_t, h_{t-1}; t; \nu}, \psi_{a_t, h_{t-1}; t; \nu}\right)_{\nu=1}^\infty$ of $K_{a_t, h_{t-1}; t}$ such that for all uniformly bounded $g_1, g_2, g_3 : \mathcal{O} \times \mathcal{H}_t \to \mathbb{R}$, $\theta \in \Theta$*

$$\sum_{\nu=1}^\infty \lambda_{a_t, h_{t-1}; t; \nu}^{-2} |\langle \mathbb{E}\{ (R_t + g_1(O_{t+1}, H_t)) \pi_{\theta_t}(A_t \mid O_t, H_{t-1}) \tag{14}$$

$$\mid O_0 = o_0, A_t = a_t, H_{t-1} = h_{t-1} \}, \psi_{a_t, h_{t-1}; t; \nu} \rangle|^2 < \infty.$$

*and*

$$\sum_{\nu=1}^{\infty} \lambda_{a_t,h_{t-1};t;\nu}^{-2} \left| \langle \mathbb{E} \{ (R_t + g_2(O_{t+1}, H_t))(\nabla_\theta \pi_{\theta_t}(A_t \mid O_t, H_{t-1}))_i + g_2(O_{t+1}, H_t)\pi_{\theta_t} \right.$$

$$\left. (A_t \mid O_t, H_{t-1}) \mid O_0 = o_0, A_t = a_t, H_{t-1} = h_{t-1} \}, \psi_{a_t,h_{t-1};t;\nu} \rangle \right|^2 < \infty, \textit{ for each } i = 1, ..., d_\Theta.$$

Next, we prove that completeness (Assumption 6) and regularities (Assumption 7) are sufficient conditions for the existence of bridge functions that satisfy Assumption 2.

*Proof.* For $t = T, \ldots, 1$, by Assumption 7(a), $K_{a_t,h_{t-1};t}$ is a compact operator for each $(a_t, h_{t-1}) \in \mathcal{A} \times \mathcal{H}_{t-1}$ (Example 2.3 in Carrasco et al. (2007)), so there exists a singular value system stated in Assumption 7(c) according to Lemma 1. Then by Assumption 3, we have $\text{Ker}\left(K_{a_t,h_{t-1};t}^*\right) = 0$, since for any $g \in \text{Ker}\left(K_{a_t,h_{t-1};t}^*\right)$, we have, by the definition of Ker, $K_{a_t,h_{t-1};t}^* g = \mathbb{E}[g(O_0, A_t, H_{t-1}) \mid O_t, A_t = a_t, H_{t-1} = h_{t-1}] = 0$, which implies that $g = 0$ a.s.. Therefore $\text{Ker}\left(K_{a_t,h_{t-1};t}^*\right) = 0$ and $\text{Ker}\left(K_{a_t,h_{t-1};t}^*\right)^\perp = \boldsymbol{L}^2\left(\mu_{O_0|A_t,H_{t-1}}(o_0 \mid a_t, h_{t-1})\right)$. Consequently, by Assumption 7(b), all the terms on the right-hand side of equations (3)(4) are actually in $\text{Ker}\left(K_{a_t,h_{t-1};t}^*\right)$ for every $a_t \in \mathcal{A}, h_{t-1} \in \mathcal{H}_{t-1}$. Therefore condition (a) in Lemma 2 has been verified. Further, condition (b) in Lemma 2 is also satisfied according to Assumption 7(c). Therefore, all the conditions in Lemma 2 are satisfied by recursively applying the above argument from $t = T$ to $t = 1$, which ensures the existence of solutions to the linear integral equations (3)(4). □

## B.2 Scale of $M_\mathcal{B}$.

In this section, we discuss the scale of $M_\mathcal{B}$ which is the upper bound for the bridge function classes. We first consider the scale of $b_{V,t}^{\pi_\theta}$. We notice that for every $t$ and any $\theta$, we need to find a solution $b_{V,t}$ that satisfies

$$\mathbb{E}[b_{V,t}(A_t, O_t, H_{t-1}) \mid O_0, A_t, H_{t-1}] = \mathbb{E}[(R_t + \sum_a b_{V,t+1}(a, O_{t+1}, H_t))\pi_{\theta_t} \mid O_0, A_t, H_{t-1}].$$

Now we consider the solution $b_{V,t}^{\pi_\theta}$. In particular, according to the proofs for identification in Appendix E, at $t = T$, we have

$$\mathbb{E}[\sum_a b_{V,T}^{\pi_\theta}(a, O_T, H_{T-1}) \mid O_0, H_{T-1}] = \mathbb{E}[R_T \mid O_0, H_{T-1}].$$

Since $|R_T| \leq 1$, we need to guarantee $|\mathbb{E}[\sum_a b_{V,T}^{\pi_\theta}(a, O_T, H_{T-1}) \mid O_0, H_{T-1}]| \leq 1$. To this end, a sufficient condition should be set as $\|\sum_a b_{V,T}^{\pi_\theta}(a, O_T, H_{T-1})\|_\infty \leq 1$. Similarly, for each $t$, we have

$$|\mathbb{E}[\sum_a b_{V,t}(a, O_t, H_{t-1}) \mid O_0, H_{t-1}]| \leq |\mathbb{E}[\sum_{j=t}^T R_j \mid O_0, H_{t-1}]| \leq T - t + 1 \qquad (15)$$

Therefore, a sufficient condition should be set as $\|\sum_a b_{V,t}(a, O_t, H_{t-1}) \mid O_0, H_{t-1}]\|_\infty \leq T-t+1$. Therefore the upper bound for the value bridge function class should scale with $T$.

Next, we consider $b_{\nabla V,t}^{\pi_\theta}$. According to equation (169) in the proof of identification result, we have

$$\mathbb{E}\left[\sum_a b_{\nabla V,t}^{\pi_\theta}(a, O_t, H_{t-1}) \mid S_t, H_{t-1}\right]$$
$$= \sum_a \mathbb{E}\left[R_t \nabla_\theta \pi_{\theta_t}(a \mid O_t, H_{t-1}) \mid S_t, H_{t-1}, A_t = a\right]$$

$$+ \sum_a \mathbb{E}\left[\sum_{a'} b^{\pi_\theta}_{\nabla V,t+1}(a', O_{t+1}, H_t)\pi_{\theta_t}(a \mid O_t, H_{t-1}) \mid S_t, H_{t-1}, A_t = a\right]$$

$$+ \sum_a \mathbb{E}\left[\sum_{a'} b^{\pi_\theta}_{V,t+1}(a', O_{t+1}, H_t)\nabla_\theta\pi_{\theta_t}(a \mid O_t, H_{t-1}) \mid S_t, H_{t-1}, A_t = a\right]$$

$$= \sum_a \mathbb{E}\left[R_t\pi_{\theta_t}(a \mid O_t, H_{t-1})\nabla_\theta \log \pi_{\theta_t}(a \mid O_t, H_{t-1}) \mid S_t, H_{t-1}, A_t = a\right] \tag{16}$$

$$+ \sum_a \mathbb{E}\left[\sum_{a'} b^{\pi_\theta}_{\nabla V,t+1}(a', O_{t+1}, H_t)\pi_{\theta_t}(a \mid O_t, H_{t-1}) \mid S_t, H_{t-1}, A_t = a\right]$$

$$+ \sum_a \mathbb{E}\left[\sum_{a'} b^{\pi_\theta}_{V,t+1}(a', O_{t+1}, H_t)\pi_{\theta_t}(a \mid O_t, H_{t-1})\nabla_\theta \log \pi_{\theta_t}(a \mid O_t, H_{t-1}) \mid S_t, H_{t-1}, A_t = a\right]$$

At $t$, we notice that $\sum_a \pi_{\theta_t}(a \mid O_t, H_{t-1}) = 1$ for each $(O_t, H_{t-1})$. In addition, we have $\|\log \pi_{\theta_t}(a \mid O_t, H_{t-1})\|_{\ell^\infty} \leq G$ by Assumption 4(e). Therefore, we have a relationship that

$$\left\|\mathbb{E}\left[\sum_a b^{\pi_\theta}_{\nabla V,t}(a, O_t, H_{t-1}) \mid S_t, H_{t-1}\right]\right\|_{\ell^\infty} \leq G \sum_{k=t}^{T}(1 + \|b^{\pi_\theta}_{V,k+1}\|_\infty) = G \sum_{k=t}^{T}(T - k + 1). \tag{17}$$

Therefore, a sufficient condition on $\sum_a b^{\pi_\theta}_{\nabla V,t}(a, O_t, H_{t-1})$ should be

$$\sup_{o_t, h_{t-1}} \left\|\sum_a b^{\pi_\theta}_{\nabla V,t}(a, o_t, h_{t-1})\right\|_{\ell^\infty} \leq G(T - t + 1)^2$$

. Furthermore, since $\|\cdot\|_{\ell^\infty} \leq \|\cdot\|_{\ell^2}$, a sufficient condition on the norm $\|\cdot\|_{\ell^2}$ could also be $\sup_{o_t, h_{t-1}} \|\sum_a b^{\pi_\theta}_{\nabla V,t}(a, o_t, h_{t-1})\|_{\ell^2} \leq G(T - t + 1)^2$.

Recall that $M_\mathcal{B}$ denotes the uniform upper bound over all $t$, therefore $M_\mathcal{B}$ scales with $T^2$ under our settings.

## B.3 SCALE OF $L$

In this section, we consider the scale of the Lipschitz constant $L$ such that

$$\|\nabla_\theta \mathcal{V}(\pi_\theta) - \nabla_{\theta'} \mathcal{V}(\pi_{\theta'})\| \leq L\|\theta - \theta'\|.$$

It suffices to consider the upper bound of the scale of the Hessian matrix of $\mathcal{V}(\pi_\theta)$: $\sup_\theta \|\nabla^2_\theta \mathcal{V}(\pi_\theta)\|$. We adopt the result from Proposition 5.2 in Xu et al. (2020) in the MDPs settings. The key to their proof is that $\nabla_\theta \log p_{\pi_\theta}(\tau) = \sum_{t=1}^{T} \nabla_\theta \log \pi_{\theta_t}$ where $\tau$ denotes the trajectory generated according to $\pi_\theta$. This result also holds under the POMDP settings by adding the unobserved $s_t$ and history $h_{t-1}$. See equation (148) for verification. Consequently, following the proof of Proposition 5.2 in Xu et al. (2020), we get $\sup_\theta \|\nabla^2_\theta \mathcal{V}(\pi_\theta)\|$ scales with $\tilde{G}^2 T^3$ where $\tilde{G} := \sup_{t,a_t,o_t,h_{t-1}} \|\nabla_\theta \log \pi_t(a_t \mid o_t, h_{t-1})\|_{\ell^2}$ scales with $\sqrt{T}$ in our settings. Therefore, we have that $L$ scales with $T^4$.

## B.4 ASSUMPTION 4(A)

Assumption 4(a) requires that the offline distribution $\mathbb{P}^{\pi^b}$ can calibrate the distribution $\mathbb{P}^{\pi_\theta}$ induced by $\pi_\theta$ for all $\theta$, which might not be satisfied in some practical scenarios. In the following, we discuss on practical scenarios where this assumption is not satisfied and propose a potential way to address this concern.

In certain real-world applications, such as the sequential multiple assignment randomized trials (SMART) designed to build optimal adaptive interventions, the assumption of full coverage is usually satisfied. This is because data collection in these trials is typically randomized, ensuring a comprehensive representation by the behavior policy. However, we note that in domains such as

electronic medical records, meeting the full coverage assumption may pose challenges due to ethical or logistical constraints.

To address scenarios where the full coverage assumption might not hold, we could incorporate the principle of pessimism into our approach. This involves penalizing state-action pairs that are rarely visited under the offline distribution. The idea of incorporating pessimism has been widely used in the offline RL literature for MDPs. For example, a practical implementation of this idea can be adapted from Zhan et al. (2022), where a regularity term is added to the objective function to measure the discrepancy between the policy of interest and the behavior policy. By identifying and estimating the gradient of this modified objective function, we could potentially provide an upper bound on suboptimality and maintain a similar theoretical result by only assuming partial coverage, i.e.

$$C_{\pi^b}^{\pi_{\theta^*}} := \max_{t=1,..,T}(\mathbb{E}^{\pi^b}[(\frac{p_t^{\pi_{\theta^*}}(S_t, H_{t-1})}{p_t^{\pi^b}(S_t, H_{t-1})\pi_t^b(A_t \mid S_t)})^2])^{\frac{1}{2}} < \infty.$$

This partial coverage assumption only needs that the offline distribution $\mathbb{P}^{\pi^b}$ can calibrate the distribution $\mathbb{P}^{\pi_{\theta^*}}$ induced by the in-class optimal policy, which is a milder condition compared to the full coverage assumption.

## B.5 ASSUMPTION 5(C)

Policies that satisfy Assumption 5(c) are named Fisher-non-degenerate policies in the field of (natural) policy gradient. This assumption was assumed originally in the pioneering works on the natural policy gradient and the natural actor-critic method. See for example Kakade (2001); Peters & Schaal (2008a); Bhatnagar et al. (2009). Recently, there is also a line of work studying the policy gradient-based algorithms by assuming Fisher-non-degenerate policies. See for example Zhang et al. (2020); Liu et al. (2020); Xu et al. (2021); Ding et al. (2022); Masiha et al. (2022); Yuan et al. (2022); Fatkhullin et al. (2023). We summarize common policy classes that satisfy this assumption: the Gaussian policies with a full row rank feature map, a subclass of neural net mean parametrization, full-rank exponential family distributions with a parametrized mean, etc. For more examples, we direct you to section 8 of Ding et al. (2022), section B.2 of Liu et al. (2020), and Appendix B of Fatkhullin et al. (2023).

## B.6 AN EXPLICIT FORM OF THE IDENTIFICATION RESULT FOR TABULAR CASES

We present a specific identification result here by taking a generic partially observable discrete contextual bandit (i.e., single stage, finite state/action spaces) as an illustrative example, which provides an explicit substantiation of Theorem 1. We first introduce some additional notations. For random variables $X, Y$ taking values on $\{x_1, ..., x_m\}$ and $\{y_1, ..., y_n\}$, we use $\mathbb{P}(X \mid Y)$ to denote a $m \times n$ matrix with $\mathbb{P}_{i,j}(X \mid Y) := p^{\pi^b}(X = x_i \mid Y = y_j)$. Also, $\mathbb{P}(X)$ denotes a column vector with $\mathbb{P}_i(X) := p^{\pi^b}(X = x_i)$. $M^\dagger$ is used to denote the Moore-Penrose inverse of a matrix $M$. Then we have the following identification results. The proof can be adapted directly from Appendix E.1.1. of Shi et al. (2022).

**Example 1.** *In a partially observable discrete contextual bandit, if* $\text{rank}(\mathbb{P}(O_1 \mid S_1)) = |\mathcal{S}|$ *and* $\text{rank}(\mathbb{P}(O_0 \mid S_1)) = |\mathcal{S}|$, *then under Assumption 1, the following results hold:*

$$\nabla_\theta \mathcal{V}(\pi_\theta) = \sum_{a_1, o_1, r_1} r_1 \nabla_\theta \pi_\theta(a_1 \mid o_1)\mathbb{P}(r_1, o_1 \mid O_0, a_1)\mathbb{P}(O_1 \mid O_0, a_1)^\dagger\mathbb{P}(O_1) \quad (18)$$

*In particular, let* $\mathbb{B}_{\nabla V,1}^{\pi_\theta}(a, O_1)$ *be a* $d_\Theta \times |\mathcal{O}|$ *matrix storing the gradient bridge value, then we have*

$$\mathbb{B}_{\nabla V,1}^{\pi_\theta}(a, O_1) = \sum_{o_1, r_1} r_1 \nabla_\theta \pi_\theta(a \mid o_1)\mathbb{P}(r_1, o_1 \mid O_0, a)\mathbb{P}(O_1 \mid O_0, a)^\dagger. \quad (19)$$

As a special case, Example 1 shows that the policy gradient can be explicitly identified as a form of matrix multiplications using observed variables in a discrete contextual bandit. We note that $\text{rank}(\mathbb{P}(O_1 \mid S_1)) = |\mathcal{S}|$, $\text{rank}(\mathbb{P}(O_0 \mid S_1)) = |\mathcal{S}|$ are sufficient conditions for guaranteeing Assumptions 2, 3 in this case. They imply that $O_1, O_0$ carry sufficient information about $S_1$ under the offline distribution and that the linear systems (discrete versions of linear integral operators

(3)(4) have solutions. Under the given assumptions, the term $\sum_{s_1} p^{\pi^b}(r_1, o_1 \mid s_1, a_1) p^{\pi^b}(s_1)$ inside $\nabla_\theta V_1^{\pi_\theta} = \sum_{r_1, a_1, o_1, s_1} r_1 p^{\pi^b}(r_1, o_1 \mid s_1, a_1) \nabla_\theta \pi_\theta(a_1 \mid o_1) p^{\pi^b}(s_1)$ can be expressed into a term that only involves observable variables.

### B.7 INCREASED COMPLEXITY DUE TO HISTORY-DEPENDENT POLICIES IN POMDPs

In this section, we discuss how the inclusion of history impacts the complexity of policy gradient ascent for confounded POMDPs under offline settings across statistical, optimization, and computational aspects. Specifically, we illustrate this with the example of a log-linear policy:

$$\pi_{\theta_t}(a_t \mid o_t, h_{t-1}) \propto \exp(\theta_t^\top \phi_t(a_t, o_t, h_{t-1})).$$

**Statistical Aspect**. In terms of statistical estimation, we examine the upper bound presented in Theorem 2 for the policy gradient estimation error. First, the dimension of the policy space $d_\Theta$ implicitly depends on $T$. At each step $t$, $\phi_t$ is a feature map from a space of dimension $t|\mathcal{A}|dim(\mathcal{O})$ to a space of dimension $d_{\Theta_t}$. To preserve information adequately, it's reasonable to assume that $d_{\Theta_t}$ grows with $t$. In contrast, in MDP settings, a fixed $d_{\Theta_t}$ assumption may suffice for all $t$. Second, the function classes for the bridge functions and test functions should also be rich enough because they are functions of histories that scale with $t$ at each stage $t$. Therefore the complexity of the function classes $\gamma(\mathcal{F}, \mathcal{B})$ also grows when the number of stages $T$ increases. These factors collectively contribute to the complexity of estimation when dealing with history-dependent policies.

**Optimization Aspect**. We discuss the assumptions in Section 6.2, which mainly affects the complexity in the optimization aspect. Regarding Assumption 5(a), when $\phi_t(a_t, o_t, h_{t-1})$ is in a compact region with $\|\phi_t(a_t, o_t, h_{t-1})\|_{\ell_2} \leq B$, it is straightforward to show that $\left\|\nabla_{\theta_t} \log \pi_{\theta_t}(a_t \mid o_t, h_{t-1}) - \nabla_{\theta_t} \log \pi_{\theta'_t}(a_t \mid o_t, h_{t-1})\right\|_{\ell^2} \leq B^2 \|\theta_t - \theta'_t\|_{\ell^2}$. This assures that the Lipschitz constant remains unaffected by the historical dependence. Assumption 5(b) requires the positive definiteness of the Fisher information matrix, where the constant $\mu$ implicitly depends on the number of stage $T$. Intuitively, obtaining a large $\mu$ becomes more challenging in the context of history-dependent policies due to the high dimensionality of the history space. A potential approach to mitigate this challenge involves mapping the history to a lower-dimensional space that retains sufficient information. For Assumption 5(c), the scale of the constant $L$ increases when considering history, compared to the standard MDP settings. See Appendix B.3 for more details. It's evident that the historical dependence amplifies the complexity through Assumptions 5(b) and 5(c). Furthermore, the dimension of the parameter space $d_\Theta$, implicitly depending on $T$, heightens the challenge of gradient ascent for the same reasons elucidated in the statistical aspect.

**Computational Aspect.** We focus on the analysis of the computational complexity of Algorithm 1 using RKHSs, Gaussian kernels, and log-linear policies, yielding a time complexity of $O(Kd_\Theta N^2 T \max\{T, N\})$. See Appendix I.2 for more details. Compared to the standard MDP settings, the introduction of history-dependence primarily increases the computational complexities in two steps: the evaluation of the empirical kernel matrix to derive the closed-form solution for the min-max optimization problem (9) and the update of the policy parameter. For the empirical kernel matrix evaluation, kernel functions must be computed in the history space, a task that scales with $T$. Furthermore, we need $d_\Theta$ operations to update each coordinate, where $d_\Theta$ implicitly depends on $T$.

## C FURTHER RESULTS RELATED TO THEOREM 2

In this section, we present two examples of Theorem 2. In particular, we consider the case when all the related function classes are VC subgraphs, which is commonly considered in parametric settings. For example, the finite-dimensional linear function classes with dimension $d$ has a VC-dimension $d+1$. Additionally, we study the case when all the function classes are RKHSs, which are commonly used in nonparametric estimation.

Before we present the main result, two more assumptions are considered. Assumption 8 is a Lipschitz assumption imposed on the policy space. The commonly-used log-linear policy class satisfies this condition. See C.3.3 in Zanette et al. (2021). Assumption 8 is a technical assumption that allows us to conveniently express the upper bound in terms of the dimension for parameter space rather than the complexity of policy space. Assumption 9 is an eigen-decay assumption imposed for the RKHSs. Intuitively, the eigen-decay rate measures the size of an RKHS.

Then we have the following two main results. The proofs and more details are provided in Appendix F.

**Theorem 4** (Policy gradient estimation error with VC dimension). *Under Assumptions 1, 2, 3, 4, 8, with probability at least $1 - \zeta$ it holds that*

$$\sup_{\theta \in \Theta} \left\| \nabla_\theta \mathcal{V}(\pi_\theta) - \widehat{\nabla_\theta \mathcal{V}(\pi_\theta)} \right\|_{\ell^2} \lesssim C_{\pi^b} \tau_{\max} M_\mathcal{B} M_\mathcal{F} d_\Theta T^{\frac{5}{2}} \sqrt{\frac{\log(T/\zeta)\, \gamma(\mathcal{F}, \mathcal{B}) \log N}{N}},$$

*where $\gamma(\mathcal{B}, \mathcal{F})$ denotes $\max \left\{ \left\{ \mathbb{V}(\mathcal{F}_j^{(t)}) \right\}_{j=1, t=1}^{d_\Theta+1, T}, \left\{ \mathbb{V}(\mathcal{B}_j^{(t)}) \right\}_{j=1, t=1}^{d_\Theta+1, T} \right\}.$*

**Theorem 5** (Policy gradient estimation error in RKHSs). *Under Assumptions 1, 2, 3, 4, 8, 9, with probability at least $1 - \zeta$ it holds that*

$$\sup_{\theta \in \Theta} \| \nabla_\theta \mathcal{V}(\pi_\theta) - \widehat{\nabla_\theta \mathcal{V}(\pi_\theta)} \|_{\ell^2}$$

$$\lesssim C_{\pi^b} \tau_{\max} M_\mathcal{B} M_\mathcal{F} h(d_\Theta) T^{\frac{5}{2}} \sqrt{\log(c_1 T/\zeta) \log N}\, N^{-\frac{1}{2 + \max\{1/\alpha_{K_\mathcal{F}}, 1/\alpha_{\min}\}}} \tag{20}$$

*where $h(d_\Theta) = \max\{d_\Theta, d_\Theta^{\frac{1+2\alpha_{K_\mathcal{F}}}{2+1/\alpha_{K_\mathcal{F}}}}, d_\Theta^{\frac{1+2\alpha_{\max}}{2+1/\alpha_{\min}}}\}$. Here $\alpha_{\max} = \max\{\alpha_{K_\mathcal{B}}, \alpha_{K_{\mathcal{G},V}}, \alpha_{K_{\mathcal{G},\nabla V}}\}$, $\alpha_{\min} = \min\{\alpha_{K_\mathcal{B}}, \alpha_{K_{\mathcal{G},V}}, \alpha_{K_{\mathcal{G},\nabla V}}\}$ defined in Assumption 9 measure the eigen-decay rates of RKHSs.*

## D  DEFINITIONS AND AUXILIARY LEMMAS

In this section, we list some definitions and auxiliary lemmas.

**Definition D.1** (Measure of ill-posedness). *At each $t$, given the bridge function class $\mathcal{B}^{(t)}$, the measure of ill-posedness is defined as*

$$\tau_t := \sup_{b \in \mathcal{B}^{(t)}} \frac{\|b(W_t)\|_{2,2}}{\|\mathbb{E}[b(W_t) \mid X_t]\|_{2,2}}. \tag{21}$$

*Let $\tau_{\max} = \max_{t=1:T} \tau_t$ be the maximum of the measure of ill-posedness across $T$ decision points.*

The following definition is adapted from Liu et al. (2020); Ding et al. (2022); Yuan et al. (2022); Fatkhullin et al. (2023) in our POMDP settings. The transferred compatible function approximation error $\varepsilon_{approx}$ is used to measure the expressive power of the policy class. It becomes small when the space of the policy parameter increases (Wang et al., 2019; Liu et al., 2020).

**Definition D.2** (Transferred compatible function approximation error). *The transferred compatible function approximation error $\varepsilon_{approx}$ is defined as*

$$\sup_{\theta \in \Theta} \sum_{t=1}^{T} \left( \mathbb{E}^{\pi_{\theta^*}} \left[ \left( \mathbb{A}_t^{\pi_\theta}(A_t, O_t, S_t, H_{t-1}) - w_t^*(\theta)^\top \nabla_{\theta_t} \log \pi_{\theta_t}(A_t \mid O_t, H_{t-1}) \right)^2 \right] \right)^{\frac{1}{2}},$$

*where $\mathbb{A}_t^{\pi_\theta}$ is the advantage function defined in Definition D.7, and $w_t^*(\theta) \in \mathbb{R}^{d_{\Theta_t}}$ is defined as $\arg\min_{w_t} \mathbb{E}^{\pi_\theta} \left[ \left( \mathbb{A}_t^{\pi_\theta}(A_t, O_t, S_t, H_{t-1}) - w_t^\top \nabla_{\theta_t} \log \pi_{\theta_t}(A_t \mid O_t, H_{t-1}) \right)^2 \right].$*

**Definition D.3** (Functional norm associated with vector-valued function class). *For any vector-valued function class $\mathcal{H} = \{h : \mathbb{R}^{d_1} \to \mathbb{R}^{d_2} \mid h = (h_1, ..., h_{d_2})^\top, h_j \in \mathcal{H}_j, \forall 1 \leq j \leq d_2\}$, a functional norm $\|\cdot\|_\mathcal{H}$ associated with $\mathcal{H}$ is defined as $\|h\|_\mathcal{H} := (\sum_{j=1}^{d_2} \|h_j\|_{\mathcal{H}_j}^2)^{\frac{1}{2}}$ where $\|\cdot\|_{\mathcal{H}_j}$ is a functional norm associated with $\mathcal{H}_j$.*

**Definition D.4** ($\mathcal{L}^2(\ell^2, \mathbb{P}^{\pi^b})$-norm). *For any vector-valued function $h \in \mathcal{H}$, the population $\mathcal{L}^2$ norm with respect to $\mathbb{P}^{\pi^b}$ is defined as $\|h\|_{2,2} := \|h\|_{\mathcal{L}^2(\ell^2, \mathbb{P}^{\pi^b})} = (\mathbb{E}_{X \sim \mathbb{P}^{\pi^b}} \|h(X)\|_{\ell^2}^2)^{1/2}$. When $h$ is real-valued, we use notations $\|h\|_2$ for simplicity.*

**Lemma 3** (Policy gradient in POMDPs). *For any $\pi_\theta \in \Pi_\Theta$, we have*

$$\nabla_\theta \mathcal{V}(\pi_\theta) = \mathbb{E}^{\pi_\theta} \left[ \sum_{t=1}^{T} R_t \sum_{j=1}^{t} \nabla_\theta \log \pi_{\theta_j}(A_j \mid O_j, H_{j-1}) \right], \tag{22}$$

where $\nabla_\theta \log \pi_{\theta_t}(a_t \mid o_t, h_{t-1})$ is a $d_\Theta$-dimensional vector with only non-zero elements in its $t$-th block for every $t$.

**Lemma 4** (Theorem 14.1 in Wainwright (2019))**.** *Given a star-shaped and b-uniformly bounded function class $\mathcal{F}$, let $\delta_n$ be any positive solution of the inequality*

$$\mathcal{R}_n(\delta; \mathcal{F}) \leq \frac{\delta^2}{b}.$$

*Then for any $t \geq \delta_n$, we have*

$$\left| \|f\|_n^2 - \|f\|_2^2 \right| \leq \frac{1}{2}\|f\|_2^2 + \frac{t^2}{2} \quad \text{for all } f \in \mathcal{F}$$

*with probability at least $1 - c_1 e^{-c_2 \frac{n t^2}{b^2}}$. Here*

$$\mathcal{R}_n(\delta; \mathcal{F}).$$

*denotes the localized population Rademacher complexity.*

The following lemma is a generalization of lemma 4 when $f$ is a vector-valued function. The idea is to incorporate a contraction inequality from Maurer (2016) for the vector-valued function. The rest proof is adapted from Wainwright (2019).

**Lemma 5.** *Given a star-shaped and 1-uniformly bounded function class $\mathcal{F}$, let $\delta_n$ be any positive solution of the inequality*
$$\overline{\mathcal{R}}_n(\delta; \mathcal{F} \mid_k) \leq \delta^2$$
*for any $k = 1, ..., d$. Then for any $t \geq \delta_n$, we have*

$$\left| \|f\|_{n,2,2}^2 - \|f\|_{2,2}^2 \right| \leq \frac{1}{2}\|f\|_{2,2}^2 + \frac{dt^2}{2} \quad \text{for all } f \in \mathcal{F} \tag{23}$$

*with probability at least $1 - c_1 e^{-c_2 n t^2}$. Here*

$$\mathcal{R}_n(\delta; \mathcal{F} \mid_k).$$

*denotes the localized population Rademacher complexity of the projection of $\mathcal{F}$ on its $k-$th coordinate.*

**Definition D.5** (Star convex hull of $\mathcal{H}$)**.** *For a function class $\mathcal{H}$, we define* $\text{star}(\mathcal{H}) := \{rh : h \in \mathcal{H}, r \in [0,1]\}$.

**Lemma 6** (Lemma 11 of Foster & Syrgkanis (2019))**.** *Consider a function class $\mathcal{F}$, with* $\sup_{f \in \mathcal{F}} \|f\|_\infty \leq 1$, *and pick any $f^\star \in \mathcal{F}$. Let $\delta_n^2 \geq \frac{4d \log(41 \log(2c_2 n))}{c_2 n}$ be any solution to the inequalities:*
$$\forall t \in \{1, \ldots, d\} : \mathcal{R}\left(\delta, \text{star}\left(\mathcal{F}\mid_t - f_t^\star\right)\right) \leq \delta^2.$$

*Moreover, assume that the loss $\ell$ is L-Lipschitz in its first argument with respect to the $\ell_2$ norm. Then for some universal constants $c_5, c_6$, with probability $1 - c_5 \exp\left(c_6 n \delta_n^2\right)$,*

$$\left| \mathbb{P}_n\left(\mathcal{L}_f - \mathcal{L}_{f^\star}\right) - \mathbb{P}\left(\mathcal{L}_f - \mathcal{L}_{f^\star}\right) \right| \leq 18Ld\delta_n \left\{ \|f - f^\star\|_{2,2} + \delta_n \right\}, \quad \forall f \in \mathcal{F}.$$

**Definition D.6** (L-smoothness)**.** *A continuously differentiable function $f : \mathbb{R}^n \to \mathbb{R}$ is L-smooth if $\nabla f$ is L-Lipschitz, i.e., for all $x, y \in domain(f)$,, it holds that $\|\nabla f(x) - \nabla f(y)\|_{\ell^2} \leq L \|x - y\|_{\ell^2}$.*

**Lemma 7** (Ascent Lemma)**.** *If the function $f : \mathcal{D} \to \mathbb{R}$ is L-smooth over a set $\mathcal{X} \subseteq \mathcal{D}$, then for any $(x, y) \in \mathcal{X}$ :*

$$f(y) \geq f(x) + \langle \nabla f(x), y - x \rangle - \frac{L}{2}\|y - x\|_2^2.$$

**Definition D.7.** *Let the value function $V$ and the action-value function $Q$ of a policy $\pi_\theta \in \Pi_\Theta$ be defined as*

$$V_t^{\pi_\theta}(o_t, s_t, h_{t-1}) = \mathbb{E}^{\pi_\theta}[\sum_{j=t}^T R_j \mid O_t = o_t, S_t = s_t, H_{t-1} = h_{t-1}], \tag{24}$$

*and*

$$Q_t^{\pi_\theta}(a_t, o_t, s_t, h_{t-1}) = \mathbb{E}^{\pi_\theta}[\sum_{j=t}^{T} R_j \mid A_t = a_t, O_t = o_t, S_t = s_t, H_{t-1} = h_{t-1}]. \qquad (25)$$

*Then the advantage function $\mathbb{A}$ is defined as*

$$\mathbb{A}_t^{\pi_\theta}(a_t, o_t, s_t, h_{t-1}) = Q_t^{\pi_\theta}(a_t, o_t, s_t, h_{t-1}) - V_t^{\pi_\theta}(o_t, s_t, h_{t-1}). \qquad (26)$$

**Lemma 8** (Performance difference lemma for POMDP). *For any two policies $\pi_\theta, \pi_{\theta'} \in \Pi_\Theta$, it holds that*

$$\mathcal{V}(\pi_\theta) - \mathcal{V}(\pi_{\theta'}) = \sum_{t=1}^{T} \mathbb{E}^{\pi_\theta} \left[ \mathbb{A}_t^{\pi_{\theta'}}(A_t, O_t, S_t, H_{t-1}) \right]. \qquad (27)$$

Remark: Lemma 8 relies on the assumption that $\{R_j\}_{j=t+1}^{T} \perp\!\!\!\perp_{\mathbb{P}^{\pi_\theta}} S_t \mid S_{t+1}, H_t$ for each $t$, which is satisfied under our POMDP settings according to Figure 1.

**Lemma 9** (Policy gradient in POMDPs (I)). *For any $\pi_\theta \in \Pi_\Theta$, we have*

$$\nabla_\theta \mathcal{V}(\pi_\theta) = \mathbb{E}^{\pi_\theta} \left[ \sum_{t=1}^{T} R_t \sum_{j=1}^{t} \nabla_\theta \log \pi_{\theta_j}(A_j \mid O_j, H_{j-1}) \right], \qquad (28)$$

*where $\nabla_\theta \log \pi_{\theta_t}(a_t \mid o_t, h_{t-1})$ is a $d_\Theta$-dimensional vector with only non-zero elements in its $t$-th block for every $t$.*

**Lemma 10** (Policy gradient for POMDPs (II)). *For any $\pi_\theta \in \Pi_\Theta$, the policy gradient can be expressed as*

$$\nabla_\theta \mathcal{V}(\pi_\theta) = \sum_{t=1}^{T} \mathbb{E}^{\pi_\theta} \left[ \nabla_\theta \log \pi_{\theta_t}(A_t \mid O_t, H_{t-1}) Q_t^{\pi_\theta}(A_t, O_t, S_t, H_{t-1}) \right]. \qquad (29)$$

*where $\nabla_\theta \log \pi_{\theta_t}(a_t \mid o_t, h_{t-1})$ is a $d_\Theta$-dimensional vector with only non-zero elements in its $t$-th block for every $t$.*

## E  PROOFS FOR POLICY GRADIENT IDENTIFICATIONS

In this section, we present a complete proof of the identification results summarized in Theorem 1. In the first part, we show that under Assumptions 1, 2, 3, we obtain another sequence of conditional moment restrictions that are projected on the unobserved $(S_t, A_t, H_{t-1})$. In the second part, we show by mathematical inductions that

$$\mathbb{E}[\sum_a b_{V,t}^{\pi_\theta}(a, O_t, H_{t-1}) \mid S_t, H_{t-1}] = \mathbb{E}^{\pi_\theta}[\sum_{j=t}^{T} R_t \mid S_t, H_{t-1}]$$

and

$$\mathbb{E} \left[ \sum_{a \in \mathcal{A}} b_{\nabla V,t}^{\pi_\theta}(a, O_t, H_{t-1}) \mid S_t, H_{t-1} \right] = \nabla_\theta \mathbb{E}^{\pi_\theta} \left[ \sum_{j=t}^{T} R_j \mid S_t, H_{t-1} \right]$$

for all $t = T, ..., 1$. In the third part, we conclude the proof of Theorem 1.

**Part I**  Suppose $\{b_{V,t}^{\pi_\theta}\}_{t=1}^{T}$ with $b_{V,T+1}^{\pi_\theta} = 0$ satisfy equation (3). Then we have

$$\mathbb{E} \left[ b_{V,t}^{\pi_\theta}(O_t, H_{t-1}, A_t) \mid O_0, H_{t-1}, A_t \right]$$

$$= \mathbb{E} \left[ \mathbb{E} \left[ b_{V,t}^{\pi_\theta}(O_t, H_{t-1}, A_t) \mid S_t, O_0, H_{t-1}, A_t \right] \mid O_0, H_{t-1}, A_t \right] \qquad (30)$$

$$= \mathbb{E} \left[ \mathbb{E} \left[ b_{V,t}^{\pi_\theta}(O_t, H_{t-1}, A_t) \mid S_t, H_{t-1}, A_t \right] \mid O_0, H_{t-1}, A_t \right] \text{ by } O_0 \perp\!\!\!\perp O_t \mid S_t, A_t, H_{t-1}$$

and

$$
\begin{aligned}
&\mathbb{E}\left[R_t \pi_{\theta_t}(a \mid O_t, H_{t-1}) + \sum_a b_{V,t+1}^{\pi_\theta}(a, O_{t+1}, H_t) \pi_{\theta_t}(A_t \mid O_t, H_{t-1}) \mid O_0, H_{t-1}, A_t\right] \\
=&\mathbb{E}\left[\mathbb{E}\left[R_t \pi_{\theta_t}(a \mid O_t, H_{t-1})\right.\right. \\
&\left.\left. + \sum_a b_{V,t+1}^{\pi_\theta}(a, O_{t+1}, H_t) \pi_{\theta_t}(A_t \mid O_t, H_{t-1}) \mid S_t, O_0, H_{t-1}, A_t\right] \mid O_0, H_{t-1}, A_t\right] \\
=&\mathbb{E}\left[\mathbb{E}\left[R_t \pi_{\theta_t}(a \mid O_t, H_{t-1})\right.\right. \\
&\left.\left. + \sum_a b_{V,t+1}^{\pi_\theta}(a, O_{t+1}, H_t) \pi_{\theta_t}(A_t \mid O_t, H_{t-1}) \mid S_t, H_{t-1}, A_t\right] \mid O_0, H_{t-1}, A_t\right] \\
&(\text{by } O_0 \perp\!\!\!\perp R_t, O_t, O_{t+1} \mid S_t, A_t, H_{t-1}).
\end{aligned}
\tag{31}
$$

Combining equations (30)(31), we have

$$
\begin{aligned}
&\mathbb{E}\left[\mathbb{E}\left[b_{V,t}^{\pi_\theta}(O_t, H_{t-1}, A_t) \mid S_t, H_{t-1}, A_t\right] \mid O_0, H_{t-1}, A_t\right] \\
=&\mathbb{E}\left[\mathbb{E}\left[R_t \pi_{\theta_t}(a \mid O_t, H_{t-1})\right.\right. \\
&\left.\left. + \sum_a b_{V,t+1}^{\pi_\theta}(a, O_{t+1}, H_t) \pi_{\theta_t}(A_t \mid O_t, H_{t-1}) \mid S_t, H_{t-1}, A_t\right] \mid O_0, H_{t-1}, A_t\right] \\
&(\text{by } (3)).
\end{aligned}
\tag{32}
$$

Therefore, by Assumption 3, we have

$$
\begin{aligned}
&\mathbb{E}\left[b_{V,t}^{\pi_\theta}(O_t, H_{t-1}, A_t) \mid S_t, H_{t-1}, A_t\right] \\
=&\mathbb{E}\left[R_t \pi_{\theta_t}(a \mid O_t, H_{t-1}) + \sum_a b_{V,t+1}^{\pi_\theta}(a, O_{t+1}, H_t) \pi_{\theta_t}(A_t \mid O_t, H_{t-1}) \mid S_t, H_{t-1}, A_t\right].
\end{aligned}
\tag{33}
$$

Equation (33) shows that the solutions to (3) also solve a similar conditional moment restriction with unobserved $S_t$. Next, we consider the gradient bridge functions.

Suppose $\{b_{\nabla V,t}^{\pi_\theta}, b_{V,t}^{\pi_\theta}\}_{t=1}^T$ with $b_{V,T+1}^{\pi_\theta} = b_{\nabla V,T+1}^{\pi_\theta} = 0$ satisfy (4). Then we have

$$
\begin{aligned}
&\mathbb{E}\left[b_{\nabla V,t}^{\pi_\theta}(O_t, H_{t-1}, A_t) \mid O_0, A_t, H_{t-1}\right] \\
=&\mathbb{E}\left[\mathbb{E}\left[b_{\nabla V,t}^{\pi_\theta}(O_t, H_{t-1}, A_t) \mid S_t, O_0, A_t, H_{t-1}\right] \mid O_0, A_t, H_{t-1}\right] \\
=&\mathbb{E}\left[\mathbb{E}\left[b_{\nabla V,t}^{\pi_\theta}(O_t, H_{t-1}, A_t) \mid S_t, A_t, H_{t-1}\right] \mid O_0, A_t, H_{t-1}\right] \text{ by } O_0 \perp\!\!\!\perp O_t \mid S_t, A_t, H_{t-1}.
\end{aligned}
\tag{34}
$$

and

$$
\begin{aligned}
&\mathbb{E}\left[(R_t + \sum_a b^{\pi_\theta}_{V,t+1}(a, O_{t+1}, H_t))\nabla_\theta \pi_{\theta_t}(A_t \mid O_t, H_{t-1}) + \sum_a b^{\pi_\theta}_{\nabla V,t+1}(a, O_{t+1}, H_t)\right. \\
&\left. \pi_{\theta_t}(A_t \mid O_t, H_{t-1}) \mid O_0, A_t, H_{t-1}\right] \\
=&\mathbb{E}\left[\mathbb{E}\left[(R_t\right.\right. \\
&\left. + \sum_a b^{\pi_\theta}_{V,t+1}(a, O_{t+1}, H_t))\nabla_\theta \pi_{\theta_t}(A_t \mid O_t, H_{t-1}) \mid S_t, O_0, A_t, H_{t-1}\right] \mid O_0, A_t, H_{t-1}\Big] \\
&+ \mathbb{E}\left[\mathbb{E}\left[\sum_a b^{\pi_\theta}_{\nabla V,t+1}(a, O_{t+1}, H_t)\pi_{\theta_t}(A_t \mid O_t, H_{t-1}) \mid S_t, O_0, A_t, H_{t-1}\right] \mid O_0, A_t, H_{t-1}\right] \\
=&\mathbb{E}\left[\mathbb{E}\left[(R_t + \sum_a b^{\pi_\theta}_{V,t+1}(a, O_{t+1}, H_t))\nabla_\theta \pi_{\theta_t}(A_t \mid O_t, H_{t-1}) \mid S_t, A_t, H_{t-1}\right] \mid O_0, A_t, H_{t-1}\right] \\
&+ \mathbb{E}\left[\mathbb{E}\left[\sum_a b^{\pi_\theta}_{\nabla V,t+1}(a, O_{t+1}, H_t)\pi_{\theta_t}(A_t \mid O_t, H_{t-1}) \mid S_t, A_t, H_{t-1}\right] \mid O_0, A_t, H_{t-1}\right] \\
&(\text{by } O_0 \perp\!\!\!\perp R_t, O_t, O_{t+1} \mid S_t, A_t, H_{t-1}).
\end{aligned}
\tag{35}
$$

Combining the above two equations, we have

$$
\begin{aligned}
&\mathbb{E}\left[\mathbb{E}\left[b^{\pi_\theta}_{\nabla V,t}(O_t, H_{t-1}, A_t) \mid S_t, A_t, H_{t-1}\right] \mid O_0, A_t, H_{t-1}\right] \\
=&\mathbb{E}\left[\mathbb{E}\left[(R_t + \sum_a b^{\pi_\theta}_{V,t+1}(a, O_{t+1}, H_t))\nabla_\theta \pi_{\theta_t}(A_t \mid O_t, H_{t-1}) \mid S_t, A_t, H_{t-1}\right] \mid O_0, A_t, H_{t-1}\right] \\
&+ \mathbb{E}\left[\mathbb{E}\left[\sum_a b^{\pi_\theta}_{\nabla V,t+1}(a, O_{t+1}, H_t)\pi_{\theta_t}(A_t \mid O_t, H_{t-1}) \mid S_t, A_t, H_{t-1}\right] \mid O_0, A_t, H_{t-1}\right] \text{ by (4).}
\end{aligned}
\tag{36}
$$

Therefore, by Assumption 3, we have

$$
\begin{aligned}
&\mathbb{E}\left[b^{\pi_\theta}_{\nabla V,t}(O_t, H_{t-1}, A_t) \mid S_t, A_t, H_{t-1}\right] \\
=&\mathbb{E}\left[(R_t + \sum_a b^{\pi_\theta}_{V,t+1}(a, O_{t+1}, H_t))\nabla_\theta \pi_{\theta_t}(A_t \mid O_t, H_{t-1}) \mid S_t, A_t, H_{t-1}\right] \\
&+ \mathbb{E}\left[\sum_a b^{\pi_\theta}_{\nabla V,t+1}(a, O_{t+1}, H_t)\pi_{\theta_t}(A_t \mid O_t, H_{t-1}) \mid S_t, A_t, H_{t-1}\right].
\end{aligned}
\tag{37}
$$

Equation (37) basically shows that the solutions to (4) also solve a similar integral equation with unobserved $S_t$.

In the rest of the appendix, we will utilize equations (33)(37) several times. Next, we move on to Part II of the proof.

**Part II** In this part, we show by mathematical inductions that $\mathbb{E}[\sum_a b^{\pi_\theta}_{V,t}(a, O_t, H_{t-1}) \mid S_t, H_{t-1}] = \mathbb{E}^{\pi_\theta}[\sum_{j=t}^T R_t \mid S_t, H_{t-1}]$ and $\mathbb{E}\left[\sum_{a \in \mathcal{A}} b^{\pi_\theta}_{\nabla V,t}(a, O_t, H_{t-1}) \mid S_t, H_{t-1}\right]$ $= \nabla_\theta \mathbb{E}^{\pi_\theta}\left[\sum_{j=t}^T R_j \mid S_t, H_{t-1}\right]$ for all $t = T, ..., 1$, which are summarized in the following lemmas.

**Lemma 11.** *Under Assumptions 1, 2, 3, it holds for all $t = T, ..., 1$ that*

$$
\mathbb{E}[\sum_a b^{\pi_\theta}_{V,t}(a, O_t, H_{t-1}) \mid S_t, H_{t-1}] = \mathbb{E}^{\pi_\theta}[\sum_{j=t}^T R_t \mid S_t, H_{t-1}].
\tag{38}
$$

**Lemma 12.** *Under assumptions Assumptions 1, 2, 3, it holds for all $t = T, ..., 1$ that*

$$\mathbb{E}\left[\sum_{a \in \mathcal{A}} b_{\nabla V, t}^{\pi_\theta}(a, O_t, H_{t-1}) \mid S_t, H_{t-1}\right] = \nabla_\theta \mathbb{E}^{\pi_\theta}\left[\sum_{j=t}^{T} R_j \mid S_t, H_{t-1}\right]. \tag{39}$$

Proofs for Lemma 11 and Lemma 12 are provided in Appendix H.

**Part III** In the third part, we conclude the proof of Theorem 1 by utilizing Lemma 11 and Lemma 12. In particular, we express the policy gradient as

$$
\begin{aligned}
&\nabla_\theta \mathcal{V}(\pi_\theta) \\
&= \nabla_\theta \mathbb{E}^{\pi_\theta}\left[\sum_{t=1}^{T} R_t\right] \quad \text{by definition of policy value} \\
&= \nabla_\theta \mathbb{E}_{S_1 \sim \nu_1}\left[\mathbb{E}^{\pi_\theta}\left[\sum_{t=1}^{T} R_t \mid S_1\right]\right] \quad \text{by the law of total expectation} \\
&= \nabla_\theta \mathbb{E}_{S_1 \sim \nu_1}\left[\mathbb{E}^{\pi_\theta}\left[\sum_{t=1}^{T} R_t \mid S_1, H_0\right]\right] \quad \text{by } H_0 = \emptyset \\
&= \mathbb{E}_{S_1 \sim \nu_1}\left[\nabla_\theta \mathbb{E}^{\pi_\theta}\left[\sum_{t=1}^{T} R_t \mid S_1, H_0\right]\right] \quad \text{by interchanging order of integration and derivative} \\
&= \mathbb{E}_{S_1 \sim \nu_1}\left[\mathbb{E}\left[\sum_{a} b_{\nabla V, 1}^{\pi_\theta}(a, O_1, H_0) \mid S_1, H_0\right]\right] \quad \text{by Lemma 12} \\
&= \mathbb{E}_{S_1 \sim \nu_1}\left[\mathbb{E}\left[\sum_{a} b_{\nabla V, 1}^{\pi_\theta}(a, O_1) \mid S_1\right]\right] \quad \text{by } H_0 = \emptyset \\
&= \mathbb{E}\left[\sum_{a} b_{\nabla V, 1}^{\pi_\theta}(a, O_1)\right].
\end{aligned} \tag{40}
$$

Similarly, we can express the policy value as

$$
\begin{aligned}
&\mathcal{V}(\pi_\theta) \\
&= \mathbb{E}^{\pi_\theta}\left[\sum_{t=1}^{T} R_t\right] \quad \text{by definition of policy value} \\
&= \mathbb{E}_{S_1 \sim \nu_1}\left[\mathbb{E}^{\pi_\theta}\left[\sum_{t=1}^{T} R_t \mid S_1\right]\right] \quad \text{by the law of total expectation} \\
&= \mathbb{E}_{S_1 \sim \nu_1}\left[\mathbb{E}^{\pi_\theta}\left[\sum_{t=1}^{T} R_t \mid S_1, H_0\right]\right] \quad \text{by } H_0 = \emptyset \\
&= \mathbb{E}_{S_1 \sim \nu_1}\left[\mathbb{E}\left[\sum_{a} b_{V, 1}^{\pi_\theta}(a, O_1, H_0) \mid S_1, H_0\right]\right] \quad \text{by Lemma 11} \\
&= \mathbb{E}_{S_1 \sim \nu_1}\left[\mathbb{E}\left[\sum_{a} b_{V, 1}^{\pi_\theta}(a, O_1) \mid S_1\right]\right] \quad \text{by } H_0 = \emptyset \\
&= \mathbb{E}\left[\sum_{a} b_{V, 1}^{\pi_\theta}(a, O_1)\right].
\end{aligned}
$$

Consequently, we have $\nabla_\theta \mathcal{V}(\pi_\theta) = \mathbb{E}\left[\sum_{a} b_{\nabla V, 1}^{\pi_\theta}(a, O_1)\right]$ and $\mathcal{V}(\pi_\theta) = \mathbb{E}\left[\sum_{a} b_{V, 1}^{\pi_\theta}(a, O_1)\right]$, and we complete the proof of Theorem 4.5.

# F  PROOFS FOR POLICY GRADIENT ESTIMATIONS

In this section, we provide a complete proof of Theorem 2.

**Proof sketch**  We briefly summarize the sketch of proofs here.  The goal is to provide $\sup_{\theta \in \Theta} \|\nabla_\theta \mathcal{V}(\pi_\theta) - \widehat{\nabla_\theta \mathcal{V}(\pi_\theta)}\|_{\ell^2}$ a finite-sample upper bound with high probability, which will appear in the suboptimality gap studied in section 6.1 for policy learning. To achieve this goal, we first decompose the $\ell^2$-norm of $\nabla_\theta \mathcal{V}(\pi_\theta) - \widehat{\nabla_\theta \mathcal{V}(\pi_\theta)}$ into summations of one-step errors mainly caused by min-max estimation procedure at each $t$. Then we provide finite-sample upper bounds for these one-step errors uniformly over all $\theta \in \Theta$ and all $t = 1, ..., T$ by adopting uniform laws of large numbers. Finally, the finite-sample upper bound on $\sup_{\theta \in \Theta} \|\nabla_\theta \mathcal{V}(\pi_\theta) - \widehat{\nabla_\theta \mathcal{V}(\pi_\theta)}\|_{\ell^2}$ can be obtained by combining the decomposition of errors and the analysis of one-step errors. In the rest of this section, we present rigorous analysis for each step.

## F.1  DECOMPOSITION OF ERROR

We let $\{b_{V,t}^{\pi_\theta}, b_{\nabla V,t}^{\pi_\theta}\}_{t=1}^T$ denote a set of bridge equations that solve the conditional moment equations (3)(4), i.e. the true bridge functions that can identify the policy values and policy gradients. Similarly, we let $\{\widehat{b}_{V,t}^{\pi_\theta}, \widehat{b}_{\nabla V,t}^{\pi_\theta}\}_{t=1}^T$ denote the set of functions returned by the Algorithm 1. Then, we consider the following decomposition of the error as

$$\nabla_\theta \mathcal{V}(\pi_\theta) - \widehat{\nabla_\theta \mathcal{V}(\pi_\theta)}$$
$$= \mathbb{E}^{\pi^b}[\sum_a b_{\nabla V,1}^{\pi_\theta}(a, O_1)] - \widehat{\mathbb{E}}^{\pi^b}[\sum_a \widehat{b}_{\nabla V,1}^{\pi_\theta}(a, O_1)]$$
$$= \mathbb{E}^{\pi^b}[\sum_a b_{\nabla V,1}^{\pi_\theta}(a, O_1)] - \mathbb{E}^{\pi^b}[\sum_a \widehat{b}_{\nabla V,1}^{\pi_\theta}(a, O_1)] + \mathbb{E}^{\pi^b}[\sum_a \widehat{b}_{\nabla V,1}^{\pi_\theta}(a, O_1)] - \widehat{\mathbb{E}}^{\pi^b}[\sum_a \widehat{b}_{\nabla V,1}^{\pi_\theta}(a, O_1)]$$
$$\tag{41}$$

The way of analyzing the second term is standard by using techniques from the empirical process theory, and we leave this term to the final. Now we consider the first term.

$$\mathbb{E}^{\pi^b}\left[\sum_a b_{\nabla V,1}^{\pi_\theta}(a, O_1) - \sum_a \widehat{b}_{\nabla V,1}(a, O_1)\right]$$
$$= \mathbb{E}^{\pi^b}\left[\frac{1}{\pi_1^b(A_1 \mid S_1)}(b_{\nabla V,1}^{\pi_\theta}(A_1, O_1) - \widehat{b}_{\nabla V,1}^{\pi_\theta}(A_1, O_1))\right] \quad \text{by } A_1 \perp\!\!\!\perp O_1 \mid S_1$$
$$= \mathbb{E}^{\pi^b}\left[\mathbb{E}^{\pi^b}\left[\frac{1}{\pi_1^b(A_1 \mid S_1)}(b_{\nabla V,1}^{\pi_\theta}(A_1, O_1) - \widehat{b}_{\nabla V,1}^{\pi_\theta}(A_1, O_1)) \mid S_1, A_1\right]\right] \tag{42}$$
$$= \mathbb{E}^{\pi^b}\left[\frac{1}{\pi_1^b(A_1 \mid S_1)}\mathbb{E}^{\pi^b}\left[b_{\nabla V,1}^{\pi_\theta}(A_1, O_1) \mid S_1, A_1\right]\right]$$
$$\quad - \mathbb{E}^{\pi^b}\left[\frac{1}{\pi_1^b(A_1 \mid S_1)}\mathbb{E}^{\pi^b}\left[\widehat{b}_{\nabla V,1}^{\pi_\theta}(A_1, O_1) \mid S_1, A_1\right]\right]$$

For the first term of the last equation, we expand it by using the conditional moment equation in the unobserved space (37), and thus we have

$$\mathbb{E}^{\pi^b}\left[\frac{1}{\pi_1^b(A_1 \mid S_1)}\mathbb{E}^{\pi^b}\left[b_{\nabla V,1}^{\pi_\theta}(A_1, O_1) \mid S_1, A_1\right]\right]$$
$$= \mathbb{E}^{\pi^b}\left[\frac{1}{\pi_1^b(A_1 \mid S_1)}\mathbb{E}^{\pi^b}\left[(R_1 + \sum_{a'} b_{V,2}^{\pi_\theta}(a', O_2, H_1))\nabla_\theta \pi_{\theta_1}(A_1 \mid O_1)\right.\right.$$
$$\left.\left. + \sum_{a'} b_{\nabla V,2}^{\pi_\theta}(a', O_2, H_1)\pi_{\theta_1}(A_1 \mid O_1) \mid S_1, A_1\right]\right]$$

$$
\begin{aligned}
=&\mathbb{E}^{\pi^b}\left[\frac{1}{\pi_1^b(A_1\mid S_1)}\mathbb{E}^{\pi^b}\left[(R_1+\sum_{a'}b_{V,2}^{\pi_\theta}(a',O_2,H_1))\nabla_\theta\pi_{\theta_1}(A_1\mid O_1)\mid S_1,A_1\right]\right]\\
&+\mathbb{E}^{\pi^b}\left[\frac{1}{\pi_1^b(A_1\mid S_1)}\mathbb{E}^{\pi^b}\left[\sum_{a'}b_{\nabla V,2}^{\pi_\theta}(a',O_2,H_1)\pi_{\theta_1}(A_1\mid O_1)\mid S_1,A_1\right]\right]\\
=&\mathbb{E}^{\pi^b}\left[\frac{1}{\pi_1^b(A_1\mid S_1)}\mathbb{E}^{\pi^b}\left[(R_1+\sum_{a'}b_{V,2}^{\pi_\theta}(a',O_2,H_1))\nabla_\theta\pi_{\theta_1}(A_1\mid O_1)\mid S_1,A_1\right]\right]\\
&-\mathbb{E}^{\pi^b}\left[\frac{1}{\pi_1^b(A_1\mid S_1)}\mathbb{E}^{\pi^b}\left[(R_1+\sum_{a'}\widehat{b}_{V,2}^{\pi_\theta}(a',O_2,H_1))\nabla_\theta\pi_{\theta_1}(A_1\mid O_1)\mid S_1,A_1\right]\right]\\
&+\mathbb{E}^{\pi^b}\left[\frac{1}{\pi_1^b(A_1\mid S_1)}\mathbb{E}^{\pi^b}\left[(R_1+\sum_{a'}\widehat{b}_{V,2}^{\pi_\theta}(a',O_2,H_1))\nabla_\theta\pi_{\theta_1}(A_1\mid O_1)\mid S_1,A_1\right]\right]\\
&+\mathbb{E}^{\pi^b}\left[\frac{1}{\pi_1^b(A_1\mid S_1)}\mathbb{E}^{\pi^b}\left[\sum_{a'}b_{\nabla V,2}^{\pi_\theta}(a',O_2,H_1)\pi_{\theta_1}(A_1\mid O_1)\mid S_1,A_1\right]\right]\\
&-\mathbb{E}^{\pi^b}\left[\frac{1}{\pi_1^b(A_1\mid S_1)}\mathbb{E}^{\pi^b}\left[\sum_{a'}\widehat{b}_{\nabla V,2}^{\pi_\theta}(a',O_2,H_1)\pi_{\theta_1}(A_1\mid O_1)\mid S_1,A_1\right]\right]\\
&+\mathbb{E}^{\pi^b}\left[\frac{1}{\pi_1^b(A_1\mid S_1)}\mathbb{E}^{\pi^b}\left[\sum_{a'}\widehat{b}_{\nabla V,2}^{\pi_\theta}(a',O_2,H_1)\pi_{\theta_1}(A_1\mid O_1)\mid S_1,A_1\right]\right]\\
=&\mathbb{E}^{\pi^b}\left[\frac{1}{\pi_1^b(A_1\mid S_1)}\mathbb{E}^{\pi^b}\left[(\sum_{a'}(b_{V,2}^{\pi_\theta}(a',O_2,H_1)-\widehat{b}_{V,2}^{\pi_\theta}(a',O_2,H_1)))\nabla_\theta\pi_{\theta_1}(A_1\mid O_1)\mid S_1,A_1\right]\right]\\
&+\mathbb{E}^{\pi^b}\left[\frac{1}{\pi_1^b(A_1\mid S_1)}\mathbb{E}^{\pi^b}\left[(R_1+\sum_{a'}\widehat{b}_{V,2}^{\pi_\theta}(a',O_2,H_1))\nabla_\theta\pi_{\theta_1}(A_1\mid O_1)\mid S_1,A_1\right]\right]\\
&+\mathbb{E}^{\pi^b}\left[\frac{1}{\pi_1^b(A_1\mid S_1)}\mathbb{E}^{\pi^b}\left[\sum_{a'}(b_{\nabla V,2}^{\pi_\theta}(a',O_2,H_1)-\widehat{b}_{\nabla V,2}^{\pi_\theta}(a',O_2,H_1))\pi_{\theta_1}(A_1\mid O_1)\mid S_1,A_1\right]\right]\\
&+\mathbb{E}^{\pi^b}\left[\frac{1}{\pi_1^b(A_1\mid S_1)}\mathbb{E}^{\pi^b}\left[\sum_{a'}\widehat{b}_{\nabla V,2}^{\pi_\theta}(a',O_2,H_1)\pi_{\theta_1}(A_1\mid O_1)\mid S_1,A_1\right]\right].
\end{aligned}
\tag{43}
$$

Now we add the second term back and have,

$$
\begin{aligned}
&\mathbb{E}^{\pi^b}\left[\sum_a b_{\nabla V,1}^{\pi_\theta}(a,O_1)-\sum_a\widehat{b}_{\nabla V,1}(a,O_1)\right]\\
=&\mathbb{E}^{\pi^b}\left[\frac{1}{\pi_1^b(A_1\mid S_1)}\mathbb{E}^{\pi^b}\left[b_{\nabla V,1}^{\pi_\theta}(A_1,O_1)\mid S_1,A_1\right]\right]\\
&-\mathbb{E}^{\pi^b}\left[\frac{1}{\pi_1^b(A_1\mid S_1)}\mathbb{E}^{\pi^b}\left[\widehat{b}_{\nabla V,1}^{\pi_\theta}(A_1,O_1)\mid S_1,A_1\right]\right]\\
=&\mathbb{E}^{\pi^b}\left[\frac{1}{\pi_1^b(A_1\mid S_1)}\mathbb{E}^{\pi^b}\left[(\sum_{a'}(b_{V,2}^{\pi_\theta}(a',O_2,H_1)-\widehat{b}_{V,2}^{\pi_\theta}(a',O_2,H_1)))\nabla_\theta\pi_{\theta_1}(A_1\mid O_1)\mid S_1,A_1\right]\right]\\
&+\mathbb{E}^{\pi^b}\left[\frac{1}{\pi_1^b(A_1\mid S_1)}\mathbb{E}^{\pi^b}\left[(R_1+\sum_{a'}\widehat{b}_{V,2}^{\pi_\theta}(a',O_2,H_1))\nabla_\theta\pi_{\theta_1}(A_1\mid O_1)\mid S_1,A_1\right]\right]\\
&+\mathbb{E}^{\pi^b}\left[\frac{1}{\pi_1^b(A_1\mid S_1)}\mathbb{E}^{\pi^b}\left[\sum_{a'}(b_{\nabla V,2}^{\pi_\theta}(a',O_2,H_1)-\widehat{b}_{\nabla V,2}^{\pi_\theta}(a',O_2,H_1))\pi_{\theta_1}(A_1\mid O_1)\mid S_1,A_1\right]\right]
\end{aligned}
\tag{44}
$$

$$+ \mathbb{E}^{\pi^b}\left[\frac{1}{\pi_1^b(A_1 \mid S_1)}\mathbb{E}^{\pi^b}\left[\sum_{a'}\widehat{b}^{\pi_\theta}_{\nabla V,2}(a',O_2,H_1)\pi_{\theta_1}(A_1 \mid O_1) \mid S_1,A_1\right]\right]$$

$$- \mathbb{E}^{\pi^b}\left[\frac{1}{\pi_1^b(A_1 \mid S_1)}\mathbb{E}^{\pi^b}\left[\widehat{b}^{\pi_\theta}_{\nabla V,1}(A_1,O_1) \mid S_1,A_1\right]\right]$$

$$= I + II + III$$

where

$$
\begin{aligned}
I = \mathbb{E}^{\pi^b}\bigg[\frac{1}{\pi_1^b(A_1 \mid S_1)}\mathbb{E}^{\pi^b}\bigg[&(R_1 + \sum_{a'}\widehat{b}^{\pi_\theta}_{V,2}(a',O_2,H_1))\nabla_\theta\pi_{\theta_1}(A_1 \mid O_1) \\
&+ \sum_{a'}\widehat{b}^{\pi_\theta}_{\nabla V,2}(a',O_2,H_1)\pi_{\theta_1}(A_1 \mid O_1) - \widehat{b}^{\pi_\theta}_{\nabla V,1}(A_1,O_1) \mid S_1,A_1\bigg]\bigg]
\end{aligned}
\tag{45}
$$

and

$$
\begin{aligned}
&II \\
&= \mathbb{E}^{\pi^b}\bigg[\frac{1}{\pi_1^b(A_1 \mid S_1)}\mathbb{E}^{\pi^b}\bigg[(\sum_{a'}(b^{\pi_\theta}_{V,2}(a',O_2,H_1) - \widehat{b}^{\pi_\theta}_{V,2}(a',O_2,H_1)))\nabla_\theta\pi_{\theta_1}(A_1 \mid O_1) \mid S_1,A_1\bigg]\bigg]
\end{aligned}
\tag{46}
$$

and

$$
\begin{aligned}
&III \\
&= \mathbb{E}^{\pi^b}\bigg[\frac{1}{\pi_1^b(A_1 \mid S_1)}\mathbb{E}^{\pi^b}\bigg[\sum_{a'}(b^{\pi_\theta}_{\nabla V,2}(a',O_2,H_1) - \widehat{b}^{\pi_\theta}_{\nabla V,2}(a',O_2,H_1))\pi_{\theta_1}(A_1 \mid O_1) \mid S_1,A_1\bigg]\bigg].
\end{aligned}
\tag{47}
$$

Intuitively, the term $I$ can be regarded as the error caused by solving equation 37, the term $II$ can be understood as the estimation error for the V-bridge function in the previous step, and term $III$ can be viewed as the estimation error for the gradient bridge function in the previous step.

We deal with the term $III$ at first. We introduce the following lemma that is useful for analyzing the term $III$.

**Lemma 13.** *For any function $f_t : \mathcal{A} \times \mathcal{O} \times \mathcal{H}_{t-1} \to \mathbb{R}^d$, the following holds:*

$$
\begin{aligned}
&\mathbb{E}^{\pi^b}\left[\frac{p_t^{\pi_\theta}(S_t,H_{t-1})}{p_t^{\pi^b}(S_t,H_{t-1})\pi_t^b(A_t \mid S_t)}f_t(A_t,O_t,H_{t-1})\right] \\
&= \mathbb{E}^{\pi^b}\left[\frac{p_{t-1}^{\pi_\theta}(S_{t-1},H_{t-2})\pi_{\theta_{t-1}}(A_{t-1} \mid O_{t-1},H_{t-2})}{p_{t-1}^{\pi^b}(S_{t-1},H_{t-2})\pi_{t-1}^b(A_{t-1} \mid S_{t-1})}\sum_a f_t(a,O_t,H_{t-1})\right].
\end{aligned}
\tag{48}
$$

Then, according to Lemma 13, we have

$$
\begin{aligned}
III &= \mathbb{E}^{\pi^b}\left[\frac{1}{\pi_1^b(A_1 \mid S_1)}\mathbb{E}^{\pi^b}\left[\sum_{a'}(b_{\nabla V,2}^{\pi_\theta}(a', O_2, H_1)\right.\right. \\
&\quad \left.\left. -\widehat{b}_{\nabla V,2}^{\pi_\theta}(a', O_2, H_1))\pi_{\theta_1}(A_1 \mid O_1) \mid S_1, A_1\right]\right] \\
&= \mathbb{E}^{\pi^b}\left[\frac{1}{\pi_1^b(A_1 \mid S_1)}\sum_{a'}(b_{\nabla V,2}^{\pi_\theta}(a', O_2, H_1)-\widehat{b}_{\nabla V,2}^{\pi_\theta}(a', O_2, H_1))\pi_{\theta_1}(A_1 \mid O_1)\right] \\
&= \mathbb{E}^{\pi^b}\left[\frac{p_2^{\pi_\theta}(S_2, H_1)}{p_2^{\pi^b}(S_2, H_1)\pi_2^b(A_2 \mid S_2)}(b_{\nabla V,2}^{\pi_\theta}(A_2, O_2, H_1)-\widehat{b}_{\nabla V,2}^{\pi_\theta}(A_2, O_2, H_1))\right] \text{ by Lemma 13} \\
&= \mathbb{E}^{\pi^b}\left[\mathbb{E}^{\pi^b}\left[\frac{p_2^{\pi_\theta}(S_2, H_1)}{p_2^{\pi^b}(S_2, H_1)\pi_2^b(A_2 \mid S_2)}(b_{\nabla V,2}^{\pi_\theta}(A_2, O_2, H_1)\right.\right. \\
&\quad \left.\left. -\widehat{b}_{\nabla V,2}^{\pi_\theta}(A_2, O_2, H_1)) \mid A_2, S_2, H_1\right]\right] \\
&= \mathbb{E}^{\pi^b}\left[\frac{p_2^{\pi_\theta}(S_2, H_1)}{p_2^{\pi^b}(S_2, H_1)\pi_2^b(A_2 \mid S_2)}\mathbb{E}^{\pi^b}\left[(b_{\nabla V,2}^{\pi_\theta}(A_2, O_2, H_1)\right.\right. \\
&\quad \left.\left. -\widehat{b}_{\nabla V,2}^{\pi_\theta}(A_2, O_2, H_1)) \mid A_2, S_2, H_1\right]\right]
\end{aligned}
$$
(49)

We notice that we by using the conditional moment equation equation 37 again for $\mathbb{E}^{\pi^b}\left[b_{\nabla V,2}^{\pi_\theta}(A_2, O_2, H_1) \mid A_2, S_2, H_1\right]$, we can further get three terms using the same strategy for analyzing $\mathbb{E}^{\pi^b}\left[\frac{1}{\pi_1^b(A_1|S_1)}(b_{\nabla V,1}^{\pi_\theta}(A_1, O_1) - \widehat{b}_{\nabla V,1}^{\pi_\theta}(A_1, O_1))\right]$ previously. Specifically, we have

$$
\begin{aligned}
III &= \mathbb{E}^{\pi^b}\left[\frac{p_2^{\pi_\theta}(S_2, H_1)}{p_2^{\pi^b}(S_2, H_1)\pi_2^b(A_2 \mid S_2)}\mathbb{E}^{\pi^b}\left[(b_{\nabla V,2}^{\pi_\theta}(A_2, O_2, H_1)\right.\right. \\
&\quad \left.\left. -\widehat{b}_{\nabla V,2}^{\pi_\theta}(A_2, O_2, H_1)) \mid A_2, S_2, H_1\right]\right] \\
&= (a) + (b) + (c)
\end{aligned}
$$
(50)

where

$$
\begin{aligned}
(a) &= \mathbb{E}^{\pi^b}\left[\frac{p_2^{\pi_\theta}(S_2, H_1)}{p_2^{\pi^b}(S_2, H_1)\pi_2^b(A_2 \mid S_2)}\mathbb{E}^{\pi^b}\left[(R_2 + \sum_{a'}\widehat{b}_{V,3}^{\pi_\theta}(a', O_3, H_2))\nabla_\theta\pi_{\theta_2}(A_2 \mid O_2, H_1)\right.\right. \\
&\quad \left.\left. + \sum_{a'}\widehat{b}_{\nabla V,3}^{\pi_\theta}(a', O_3, H_2)\pi_{\theta_2}(A_2 \mid O_2, H_1) - \widehat{b}_{\nabla V,2}^{\pi_\theta}(A_2, O_2, H_1) \mid S_2, A_2, H_1\right]\right]
\end{aligned}
$$
(51)

and

$$
\begin{aligned}
&(b) \\
&= \mathbb{E}^{\pi^b}\left[\frac{p_2^{\pi_\theta}(S_2, H_1)}{p_2^{\pi^b}(S_2, H_1)\pi_2^b(A_2 \mid S_2)}\mathbb{E}^{\pi^b}\left[(\sum_{a'}(b_{V,3}^{\pi_\theta}(a', O_3, H_2)-\widehat{b}_{V,3}^{\pi_\theta}(a', O_3, H_2)))\right.\right. \\
&\quad \left.\left. \nabla_\theta\pi_{\theta_2}(A_2 \mid O_2, H_1) \mid S_2, A_2, H_1\right]\right]
\end{aligned}
$$
(52)

and

$$
\begin{aligned}
&(c) \\
&= \mathbb{E}^{\pi^b}\left[\frac{p_2^{\pi_\theta}(S_2, H_1)}{p_2^{\pi^b}(S_2, H_1)\pi_2^b(A_2 \mid S_2)}\mathbb{E}^{\pi^b}\left[\sum_{a'}(b_{\nabla V,3}^{\pi_\theta}(a', O_3, H_2)-\widehat{b}_{\nabla V,3}^{\pi_\theta}(a', O_3, H_2))\right.\right. \\
&\quad \left.\left. \pi_{\theta_2}(A_2 \mid O_2, H_1) \mid S_2, A_2, H_1\right]\right].
\end{aligned}
$$
(53)

Then we can analyze term (c) by Lemma 13 again.

For general $t$, we define $\kappa_{\pi^b,t}^{\pi_\theta}(S_t, H_{t-1})$ as $\frac{p_t^{\pi_\theta}(S_t, H_{t-1})}{p_t^{\pi^b}(S_t, H_{t-1})}$, which denotes the density ratio at time $t$. Then, by induction, we have the following decomposition:

$$\mathbb{E}^{\pi^b}\left[\sum_a b_{\nabla V,1}^{\pi_\theta}(a, O_1) - \sum_a \widehat{b}_{\nabla V,1}(a, O_1)\right] = \sum_{t=1}^T \epsilon_t + \sum_{t=1}^{T-1} \varepsilon_t \tag{54}$$

where

$$\epsilon_t := \mathbb{E}^{\pi^b}\left[\frac{\kappa_{\pi^b,t}^{\pi_\theta}(S_t, H_{t-1})}{\pi_t^b(A_t \mid S_t)}\mathbb{E}^{\pi^b}\left[(R_t + \sum_{a'}\widehat{b}_{V,t+1}^{\pi_\theta}(a', O_{t+1}, H_t))\nabla_\theta \pi_{\theta_t}(A_t \mid O_t, H_{t-1})\right.\right.$$

$$\left.\left. + \sum_{a'}\widehat{b}_{\nabla V,t+1}^{\pi_\theta}(a', O_{t+1}, H_t)\pi_{\theta_t}(A_t \mid O_t, H_{t-1}) - \widehat{b}_{\nabla V,t}^{\pi_\theta}(A_t, O_t, H_{t-1}) \mid S_t, A_t, H_{t-1}\right]\right] \tag{55}$$

and

$$\varepsilon_t :=$$

$$\mathbb{E}^{\pi^b}\left[\frac{\kappa_{\pi^b,t}^{\pi_\theta}(S_t, H_{t-1})}{\pi_t^b(A_t \mid S_t)}\mathbb{E}^{\pi^b}\left[(\sum_{a'}(b_{V,t+1}^{\pi_\theta}(a', O_{t+1}, H_t) - \widehat{b}_{V,t+1}^{\pi_\theta}(a', O_{t+1}, H_t)))\right.\right. \tag{56}$$

$$\left.\left.\nabla_\theta \pi_{\theta_t}(A_t \mid O_t, H_{t-1}) \mid A_t, S_t, H_{t-1}\right]\right].$$

Now we analyze $\varepsilon_t$ using Lemma 13 again. Since $H_{t-1} = (A_{t-1}, O_{t-1}, H_{t-2})$, we can define a function as

$$f_t(A_t, O_t, H_{t-1}) = (b_{V,t}^{\pi_\theta}(A_t, O_t, H_{t-1}) - \widehat{b}_{V,t}^{\pi_\theta}(A_t, O_t, H_{t-1}))\nabla_\theta \log \pi_{\theta_{t-1}}(A_{t-1} \mid O_{t-1}, H_{t-2}).$$

By using the operation $\pi_\theta \nabla_\theta \log \pi_\theta = \nabla_\theta \pi_\theta$, we can express $\varepsilon_t$ as

$$\varepsilon_t$$

$$=\mathbb{E}^{\pi^b}\left[\frac{\kappa_{\pi^b,t}^{\pi_\theta}(S_t, H_{t-1})}{\pi_t^b(A_t \mid S_t)}\mathbb{E}^{\pi^b}\left[(\sum_{a'} f_{t+1}(a', O_{t+1}, H_t))\pi_{\theta_t}(A_t \mid O_t, H_{t-1}) \mid A_t, S_t, H_{t-1}\right]\right]$$

$$=\mathbb{E}^{\pi^b}\left[\frac{\kappa_{\pi^b,t}^{\pi_\theta}(S_t, H_{t-1})}{\pi_t^b(A_t \mid S_t)}(\sum_{a'} f_{t+1}(a', O_{t+1}, H_t))\pi_{\theta_t}(A_t \mid O_t, H_{t-1})\right]$$

$$=\mathbb{E}^{\pi^b}\left[\frac{\kappa_{\pi^b,t+1}^{\pi_\theta}(S_{t+1}, H_t)}{\pi_t^b(A_{t+1} \mid S_{t+1})}f_{t+1}(A_{t+1}, O_{t+1}, H_t)\right] \quad \text{by Lemma 13}$$

$$=\mathbb{E}^{\pi^b}\left[\frac{\kappa_{\pi^b,t+1}^{\pi_\theta}(S_{t+1}, H_t)}{\pi_t^b(A_{t+1} \mid S_{t+1})}(b_{V,t+1}^{\pi_\theta}(A_{t+1}, O_{t+1}, H_t)\right.$$

$$\left. - \widehat{b}_{V,t+1}^{\pi_\theta}(A_{t+1}, O_{t+1}, H_t))\nabla_\theta \log \pi_{\theta_t}(A_t \mid O_t, H_{t-1})\right] \tag{57}$$

$$=\mathbb{E}^{\pi^b}\left[\mathbb{E}^{\pi^b}\left[\frac{\kappa_{\pi^b,t+1}^{\pi_\theta}(S_{t+1}, H_t)}{\pi_t^b(A_{t+1} \mid S_{t+1})}(b_{V,t+1}^{\pi_\theta}(A_{t+1}, O_{t+1}, H_t)-\right.\right.$$

$$\left.\left.\widehat{b}_{V,t+1}^{\pi_\theta}(A_{t+1}, O_{t+1}, H_t))\nabla_\theta \log \pi_{\theta_t}(A_t \mid O_t, H_{t-1}) \mid S_{t+1}, A_{t+1}, H_t\right]\right]$$

$$=\mathbb{E}^{\pi^b}\left[\frac{\kappa_{\pi^b,t+1}^{\pi_\theta}(S_{t+1}, H_t)}{\pi_t^b(A_{t+1} \mid S_{t+1})}\nabla_\theta \log \pi_{\theta_t}(A_t \mid O_t, H_{t-1})\mathbb{E}^{\pi^b}\left[(b_{V,t+1}^{\pi_\theta}(A_{t+1}, O_{t+1}, H_t)-\right.\right.$$

$$\left.\left.\widehat{b}_{V,t+1}^{\pi_\theta}(A_{t+1}, O_{t+1}, H_t)) \mid S_{t+1}, A_{t+1}, H_t\right]\right].$$

Then we can consider

$$\mathbb{E}^{\pi^b}\left[(b_{V,t+1}^{\pi_\theta}(A_{t+1},O_{t+1},H_t)-\widehat{b}_{V,t+1}^{\pi_\theta}(A_{t+1},O_{t+1},H_t))\mid S_{t+1},A_{t+1},H_t\right]$$

$$=\mathbb{E}^{\pi^b}\left[R_{t+1}\pi_{\theta_{t+1}}(A_{t+1}\mid O_{t+1},H_t)\right.$$

$$\left.+\sum_{a'}b_{V,t+2}^{\pi_\theta}(a',O_{t+2},H_{t+1})\pi_{\theta_{t+1}}(A_{t+1}\mid O_{t+1},H_t)\mid S_{t+1},H_t,A_{t+1}\right]$$

$$-\mathbb{E}^{\pi^b}\left[\widehat{b}_{V,t+1}^{\pi_\theta}(A_{t+1},O_{t+1},H_t))\mid S_{t+1},A_{t+1},H_t\right]\text{ by equation (33)}$$

$$=\mathbb{E}^{\pi^b}\left[R_{t+1}\pi_{\theta_{t+1}}(A_{t+1}\mid O_{t+1},H_t)\right.$$

$$\left.+\sum_{a'}b_{V,t+2}^{\pi_\theta}(a',O_{t+2},H_{t+1})\pi_{\theta_{t+1}}(A_{t+1}\mid O_{t+1},H_t)\mid S_{t+1},H_t,A_{t+1}\right]$$

$$-\mathbb{E}^{\pi^b}\left[R_{t+1}\pi_{\theta_{t+1}}(A_{t+1}\mid O_{t+1},H_t)\right.$$

$$\left.+\sum_{a'}\widehat{b}_{V,t+2}^{\pi_\theta}(a',O_{t+2},H_{t+1})\pi_{\theta_{t+1}}(A_{t+1}\mid O_{t+1},H_t)\mid S_{t+1},H_t,A_{t+1}\right]$$

$$+\mathbb{E}^{\pi^b}\left[R_{t+1}\pi_{\theta_{t+1}}(A_{t+1}\mid O_{t+1},H_t)\right.$$

$$\left.+\sum_{a'}\widehat{b}_{V,t+2}^{\pi_\theta}(a',O_{t+2},H_{t+1})\pi_{\theta_{t+1}}(A_{t+1}\mid O_{t+1},H_t)\mid S_{t+1},H_t,A_{t+1}\right]$$

$$-\mathbb{E}^{\pi^b}\left[\widehat{b}_{V,t+1}^{\pi_\theta}(A_{t+1},O_{t+1},H_t))\mid S_{t+1},A_{t+1},H_t\right]$$

$$=\mathbb{E}^{\pi^b}\left[\sum_{a'}(b_{V,t+2}^{\pi_\theta}(a',O_{t+2},H_{t+1})\right.$$

$$\left.-\widehat{b}_{V,t+2}^{\pi_\theta}(a',O_{t+2},H_{t+1}))\pi_{\theta_{t+1}}(A_{t+1}\mid O_{t+1},H_t)\mid S_{t+1},H_t,A_{t+1}\right]$$

$$+\mathbb{E}^{\pi^b}\left[R_{t+1}\pi_{\theta_{t+1}}(A_{t+1}\mid O_{t+1},H_t)\right.$$

$$\left.+\sum_{a'}\widehat{b}_{V,t+2}^{\pi_\theta}(a',O_{t+2},H_{t+1})\pi_{\theta_{t+1}}(A_{t+1}\mid O_{t+1},H_t)\mid S_{t+1},H_t,A_{t+1}\right]$$

$$-\mathbb{E}^{\pi^b}\left[\widehat{b}_{V,t+1}^{\pi_\theta}(A_{t+1},O_{t+1},H_t))\mid S_{t+1},A_{t+1},H_t\right]\tag{58}$$

Therefore,

$$\varepsilon_t$$

$$=\mathbb{E}^{\pi^b}\left[\frac{\kappa_{\pi^b,t+1}^{\pi_\theta}(S_{t+1},H_t)}{\pi_t^b(A_{t+1}\mid S_{t+1})}\nabla_\theta\log\pi_{\theta_t}(A_t\mid O_t,H_{t-1})\mathbb{E}^{\pi^b}\left[(b_{V,t+1}^{\pi_\theta}(A_{t+1},O_{t+1},H_t)-\right.\right.$$

$$\left.\left.\widehat{b}_{V,t+1}^{\pi_\theta}(A_{t+1},O_{t+1},H_t))\mid S_{t+1},A_{t+1},H_t\right]\right]$$

$$=\mathbb{E}^{\pi^b}\left[\frac{\kappa_{\pi^b,t+1}^{\pi_\theta}(S_{t+1},H_t)}{\pi_t^b(A_{t+1}\mid S_{t+1})}\nabla_\theta\log\pi_{\theta_t}(A_t\mid O_t,H_{t-1})\mathbb{E}^{\pi^b}\left[R_{t+1}\pi_{\theta_{t+1}}(A_{t+1}\mid O_{t+1},H_t)\right.\right.$$

$$+\sum_{a'}\widehat{b}_{V,t+2}^{\pi_\theta}(a',O_{t+2},H_{t+1})\pi_{\theta_{t+1}}(A_{t+1}\mid O_{t+1},H_t)$$

$$\left.\left.-\widehat{b}_{V,t+1}^{\pi_\theta}(A_{t+1},O_{t+1},H_t))\mid S_{t+1},A_{t+1},H_t\right]\right]$$

$$+\mathbb{E}^{\pi^b}\left[\frac{\kappa_{\pi^b,t+1}^{\pi_\theta}(S_{t+1},H_t)}{\pi_t^b(A_{t+1}\mid S_{t+1})}\nabla_\theta\log\pi_{\theta_t}(A_t\mid O_t,H_{t-1})\mathbb{E}^{\pi^b}\left[\sum_{a'}(b_{V,t+2}^{\pi_\theta}(a',O_{t+2},H_{t+1})\right.\right.$$

$$-\widehat{b}_{V,t+2}^{\pi_\theta}(a',O_{t+2},H_{t+1}))\pi_{\theta_{t+1}}(A_{t+1}\mid O_{t+1},H_t)\mid S_{t+1},A_{t+1},H_t\Big]\Big]. \tag{59}$$

The first term can be understood as the estimation error for the min-max estimator, and the second term is caused by the estimation error for the value bridge function from the last step. Notice that the second term can be analyzed by using Lemma 13 again by setting

$$\begin{aligned}
&f_{t+2}(A_{t+2},O_{t+2},H_{t+1})\\
&:=(b_{V,t+2}^{\pi_\theta}(A_{t+2},O_{t+2},H_{t+1})-\widehat{b}_{V,t+2}^{\pi_\theta}(A_{t+2},O_{t+2},H_{t+1}))\nabla_\theta\log\pi_{\theta_t}(A_t\mid O_t,H_{t-1}).
\end{aligned} \tag{60}$$

By induction, we have

$$\varepsilon_t=\sum_{j=t+1}^{T}e_j \tag{61}$$

where

$$\begin{aligned}
e_j=&\mathbb{E}^{\pi^b}\Bigg[\frac{\kappa_{\pi^b,j}^{\pi_\theta}(S_j,H_{j-1})}{\pi_j^b(A_j\mid S_j)}\nabla_\theta\log\pi_{\theta_t}(A_t\mid O_t,H_{t-1})\mathbb{E}^{\pi^b}\Big[R_j\pi_{\theta_j}(A_j\mid O_j,H_{j-1})\\
&+\sum_{a'}\widehat{b}_{V,j+1}^{\pi_\theta}(a',O_{j+1},H_j)\pi_{\theta_j}(A_j\mid O_j,H_{j-1})-\widehat{b}_{V,j}^{\pi_\theta}(A_j,O_j,H_{j-1}))\mid S_j,A_j,H_{j-1}\Big]\Bigg].
\end{aligned} \tag{62}$$

We note that $\nabla_\theta\log\pi_{\theta_t}(A_t\mid O_t,H_{t-1})$ is always measurable with repect to the sigma-field generated by $H_{j-1}$ for each $j\geq t+1$. Therefore the term $\nabla_\theta\log\pi_{\theta_t}(A_t\mid O_t,H_{t-1})$ is always kept for each $j\geq t+1$ in $e_j$.

In summary, we can decompose the policy gradient as

$$\mathbb{E}^{\pi^b}\Bigg[\sum_a b_{\nabla V,1}^{\pi_\theta}(a,O_1)-\sum_a\widehat{b}_{\nabla V,1}^{\pi_\theta}(a,O_1)\Bigg]=\sum_{t=1}^{T}\epsilon_t+\sum_{t=1}^{T-1}\sum_{j=t+1}^{T}e_j \tag{63}$$

where

$$\begin{aligned}
\epsilon_t=&\mathbb{E}^{\pi^b}\Bigg[\frac{\kappa_{\pi^b,t}^{\pi_\theta}(S_t,H_{t-1})}{\pi_t^b(A_t\mid S_t)}\mathbb{E}^{\pi^b}\Big[(R_t+\sum_{a'}\widehat{b}_{V,t+1}^{\pi_\theta}(a',O_{t+1},H_t))\nabla_\theta\pi_{\theta_t}(A_t\mid O_t,H_{t-1})\\
&+\sum_{a'}\widehat{b}_{\nabla V,t+1}^{\pi_\theta}(a',O_{t+1},H_t)\pi_{\theta_t}(A_t\mid O_t,H_{t-1})-\widehat{b}_{\nabla V,t}^{\pi_\theta}(A_t,O_t,H_{t-1})\mid S_t,A_t,H_{t-1}\Big]\Bigg]
\end{aligned} \tag{64}$$

and

$$\begin{aligned}
e_j=&\mathbb{E}^{\pi^b}\Bigg[\frac{\kappa_{\pi^b,j}^{\pi_\theta}(S_j,H_{j-1})}{\pi_j^b(A_j\mid S_j)}\nabla_\theta\log\pi_{\theta_t}(A_t\mid O_t,H_{t-1})\mathbb{E}^{\pi^b}\Big[R_j\pi_{\theta_j}(A_j\mid O_j,H_{j-1})\\
&+\sum_{a'}\widehat{b}_{V,j+1}^{\pi_\theta}(a',O_{j+1},H_j)\pi_{\theta_j}(A_j\mid O_j,H_{j-1})-\widehat{b}_{V,j}^{\pi_\theta}(A_j,O_j,H_{j-1}))\mid S_j,A_j,H_{j-1}\Big]\Bigg].
\end{aligned} \tag{65}$$

$\epsilon_t$ denotes the one-step error caused by min-max estimation for the conditional operator for *gradient* bridge at time $t$, while $e_t$ denotes the one-step error caused by min-max estimation for the conditional operator for *value* bridge at time $t$.

In the next section, we upper bound them in terms of density ratio, one-step error and ill-posedness.

### F.2 Bounds for the decomposition

Recall from the previous section that

$$
\begin{aligned}
&\nabla_\theta \mathcal{V}(\pi_\theta) - \widehat{\nabla_\theta \mathcal{V}(\pi_\theta)} \\
=&\mathbb{E}^{\pi^b}[\sum_a b_{\nabla V,1}^{\pi_\theta}(a, O_1)] - \mathbb{E}^{\pi^b}[\sum_a \widehat{b}_{\nabla V,1}^{\pi_\theta}(a, O_1)] + \mathbb{E}^{\pi^b}[\sum_a \widehat{b}_{\nabla V,1}^{\pi_\theta}(a, O_1)] - \widehat{\mathbb{E}}^{\pi^b}[\sum_a \widehat{b}_{\nabla V,1}^{\pi_\theta}(a, O_1)] \\
=&\sum_{t=1}^T \epsilon_t + \sum_{t=1}^{T-1} \sum_{j=t+1}^T e_j + \mathbb{E}^{\pi^b}[\sum_a \widehat{b}_{\nabla V,1}^{\pi_\theta}(a, O_1)] - \widehat{\mathbb{E}}^{\pi^b}[\sum_a \widehat{b}_{\nabla V,1}^{\pi_\theta}(a, O_1)] \text{ by equation (63).}
\end{aligned}
$$
$$(66)$$

Then by the triangular inequality, we have

$$
\begin{aligned}
&\|\nabla_\theta \mathcal{V}(\pi_\theta) - \widehat{\nabla_\theta \mathcal{V}(\pi_\theta)}\|_{\ell^2} \\
=&\|\sum_{t=1}^T \epsilon_t + \sum_{t=1}^{T-1} \sum_{j=t+1}^T e_j + \mathbb{E}^{\pi^b}[\sum_a \widehat{b}_{\nabla V,1}^{\pi_\theta}(a, O_1)] - \widehat{\mathbb{E}}^{\pi^b}[\sum_a \widehat{b}_{\nabla V,1}^{\pi_\theta}(a, O_1)]\|_{\ell^2} \\
\leq&\sum_{t=1}^T \|\epsilon_t\|_{\ell^2} + \sum_{t=1}^{T-1} \sum_{j=t+1}^T \|e_j\|_{\ell^2} + \|\mathbb{E}^{\pi^b}[\sum_a \widehat{b}_{\nabla V,1}^{\pi_\theta}(a, O_1)] - \widehat{\mathbb{E}}^{\pi^b}[\sum_a \widehat{b}_{\nabla V,1}^{\pi_\theta}(a, O_1)]\|_{\ell^2}.
\end{aligned}
$$
$$(67)$$

The third term can be upper bounded by the uniform law of large numbers according to the empirical processes which involve the size of $\mathcal{B}^{(1)}$. In the following, we consider the first term and the second term.

By Assumption 4(b), for each $\widehat{b}_{V,t+1}^{\pi_\theta}, \widehat{b}_{\nabla V,t+1}^{\pi_\theta}$ and any $\theta$, there exists a solution $b_{V,t}^*(\widehat{b}_{V,t+1}^{\pi_\theta}, \theta), b_{\nabla V,t}^*(\widehat{b}_{V,t+1}^{\pi_\theta}, \widehat{b}_{\nabla V,t+1}^{\pi_\theta}, \theta)$ to the conditional moment equations (3)(4). By Assumptions 1, 3, and the arguments in Part I in the proof of Theorem 1 (Appendix E), $b_{V,t}^*(\widehat{b}_{V,t+1}^{\pi_\theta}, \theta)$, $b_{\nabla V,t}^*(\widehat{b}_{V,t+1}^{\pi_\theta}, \widehat{b}_{\nabla V,t+1}^{\pi_\theta}, \theta)$ and $(\widehat{b}_{V,t+1}^{\pi_\theta}, \widehat{b}_{\nabla V,t+1}^{\pi_\theta}, \theta)$ also satisfies the conditional moment equations that depend on the latent states (33)(37). Therefore, we have

$$
\begin{aligned}
\epsilon_t =&\mathbb{E}^{\pi^b}\left[ \frac{\kappa_{\pi^b,t}^{\pi_\theta}(S_t, H_{t-1})}{\pi_t^b(A_t \mid S_t)} \mathbb{E}^{\pi^b}\left[ (R_t + \sum_{a'} \widehat{b}_{V,t+1}^{\pi_\theta}(a', O_{t+1}, H_t)) \nabla_\theta \pi_{\theta_t}(A_t \mid O_t, H_{t-1}) \right.\right. \\
&\left.\left. + \sum_{a'} \widehat{b}_{\nabla V,t+1}^{\pi_\theta}(a', O_{t+1}, H_t) \pi_{\theta_t}(A_t \mid O_t, H_{t-1}) - \widehat{b}_{\nabla V,t}^{\pi_\theta}(A_t, O_t, H_{t-1}) \mid S_t, A_t, H_{t-1} \right]\right] \\
=&\mathbb{E}^{\pi^b}\left[ \frac{\kappa_{\pi^b,t}^{\pi_\theta}(S_t, H_{t-1})}{\pi_t^b(A_t \mid S_t)} \mathbb{E}^{\pi^b}\left[ b_{\nabla V,t}^*(\widehat{b}_{V,t+1}^{\pi_\theta}, \widehat{b}_{\nabla V,t+1}^{\pi_\theta}, \theta)(A_t, O_t, H_{t-1}) \right.\right. \\
&\left.\left. - \widehat{b}_{\nabla V,t}^{\pi_\theta}(A_t, O_t, H_{t-1}) \mid S_t, A_t, H_{t-1} \right]\right] \text{ by plugging in the solution to (37)} \\
=&\mathbb{E}^{\pi^b}\left[ \frac{\kappa_{\pi^b,t}^{\pi_\theta}(S_t, H_{t-1})}{\pi_t^b(A_t \mid S_t)} (b_{\nabla V,t}^*(\widehat{b}_{V,t+1}^{\pi_\theta}, \widehat{b}_{\nabla V,t+1}^{\pi_\theta}, \theta)(A_t, O_t, H_{t-1}) - \widehat{b}_{\nabla V,t}^{\pi_\theta}(A_t, O_t, H_{t-1})) \right]
\end{aligned}
$$
$$(68)$$

and

$$
\begin{aligned}
e_j =&\mathbb{E}^{\pi^b}\left[ \frac{\kappa_{\pi^b,j}^{\pi_\theta}(S_j, H_{j-1})}{\pi_j^b(A_j \mid S_j)} \nabla_\theta \log \pi_{\theta_t}(A_t \mid O_t, H_{t-1}) \mathbb{E}^{\pi^b}\left[ R_j \pi_{\theta_j}(A_j \mid O_j, H_{j-1}) \right.\right. \\
&\left.\left. + \sum_{a'} \widehat{b}_{V,j+1}^{\pi_\theta}(a', O_{j+1}, H_j) \pi_{\theta_j}(A_j \mid O_j, H_{j-1}) - \widehat{b}_{V,j}^{\pi_\theta}(A_j, O_j, H_{j-1})) \mid S_j, A_j, H_{j-1} \right]\right]. \\
=&\mathbb{E}^{\pi^b}\left[ \frac{\kappa_{\pi^b,j}^{\pi_\theta}(S_j, H_{j-1})}{\pi_j^b(A_j \mid S_j)} \nabla_\theta \log \pi_{\theta_t}(A_t \mid O_t, H_{t-1}) \mathbb{E}^{\pi^b}\left[ b_{V,j}^*(\widehat{b}_{V,j+1}^{\pi_\theta}, \theta)(A_j, O_j, H_{j-1}) \right.\right.
\end{aligned}
$$

$$-\widehat{b}_{V,j}^{\pi_\theta}(A_j, O_j, H_{j-1})) \mid S_j, A_j, H_{j-1}\Big]\Big] \text{ by plugging in the solution to (33)}$$

$$=\mathbb{E}^{\pi^b}\left[\frac{\kappa_{\pi^b,j}^{\pi_\theta}(S_j, H_{j-1})}{\pi_j^b(A_j \mid S_j)}\nabla_\theta \log \pi_{\theta_t}(A_t \mid O_t, H_{t-1})\right.$$

$$\left.(b_{V,j}^*(\widehat{b}_{V,j+1}^{\pi_\theta},\theta)(A_j, O_j, H_{j-1}) - \widehat{b}_{V,j}^{\pi_\theta}(A_j, O_j, H_{j-1}))\right]. \tag{69}$$

Then we upper bound $\|\epsilon_t\|_{\ell^2}$ and $\|e_t\|_{\ell^2}$. For clarity, we write $b_{V,t}^*, b_{\nabla V,t}^*$ to denote $b_{V,t}^*(\widehat{b}_{V,t+1}^{\pi_\theta},\theta), b_{\nabla V,t}^*(\widehat{b}_{V,t+1}^{\pi_\theta}, \widehat{b}_{\nabla V,t+1}^{\pi_\theta},\theta)$ if there is no confusion.

$$\|\epsilon_t\|_{\ell^2}$$

$$=\sqrt{\sum_{i=1}^{d_\Theta}\epsilon_{t,i}^2}$$

$$\leq\left(\sum_{i=1}^{d_\Theta}\mathbb{E}^{\pi^b}\left[\left(\frac{\kappa_{\pi^b,t}^{\pi_\theta}(S_t, H_{t-1})}{\pi_t^b(A_t \mid S_t)}\right)^2\right]\mathbb{E}^{\pi^b}\left[\left(b_{\nabla V,t,i}^*(A_t, O_t, H_{t-1}) - \widehat{b}_{\nabla V,t,i}^{\pi_\theta}(A_t, O_t, H_{t-1})\right)^2\right]\right)^{\frac{1}{2}}$$

$$=\left(\mathbb{E}^{\pi^b}\left[\left(\frac{\kappa_{\pi^b,t}^{\pi_\theta}(S_t, H_{t-1})}{\pi_t^b(A_t \mid S_t)}\right)^2\right]\right)^{\frac{1}{2}}\left(\sum_{i=1}^{d_\Theta}\mathbb{E}^{\pi^b}\left[\left(b_{\nabla V,t,i}^*(A_t, O_t, H_{t-1})\right.\right.\right.$$

$$\left.\left.\left.-\widehat{b}_{\nabla V,t,i}^{\pi_\theta}(A_t, O_t, H_{t-1})\right)^2\right]\right)^{\frac{1}{2}}$$

$$=\left(\mathbb{E}^{\pi^b}\left[\left(\frac{\kappa_{\pi^b,t}^{\pi_\theta}(S_t, H_{t-1})}{\pi_t^b(A_t \mid S_t)}\right)^2\right]\right)^{\frac{1}{2}}\left(\mathbb{E}^{\pi^b}\left[\sum_{i=1}^{d_\Theta}\left(b_{\nabla V,t,i}^*(A_t, O_t, H_{t-1})\right.\right.\right.$$

$$\left.\left.\left.-\widehat{b}_{\nabla V,t,i}^{\pi_\theta}(A_t, O_t, H_{t-1})\right)^2\right]\right)^{\frac{1}{2}}$$

$$=\left(\mathbb{E}^{\pi^b}\left[\left(\frac{\kappa_{\pi^b,t}^{\pi_\theta}(S_t, H_{t-1})}{\pi_t^b(A_t \mid S_t)}\right)^2\right]\right)^{\frac{1}{2}}\|b_{\nabla V,t}^*(A_t, O_t, H_{t-1}) - \widehat{b}_{\nabla V,t}^{\pi_\theta}(A_t, O_t, H_{t-1})\|_{\mathcal{L}^2(\ell^2, \mathbb{P}^{\pi^b})}$$

$$\leq C_{\pi^b}\|b_{\nabla V,t}^*(A_t, O_t, H_{t-1}) - \widehat{b}_{\nabla V,t}^{\pi_\theta}(A_t, O_t, H_{t-1})\|_{\mathcal{L}^2(\ell^2, \mathbb{P}^{\pi^b})} \text{ by Assumption 4(a)}$$

$$\leq C_{\pi^b}\tau_t\|\mathbb{E}^{\pi^b}\left[b_{\nabla V,t}^*(A_t, O_t, H_{t-1}) - \widehat{b}_{\nabla V,t}^{\pi_\theta}(A_t, O_t, H_{t-1}) \mid O_0, A_t, H_{t-1}\right]\|_{\mathcal{L}^2(\ell^2, \mathbb{P}^{\pi^b})}$$

by Definition D.1.

where the term $\|\mathbb{E}^{\pi^b}\left[b_{\nabla V,t}^*(A_t, O_t, H_{t-1}) - \widehat{b}_{\nabla V,t}^{\pi_\theta}(A_t, O_t, H_{t-1}) \mid O_0, A_t, H_{t-1}\right]\|_{\mathcal{L}^2(\ell^2, \mathbb{P}^{\pi^b})}$ is the projected residual mean squared error (RMSE) for the min-max estimation operator (Dikkala et al., 2020). We use the notation $\mathbb{T}_{X_t}$ as a projection operator into the space generated by $X_t = (O_0, A_t, H_{t-1})$. And the projected RMSE can be denoted as

$$\left\|\mathbb{T}_{X_t}\left[b_{\nabla V,t}^*(\widehat{b}_{V,t+1}^{\pi_\theta}, \widehat{b}_{\nabla V,t+1}^{\pi_\theta},\theta) - \widehat{b}_{\nabla V,t}^{\pi_\theta}\right]\right\|_{\mathcal{L}^2(\ell^2, \mathbb{P}^{\pi^b})}.$$

For the second term, we have $\|e_j\|_{\ell^2} = \sqrt{\sum_{i=1}^{d_\Theta}e_{j,i}^2}$, where

$$e_{j,i}^2$$

$$= \left( \mathbb{E}^{\pi^b} [\cdots] \right)^2$$

$$\leq \left( \mathbb{E}^{\pi^b} [|\cdots|] \right)^2$$

$$\leq \left( \sup_{t, \theta_t, a_t, o_t, h_{t-1}} \|\nabla_\theta \log \pi_{\theta_t}(a_t \mid o_t, h_{t-1})\|_{\ell^\infty} \mathbb{E}^{\pi^b} [|\cdots|] \right)^2$$

$$= G^2 \left( \mathbb{E}^{\pi^b} \left[ |\frac{\kappa_{\pi^b, j}^{\pi_\theta}(S_j, H_{j-1})}{\pi_j^b(A_j \mid S_j)} (b_{V,j}^*(A_j, O_j, H_{j-1}) - \widehat{b}_{V,j}^{\pi_\theta}(A_j, O_j, H_{j-1}))| \right] \right)^2$$

by Assumption 4(e)

which does not depend on the index $i$. Therefore we have

$$\|e_j\|_{\ell^2}$$

$$\leq G \sqrt{\sum_{i=t}^T d_{\Theta_i}} \mathbb{E}^{\pi^b} \left[ |\frac{\kappa_{\pi^b, j}^{\pi_\theta}(S_j, H_{j-1})}{\pi_j^b(A_j \mid S_j)} (b_{V,j}^*(A_j, O_j, H_{j-1}) - \widehat{b}_{V,j}^{\pi_\theta}(A_j, O_j, H_{j-1}))| \right]$$

$$\leq G \sqrt{\sum_{i=t}^T d_{\Theta_i}} \left( \mathbb{E}^{\pi^b} \left[ \left( \frac{\kappa_{\pi^b, j}^{\pi_\theta}(S_j, H_{j-1})}{\pi_t^b(A_j \mid S_j)} \right)^2 \right] \right)^{\frac{1}{2}}$$

$$\left( \mathbb{E}^{\pi^b} \left[ (b_{V,j}^*(A_j, O_j, H_{j-1}) - \widehat{b}_{V,j}^{\pi_\theta}(A_j, O_j, H_{j-1}))^2 \right] \right)^{\frac{1}{2}}$$

(Cauchy-Schwartz inequality)

$$\leq G \sqrt{\sum_{i=t}^T d_{\Theta_i}} \left( \mathbb{E}^{\pi^b} \left[ \left( \frac{\kappa_{\pi^b, j}^{\pi_\theta}(S_j, H_{j-1})}{\pi_t^b(A_j \mid S_j)} \right)^2 \right] \right)^{\frac{1}{2}}$$

$$\|b_{V,j}^*(A_j, O_j, H_{j-1}) - \widehat{b}_{V,j}^{\pi_\theta}(A_j, O_j, H_{j-1})\|_{\mathcal{L}^2(\mathbb{P}^{\pi^b})}$$

$$\leq C_{\pi^b} G \sqrt{\sum_{i=t}^T d_{\Theta_i}} \|b_{V,j}^*(A_j, O_j, H_{j-1}) - \widehat{b}_{V,j}^{\pi_\theta}(A_j, O_j, H_{j-1})\|_{\mathcal{L}^2(\mathbb{P}^{\pi^b})}$$

$$= C_{\pi^b} G \sqrt{\sum_{i=t}^T d_{\Theta_i} \tau_j} \|\mathbb{E}^{\pi^b} \left[ b_{V,j}^*(A_j, O_j, H_{j-1}) - \widehat{b}_{V,j}^{\pi_\theta}(A_j, O_j, H_{j-1}) \mid O_0, A_j, H_{j-1} \right] \|_{\mathcal{L}^2(\mathbb{P}^{\pi^b})}.$$

Similarly, we can use the notation for the projected RMSE:

$$\left\| \mathbb{T}_{X_j} \left[ b_{V,j}^*(\widehat{b}_{V,j+1}^{\pi_\theta}, \theta) - \widehat{b}_{V,j}^{\pi_\theta} \right] \right\|_{\mathcal{L}^2(\mathbb{P}^{\pi^b})}.$$

In summary, we have the upper bound for $\|\nabla_\theta \mathcal{V}(\pi_\theta) - \widehat{\nabla_\theta \mathcal{V}(\pi_\theta)}\|_{\ell^2}$:

$$\|\nabla_\theta \mathcal{V}(\pi_\theta) - \widehat{\nabla_\theta \mathcal{V}(\pi_\theta)}\|_{\ell^2}$$

$$\leq \sum_{t=1}^T \|\epsilon_t\|_{\ell^2} + \sum_{t=1}^{T-1} \sum_{j=t+1}^T \|e_j\|_{\ell^2} + \|\mathbb{E}^{\pi^b}[\sum_a \widehat{b}_{\nabla V,1}^{\pi_\theta}(a, O_1)] - \widehat{\mathbb{E}}^{\pi^b}[\sum_a \widehat{b}_{\nabla V,1}^{\pi_\theta}(a, O_1)]\|_{\ell^2}$$

$$\leq \sum_{t=1}^T C_{\pi^b} \tau_t \left\| \mathbb{T}_{X_t} \left[ b_{\nabla V,t}^*(\widehat{b}_{\nabla V,t+1}^{\pi_\theta}, \widehat{b}_{\nabla V,t+1}^{\pi_\theta}, \theta) - \widehat{b}_{\nabla V,t}^{\pi_\theta} \right] \right\|_{\mathcal{L}^2(\ell^2, \mathbb{P}^{\pi^b})}$$

$$+ \sum_{t=1}^{T-1} \sum_{j=t+1}^{T} C_{\pi^b} G \sqrt{\sum_{i=t}^{T} d_{\Theta_i} \tau_j} \left\| \mathbb{T}_{X_j} \left[ b_{V,j}^*(\widehat{b}_{V,j+1}^{\pi_\theta}, \theta) - \widehat{b}_{V,j}^{\pi_\theta} \right] \right\|_{\mathcal{L}^2(\mathbb{P}^{\pi^b})}$$

$$+ \| \mathbb{E}^{\pi^b} [\sum_a \widehat{b}_{\nabla V,1}^{\pi_\theta}(a, O_1)] - \widehat{\mathbb{E}}^{\pi^b} [\sum_a \widehat{b}_{\nabla V,1}^{\pi_\theta}(a, O_1)] \|_{\ell^2}. \tag{70}$$

All the undetermined terms are one-step errors, and we provide finite-sample error bounds on each three in the next section.

### F.3 ONE-STEP ESTIMATION

In this section, we upper bounds three terms from the previous section separately:
$\left\| \mathbb{T}_{X_t} \left[ b_{\nabla V,t}^*(\widehat{b}_{V,t+1}^{\pi_\theta}, \widehat{b}_{\nabla V,t+1}^{\pi_\theta}, \theta) - \widehat{b}_{\nabla V,t}^{\pi_\theta} \right] \right\|_{\mathcal{L}^2(\ell^2, \mathbb{P}^{\pi^b})}$, $\left\| \mathbb{T}_{X_j} \left[ b_{V,j}^*(\widehat{b}_{V,j+1}^{\pi_\theta}, \theta) - \widehat{b}_{V,j}^{\pi_\theta} \right] \right\|_{\mathcal{L}^2(\mathbb{P}^{\pi^b})}$, and
$\| \mathbb{E}^{\pi^b} [\sum_a \widehat{b}_{\nabla V,1}^{\pi_\theta}(a, O_1)] - \widehat{\mathbb{E}}^{\pi^b} [\sum_a \widehat{b}_{\nabla V,1}^{\pi_\theta}(a, O_1)] \|_{\ell^2}$.

We first introduce two concepts from the empirical process that are used to measure the size of function classes. The following definition is adapted from Wainwright (2019) and Foster & Syrgkanis (2019).

**Definition F.1** (Localize population Rademacher complexity and critical radius.)*. Given any real-valued function class $\mathcal{G}$ defined over a random vector $X$ and any radius $\delta > 0$, the local population Rademacher complexity is given by*

$$\mathcal{R}_n(\mathcal{G}, \delta) = \mathbb{E}_{\epsilon, X} [ \sup_{g \in \mathcal{G}: \|g\|_{2,2} \le \delta} |n^{-1} \sum_{i=1}^{n} \epsilon_i g(X_i)| ],$$

*where $\{X_i\}_{i=1}^{n}$ are i.i.d. copies of $X$ and $\{\epsilon_i\}_{i=1}^{n}$ are i.i.d. Rademacher random variables taking values in $\{-1, +1\}$ with equal probability. Further, assume that $\mathcal{G}$ is a 1-uniformly bounded vector-valued function class $\{g: \mathcal{X} \to \mathbb{R}^d, \sup_x \|g(x)\|_{\ell^2} \le 1\}$. Let $\mathcal{G}|_k$ denotes the k-th coordinate projection of $\mathcal{G}$. Assume that $\mathcal{G}|_k$ is a star-shaped function class, i.e. $\alpha g_k \in \mathcal{G}|_k$ for any $g_k \in \mathcal{G}|_k$ and scalar $\alpha \in [-1, 1]$. Then the critical radius of $\mathcal{G}$, denoted by $\delta_n$, is the solution to the inequality*

$$\max_{k=1,\dots,d} \mathcal{R}_n(\mathcal{G}|_k, \delta) \le \delta^2.$$

In this work, critical radius is used in the theoretical analysis to measure the size of function classes, which provides a way to get a *uniform* law of large numbers with a convergence rate at each time $t$. At each $t$, the uniformness comes from test functions $f \in \mathcal{F}^{(t)}$, estimated $\widehat{b}_{t+1}^{\pi_\theta} \in \mathcal{B}^{(t+1)}$ from the previous iteration $t + 1$, and the policy parameter $\theta \in \Theta$. Compared to the well-known VC-dimension or Rademacher complexity that is also used the measure the size of function classes, this localized version potentially provides optimal rates by utilizing local information.

In the following parts, we utilize critical radius to obtain the convergence rate of one-step estimation errors.

**Part I** We first upper bound the one-step estimation error about the conditional moment operator on the gradient bridge functions $\left\| \mathbb{T}_{X_t} \left[ b_{\nabla V,t}^*(\widehat{b}_{V,t+1}^{\pi_\theta}, \widehat{b}_{\nabla V,t+1}^{\pi_\theta}, \theta) - \widehat{b}_{\nabla V,t}^{\pi_\theta} \right] \right\|_{\mathcal{L}^2(\ell^2, \mathbb{P}^{\pi^b})}$. The proof techniques are adapted from Dikkala et al. (2020); Miao et al. (2022); Lu et al. (2022). The difference is that we develop an upper bound that is uniformly over $(\widehat{b}_{V,t+1}^{\pi_\theta}, \widehat{b}_{\nabla V,t+1}^{\pi_\theta}, \theta)$, and we are dealing with a random vector that needs a vector-formed uniform law.

We let $\mathcal{B}_V^{(t)}$ and $\mathcal{B}_{\nabla V}^{(t)}$ denote the function space that contain value bridge and gradient bridge respectively. Then $\mathcal{B}^{(t)} = \mathcal{B}_{\nabla V}^{(t)} \times \mathcal{B}_V^{(t)}$. We aim to upper bound the following term:

$$\sup_{\theta \in \Theta} \sup_{b_{V,t+1} \in \mathcal{B}_V^{(t+1)}, b_{\nabla V,t+1} \in \mathcal{B}_{\nabla V}^{(t+1)}} \left\| \mathbb{T}_{X_t} \left[ b_{\nabla V,t}^*(b_{V,t+1}, b_{\nabla V,t+1}, \theta) - \widehat{b}_{\nabla V,t} \right] \right\|_{\mathcal{L}^2(\ell^2, \mathbb{P}^{\pi^b})}$$

where $\widehat{b}_{\nabla V,t}$ is denoted as the min-max estimator when plugging in $(b_{V,t+1}, b_{\nabla V,t+1}, \theta)$ in equation 9, and $b_{\nabla V,t}^*(b_{V,t+1}, b_{\nabla V,t+1}, \theta)$ denotes the solution to (37) by Assumption 4(b).

In the proof, we assume that the function classes are already norm-constrained for clarity, and there is no penalty term related to the norm of functions in the definition of loss functions. Furthermore, we assume that $M_{\mathcal{F}} = 1$ without loss of generality. In addition, for simplicity, we abuse the notations of loss functions used in section 5. Specifically, we are focusing on the loss functions for the gradient bridge functions here.

Consider $\Psi_{t,N}(b_{\nabla V,t}, b_{V,t+1}, b_{\nabla V,t+1}, \theta, f)$ which is defined as

$$\frac{1}{N}\sum_{n=1}^{N}\left[\left(m_{\nabla V}(Z_{t,n}; b_{\nabla V,t}, b_{V,t+1}, b_{\nabla V,t+1}, \theta)\right)^T f(X_{t,n})\right].$$

We use $\Psi_t(b_{\nabla V,t}, b_{V,t+1}, b_{\nabla V,t+1}, \theta, f)$ to denotes the population version which is

$$\mathbb{E}^{\pi^b}\left[\left(m_{\nabla V}(Z_t; b_{\nabla V,t}, b_{V,t+1}, b_{\nabla V,t+1}, \theta)\right)^T f(X_t)\right].$$

Further, we use $\Psi_{t,N}^{\lambda}(b_{\nabla V,t}, b_{V,t+1}, b_{\nabla V,t+1}, \theta, f)$ to denote

$$\frac{1}{N}\sum_{n=1}^{N}\left[\left(m_{\nabla V}(Z_{t,n}; b_{\nabla V,t}, b_{V,t+1}, b_{\nabla V,t+1}, \theta)\right)^T f(X_{t,n})\right] - \lambda\|f\|_{N,2,2}^2.$$

Similarly, $\Psi_t^{\lambda}(b_{\nabla V,t}, b_{V,t+1}, b_{\nabla V,t+1}, \theta, f)$ is used to denote

$$\mathbb{E}^{\pi^b}\left[\left(m_{\nabla V}(Z_t; b_{\nabla V,t}, b_{V,t+1}, b_{\nabla V,t+1}, \theta)\right)^T f(X_t)\right] - \lambda\|f\|_{2,2}^2.$$

Then the min-max estimator is defined as

$$\widehat{b}_{\nabla V,t} = \underset{b_{\nabla V,t}\in\mathcal{B}_{\nabla V}^{(t)}}{\arg\min}\ \underset{f\in\mathcal{F}^{(t)}}{\sup}\ \Psi_{t,N}^{\lambda}(b_{\nabla V,t}, b_{V,t+1}, b_{\nabla V,t+1}, \theta, f)$$

given any $b_{V,t+1}\in\mathcal{B}_V^{(t+1)}, b_{\nabla V,t+1}\in\mathcal{B}_{\nabla V}^{(t+1)}, \theta\in\Theta$.

The true $\widehat{b}_{\nabla V,t}^*$ (depending on $(b_{V,t+1}, b_{\nabla V,t+1}, \theta)$) satisfies that

$$\Psi_t(b_{\nabla V,t}^*, b_{V,t+1}, b_{\nabla V,t+1}, \theta, f) = 0$$

given any $b_{V,t+1}\in\mathcal{B}_V^{(t+1)}, b_{\nabla V,t+1}\in\mathcal{B}_{\nabla V}^{(t+1)}, \theta\in\Theta$.

Furthermore, in order to get a clear uniform bound with respect to $\theta$, we introduce the following assumption:

**Assumption 8.** *There exists a constant $\tilde{\beta}$ such that the following holds for all $t = 1, ..., T$*

$$\sum_{a\in\mathcal{A}}\left|\pi_{\theta_t'}(a\mid o_t, h_{t-1}) - \pi_{\theta_t}(a\mid o_t, h_{t-1})\right| \leq \tilde{\beta}\|\theta - \theta'\|_{\ell^2}, \quad \textit{for all } o_t, h_{t-1}\in\mathcal{O}\times\mathcal{H}_{t-1}.$$

Then we start our proof. The key term that is used in the proof is called a sup-loss which is defined as

$$\underset{f\in\mathcal{F}^{(t)}}{\sup}\ \Psi_{t,N}(\widehat{b}_{\nabla V,t}, b_{V,t+1}, b_{\nabla V,t+1}, \theta, f) - \Psi_{t,N}(b_{\nabla V,t}^*, b_{V,t+1}, b_{\nabla V,t+1}, \theta, f) - 2\lambda\|f\|_{N,2,2}^2.$$

$$(71)$$

In our proof, we will upper bound and lower bound the sup-loss. We will show that the upper bound of the sup-loss is a small term that converges to 0, and the lower bound turns out to be an projected RMSE. In this way, we provide a finite-sample upper bound for the projected RMSE. We highlight that the described upper bound and lower bound should hold *uniformly* for any $b_{V,t+1}\in\mathcal{B}_V^{(t+1)}, b_{\nabla V,t+1}\in\mathcal{B}_{\nabla V}^{(t+1)}, \theta\in\Theta$.

**Upper bound the sup-loss (71).** We first consider the following decomposition. For any $b_{V,t+1} \in \mathcal{B}_V^{(t+1)}, b_{\nabla V,t+1} \in \mathcal{B}_{\nabla V}^{(t+1)}, \theta \in \Theta$, we have

$$
\Psi_{t,N}^\lambda(\widehat{b}_{\nabla V,t}, b_{V,t+1}, b_{\nabla V,t+1}, \theta, f)
$$

$$
= \Psi_{t,N}(\widehat{b}_{\nabla V,t}, b_{V,t+1}, b_{\nabla V,t+1}, \theta, f) - \lambda\|f\|_{N,2,2}^2
$$

$$
= \Psi_{t,N}(\widehat{b}_{\nabla V,t}, b_{V,t+1}, b_{\nabla V,t+1}, \theta, f) - \Psi_{t,N}(b_{\nabla V,t}^*, b_{V,t+1}, b_{\nabla V,t+1}, \theta, f)
$$

$$
\quad + \Psi_{t,N}(b_{\nabla V,t}^*, b_{V,t+1}, b_{\nabla V,t+1}, \theta, f) - \lambda\|f\|_{N,2,2}^2
$$

$$
\geq \Psi_{t,N}(\widehat{b}_{\nabla V,t}, b_{V,t+1}, b_{\nabla V,t+1}, \theta, f) - \Psi_{t,N}(b_{\nabla V,t}^*, b_{V,t+1}, b_{\nabla V,t+1}, \theta, f) - 2\lambda\|f\|_{N,2,2}^2
$$

$$
\quad + \inf_{f\in\mathcal{F}^{(t)}} \left\{ \Psi_{t,N}(b_{\nabla V,t}^*, b_{V,t+1}, b_{\nabla V,t+1}, \theta, f) + \lambda\|f\|_{N,2,2}^2 \right\}
$$

$$
= \Psi_{t,N}(\widehat{b}_{\nabla V,t}, b_{V,t+1}, b_{\nabla V,t+1}, \theta, f) - \Psi_{t,N}(b_{\nabla V,t}^*, b_{V,t+1}, b_{\nabla V,t+1}, \theta, f) - 2\lambda\|f\|_{N,2,2}^2
$$

$$
\quad - \inf_{f\in\mathcal{F}^{(t)}} \left\{ -\Psi_{t,N}(b_{\nabla V,t}^*, b_{V,t+1}, b_{\nabla V,t+1}, \theta, -f) + \lambda\|f\|_{N,2,2}^2 \right\} \text{ (by symmetry and shapes of } \Psi_{t,N})
$$

$$
= \Psi_{t,N}(\widehat{b}_{\nabla V,t}, b_{V,t+1}, b_{\nabla V,t+1}, \theta, f) - \Psi_{t,N}(b_{\nabla V,t}^*, b_{V,t+1}, b_{\nabla V,t+1}, \theta, f) - 2\lambda\|f\|_{N,2,2}^2
$$

$$
\quad + \inf_{f\in\mathcal{F}^{(t)}} \left\{ -\Psi_{t,N}(b_{\nabla V,t}^*, b_{V,t+1}, b_{\nabla V,t+1}, \theta, f) + \lambda\|f\|_{N,2,2}^2 \right\} \text{ (by symmetry of } \mathcal{F})
$$

$$
= \Psi_{t,N}(\widehat{b}_{\nabla V,t}, b_{V,t+1}, b_{\nabla V,t+1}, \theta, f) - \Psi_{t,N}(b_{\nabla V,t}^*, b_{V,t+1}, b_{\nabla V,t+1}, \theta, f) - 2\lambda\|f\|_{N,2,2}^2
$$

$$
\quad - \sup_{f\in\mathcal{F}^{(t)}} \left\{ \Psi_{t,N}(b_{\nabla V,t}^*, b_{V,t+1}, b_{\nabla V,t+1}, \theta, f) - \lambda\|f\|_{N,2,2}^2 \right\}
$$

$$
= \Psi_{t,N}(\widehat{b}_{\nabla V,t}, b_{V,t+1}, b_{\nabla V,t+1}, \theta, f) - \Psi_{t,N}(b_{\nabla V,t}^*, b_{V,t+1}, b_{\nabla V,t+1}, \theta, f) - 2\lambda\|f\|_{N,2,2}^2
$$

$$
\quad - \sup_{f\in\mathcal{F}^{(t)}} \Psi_{t,N}^\lambda(b_{\nabla V,t}^*, b_{V,t+1}, b_{\nabla V,t+1}, \theta, f).
$$

$$(72)$$

Taking $\sup_{f\in\mathcal{F}^{(t)}}$ on both sides of the inequality, we get

$$
\sup_{f\in\mathcal{F}^{(t)}} \Psi_{t,N}^\lambda(\widehat{b}_{\nabla V,t}, b_{V,t+1}, b_{\nabla V,t+1}, \theta, f)
$$

$$
\geq \sup_{f\in\mathcal{F}^{(t)}} \Psi_{t,N}(\widehat{b}_{\nabla V,t}, b_{V,t+1}, b_{\nabla V,t+1}, \theta, f) - \Psi_{t,N}(b_{\nabla V,t}^*, b_{V,t+1}, b_{\nabla V,t+1}, \theta, f) - 2\lambda\|f\|_{N,2,2}^2
$$

$$
\quad - \sup_{f\in\mathcal{F}^{(t)}} \Psi_{t,N}^\lambda(b_{\nabla V,t}^*, b_{V,t+1}, b_{\nabla V,t+1}, \theta, f).
$$

$$(73)$$

Rearranging, we have

$$
\sup_{f\in\mathcal{F}^{(t)}} \Psi_{t,N}(\widehat{b}_{\nabla V,t}, b_{V,t+1}, b_{\nabla V,t+1}, \theta, f) - \Psi_{t,N}(b_{\nabla V,t}^*, b_{V,t+1}, b_{\nabla V,t+1}, \theta, f) - 2\lambda\|f\|_{N,2,2}^2
$$

$$
\leq \sup_{f\in\mathcal{F}^{(t)}} \Psi_{t,N}^\lambda(\widehat{b}_{\nabla V,t}, b_{V,t+1}, b_{\nabla V,t+1}, \theta, f) + \sup_{f\in\mathcal{F}^{(t)}} \Psi_{t,N}^\lambda(b_{\nabla V,t}^*, b_{V,t+1}, b_{\nabla V,t+1}, \theta, f)
$$

$$
\leq 2 \sup_{f\in\mathcal{F}^{(t)}} \Psi_{t,N}^\lambda(b_{\nabla V,t}^*, b_{V,t+1}, b_{\nabla V,t+1}, \theta, f) \text{ (by definition of } \widehat{b}_{\nabla V,t}).
$$

$$(74)$$

It suffices to upper bound $\sup_{f\in\mathcal{F}^{(t)}} \Psi_{t,N}^\lambda(b_{\nabla V,t}^*, b_{V,t+1}, b_{\nabla V,t+1}, \theta, f)$ uniformly for $b_{V,t+1} \in \mathcal{B}_V^{(t+1)}, b_{\nabla V,t+1} \in \mathcal{B}_{\nabla V}^{(t+1)}, \theta \in \Theta$. To this end, we apply two uniform laws from the empirical processes theory (Wainwright, 2019; Foster & Syrgkanis, 2019).

For any $b_{V,t+1} \in \mathcal{B}_V^{(t+1)}, b_{\nabla V,t+1} \in \mathcal{B}_{\nabla V}^{(t+1)}, \theta \in \Theta$, we have

$$
\sup_{f\in\mathcal{F}^{(t)}} \Psi_{t,N}^\lambda(b_{\nabla V,t}^*, b_{V,t+1}, b_{\nabla V,t+1}, \theta, f)
$$

$$
= \sup_{f\in\mathcal{F}^{(t)}} \left\{ \Psi_{t,N}(b_{\nabla V,t}^*, b_{V,t+1}, b_{\nabla V,t+1}, \theta, f) - \lambda\|f\|_{N,2,2}^2 \right\}.
$$

$$(75)$$

To upper bound $\sup_{f \in \mathcal{F}^{(t)}} \Psi_{t,N}^{\lambda}(b_{\nabla V,t}^*, b_{V,t+1}, b_{\nabla V,t+1}, \theta, f)$, it it sufficient to upper bound $\Psi_{t,N}(b_{\nabla V,t}^*, b_{V,t+1}, b_{\nabla V,t+1}, \theta, f)$ and lower bound $\|f\|_{N,2,2}^2$ uniformly for all $f \in \mathcal{F}^{(t)}$. To this end, we apply Lemma 5 (vector-valued version of Theorem 14.1 of Wainwright (2019)) and Lemma 6 (Lemma 11 of Foster & Syrgkanis (2019)) that are able to relate the empirical version to the population version uniformly.

For the first term $\Psi_{t,N}(b_{\nabla V,t}^*, b_{V,t+1}, b_{\nabla V,t+1}, \theta, f)$, we have that it is equal to $\frac{1}{N} \sum_{n=1}^{N}$ $\left[ (m_{\nabla V}(Z_{t,n}; b_{\nabla V,t}, b_{V,t+1}, b_{\nabla V,t+1}, \theta))^T f(X_{t,n}) \right]$. We apply Lemma 6 here. We let $\mathcal{L}_f$ in Lemma 6 be set as $\mathcal{L}_g := (m_{\nabla V}(Z_t; b_{\nabla V,t}, b_{V,t+1}, b_{\nabla V,t+1}, \theta))^T f(X_t)$ where $g = \frac{1}{2(G+1)M_{\mathcal{B}}}$ $(m_{\nabla V}(Z_t; b_{\nabla V,t}, b_{V,t+1}, b_{\nabla V,t+1}, \theta))^T f(X_t)$. By the conditional moment equation for the gradient bridge function equation 4, we have $|g|_{\infty} \leq \frac{1}{2(G+1)M_{\mathcal{B}}} \|m_{\nabla V}\|_{\infty,2} \|f\|_{\infty,2} \leq \frac{1}{2(G+1)M_{\mathcal{B}}} \|m_{\nabla V}\|_{\infty,2} \leq 1$.

In particular, we have

$$
\begin{aligned}
|g|_{\infty} \leq & \frac{1}{2(G+1)M_{\mathcal{B}}} \|m_{\nabla V}\|_{\infty,2} \|f\|_{\infty,2} \\
\leq & \frac{1}{2(G+1)M_{\mathcal{B}}} \|m_{\nabla V}\|_{\infty,2} \text{ by Assumption 4(d)} \\
= & \frac{1}{2(G+1)M_{\mathcal{B}}} \| b_{\nabla V,t} - \left( R_t + \sum_{a' \in \mathcal{A}} b_{V,t+1}(a', O_{t+1}, H_t) \right) \nabla_{\theta} \pi_{\theta_t} \\
& - \sum_{a' \in \mathcal{A}} b_{\nabla V,t+1}(a', O_{t+1}, H_t) \pi_{\theta_t} \|_{\infty,2} \\
= & \frac{1}{2(G+1)M_{\mathcal{B}}} \| b_{\nabla V,t} - \left( R_t + \sum_{a' \in \mathcal{A}} b_{V,t+1}(a', O_{t+1}, H_t) \right) \pi_{\theta_t} \nabla_{\theta} \log \pi_{\theta_t} \\
& - \sum_{a' \in \mathcal{A}} b_{\nabla V,t+1}(a', O_{t+1}, H_t) \pi_{\theta_t} \|_{\infty,2} \\
\leq & \frac{1}{2(G+1)M_{\mathcal{B}}} \| b_{\nabla V,t} \|_{\infty,2} \\
& + \frac{1}{2(G+1)M_{\mathcal{B}}} \| \left( R_t + \sum_{a' \in \mathcal{A}} b_{V,t+1}(a', O_{t+1}, H_t) \right) \pi_{\theta_t} \nabla_{\theta} \log \pi_{\theta_t} \|_{\infty,2} \\
& + \frac{1}{2(G+1)M_{\mathcal{B}}} \| \sum_{a' \in \mathcal{A}} b_{\nabla V,t+1}(a', O_{t+1}, H_t) \pi_{\theta_t} \|_{\infty,2} \\
\leq & \frac{1}{2(G+1)M_{\mathcal{B}}} \| b_{\nabla V,t} \|_{\infty,2} \\
& + \frac{1}{2(G+1)M_{\mathcal{B}}} G(1 + \| \left( \sum_{a' \in \mathcal{A}} b_{V,t+1}(a', O_{t+1}, H_t) \right) \|_{\infty}) \\
& + \frac{1}{2(G+1)M_{\mathcal{B}}} \| \sum_{a' \in \mathcal{A}} b_{\nabla V,t+1}(a', O_{t+1}, H_t) \|_{\infty,2} \\
\leq & \frac{1}{2(G+1)M_{\mathcal{B}}} M_{\mathcal{B}} + \frac{1}{2(G+1)M_{\mathcal{B}}} G M_{\mathcal{B}} + \frac{1}{2(G+1)M_{\mathcal{B}}} M_{\mathcal{B}} \\
\leq & 1. \quad (76)
\end{aligned}
$$

Also, $\mathcal{L}_g$ is trivially Lipschitz continuous w.r.t. g with constant $2(G+1)M_{\mathcal{B}}$. Let $g^*$ be 0, and Lemma 6 shows that with probability at least $1 - \zeta$, it holds uniformly for all $f \in \mathcal{F}^{(t)}$, $b_{V,t+1} \in$

$\mathcal{B}_V^{(t+1)}, b_{\nabla V, t+1} \in \mathcal{B}_{\nabla V}^{(t+1)}$ that

$$
\begin{aligned}
&|\Psi_{t,N}(b_{\nabla V,t}^*, b_{V,t+1}, b_{\nabla V,t+1}, \theta, f) - \Psi_t(b_{\nabla V,t}^*, b_{V,t+1}, b_{\nabla V,t+1}, \theta, f)| \\
&\leq 18(2(G+1)M_{\mathcal{B}})\delta_{\Omega_{t,\theta}}(\|g\|_2 + \delta_{\Omega_{t,\theta}}) \\
&\leq 18(2(G+1)M_{\mathcal{B}})\delta_{\Omega_{t,\theta}}(\|f\|_{2,2} + \delta_{\Omega_{t,\theta}}).
\end{aligned}
\tag{77}
$$

for a fixed $\theta$, where $\delta_{\Omega_{t,\theta}} = \delta_{N,\Omega_{t,\theta}} + c_0\sqrt{\frac{\log(c_1/\zeta)}{N}}$, and $\delta_{N,\Omega}$ is the upper bound on the critical radius of the function class

$$
\begin{aligned}
\Omega_{t,\theta} :=\{&\frac{r}{2(G+1)M_{\mathcal{B}}} m_{\nabla V}(z_t; b_{\nabla V,t}, b_{\nabla V,t+1}, b_{V,t+1}, \theta)^T f(x_t) : \mathcal{Z}_t \times \mathcal{X}_t \to \mathbb{R} \\
&\text{for all } r \in [-1,1], b_{\nabla V,t} \in \mathcal{B}_{\nabla V}^{(t)}, b_{V,t+1} \in \mathcal{B}_V^{(t+1)}, b_{\nabla V,t+1} \in \mathcal{B}_{\nabla V}^{(t+1)}\}.
\end{aligned}
\tag{78}
$$

To tackle the uniformness with respect to $\theta$, we consider an $\epsilon$-net of $\Theta_t$ in the Euclidean space. In particular, according to Example 5.8 of Wainwright (2019), if $\Theta_t$ is bounded with radius $R$ for all $t$, then the covering number of $\Theta_t$ in the $\ell^2$ norm is not greater than $(1 + \frac{2R}{\epsilon})^{d_{\Theta_t}}$. We let $N(\epsilon, \Theta_t, \ell^2)$ denote the corresponding covering number. In addition, the uniformness with respect to $t$ should also be considered.

Below, we apply a standard union bound with the Lipschitz property of $\Psi_{t,N}$ to derive a uniform law concerning $\theta$ and $t$.

We first consider the Lipschitz property for the population version $\Psi_t$:

$$
\begin{aligned}
&|\Psi_t(b_{\nabla V,t}^*, b_{V,t+1}, b_{\nabla V,t+1}, \theta, f) - \Psi_t(b_{\nabla V,t}^*, b_{V,t+1}, b_{\nabla V,t+1}, \theta', f)| \\
&=|(b_{\nabla V,t} - (R_t + b_{V,t+1})\nabla_\theta \pi_{\theta_t} - b_{\nabla V,t+1}\pi_{\theta_t})^\top f - (b_{\nabla V,t} - (R_t + b_{V,t+1})\nabla_{\theta'}\pi_{\theta_t'} - b_{\nabla V,t+1}\pi_{\theta_t'})^\top f| \\
&=|(R_t + b_{V,t+1})(\nabla_\theta \pi_{\theta_t} - \nabla_{\theta'}\pi_{\theta_t'})^\top f + (b_{\nabla V,t+1}\pi_{\theta_t} - b_{\nabla V,t+1}\pi_{\theta_t'})^\top f| \\
&\leq(\|(R_t + b_{V,t+1})(\nabla_\theta \pi_{\theta_t} - \nabla_{\theta'}\pi_{\theta_t'})\|_{\ell^2} + \|(b_{\nabla V,t+1}\pi_{\theta_t} - b_{\nabla V,t+1}\pi_{\theta_t'})\|_{\ell^2})\|f\|_{\ell^2} \\
&\leq M_{\mathcal{B}}\|\nabla_\theta \pi_{\theta_t} - \nabla_{\theta'}\pi_{\theta_t'}\|_{\ell^2} + M_{\mathcal{B}}|\pi_{\theta_t} - \pi_{\theta_t'}| \\
&= M_{\mathcal{B}}\|\pi_{\theta_t}\nabla_\theta \log \pi_{\theta_t} - \pi_{\theta_t'}\nabla_{\theta'}\log \pi_{\theta_t'}\|_{\ell^2} + M_{\mathcal{B}}|\pi_{\theta_t} - \pi_{\theta_t'}| \\
&= M_{\mathcal{B}}\|\pi_{\theta_t}\nabla_\theta \log \pi_{\theta_t} - \pi_{\theta_t}\nabla_{\theta'}\log \pi_{\theta_t'} + \pi_{\theta_t}\nabla_{\theta'}\log \pi_{\theta_t'} - \pi_{\theta_t'}\nabla_{\theta'}\log \pi_{\theta_t'}\|_{\ell^2} + M_{\mathcal{B}}|\pi_{\theta_t} - \pi_{\theta_t'}| \\
&\leq M_{\mathcal{B}}(\|\nabla_\theta \log \pi_{\theta_t} - \nabla_{\theta'}\log \pi_{\theta_t'}\|_{\ell^2} + \|\pi_{\theta_t}\nabla_{\theta'}\log \pi_{\theta_t'} - \pi_{\theta_t'}\nabla_{\theta'}\log \pi_{\theta_t'}\|_{\ell^2}) + M_{\mathcal{B}}|\pi_{\theta_t} - \pi_{\theta_t'}| \\
&\leq M_{\mathcal{B}}(\|\nabla_\theta \log \pi_{\theta_t} - \nabla_{\theta'}\log \pi_{\theta_t'}\|_{\ell^2} + |\pi_{\theta_t} - \pi_{\theta_t'}|\|\nabla_{\theta'}\log \pi_{\theta_t'}\|_{\ell^2}) + M_{\mathcal{B}}|\pi_{\theta_t} - \pi_{\theta_t'}|
\end{aligned}
\tag{79}
$$

By Assumption 5(b), we have

$$
\|\nabla_\theta \log \pi_{\theta_t} - \nabla_{\theta'}\log \pi_{\theta_t'}\|_{\ell^2} \leq \beta\|\theta_t - \theta_t'\|_{\ell^2}.
\tag{80}
$$

By Assumption 8, we have

$$
|\pi_{\theta_t} - \pi_{\theta_t'}| \leq \tilde{\beta}\|\theta_t - \theta_t'\|_{\ell^2}.
\tag{81}
$$

for some constant $\tilde{\beta}$.

By Assumption 4(e), we have

$$
\|\nabla_{\theta'}\log \pi_{\theta_t'}\|_{\ell^2} \leq \sqrt{d_{\Theta_t}}\|\nabla_{\theta'}\log \pi_{\theta_t'}\|_{\ell^\infty} \leq \sqrt{d_{\Theta_t}}G.
\tag{82}
$$

Therefore

$$
\begin{aligned}
&|\Psi_t(b_{\nabla V,t}^*, b_{V,t+1}, b_{\nabla V,t+1}, \theta, f) - \Psi_t(b_{\nabla V,t}^*, b_{V,t+1}, b_{\nabla V,t+1}, \theta', f)| \\
&\leq M_{\mathcal{B}}(\|\nabla_\theta \log \pi_{\theta_t} - \nabla_{\theta'}\log \pi_{\theta_t'}\|_{\ell^2} + |\pi_{\theta_t} - \pi_{\theta_t'}|\|\nabla_{\theta'}\log \pi_{\theta_t'}\|_{\ell^2}) + M_{\mathcal{B}}|\pi_{\theta_t} - \pi_{\theta_t'}| \\
&\leq M_{\mathcal{B}}\beta\|\theta_t - \theta_t'\|_{\ell^2} + M_{\mathcal{B}}\tilde{\beta}\sqrt{d_{\Theta_t}}G\|\theta_t - \theta_t'\|_{\ell^2} + M_{\mathcal{B}}\tilde{\beta}\|\theta_t - \theta_t'\|_{\ell^2} \\
&\leq c_{\beta,\tilde{\beta},G}M_{\mathcal{B}}\sqrt{d_{\Theta_t}}\|\theta_t - \theta_t'\|_{\ell^2}.
\end{aligned}
\tag{83}
$$

By a similar argument, we can also show that

$$
\begin{aligned}
&|\Psi_{t,N}(b^*_{\nabla V,t}, b_{V,t+1}, b_{\nabla V,t+1}, \theta, f) - \Psi_{t,N}(b^*_{\nabla V,t}, b_{V,t+1}, b_{\nabla V,t+1}, \theta', f)| \\
&\leq c_{\beta,\tilde{\beta},G} M_{\mathcal{B}} \sqrt{d_{\Theta_t}} \|\theta_t - \theta'_t\|_{\ell^2}.
\end{aligned}
\tag{84}
$$

Next, we apply a union bound given a $\frac{\epsilon}{\sqrt{d_{\Theta_t}}}$-net of $\Theta_t$. For each $\theta_t$, we are able to find $\theta'_t$ in this $\frac{\epsilon}{\sqrt{d_{\Theta_t}}}$-net such that the following holds uniformly w.r.t. $t = 1, ..., T, \theta_t \in \Theta_t, f, b_{V,t+1}, b_{\nabla V,t+1}$.

$$
\begin{aligned}
&|\Psi_{t,N}(b^*_{\nabla V,t}, b_{V,t+1}, b_{\nabla V,t+1}, \theta, f) - \Psi_t(b^*_{\nabla V,t}, b_{V,t+1}, b_{\nabla V,t+1}, \theta, f)| \\
&\leq |\Psi_{t,N}(b^*_{\nabla V,t}, b_{V,t+1}, b_{\nabla V,t+1}, \theta, f) - \Psi_{t,N}(b^*_{\nabla V,t}, b_{V,t+1}, b_{\nabla V,t+1}, \theta', f)| \\
&\quad + |\Psi_{t,N}(b^*_{\nabla V,t}, b_{V,t+1}, b_{\nabla V,t+1}, \theta', f) - \Psi_t(b^*_{\nabla V,t}, b_{V,t+1}, b_{\nabla V,t+1}, \theta', f)| \\
&\quad + |\Psi_t(b^*_{\nabla V,t}, b_{V,t+1}, b_{\nabla V,t+1}, \theta', f) - \Psi_t(b^*_{\nabla V,t}, b_{V,t+1}, b_{\nabla V,t+1}, \theta, f)| \\
&\leq 36(G+1)M_{\mathcal{B}}(\sup_{t,\theta} \delta_{N,\Omega_{t,\theta}} + c_0 \sqrt{\frac{\log(c_1 TN(\frac{\epsilon}{\sqrt{d_{\Theta_t}}}, \Theta_t, \ell^2)/\zeta)}{N}})(\|f\|_{2,2} \\
&\quad + (\sup_{t,\theta} \delta_{N,\Omega_{t,\theta}} + c_0 \sqrt{\frac{\log(c_1 TN(\frac{\epsilon}{\sqrt{d_{\Theta_t}}}, \Theta_t, \ell^2)/\zeta)}{N}})) + 2c_{\beta,\tilde{\beta},G} M_{\mathcal{B}}\epsilon
\end{aligned}
\tag{85}
$$

with probability at least $1 - \zeta$.

For simplicity, we let $\delta_\Omega$ denote $\sup_{t,\theta} \delta_{N,\Omega_{t,\theta}} + c_0 \sqrt{\frac{\log(c_1 TN(\frac{\epsilon}{\sqrt{d_{\Theta_t}}}, \Theta_t, \ell^2)/\zeta)}{N}}$. Then we have

$$
\begin{aligned}
&|\Psi_{t,N}(b^*_{\nabla V,t}, b_{V,t+1}, b_{\nabla V,t+1}, \theta, f) - \Psi_t(b^*_{\nabla V,t}, b_{V,t+1}, b_{\nabla V,t+1}, \theta, f)| \\
&\leq 36(G+1)M_{\mathcal{B}}\delta_\Omega(\|f\|_{2,2} + \delta_\Omega) + 2c_{\beta,\tilde{\beta},G} M_{\mathcal{B}}\epsilon
\end{aligned}
\tag{86}
$$

uniformly for all parameters with probability at least $1 - \zeta$.

Next we consider the term $\|f\|^2_{N,2,2}$. By applying Lemma 5 directly, and let $\delta_{\mathcal{F}} = \delta_{\mathcal{N},\mathcal{F}} + c_0 \sqrt{\frac{\log(c_1\zeta)}{N}}$ where $\delta_{\mathcal{N},\mathcal{F}}$ denotes the critical radius of $\mathcal{F}^{(t)}$ for all $t$. With probability at least $1 - \zeta$, it holds uniformly for all $f \in \mathcal{F}^{(t)}$ that

$$
|\|f\|^2_{N,2,2} - \|f\|^2_{2,2}| \leq \frac{1}{2}\|f\|^2_{2,2} + \frac{1}{2}d_\Theta \delta^2_{\mathcal{F}}.
\tag{87}
$$

Given (86)(87), with probability at least $1 - 2\zeta$, it holds uniformly for all $f \in \mathcal{F}^{(t)}, b_{V,t+1} \in \mathcal{B}_V^{(t+1)}, b_{\nabla V,t+1} \in \mathcal{B}_{\nabla V}^{(t+1)}, \theta \in \Theta$ that

$$
\begin{aligned}
&\sup_{f \in \mathcal{F}^{(t)}} \Psi^\lambda_{t,N}(b^*_{\nabla V,t}, b_{V,t+1}, b_{\nabla V,t+1}, \theta, f) \\
&= \sup_{f \in \mathcal{F}^{(t)}} \{\Psi_{t,N}(b^*_{\nabla V,t}, b_{V,t+1}, b_{\nabla V,t+1}, \theta, f) - \lambda\|f\|^2_{N,2,2}\} \\
&\leq \sup_{f \in \mathcal{F}^{(t)}} \{\Psi_t(b^*_{\nabla V,t}, b_{V,t+1}, b_{\nabla V,t+1}, \theta, f) + 18(2(G+1)M_{\mathcal{B}})\delta_\Omega(\|f\|_{2,2} + \delta_\Omega) \\
&\quad - \lambda\|f\|^2_{N,2,2}\} + 2c_{\beta,\tilde{\beta},G} M_{\mathcal{B}}\epsilon \text{ by (86)} \\
&\leq \sup_{f \in \mathcal{F}^{(t)}} \{\Psi_t(b^*_{\nabla V,t}, b_{V,t+1}, b_{\nabla V,t+1}, \theta, f) + 18(2(G+1)M_{\mathcal{B}})\delta_{\Omega_{t,\theta}}(\|f\|_{2,2} + \delta_\Omega) \\
&\quad - \lambda\|f\|^2_{2,2} + \frac{1}{2}\lambda\|f\|^2_{2,2} + \lambda\frac{1}{2}d_\Theta \delta^2_{\mathcal{F}}\} + 2c_{\beta,\tilde{\beta},G} M_{\mathcal{B}}\epsilon \text{ by (87)} \\
&\leq \sup_{f \in \mathcal{F}^{(t)}} \{\Psi_t(b^*_{\nabla V,t}, b_{V,t+1}, b_{\nabla V,t+1}, \theta, f)\} + 2c_{\beta,\tilde{\beta},G} M_{\mathcal{B}}\epsilon \\
&\quad + \sup_{f \in \mathcal{F}^{(t)}} \{18(2(G+1)M_{\mathcal{B}})\delta_\Omega(\|f\|_{2,2} + \delta_\Omega) - \lambda\|f\|^2_{2,2} + \frac{1}{2}\lambda\|f\|^2_{2,2} + \lambda\frac{1}{2}d_\Theta \delta^2_{\mathcal{F}}\}
\end{aligned}
$$

$$= \sup_{f \in \mathcal{F}^{(t)}} \{18(2(G+1)M_{\mathcal{B}})\delta_{\Omega}(\|f\|_{2,2} + \delta_{\Omega}) - \lambda\|f\|_{2,2}^2 + \frac{1}{2}\lambda\|f\|_{2,2}^2 + \lambda\frac{1}{2}d_{\Theta}\delta_{\mathcal{F}}^2\} + 2c_{\beta,\tilde{\beta},G}M_{\mathcal{B}}\epsilon$$

$$\leq \frac{1}{2\lambda}18^2(2(G+1)M_{\mathcal{B}})^2\delta_{\Omega}^2 + 18(2(G+1)M_{\mathcal{B}})\delta_{\Omega}^2 + \frac{\lambda}{2}d_{\Theta}\delta_{\mathcal{F}}^2 + 2c_{\beta,\tilde{\beta},G}M_{\mathcal{B}}\epsilon \tag{88}$$

where the second equality is due to $\Psi_t(b^*_{\nabla V,t}, b_{V,t+1}, b_{\nabla V,t+1}, \theta, f) = 0$ by Assumption 4(b), and the final inequality is due to the fact $\sup_{\|f\|_{2,2}} \{a\|f\|_{2,2} - b\|f\|_{2,2}^2\} \leq a^2/4b$.

**Lower bound the sup loss equation 71** To begin with, we define a function $f_t = \mathbb{T}_{X_t}[\widehat{b}_{\nabla V,t} - b^*_{\nabla V,t}(b_{\nabla V,t+1}, b_{V,t+1}, \theta)] = \mathbb{E}^{\pi^b}[\widehat{b}_{\nabla V,t}(W_t) - b^*_{\nabla V,t}(b_{\nabla V,t+1}, b_{V,t+1}, \theta)(W_t) \mid X_t]$ where $W_t := (A_t, O_t, H_{t-1})$. Then $f_t \in \mathcal{F}^{(t)}$ by Assumption 4(c).

We need another localized uniform law for lower bounding the empirical sup-loss. We first define a function class

$$\Xi_t = \left\{ (w_t, x_t) \mapsto r\frac{1}{2M_{\mathcal{B}}} \left[ b_{\nabla V,t} - b^*_{\nabla V,t}(b_{\nabla V,t+1}, b_{V,t+1}, \theta) \right](w_t)^T f(x_t); \right.$$
$$\left. b_{\nabla V,t} \in \mathcal{B}^{(t)}_{\nabla V}, f \in \mathcal{F}^{(t)}, b_{V,t+1} \in \mathcal{B}^{(t+1)}_V, b_{\nabla V,t+1} \in \mathcal{B}^{(t+1)}_{\nabla V}, r \in [-1, 1], \theta \in \Theta \right\}. \tag{89}$$

We then notice that $\Psi_{t,N}(\widehat{b}_{\nabla V,t}, b_{V,t+1}, b_{\nabla V,t+1}, \theta, f) - \Psi_{t,N}(b^*_{\nabla V,t}, b_{V,t+1}, b_{\nabla V,t+1}, \theta, f)$ can be written as

$$\frac{1}{N}\sum_{n=1}^{N}(\widehat{b}_{\nabla V,t}(W_{t,n}) - b^*_{\nabla V,t}(W_{t,n}))^T f(X_{t,n}). \tag{90}$$

By Lemma 6, we have a uniform law for

$$\Psi_{t,N}(b_{\nabla V,t}, b_{V,t+1}, b_{\nabla V,t+1}, \theta, f) - \Psi_{t,N}(b^*_{\nabla V,t}, b_{V,t+1}, b_{\nabla V,t+1}, \theta, f)$$
$$= \frac{1}{N}\sum_{n=1}^{N}(b_{\nabla V,t}(W_{t,n}) - b^*_{\nabla V,t}(W_{t,n}))^T f(X_{t,n}) \tag{91}$$

over the function space $\Xi_{t,\theta}$.

Specifically, we let $\mathcal{L}_f$ in Lemma 6 be set as $\mathcal{L}_g := (b_{\nabla V,t}(W_t) - b^*_{\nabla V,t}(W_t))^T f(X_t)$ where $g = \frac{1}{2M_{\mathcal{B}}}(b_{\nabla V,t}(W_t) - b^*_{\nabla V,t}(W_t))^T f(X_t)$. We have $|g|_\infty \leq \frac{1}{2M_{\mathcal{B}}}\|b_{\nabla V,t}(w_t) - b^*_{\nabla V,t}(w_t)\|_{\infty,2}\|f\|_{\infty,2} \leq \frac{1}{2M_{\mathcal{B}}}(\|b_{\nabla V,t}\|_{\infty,2} + \|b^*_{\nabla V,t}\|_{\infty,2}) \leq 1$. Also, $\mathcal{L}_g$ is trivially Lipschitz continuous w.r.t. g with constant $2M_{\mathcal{B}}$. Let $g^*$ be 0, and Lemma 6 shows that with probability at least $1 - \zeta$, it holds uniformly for all $t = 1, ..., T$, $f \in \mathcal{F}^{(t)}, b_{V,t+1} \in \mathcal{B}^{(t+1)}_V, b_{\nabla V,t+1} \in \mathcal{B}^{(t+1)}_{\nabla V}, \theta \in \Theta$ that

$$|\left(\Psi_{t,N}(b_{\nabla V,t}, b_{V,t+1}, b_{\nabla V,t+1}, \theta, f) - \Psi_{t,N}(b^*_{\nabla V,t}, b_{V,t+1}, b_{\nabla V,t+1}, \theta, f)\right)$$
$$- \left(\Psi_t(b_{\nabla V,t}, b_{V,t+1}, b_{\nabla V,t+1}, \theta, f) - \Psi_t(b^*_{\nabla V,t}, b_{V,t+1}, b_{\nabla V,t+1}, \theta, f)\right)|$$
$$\leq 36M_{\mathcal{B}}\delta_{\Xi}(\|g\|_2 + \delta_{\Xi}) \tag{92}$$
$$\leq 36M_{\mathcal{B}}\delta_{\Xi}(\|f\|_{2,2} + \delta_{\Xi}).$$

where $\delta_{\Xi} = \delta_{N,\Xi} + c_0\sqrt{\frac{\log(c_1 T/\zeta)}{N}}$, and $\delta_{N,\Xi}$ is the upper bound on the critical radius of the function class $\Xi$.

Now we are ready to lower bound the sup-loss (71). Since $f_t = \mathbb{E}^{\pi^b}[\widehat{b}_{\nabla V,t}(W_t) - b^*_{\nabla V,t}(b_{\nabla V,t+1}, b_{V,t+1}, \theta)(W_t) \mid X_t]$ is assumed to be in $\mathcal{F}^{(t)}$ by Assumption 4(C), we also have $\frac{1}{2}f_t \in \mathcal{F}^{(t)}$ by star-shaped of $\mathcal{F}^{(t)}$. Therefore, we have

$$\sup_{f \in \mathcal{F}^{(t)}} \Psi_{t,N}(\widehat{b}_{\nabla V,t}, b_{V,t+1}, b_{\nabla V,t+1}, \theta, f) - \Psi_{t,N}(b^*_{\nabla V,t}, b_{V,t+1}, b_{\nabla V,t+1}, \theta, f) - 2\lambda\|f\|_{N,2,2}^2$$

$$\geq \Psi_{t,N}(\widehat{b}_{\nabla V,t}, b_{V,t+1}, b_{\nabla V,t+1}, \theta, \frac{1}{2}f_t) - \Psi_{t,N}(b^*_{\nabla V,t}, b_{V,t+1}, b_{\nabla V,t+1}, \theta, \frac{1}{2}f_t) - 2\lambda\|\frac{1}{2}f_t\|^2_{N,2,2}$$

$$\geq \Psi_t(\widehat{b}_{\nabla V,t}, b_{V,t+1}, b_{\nabla V,t+1}, \theta, \frac{1}{2}f_t) - \Psi_t(b^*_{\nabla V,t}, b_{V,t+1}, b_{\nabla V,t+1}, \theta, \frac{1}{2}f_t)$$
$$- 36M_\mathcal{B}\delta_\Xi(\|\frac{1}{2}f_t\|_{2,2} + \delta_\Xi) - \frac{\lambda}{2}\|f_t\|^2_{N,2,2} \text{ by (92)}$$

$$\geq \Psi_t(\widehat{b}_{\nabla V,t}, b_{V,t+1}, b_{\nabla V,t+1}, \theta, \frac{1}{2}f_t) - \Psi_t(b^*_{\nabla V,t}, b_{V,t+1}, b_{\nabla V,t+1}, \theta, \frac{1}{2}f_t)$$
$$- 18M_\mathcal{B}\delta_\Xi\|f_t\|_{2,2} - 36M_\mathcal{B}\delta^2_\Xi - \frac{\lambda}{2}(\|f_t\|^2_{2,2} + \frac{1}{2}\|f_t\|^2_{2,2} + \frac{1}{2}d_\Theta\delta^2_\mathcal{F}) \text{ by (87)}$$

$$= \frac{1}{2}\|f_t\|^2_{2,2} - 18M_\mathcal{B}\delta_\Xi\|f_t\|_{2,2} - 36M_\mathcal{B}\delta^2_\Xi - \frac{\lambda}{2}(\|f_t\|^2_{2,2} + \frac{1}{2}\|f_t\|^2_{2,2} + \frac{1}{2}d_\Theta\delta^2_\mathcal{F})$$

$$= (\frac{1}{2} - \frac{3\lambda}{4})\|f_t\|^2_{2,2} - 18M_\mathcal{B}\delta_\Xi\|f_t\|_{2,2} - 36M_\mathcal{B}\delta^2_\Xi - \frac{\lambda}{4}d_\Theta\delta^2_\mathcal{F} \tag{93}$$

where the first equality is due to

$$\Psi_t(\widehat{b}_{\nabla V,t}, b_{V,t+1}, b_{\nabla V,t+1}, \theta, \frac{1}{2}f_t) - \Psi_t(b^*_{\nabla V,t}, b_{V,t+1}, b_{\nabla V,t+1}, \theta, \frac{1}{2}f_t)$$
$$= \mathbb{E}^{\pi^b}\left[\frac{1}{2}(\widehat{b}_{\nabla V,t}(W_t) - b^*_{\nabla V,t}(W_t))^T f_t(X_t)\right]$$
$$= \mathbb{E}^{\pi^b}\left[\frac{1}{2}\mathbb{E}^{\pi^b}[(\widehat{b}_{\nabla V,t}(W_t) - b^*_{\nabla V,t}(W_t))^T \mid X_t]f_t(X_t)\right] \tag{94}$$
$$= \frac{1}{2}\mathbb{E}^{\pi^b}\left[\|f_t\|^2_{\ell 2}\right]$$
$$= \frac{1}{2}\|f_t\|^2_{2,2}$$

**Combining upper bound and lower bound** By combining (74)(88)(93), we have with probability at least $1 - 3\zeta$ (by recalling that we applied three uniform laws), it holds uniformly for all $f \in \mathcal{F}^{(t)}$, $b_{V,t+1} \in \mathcal{B}_V^{(t+1)}, b_{\nabla V,t+1} \in \mathcal{B}_{\nabla V}^{(t+1)}, \theta \in \Theta$ that

$$(\frac{1}{2} - \frac{3\lambda}{4})\|f_t\|^2_{2,2} - 18M_\mathcal{B}\delta_\Xi\|f_t\|_{2,2} - 36M_\mathcal{B}\delta^2_\Xi - \frac{\lambda}{4}d_\Theta\delta^2_\mathcal{F}$$
$$\leq \text{sup-loss (71) by (93)} \tag{95}$$
$$\leq 2 \sup_{f \in \mathcal{F}^{(t)}} \Psi^\lambda_{t,N}(b^*_{\nabla V,t}, b_{V,t+1}, b_{\nabla V,t+1}, \theta, f) \text{ by (74)}$$
$$\leq \frac{1}{\lambda}18^2(2(G+1)M_\mathcal{B})^2\delta^2_\Omega + 36(2(G+1)M_\mathcal{B})\delta^2_\Omega + \lambda d_\Theta\delta^2_\mathcal{F} + 2c_{\beta,\tilde{\beta},G}M_\mathcal{B}\epsilon \text{ by (88)}$$

By rearranging, we have
$$a\|f_t\|^2_{2,2} - b\|f_t\|_{2,2} - c \leq 0 \tag{96}$$
where
$$a = \frac{1}{2} - \frac{3\lambda}{4}, \tag{97}$$
$$b = 18M_\mathcal{B}\delta_\Xi, \tag{98}$$
and
$$c = 36M_\mathcal{B}\delta^2_\Xi + \frac{5\lambda}{4}d_\Theta\delta^2_\mathcal{F}$$
$$+ \left\{\frac{1}{\lambda}18^2(2(G+1)M_\mathcal{B})^2 + 36(2(G+1)M_\mathcal{B})\right\}\delta^2_\Omega + 2c_{\beta,\tilde{\beta},G}M_\mathcal{B}\epsilon. \tag{99}$$

By solving the quadratic inequality (96), and setting $\lambda < \frac{2}{3}$ for guaranteeing $a > 0$, we have

$$\|f_t\|_{2,2} \le \frac{b}{2a} + \sqrt{\frac{b^2}{4a^2} + \frac{c}{a}} \le \frac{b}{2a} + \frac{b}{2a} + \sqrt{\frac{c}{a}} = \frac{b}{a} + \sqrt{\frac{c}{a}}. \tag{100}$$

Since $\|f_t\|_{2,2}$ is exactly the projected RMSE by definition, we have with probability at least $1 - 3\zeta$ that

$$\sup_t \sup_{\theta_t \in \Theta_t} \sup_{b_{V,t+1} \in \mathcal{B}_V^{(t+1)}, b_{\nabla V,t+1} \in \mathcal{B}_{\nabla V}^{(t+1)}} \left\| \mathbb{T}_{X_t} \left[ b^*_{\nabla V,t}(b_{V,t+1}, b_{\nabla V,t+1}, \theta) - \widehat{b}_{\nabla V,t} \right] \right\|_{\mathcal{L}^2(\ell^2, \mathbb{P}^{\pi^b})}$$

$$\le \frac{b}{a} + \sqrt{\frac{c}{a}}$$

$$= \frac{1}{\frac{1}{2} - \frac{3\lambda}{4}} 18 M_\mathcal{B} \delta_\Xi + \sqrt{\frac{1}{\frac{1}{2} - \frac{3\lambda}{4}} \left( 36 M_\mathcal{B} \delta_\Xi^2 + \frac{5\lambda}{4} d_\Theta \delta_\mathcal{F}^2 \right.}$$

$$\left. + \left\{ \frac{1}{\lambda} 18^2 (2(G+1) M_\mathcal{B})^2 + 36(2(G+1) M_\mathcal{B}) \right\} \delta_\Omega^2 + 2 c_{\beta, \tilde{\beta}, G} M_\mathcal{B} \epsilon \right)^{\frac{1}{2}}$$

$$\lesssim O \left( M_\mathcal{B} \delta_\Xi + \sqrt{d_\Theta} \delta_\mathcal{F} + (2(G+1) M_\mathcal{B}) \delta_\Omega + \sqrt{M_\mathcal{B} \epsilon} \right). \tag{101}$$

where $\delta_\Omega$ denotes $\sup_{t,\theta} \delta_{N,\Omega_{t,\theta}} + c_0 \sqrt{\frac{\log(c_1 T N(\frac{\epsilon}{\sqrt{d_{\Theta_t}}}, \Theta_t, \ell^2)/\zeta)}{N}}$, and $\delta_\Xi = \delta_{N,\Xi} + c_0 \sqrt{\frac{\log(c_1 T/\zeta)}{N}}$.

By setting $\epsilon = \frac{1}{n^2}$, and the fact that $\log N(\frac{\epsilon}{\sqrt{d_{\Theta_t}}}, \Theta_t, \ell^2) \le d_{\Theta_t} \log(1 + \frac{2}{\frac{1}{n^2}/\sqrt{d_{\Theta_t}}})$, we have

$$\delta_\Omega \lesssim \sup_{t,\theta} \delta_{N,\Omega_{t,\theta}} + c_0 \sqrt{\frac{\log(T) + d_{\Theta_t}^2 + \log(1 + \frac{c_1 N}{\zeta})}{N}}. \tag{102}$$

Furthermore, $\sqrt{\epsilon} \sim \frac{1}{N}$ is a high-order term and therefore can be dropped.

**Part II.** In this part, we aim to upper bound the estimation error

$$\left\| \mathbb{T}_{X_j} \left[ b^*_{V,j}(\widehat{b}_{V,j+1}^{\pi_\theta}, \theta) - \widehat{b}_{V,j}^{\pi_\theta} \right] \right\|_{\mathcal{L}^2(\mathbb{P}^{\pi^b})}$$

caused by the value bridge functions.

The idea is similar to the previous section. By upper bounding the sup-loss, we have a convergence rate involving the critical radius of function classes. By lower bounding the sup-loss, we have the projected RMSE as the lower bound. By combining the upper bound and lower bound of the sup-loss, we finally provide an upper bound for the projected RMSE. Three applications of uniform laws are also involved. We omit the proof here as it can be viewed as a special case of Part I.

We state the result here. Let $M_V$ denote the upper bound for $\mathcal{B}_V^{(t)}$. With probability at least $1 - 3\zeta$, the projected RMSE is upper bounded by

$$\sup_{\theta \in \Theta} \sup_{b_{V,t+1} \in \mathcal{B}_V^{(t+1)}} \left\| \mathbb{T}_{X_j} \left[ b^*_{V,j}(b_{V,j+1}^{\pi_\theta}, \theta) - \widehat{b}_{V,j}^{\pi_\theta} \right] \right\|_{\mathcal{L}^2(\mathbb{P}^{\pi^b})}$$

$$\lesssim O \left( M_V \delta_\aleph + \delta_{\mathcal{F}|_{d_\Theta + 1}} + M_V \delta_\Upsilon \right). \tag{103}$$

where

$$\aleph = \left\{ (w_t, x_t) \mapsto r \frac{1}{2M_V} \left[ b_{V,t} - b^*_{V,t}(b_{V,t+1}, \theta) \right] (w_t) f(x_t); \right.$$

$$\left. b_{V,t} \in \mathcal{B}_V^{(t)}, f \in \mathcal{F}^{(t)}|_{d_\Theta + 1}, b_{V,t+1} \in \mathcal{B}_V^{(t+1)}, \theta \in \Theta, r \in [-1, 1] \right\} \tag{104}$$

and

$$
\begin{aligned}
\Upsilon = \{(z_t, x_t) \mapsto \frac{r}{2M_V}[b_{V,t}(a_t, o_t, h_{t-1}) - R_t \pi_{\theta_t}(a_t \mid o_t, h_{t-1}) \\
- \sum_a b_{V,t+1}(a, o_{t+1}, h_t)\pi_{\theta_t}(a_t \mid o_t, h_{t-1})]f(x_t); \\
\text{for all } r \in [-1, 1], b_{V,t} \in \mathcal{B}_V^{(t)}, b_{V,t+1} \in \mathcal{B}_V^{(t+1)}, \theta \in \Theta\}.
\end{aligned}
\tag{105}
$$

Here $\delta_\aleph = \delta_{N,\aleph} + c_0\sqrt{\frac{\log(c_1/\zeta)}{N}}$, $\delta_{\mathcal{F}|_{d_\Theta+1}} = \delta_{N,\delta_{\mathcal{F}|_{d_\Theta+1}}} + c_0\sqrt{\frac{\log(c_1/\zeta)}{N}}$, and $\delta_\Upsilon = \delta_{N,\Upsilon} + c_0\sqrt{\frac{\log(c_1/\zeta)}{N}}$ for some constants $c_0, c_1$, where $\delta_{N,\aleph}, \delta_{N,\delta_{\mathcal{F}|_{d_\Theta+1}}}, \delta_{N,\Upsilon}$ are the critical radius of $\aleph, \mathcal{F}|_{d_\Theta+1}, \Upsilon$ respectively.

**Part III.** Finally, we consider $\|\mathbb{E}^{\pi^b}[\sum_a \widehat{b}_{\nabla V,1}^{\pi_\theta}(a, O_1)] - \widehat{\mathbb{E}}^{\pi^b}[\sum_a \widehat{b}_{\nabla V,1}^{\pi_\theta}(a, O_1)]\|_{\ell^2}$.

By applying Lemma 6, with probability at least $1 - \zeta$, we have

$$
\sup_{b_{\nabla V,1} \in \mathcal{B}_{\nabla V}^{(1)}} \|\mathbb{E}^{\pi^b}[\sum_a b_{\nabla V,1}(a, O_1)] - \widehat{\mathbb{E}}^{\pi^b}[\sum_a b_{\nabla V,1}(a, O_1)]\|_{\ell^2} \lesssim O\left(M_\mathcal{B}\sqrt{d_\Theta}\delta_{\mathcal{B}_{\nabla V}^{(1)}}\right).
\tag{106}
$$

where $\delta_{\mathcal{B}_{\nabla V}^{(1)}} = \delta_{N,\mathcal{B}_{\nabla V}^{(1)}} + c_0\sqrt{\frac{\log(c_1/\zeta)}{N}}$ and $\delta_{N,\mathcal{B}_{\nabla V}^{(1)}}$ denots the critical radius of $\mathcal{B}_{\nabla V}^{(1)}$.

### F.4 BOUNDS FOR CRITICAL RADIUS AND ONE-STEP ERRORS

According to the previous section, it suffices to upper bound the critical radius for function classes $\mathcal{F}, \Omega, \Xi, \aleph, \Upsilon$. We assume that for each $\mathcal{B}_V^{(t)}$, there is a function space $\mathcal{G}_V^{(t)}$ defined as

$$
\mathcal{G}_V^{(t)} := \{(o_t, h_{t-1}) \mapsto g(o_t, h_{t-1}); g(o_t, h_{t-1}) = \sum_a b_{V,t}(a, o_t, h_{t-1}), b_{V,t} \in \mathcal{B}_V^{(t)}\}.
\tag{107}
$$

Similarly, define

$$
\mathcal{G}_{\nabla V}^{(t)} := \{(o_t, h_{t-1}) \mapsto g(o_t, h_{t-1}); g(o_t, h_{t-1}) = \sum_a b_{\nabla V,t}(a, o_t, h_{t-1}), b_{\nabla V,t} \in \mathcal{B}_{\nabla V}^{(t)}\}.
\tag{108}
$$

#### F.4.1 VC CLASSES

We first consider $\Omega_{t,\theta}$ at each $t, \theta$, which contains the function of the form $r\frac{1}{2(G+1)M_\mathcal{B}}(b_{\nabla V,t} - R_t\nabla_\theta\pi_{\theta_t} - g_{V,t+1}\nabla_\theta\pi_{\theta_t} - \pi_{\theta_t}g_{\nabla V,t+1})^T f_t$ for any $r \in [-1, 1]$, $f_t \in \mathcal{F}^{(t)}$, $\theta_t \in \Theta_t$, $b_{\nabla V,t} \in \mathcal{B}_{\nabla V}^{(t)}$, $g_{\nabla V,t+1} \in \mathcal{G}_{\nabla V}^{(t+1)}$, and $g_{V,t+1} \in \mathcal{G}_V^{(t+1)}$. For simplicity, we assume that all these function classes are extended to their star convex hull. Then we can drop the $r \in [-1, 1]$ here. Also, $2(G+1)M_\mathcal{B}$ is a scaling constant as defined in the previous section. Without loss of generality, we drop the constant $M_\mathcal{B}$ and assume that all functions involved here are uniformly bounded by 1.

Suppose that $\mathcal{F}^{(t),\epsilon}, \mathcal{B}_{\nabla V}^{(t),\epsilon}, \mathcal{G}_{\nabla V}^{(t+1),\epsilon}, \mathcal{G}_V^{(t+1),\epsilon}$ are the $\epsilon$-coverings of the original spaces in the $\ell^\infty$ norm. Then there exist $(\tilde{f}_t, \tilde{b}_{\nabla V,t}, \tilde{g}_{\nabla V,t+1}, \tilde{g}_{V,t+1}) \in \mathcal{F}^{(t),\epsilon}, \mathcal{B}_{\nabla V}^{(t),\epsilon}, \mathcal{G}_{\nabla V}^{(t+1),\epsilon}, \mathcal{G}_V^{(t+1),\epsilon}$ such that the following holds.

For any fixed $\theta$, we have

$$
\begin{aligned}
&\|(b_{\nabla V,t} - R_t\nabla_\theta\pi_{\theta_t} - g_{V,t+1}\nabla_\theta\pi_{\theta_t} - \pi_{\theta_t}g_{\nabla V,t+1})^T f_t \\
&\quad - (\tilde{b}_{\nabla V,t} - R_t\nabla_\theta\pi_{\tilde{\theta}_t} - \tilde{g}_{V,t+1}\nabla_\theta\pi_{\tilde{\theta}_t} - \pi_{\tilde{\theta}_t}\tilde{g}_{\nabla V,t+1})^T \tilde{f}_t\|_\infty \\
&\leq \|(b_{\nabla V,t} - R_t\nabla_\theta\pi_{\theta_t} - g_{V,t+1}\nabla_\theta\pi_{\theta_t} - \pi_{\theta_t}g_{\nabla V,t+1})^T (f_t - \tilde{f}_t)\|_\infty \\
&\quad + \|(b_{\nabla V,t} - R_t\nabla_\theta\pi_{\theta_t} - g_{V,t+1}\nabla_\theta\pi_{\theta_t} - \pi_{\theta_t}g_{\nabla V,t+1})^T \tilde{f}_t - \\
&\quad (\tilde{b}_{\nabla V,t} - R_t\nabla_\theta\pi_{\tilde{\theta}_t} - \tilde{g}_{V,t+1}\nabla_\theta\pi_{\tilde{\theta}_t} - \pi_{\tilde{\theta}_t}\tilde{g}_{\nabla V,t+1})^T \tilde{f}_t\|_\infty
\end{aligned}
$$

$$\leq \|(f_t - \tilde{f}_t)\|_{\infty,2} + \|(b_{\nabla V,t} - R_t \nabla_\theta \pi_{\theta_t} - g_{V,t+1} \nabla_\theta \pi_{\theta_t} - \pi_{\theta_t} g_{\nabla V,t+1}) -$$
$$(\tilde{b}_{\nabla V,t} - R_t \nabla_\theta \pi_{\tilde{\theta}_t} - \tilde{g}_{V,t+1} \nabla_\theta \pi_{\tilde{\theta}_t} - \pi_{\tilde{\theta}_t} \tilde{g}_{\nabla V,t+1})\|_{\infty,2}$$
$$\leq \|(f_t - \tilde{f}_t)\|_{\infty,2} + \|b_{\nabla V,t} - \tilde{b}_{\nabla V,t}\|_{\infty,2} + G\sqrt{d_{\Theta_t}}\|g_{V,t+1} - \tilde{g}_{V,t+1}\|_\infty + \|g_{\nabla V,t+1} - \tilde{g}_{\nabla V,t+1}\|_{\infty,2}$$
$$\leq \sqrt{d_\Theta}\|(f_t - \tilde{f}_t)\|_{\infty,\infty} + \sqrt{d_\Theta}\|b_{\nabla V,t} - \tilde{b}_{\nabla V,t}\|_{\infty,\infty}$$
$$+ G\sqrt{d_{\Theta_t}}\|g_{V,t+1} - \tilde{g}_{V,t+1}\|_\infty + \sqrt{d_\Theta}\|g_{\nabla V,t+1} - \tilde{g}_{\nabla V,t+1}\|_{\infty,\infty}$$
$$\leq c_1 \sqrt{d_\Theta}\epsilon$$

for some constant $c_1$.

We notice that the coverings involved above do not depend on $\theta$. Therefore, $\mathcal{F}^{(t),\epsilon/c_1\sqrt{d_\Theta}} \times \mathcal{B}_{\nabla V}^{(t),\epsilon/c_1\sqrt{d_\Theta}} \times \mathcal{G}_{\nabla V}^{(t+1),\epsilon/c_1\sqrt{d_\Theta}} \times \mathcal{G}_V^{(t+1),\epsilon/c_1\sqrt{d_\Theta}}$ is a net that covers $\Omega_{t,\theta}$ for every $\theta_t \in \Theta_t$.

Let $\mathbb{V}$ denote the VC-dimension of a function space. Then we have the following result.

According to Corollary 14.3 in Wainwright (2019), C.2 in Dikkala et al. (2020), Lemma D.4, D.5 in Miao et al. (2022), we have with probability at least $1 - \zeta$ that the following holds

$$\sup_\theta \delta_{N,\Omega_{t,\theta}} \lesssim d_\Theta \sqrt{\frac{\gamma(\mathcal{B},\mathcal{F})}{N}} + \sqrt{\frac{\log(1/\zeta)}{N}}$$

where $\gamma(\mathcal{B},\mathcal{F})$ denotes $\max\left\{\left\{\mathbb{V}(\mathcal{F}_j^{(t)})\right\}_{j=1,t=1}^{d_\Theta+1,T}, \left\{\mathbb{V}(\mathcal{B}_j^{(t)})\right\}_{j=1,t=1}^{d_\Theta+1,T}\right\}$.

Similarly, we can get upper bound the critical radius for all the other involved function spaces by using the same strategies. However, we point out that $\delta_\Omega$ is the dominating term. In particular, by using covering numbers, we have

$$\delta_{N,\Xi_t} \lesssim d_\Theta \sqrt{\frac{\gamma(\mathcal{B},\mathcal{F})}{N}} + \sqrt{\frac{\log(1/\zeta)}{N}}$$

and

$$\delta_{N,\mathcal{F}^{(t)}} \lesssim \sqrt{\frac{\gamma(\mathcal{F})}{N}} + \sqrt{\frac{\log(1/\zeta)}{N}}.$$

For more details regarding bounding critical radius by VC-dimension, readers can also refer to Lemma D.5 and Example 1 in Miao et al. (2022), which focuses on the off-policy evaluation given one fixed policy. As a side product, our result for estimating the V-bridge functions recovers results in Miao et al. (2022). Moreover, we have an extra $\sqrt{d_{\Theta_t}}$ used for policy optimization.

Recall that

$$\delta_\Omega \lesssim \sup_{t,\theta} \delta_{N,\Omega_{t,\theta}} + c_0 \sqrt{\frac{\log(T/\zeta) + d_{\Theta_t}^2 + \log N}{N}}. \tag{109}$$

Consequently, we have with probability at least $1 - \gamma$

$$\left\|\mathbb{T}_{X_t}\left[b_{\nabla V,t}^*(\widehat{b}_{V,t+1}^{\pi_\theta}, \widehat{b}_{\nabla V,t+1}^{\pi_\theta}, \theta) - \widehat{b}_{\nabla V,t}^{\pi_\theta}\right]\right\|_{\mathcal{L}^2(\ell^2, \mathbb{P}^{\pi^b})} \lesssim M_{\mathcal{B}} d_\Theta T^{\frac{1}{2}} \sqrt{\frac{\log(c_1 T/\zeta)\gamma(\mathcal{F},\mathcal{B})\log N}{N}}$$

where $T^{\frac{1}{2}}$ comes from $T$ applications of Corollary 14.3 in Wainwright (2019).

By a similar argument, we have with probability at least $1 - \zeta$,

$$\left\|\mathbb{T}_{X_j}\left[b_{V,j}^*(\widehat{b}_{V,j+1}^{\pi_\theta}, \theta) - \widehat{b}_{V,j}^{\pi_\theta}\right]\right\|_{\mathcal{L}^2(\mathbb{P}^{\pi^b})} \lesssim \sqrt{M_{\mathcal{B}}} T^{\frac{1}{2}} \sqrt{d_\Theta} \sqrt{\frac{\log(c_1 T/\zeta)\gamma(\mathcal{F},\mathcal{B})\log N}{N}}.$$

When $\mathcal{B}$ and $\mathcal{F}$ are not norm-constrained (Miao et al., 2022), we need to consider $\|\cdot\|_{\mathcal{B}}$ and $\|\cdot\|_{\mathcal{F}}$. In this case, with an extra requirement of boundedness of conditional moment operators w.r.t. $\|\cdot\|_{\mathcal{B}}$ and $\|\cdot\|_{\mathcal{F}}$, an extra factor $T$ occurs. In this case, we have

$$\left\|\mathbb{T}_{X_t}\left[b_{\nabla V,t}^*(\widehat{b}_{V,t+1}^{\pi_\theta}, \widehat{b}_{\nabla V,t+1}^{\pi_\theta}, \theta) - \widehat{b}_{\nabla V,t}^{\pi_\theta}\right]\right\|_{\mathcal{L}^2(\ell^2, \mathbb{P}^{\pi^b})} \lesssim M_{\mathcal{B}} d_\Theta T^{\frac{3}{2}} \sqrt{\frac{\log(c_1 T/\zeta)\gamma(\mathcal{F},\mathcal{B})\log N}{N}}$$

and

$$\left\| \mathbb{T}_{X_j} \left[ b_{V,j}^* (\widehat{b}_{V,j+1}^{\pi_\theta}, \theta) - \widehat{b}_{V,j}^{\pi_\theta} \right] \right\|_{\mathcal{L}^2(\mathbb{P}^{\pi^b})} \lesssim \sqrt{M_\mathcal{B}} T^{\frac{3}{2}} \sqrt{d_\Theta} \sqrt{\frac{\log (c_1 T/\zeta) \, \gamma(\mathcal{F},\mathcal{B}) \log N}{N}}.$$

where $\sqrt{M_\mathcal{B}}$ is due to Appendix B.2 which implies $M_V \asymp \sqrt{M_\mathcal{B}}$.

### F.4.2 RKHSs

In this section, we consider the case that $\left\{ \mathcal{F}_j^{(t)} \right\}_{j=1,t=1}^{d_\Theta+1,T}$, $\left\{ \mathcal{B}_j^{(t)} \right\}_{j=1,t=1}^{d_\Theta+1,T}$, $\left\{ \mathcal{G}_j^{(t)} \right\}_{j=1,t=1}^{d_\Theta+1,T}$ are all RKHS. Typically, we need to consider sums of reproducing kernels (12.4.1 in Wainwright (2019)) and tensor products of reproducing kernels (12.4.2 in Wainwright (2019)).

We consider $\Omega_{t,\theta}$ first, which contains the function of the form $r \frac{1}{2(G+1)M_\mathcal{B}} (b_{\nabla V,t} - R_t \nabla_\theta \pi_{\theta_t} - g_{V,t+1} \nabla_\theta \pi_{\theta_t} - \pi_{\theta_t} g_{\nabla V,t+1})^T f_t$ for any $r \in [-1,1]$, $f_t \in \mathcal{F}^{(t)}$, $\theta_t \in \Theta_t$, $b_{\nabla V,t} \in \mathcal{B}_{\nabla V}^{(t)}$, $g_{\nabla V,t+1} \in \mathcal{G}_{\nabla V}^{(t+1)}$, and $g_{V,t+1} \in \mathcal{G}_V^{(t+1)}$. For simplicity, we assume that all these function classes are extended to their star convex hull. Then we can drop the $r \in [-1,1]$ and $\pi_\theta$ here. Since $\nabla_\theta \pi_\theta = \pi_\theta \nabla_\theta \log \pi_\theta$ which is also bounded by $G$, we can let $\tilde{\mathcal{G}}_V^{(t+1)} = \{G \cdot g : g \in \mathcal{G}_V^{(t+1)}\}$. Also, $2(G+1)M_\mathcal{B}$ is a scaling constant as defined in the previous section. Without loss of generality, we drop the constant $M_\mathcal{B}$ and assume that all functions involved here are uniformly bounded by 1.

Then $\Omega_{t,\theta}$ can be expressed as

$$\{(h_1 - h_2 - h_3)^\top f; f \in \mathcal{F}^{(t)}, h_1 \in \text{star}(\mathcal{B}_{\nabla V}^{(t)}), h_3 \in \text{star}(\mathcal{G}_{\nabla V}^{(t+1)}), h_{2,j} \in \tilde{\mathcal{G}}_V^{(t+1)}, \forall j = 1, ..., d_\Theta\}.$$

We assume that each $\text{star}(\mathcal{B}_{\nabla V}^{(t)})_j$ is an RKHS with a common kernel $K_\mathcal{B}$ for each $j = 1, ..., d_\Theta$. Similarly, $\text{star}(\mathcal{G}_{\nabla V}^{(t+1)})_j$ has a common kernel $K_{\mathcal{G},\nabla V}$ for each $j = 1, ..., d_\Theta$. $\tilde{\mathcal{G}}_V^{(t+1)}$ has a kernel $K_{\mathcal{G},V}$. Finally, the kernel for the test functions is denoted as $K_\mathcal{F}$.

Then according to 12.4.1 and 12.4.2 in Wainwright (2019), $\Omega_{t,\theta}$ can be expanded as an RKHS with a kernel $K_\Omega := \sum_{j=1}^{d_\Theta} (K_\mathcal{B} + K_{\mathcal{G},\nabla V} + K_{\mathcal{G},V}) \otimes K_\mathcal{F}$.

We use $\{\lambda_j(\cdot)\}_{j=1}^\infty$ to denote a sequence of decreasing sorted eigenvalues of a kernel. Then according to Corollary 14.5 of Wainwright (2019), the localized population Rademacher complexity should satisfy

$$\mathcal{R}_N(\delta; \Omega_{t,\theta}) \lesssim \sqrt{\frac{1}{N}} \sqrt{\sum_{j=1}^\infty \min \{\lambda_j(K_\Omega), \delta^2\}}. \tag{110}$$

Furthermore, by Weyl's inequality, we have

$$\lambda_j(K_\Omega) = \lambda_j(\sum_{i=1}^{d_\Theta} (K_\mathcal{B} + K_{\mathcal{G},\nabla V} + K_{\mathcal{G},V}) \otimes K_\mathcal{F})$$

$$\leq \sum_{i=1}^{d_\Theta} \lambda_{\lfloor \frac{j}{d_\Theta} \rfloor}((K_\mathcal{B} + K_{\mathcal{G},\nabla V} + K_{\mathcal{G},V}) \otimes K_\mathcal{F})$$

$$\leq d_\Theta \max\{\lambda_{\lfloor \frac{j}{d_\Theta} \rfloor}(K_\mathcal{B} + K_{\mathcal{G},\nabla V} + K_{\mathcal{G},V})\lambda_1(K_\mathcal{F}), \lambda_1(K_\mathcal{B} + K_{\mathcal{G},\nabla V} + K_{\mathcal{G},V})\lambda_{\lfloor \frac{j}{d_\Theta} \rfloor}(K_\mathcal{F})\}. \tag{111}$$

We consider the following assumption for measuring the size of RKHSs.

**Assumption 9** (Polynomial eigen-decay RKHS kernels). *There exist constants* $\alpha_{K_\mathcal{B}} > \frac{1}{2}$, $\alpha_{K_{\mathcal{G},\nabla V}} > \frac{1}{2}$, $\alpha_{K_{\mathcal{G},V}} > \frac{1}{2}$, $\alpha_{K_\mathcal{F}} > \frac{1}{2}$, $c > 0$ *such that* $\lambda_j(K_\mathcal{B}) < cj^{-2\alpha_{K_\mathcal{B}}}$, $\lambda_j(K_{\mathcal{G},\nabla V}) < cj^{-2\alpha_{K_{\mathcal{G},\nabla V}}}$, $\lambda_j(K_{\mathcal{G},V}) < cj^{-2\alpha_{K_{\mathcal{G},V}}}$, *and* $\lambda_j(K_\mathcal{F}) < cj^{-2\alpha_{K_\mathcal{F}}}$ *for every* $j = 1, 2, ....$

We note that the polynomial eigen-decay rate for RKHSs is commonly considered in practice (e.g. $\alpha$-order Sobolev space).

Under Assumption 9, we have

$$
\lambda_j(K_\Omega)
$$
$$
\lesssim d_\Theta \max\{\lambda_{\lfloor \frac{j}{3d_\Theta} \rfloor}(K_\mathcal{B}) + \lambda_{\lfloor \frac{j}{3d_\Theta} \rfloor}(K_{\mathcal{G},\nabla V}) + \lambda_{\lfloor \frac{j}{3d_\Theta} \rfloor}(K_{\mathcal{G},V}), \lambda_{\lfloor \frac{j}{d_\Theta} \rfloor}(K_\mathcal{F})\}
$$
$$
\lesssim d_\Theta \max\{(3d_\Theta)^{2\max\{\alpha_{K_\mathcal{B}},\alpha_{K_{\mathcal{G},V}},\alpha_{K_{\mathcal{G},\nabla V}}\}}(cj^{-2\alpha_{K_\mathcal{B}}} + cj^{-2\alpha_{K_{\mathcal{G},\nabla V}}} + cj^{-2\alpha_{K_{\mathcal{G},V}}}), d_\Theta^{2\alpha_{K_\mathcal{F}}} cj^{-2\alpha_{K_\mathcal{F}}}\}. \tag{112}
$$

For clarity, we let $\alpha_{\max}$ denote $\max\{\alpha_{K_\mathcal{B}}, \alpha_{K_{\mathcal{G},V}}, \alpha_{K_{\mathcal{G},\nabla V}}\}$ and let $\alpha_{\min}$ denote $\min\{\alpha_{K_\mathcal{B}}, \alpha_{K_{\mathcal{G},V}}, \alpha_{K_{\mathcal{G},\nabla V}}\}$. Then we have

$$
\lambda_j(K_\Omega) \lesssim \max\{d_\Theta^{1+2\alpha_{\max}} j^{-2\alpha_{\min}}, d_\Theta^{1+2\alpha_{K_\mathcal{F}}} j^{-2\alpha_{K_\mathcal{F}}}\}. \tag{113}
$$

Consequently, according to equation 110equation 113, sufficient conditions for

$$
\mathcal{R}_N(\delta; \Omega_{t,\theta}) \lesssim \delta^2
$$

are

$$
\sqrt{\frac{1}{N}} \sqrt{\sum_{j=1}^\infty \min\left\{d_\Theta^{1+2\alpha_{\max}} j^{-2\alpha_{\min}}, \delta^2\right\}} \lesssim \delta^2 \text{ if } \alpha_{K_\mathcal{F}} > \alpha_{\min}
$$

and

$$
\sqrt{\frac{1}{N}} \sqrt{\sum_{j=1}^\infty \min\left\{d_\Theta^{1+2\alpha_{K_\mathcal{F}}} j^{-2\alpha_{K_\mathcal{F}}}, \delta^2\right\}} \lesssim \delta^2 \text{ if } \alpha_{K_\mathcal{F}} \le \alpha_{\min}.
$$

By some direct calculations, we have

$$
\sqrt{\frac{1}{N}} \sqrt{\sum_{j=1}^\infty \min\left\{d_\Theta^{1+2\alpha_{\max}} j^{-2\alpha_{\min}}, \delta^2\right\}} \lesssim \sqrt{\frac{d_\Theta^{1+2\alpha_{\max}}}{N}} \delta^{1-\frac{1}{2\alpha_{\min}}}, \tag{114}
$$

and

$$
\sqrt{\frac{1}{N}} \sqrt{\sum_{j=1}^\infty \min\left\{d_\Theta^{1+2\alpha_{K_\mathcal{F}}} j^{-2\alpha_{K_\mathcal{F}}}, \delta^2\right\}} \lesssim \sqrt{\frac{d_\Theta^{1+2\alpha_{K_\mathcal{F}}}}{N}} \delta^{1-\frac{1}{2\alpha_{K_\mathcal{F}}}}. \tag{115}
$$

Therefore, we have $\delta_{N,\Omega_t} \asymp \dfrac{d_\Theta^{\frac{1+2\alpha_{K_\mathcal{F}}}{2+1/\alpha_{K_\mathcal{F}}}}}{N^{\frac{1}{2+1/\alpha_{K_\mathcal{F}}}}} \mathbb{I}(\alpha_{K_\mathcal{F}} \le \alpha_{\min}) + \dfrac{d_\Theta^{\frac{1+2\alpha_{\max}}{2+1/\alpha_{\min}}}}{N^{\frac{1}{2+1/\alpha_{\min}}}} \mathbb{I}(\alpha_{K_\mathcal{F}} > \alpha_{\min})$.

We notice that $\dfrac{d_\Theta^{\frac{1+2\alpha_{K_\mathcal{F}}}{2+1/\alpha_{K_\mathcal{F}}}}}{N^{\frac{1}{2+1/\alpha_{K_\mathcal{F}}}}} \mathbb{I}(\alpha_{K_\mathcal{F}} \le \alpha_{\min}) + \dfrac{d_\Theta^{\frac{1+2\alpha_{\max}}{2+1/\alpha_{\min}}}}{N^{\frac{1}{2+1/\alpha_{\min}}}} \mathbb{I}(\alpha_{K_\mathcal{F}} > \alpha_{\min})$ is also an asymptotic upper bound for $\Xi_t$ and $\mathcal{F}^{(t)}$.

Consequently, we have

$$
\left\| \mathbb{T}_{X_t} \left[ b_{\nabla V,t}^*(\widehat{b}_{V,t+1}^{\pi_\theta}, \widehat{b}_{\nabla V,t+1}^{\pi_\theta}, \theta) - \widehat{b}_{\nabla V,t}^{\pi_\theta} \right] \right\|_{\mathcal{L}^2(\ell^2, \mathbb{P}^{\pi^b})} \lesssim M_\mathcal{B} h(d_\Theta) T^{\frac{3}{2}} \sqrt{\log(c_1 T/\zeta) \log N} \frac{1}{N^{\frac{1}{2+\max\{1/\alpha_{K_\mathcal{F}}, 1/\alpha_{\min}\}}}}
$$

where $h(d_\Theta) = \max\{d_\Theta, d_\Theta^{\frac{1+2\alpha_{K_\mathcal{F}}}{2+1/\alpha_{K_\mathcal{F}}}}, d_\Theta^{\frac{1+2\alpha_{\max}}{2+1/\alpha_{\min}}}\}$ with $\alpha_{\max} = \max\{\alpha_{K_\mathcal{B}}, \alpha_{K_{\mathcal{G},V}}, \alpha_{K_{\mathcal{G},\nabla V}}\}$ and $\alpha_{\min} = \min\{\alpha_{K_\mathcal{B}}, \alpha_{K_{\mathcal{G},V}}, \alpha_{K_{\mathcal{G},\nabla V}}\}$.

Similarly, we have

$$
\left\| \mathbb{T}_{X_j} \left[ b_{V,j}^*(\widehat{b}_{V,j+1}^{\pi_\theta}, \theta) - \widehat{b}_{V,j}^{\pi_\theta} \right] \right\|_{\mathcal{L}^2(\mathbb{P}^{\pi^b})} \lesssim \sqrt{M_\mathcal{B}} T^{\frac{3}{2}} \sqrt{d_\Theta} \sqrt{\log(c_1 T/\zeta) \log N} \frac{1}{N^{\frac{1}{2+\max\{1/\alpha_{K_\mathcal{F}}, 1/\alpha_{\min}\}}}}.
$$

### F.5 POLICY GRADIENT ESTIMATION ERRORS

Recall that we have

$$
\begin{aligned}
&\|\nabla_\theta \mathcal{V}(\pi_\theta) - \widehat{\nabla_\theta \mathcal{V}(\pi_\theta)}\|_{\ell^2} \\
&\leq \sum_{t=1}^{T} C_{\pi^b} \tau_t \left\| \mathbb{T}_{X_t} \left[ b^*_{\nabla V, t}(\widehat{b}^{\pi_\theta}_{V, t+1}, \widehat{b}^{\pi_\theta}_{\nabla V, t+1}, \theta) - \widehat{b}^{\pi_\theta}_{\nabla V, t} \right] \right\|_{\mathcal{L}^2(\ell^2, \mathbb{P}^{\pi^b})} \\
&\quad + \sum_{t=1}^{T-1} \sum_{j=t+1}^{T} C_{\pi^b} G \sqrt{\sum_{i=t}^{T} d_{\Theta_i}} \tau_j \left\| \mathbb{T}_{X_j} \left[ b^*_{V, j}(\widehat{b}^{\pi_\theta}_{V, j+1}, \theta) - \widehat{b}^{\pi_\theta}_{V, j} \right] \right\|_{\mathcal{L}^2(\mathbb{P}^{\pi^b})} \\
&\quad + \|\mathbb{E}^{\pi^b}[\sum_a \widehat{b}^{\pi_\theta}_{\nabla V, 1}(a, O_1)] - \widehat{\mathbb{E}}^{\pi^b}[\sum_a \widehat{b}^{\pi_\theta}_{\nabla V, 1}(a, O_1)]\|_{\ell^2}.
\end{aligned}
\tag{116}
$$

Plugging in the results from section F.3, we have that with probability at least $1 - \zeta$,

$$
\begin{aligned}
&\|\nabla_\theta \mathcal{V}(\pi_\theta) - \widehat{\nabla_\theta \mathcal{V}(\pi_\theta)}\|_{\ell^2} \\
&\lesssim C_{\pi^b} \tau_{\max} M_{\mathcal{B}} d_\Theta T^{\frac{5}{2}} \sqrt{\frac{\log(T/\zeta)\, \gamma(\mathcal{F}, \mathcal{B}) \log N}{N}} \\
&\quad + C_{\pi^b} G \sqrt{\sum_{i=t}^{T} d_{\Theta_i}} \tau_{\max} \sqrt{M_{\mathcal{B}}} T^{\frac{7}{2}} \sqrt{d_\Theta} \sqrt{\frac{\log(T/\zeta)\, \gamma(\mathcal{F}, \mathcal{B}) \log N}{N}} \\
&\lesssim C_{\pi^b} \tau_{\max} M_{\mathcal{B}} d_\Theta T^{\frac{5}{2}} \sqrt{\frac{\log(T/\zeta)\, \gamma(\mathcal{F}, \mathcal{B}) \log N}{N}} \text{ since } M_{\mathcal{B}} \asymp T^2,
\end{aligned}
\tag{117}
$$

where $\gamma(\mathcal{B}, \mathcal{F})$ denotes $\max \left\{ \left\{ \mathbb{V}(\mathcal{F}_j^{(t)}) \right\}_{j=1, t=1}^{d_\Theta+1, T}, \left\{ \mathbb{V}(\mathcal{B}_j^{(t)}) \right\}_{j=1, t=1}^{d_\Theta+1, T} \right\}$.

In the more general nonparametric function classes RKHSs, we have

$$
\begin{aligned}
&\|\nabla_\theta \mathcal{V}(\pi_\theta) - \widehat{\nabla_\theta \mathcal{V}(\pi_\theta)}\|_{\ell^2} \\
&\lesssim C_{\pi^b} \tau_{\max} M_{\mathcal{B}} h(d_\Theta) T^{\frac{5}{2}} \sqrt{\log(c_1 T/\zeta) \log N} \frac{1}{N^{\frac{1}{2+\max\{1/\alpha_{K_{\mathcal{F}}}, 1/\alpha_{\min}\}}}} \\
&\quad + C_{\pi^b} G \sqrt{\sum_{i=t}^{T} d_{\Theta_i}} \tau_{\max} \sqrt{M_{\mathcal{B}}} T^{\frac{7}{2}} \sqrt{d_\Theta} \sqrt{\log(c_1 T/\zeta) \log N} \frac{1}{N^{\frac{1}{2+\max\{1/\alpha_{K_{\mathcal{F}}}, 1/\alpha_{\min}\}}}} \\
&\lesssim C_{\pi^b} \tau_{\max} M_{\mathcal{B}} h(d_\Theta) T^{\frac{5}{2}} \sqrt{\log(c_1 T/\zeta) \log N}\, N^{-\frac{1}{2+\max\{1/\alpha_{K_{\mathcal{F}}}, 1/\alpha_{\min}\}}}
\end{aligned}
\tag{118}
$$

where $h(d_\Theta) = \max\{d_\Theta, d_\Theta^{\frac{1+2\alpha_{K_{\mathcal{F}}}}{2+1/\alpha_{K_{\mathcal{F}}}}}, d_\Theta^{\frac{1+2\alpha_{\max}}{2+1/\alpha_{\min}}}\}$. Here $\alpha_{\max} = \max\{\alpha_{K_{\mathcal{B}}}, \alpha_{K_{\mathcal{G}, V}}, \alpha_{K_{\mathcal{G}, \nabla V}}\}$, $\alpha_{\min} = \min\{\alpha_{K_{\mathcal{B}}}, \alpha_{K_{\mathcal{G}, V}}, \alpha_{K_{\mathcal{G}, \nabla V}}\}$ measure the eigen-decay rates of RKHSs. See section F.4.2 for details.

## G  PROOFS FOR SUBOPTIMALITY

In this section, we present a complete proof of Theorem 3. We first assume that we have access to an *oracle* policy gradient that depends on the unobserved states. Then, by borrowing the analysis of global convergence of natural policy gradient (NPG) algorithm, we establish a global convergence result of policy gradient methods for finite-horizon POMDP with history-dependent policies. Then we establish an upper bound on the suboptimality of $\widehat{\pi}$ by replacing the oracle policy gradient with the estimated policy gradient.

### G.1 An auxiliary proposition with oracle policy gradients

In this section, we present the global convergence of the proposed gradient ascent algorithm when the *oracle* policy gradient can be accessed in the finite-horizon POMDP. The idea is motivated by Liu et al. (2020) which is focused on the infinite-horizon MDP case.

**Proposition 1.** *Given an initial* $\theta^{(0)}$*, let* $\{\theta^{(k)}\}_{k=1}^{K-1}$ *be generated by* $\theta^{(k+1)} = \theta^{(k)} + \eta \nabla_{\theta^{(k)}} \mathcal{V}(\pi_{\theta^{(k)}})$. *Then, under Assumptions 4(e) and 5(a)-(b), we have*

$$
\mathcal{V}(\pi_{\theta^*}) - \frac{1}{K} \sum_{k=0}^{K-1} \mathcal{V}(\pi_{\theta^{(k)}})
$$

$$
\leq G \frac{1}{K} \sum_{k=0}^{K-1} \sum_{t=1}^{T} \sqrt{d_{\Theta_t}} \|(w_{*,t}^{(k)}(\theta^{(k)}) - \nabla_{\theta_t^{(k)}} \mathcal{V}(\pi_{\theta^{(k)}}))\|_{\ell^2} + \frac{\beta \eta}{2} \frac{1}{K} \sum_{k=0}^{K-1} \|\nabla_{\theta^{(k)}} \mathcal{V}(\pi_{\theta^{(k)}})\|_{\ell^2}^2 \quad (119)
$$

$$
+ \frac{1}{K\eta} \sum_{t=1}^{T} \mathbb{E}_{(o_t, h_{t-1}) \sim \mathbb{P}_t^{\pi_{\theta^*}}} \left[ \mathrm{KL}(\pi^*(\cdot \mid o_t, h_{t-1}) \| \pi_{\theta^{(0)}}(\cdot \mid o_t, h_{t-1})) \right] + \varepsilon_{approx}.
$$

In Proposition 1, the first term measures the distance between the NPG update direction and the policy gradient (PG) update direction, which can be essentially upper bounded by the norm of PG according to Assumption 5(c). Therefore, according to the PG update rule, the first term converges to 0 because $\theta^{(k)}$ converges to stationary points. For the same reason, the second term also converges to 0. The third term is $O(\frac{1}{K})$, and the final term is an approximation error.

Next, we present a proof of Proposition 1.

*proof of Proposition 1.* We start from the performance difference lemma 8 for POMDP, which shows that

$$
\mathcal{V}(\pi_{\theta^*}) - \mathcal{V}(\pi_{\theta^{(k)}}) = \sum_{t=1}^{T} \mathbb{E}_{(a_t, o_t, s_t, h_{t-1}) \sim \mathbb{P}_t^{\pi_{\theta^*}}} \left[ A_t^{\pi_{\theta^{(t)}}}(a_t, o_t, s_t, h_{t-1}) \right]. \quad (120)
$$

Let $w_*^{(k)}$ be the minimizer of the loss function defined in Definition D.2 for a given $\theta^{(k)}$, i.e.

$$
w_*^{(k)} := \arg\min_w \sum_{t=1}^{T} \left( \mathbb{E}_{(a_t, o_t, s_t, h_{t-1}) \sim \mathbb{P}_t^{\pi_{\theta^{(k)}}}} \left[ \left( A_t^{\pi_{\theta^{(k)}}}(a_t, o_t, s_t, h_{t-1}) \right. \right. \right.
$$

$$
\left. \left. \left. - w_t^\top \nabla_{\theta_t^{(k)}} \log \pi_{\theta_t^{(k)}}(a_t \mid o_t, h_{t-1}) \right)^2 \right] \right)^{\frac{1}{2}}. \quad (121)
$$

Then we have a decomposition of the difference between policy values.

$$
\mathcal{V}(\pi_{\theta^*}) - \mathcal{V}(\pi_{\theta^{(k)}})
$$

$$
= \sum_{t=1}^{T} \mathbb{E}_{(a_t, o_t, s_t, h_{t-1}) \sim \mathbb{P}_t^{\pi_{\theta^*}}} \left[ A_t^{\pi_{\theta^{(k)}}}(a_t, o_t, s_t, h_{t-1}) \right]
$$

$$
= \sum_{t=1}^{T} \mathbb{E}_{(a_t, o_t, s_t, h_{t-1}) \sim \mathbb{P}_t^{\pi_{\theta^*}}} \left[ A_t^{\pi_{\theta^{(k)}}}(a_t, o_t, s_t, h_{t-1}) - (w_{*,t}^{(k)})^\top \nabla_\theta \log \pi_{\theta_t}^{(k)} \right]
$$

$$
+ \sum_{t=1}^{T} \mathbb{E}_{(a_t, o_t, s_t, h_{t-1}) \sim \mathbb{P}_t^{\pi_{\theta^*}}} \left[ (w_{*,t}^{(k)})^\top \nabla_\theta \log \pi_{\theta_t}^{(k)} \right]
$$

$$
= \underbrace{\sum_{t=1}^{T} \mathbb{E}_{(a_t, o_t, s_t, h_{t-1}) \sim \mathbb{P}_t^{\pi_{\theta^*}}} \left[ A_t^{\pi_{\theta^{(k)}}}(a_t, o_t, s_t, h_{t-1}) - (w_{*,t}^{(k)})^\top \nabla_\theta \log \pi_{\theta_t}^{(k)} \right]}_{(a)}
$$

$$+ \underbrace{\sum_{t=1}^{T} \mathbb{E}_{(a_t, o_t, s_t, h_{t-1}) \sim \mathbb{P}_t^{\pi_{\theta^*}}} \left[ (w_{*,t}^{(k)} - w_t^{(k)})^\top \nabla_\theta \log \pi_{\theta_t}^{(k)} \right]}_{(b)}$$

$$+ \underbrace{\sum_{t=1}^{T} \mathbb{E}_{(a_t, o_t, s_t, h_{t-1}) \sim \mathbb{P}_t^{\pi_{\theta^*}}} \left[ (w_t^{(k)})^\top \nabla_\theta \log \pi_{\theta_t}^{(k)} \right]}_{(c)}. \tag{122}$$

We analyze these three terms separately. For the first term (a), by applying Jensen's inequality $\mathbb{E}[X] \leq \sqrt{\mathbb{E}[X^2]}$ for each $t = 1, ..., T$, we have

$$(a)$$

$$= \sum_{t=1}^{T} \mathbb{E}_{(a_t, o_t, s_t, h_{t-1}) \sim \mathbb{P}_t^{\pi_{\theta^*}}} \left[ A_t^{\pi_{\theta^{(k)}}}(a_t, o_t, s_t, h_{t-1}) - (w_{*,t}^{(k)})^\top \nabla_\theta \log \pi_{\theta_t}^{(k)} \right]$$

$$\leq \sum_{t=1}^{T} \left( \mathbb{E}_{(a_t, o_t, s_t, h_{t-1}) \sim \mathbb{P}_t^{\pi_{\theta^*}}} \left[ \left( A_t^{\pi_{\theta^{(k)}}}(a_t, o_t, s_t, h_{t-1}) - (w_{*,t}^{(k)})^\top \nabla_\theta \log \pi_{\theta_t}^{(k)} \right)^2 \right] \right)^{\frac{1}{2}}$$

$$\leq \varepsilon_{approx} \text{ (by Definition D.2)}. \tag{123}$$

For the second term (b), by Assumption 4(e), we have

$$(b)$$

$$= \sum_{t=1}^{T} \mathbb{E}_{(a_t, o_t, s_t, h_{t-1}) \sim \mathbb{P}_t^{\pi_{\theta^*}}} \left[ (w_{*,t}^{(k)} - w_t^{(k)})^\top \nabla_\theta \log \pi_{\theta_t}^{(k)} \right]$$

$$\leq \sum_{t=1}^{T} \mathbb{E}_{(a_t, o_t, s_t, h_{t-1}) \sim \mathbb{P}_t^{\pi_{\theta^*}}} \left[ |(w_{*,t}^{(k)} - w_t^{(k)})^\top \nabla_\theta \log \pi_{\theta_t}^{(k)}| \right]$$

$$\leq \sum_{t=1}^{T} \mathbb{E}_{(a_t, o_t, s_t, h_{t-1}) \sim \mathbb{P}_t^{\pi_{\theta^*}}} \left[ \|(w_{*,t}^{(k)} - w_t^{(k)})\|_{\ell^2} \|\nabla_\theta \log \pi_{\theta_t}^{(k)}\|_{\ell^2} \right] \text{ (Cauchy-Schwartz)}$$

$$\leq \sum_{t=1}^{T} \mathbb{E}_{(a_t, o_t, s_t, h_{t-1}) \sim \mathbb{P}_t^{\pi_{\theta^*}}} \left[ \|(w_{*,t}^{(k)} - w_t^{(k)})\|_{\ell^2} \sqrt{d_{\Theta_t}} \|\nabla_\theta \log \pi_{\theta_t}^{(k)}\|_{\ell^\infty} \right]$$

$$\leq G \sum_{t=1}^{T} \sqrt{d_{\Theta_t}} \|(w_{*,t}^{(k)} - w_t^{(k)})\|_{\ell^2} \text{ (Assumption 4(e))}.$$

For the third term $(c) = \sum_{t=1}^{T} \mathbb{E}_{(a_t, o_t, s_t, h_{t-1}) \sim \mathbb{P}_t^{\pi_{\theta^*}}} \left[ (w_t^{(k)})^\top \nabla_\theta \log \pi_{\theta_t}^{(k)} \right]$, we employ $\beta$-smoothness of $\log(\pi_\theta)$, i.e. Assumption 5(b). According to Assumption 5(b) and by Taylor's theorem, it holds that

$$\left| \log \pi_{\theta_t'}(a_t \mid o_t, h_{t-1}) - \log \pi_{\theta_t}(a_t \mid o_t, h_{t-1}) - \nabla_\theta \log \pi_{\theta_t}(a_t \mid o_t, h_{t-1})^\top (\theta_t' - \theta_t) \right| \leq \frac{\beta}{2} \|\theta_t - \theta_t'\|_{\ell^2}^2$$

for each $a_t, o_t, h_{t-1} \in \mathcal{A} \times \mathcal{O} \times \mathcal{H}_{t-1}$.

Let $\theta_t'$ be $\theta_t^{(k+1)}$ and $\theta_t$ be $\theta_t^{(k)}$ here. We then have

$$\nabla_\theta \log \pi_{\theta_t^{(k)}}(a_t \mid o_t, h_{t-1})^\top \left( \theta_t^{(k+1)} - \theta_t^{(k)} \right) \leq \frac{\beta}{2} \|\theta_t^{(k+1)} - \theta_t^{(k)}\|_{\ell^2}^2 + \log \frac{\pi_{\theta_t^{(k+1)}}(a_t \mid o_t, h_{t-1})}{\pi_{\theta_t^{(k)}}(a_t \mid o_t, h_{t-1})}. \tag{124}$$

Recall that $\theta_t^{(k+1)} - \theta_t^{(k)} = \eta w_t^{(k)}$ by definition. Plugging this term into the above equation and we have

$$\eta \nabla_{\theta_t} \log \pi_{\theta_t^{(k)}} (a_t \mid o_t, h_{t-1})^\top \left( w_t^{(k)} \right) \leq \eta^2 \frac{\beta}{2} \|w_t^{(k)}\|_{\ell^2}^2 + \log \frac{\pi_{\theta_t^{(k+1)}} (a_t \mid o_t, h_{t-1})}{\pi_{\theta_t^{(k)}} (a_t \mid o_t, h_{t-1})}. \quad (125)$$

The third term $\sum_{t=1}^T \mathbb{E}_{(a_t, o_t, s_t, h_{t-1}) \sim \mathbb{P}_t^{\pi_{\theta^*}}} \left[ (w_t^{(k)})^\top \nabla_{\theta_t} \log \pi_{\theta_t}^{(k)} \right]$ can then be upper bounded by

$$(c)$$

$$= \sum_{t=1}^T \mathbb{E}_{(a_t, o_t, s_t, h_{t-1}) \sim \mathbb{P}_t^{\pi_{\theta^*}}} \left[ (w_t^{(k)})^\top \nabla_{\theta_t} \log \pi_{\theta_t}^{(k)} \right]$$

$$\leq \frac{1}{\eta} \sum_{t=1}^T \mathbb{E}_{(a_t, o_t, s_t, h_{t-1}) \sim \mathbb{P}_t^{\pi_{\theta^*}}} \left[ \eta^2 \frac{\beta}{2} \|w_t^{(k)}\|_{\ell^2}^2 + \log \frac{\pi_{\theta_t^{(k+1)}} (a_t \mid o_t, h_{t-1})}{\pi_{\theta_t^{(k)}} (a_t \mid o_t, h_{t-1})} \right] \quad \text{by equation 125}$$

$$= \frac{1}{\eta} \sum_{t=1}^T \mathbb{E}_{(o_t, h_{t-1}) \sim \mathbb{P}_t^{\pi_{\theta^*}}} \left[ \mathrm{KL}(\pi^*(\cdot \mid o_t, h_{t-1}) \| \pi_{\theta^{(k)}}(\cdot \mid o_t, h_{t-1})) \right.$$

$$\left. - \mathrm{KL}(\pi^*(\cdot \mid o_t, h_{t-1}) \| \pi_{\theta^{(k+1)}}(\cdot \mid o_t, h_{t-1})) \right] + \frac{\beta \eta}{2} \sum_{t=1}^T \|w_t^{(k)}\|_{\ell^2}^2$$

$$(126)$$

where we use $\mathrm{KL}(p \| q) = \mathbb{E}_{x \sim p}[-\log \frac{q(x)}{p(x)}]$ in the last step.

Now we can combine these three terms (a)(b)(c) together and average them with respect to $k$. Then we have

$$\mathcal{V}(\pi_{\theta^*}) - \frac{1}{K} \sum_{k=0}^{K-1} \mathcal{V}(\pi_{\theta^{(k)}})$$

$$= \frac{1}{K} \sum_{k=0}^{K-1} (\mathcal{V}(\pi_{\theta^*}) - \mathcal{V}(\pi_{\theta^{(k)}}))$$

$$\leq \varepsilon_{approx} + G \frac{1}{K} \sum_{k=0}^{K-1} \sum_{t=1}^T \sqrt{d_{\Theta_t}} \|(w_{*,t}^{(k)} - w_t^{(k)})\|_{\ell^2}$$

$$+ \underbrace{\frac{1}{\eta} \frac{1}{K} \sum_{k=0}^{K-1} \sum_{t=1}^T \mathbb{E}^{\pi_{\theta^*}} \left[ \mathrm{KL}(\pi^*(\cdot \mid O_t, H_{t-1}) \| \pi_{\theta^{(k)}}(\cdot \mid O_t, H_{t-1})) - \mathrm{KL}(\pi^*(\cdot \mid O_t, H_{t-1}) \| \pi_{\theta^{(k+1)}}(\cdot \mid O_t, H_{t-1})) \right]}_{\mathrm{I}}$$

$$+ \frac{\eta}{2} \frac{1}{K} \sum_{k=0}^{K-1} \sum_{t=1}^T \beta \|w_t^{(k)}\|_{\ell^2}^2$$

$$(127)$$

where the third term can be simplified by changing the order of summation:

$$\mathrm{I}$$

$$= \frac{1}{\eta} \frac{1}{K} \sum_{k=0}^{K-1} \sum_{t=1}^T \mathbb{E}_{(o_t, h_{t-1}) \sim \mathbb{P}_t^{\pi_{\theta^*}}} \left[ \mathrm{KL}(\pi^*(\cdot \mid o_t, h_{t-1}) \| \pi_{\theta^{(k)}}(\cdot \mid o_t, h_{t-1})) \right.$$

$$\left. - \mathrm{KL}(\pi^*(\cdot \mid o_t, h_{t-1}) \| \pi_{\theta^{(k+1)}}(\cdot \mid o_t, h_{t-1})) \right]$$

$$= \frac{1}{\eta} \sum_{t=1}^T \frac{1}{K} \sum_{k=0}^{K-1} \mathbb{E}_{(o_t, h_{t-1}) \sim \mathbb{P}_t^{\pi_{\theta^*}}} \left[ \mathrm{KL}(\pi^*(\cdot \mid o_t, h_{t-1}) \| \pi_{\theta^{(k)}}(\cdot \mid o_t, h_{t-1})) \right.$$

$$\left. - \mathrm{KL}(\pi^*(\cdot \mid o_t, h_{t-1}) \| \pi_{\theta^{(k+1)}}(\cdot \mid o_t, h_{t-1})) \right]$$

$$= \frac{1}{K\eta} \sum_{t=1}^{T} \mathbb{E}_{(o_t, h_{t-1}) \sim \mathbb{P}_t^{\pi_{\theta^*}}} \left[ \mathrm{KL}(\pi^*(\cdot \mid o_t, h_{t-1}) \| \pi_{\theta^{(0)}}(\cdot \mid o_t, h_{t-1})) \right] \text{ (by telescoping)}.$$

In summary, we have

$$\mathcal{V}(\pi_{\theta^*}) - \frac{1}{K} \sum_{k=0}^{K-1} \mathcal{V}(\pi_{\theta^{(k)}})$$

$$\leq \varepsilon_{approx} + G \frac{1}{K} \sum_{k=0}^{K-1} \sum_{t=1}^{T} \sqrt{d_{\Theta_t}} \|(w_{*,t}^{(k)} - w_t^{(k)})\|_{\ell^2} + \frac{\beta\eta}{2} \frac{1}{K} \sum_{k=0}^{K-1} \|w^{(k)}\|_{\ell^2}^2$$

$$+ \frac{1}{K\eta} \sum_{t=1}^{T} \mathbb{E}_{(o_t, h_{t-1}) \sim \mathbb{P}_t^{\pi_{\theta^*}}} \left[ \mathrm{KL}(\pi^*(\cdot \mid o_t, h_{t-1}) \| \pi_{\theta^{(0)}}(\cdot \mid o_t, h_{t-1})) \right].$$

$\square$

## G.2 PROOF OF SUBOPTIMALITY

In this section, we present a proof for Theorem 3 by leveraging Proposition 1 and Theorem 2.

For clarity, we let let $\widehat{E}_k := \nabla_{\theta^{(k)}} \mathcal{V}(\pi_{\theta^{(k)}}) - \nabla_{\theta^{(k)}} \widehat{\mathcal{V}(\pi_{\theta^{(k)}})}$ where $\nabla_{\theta^{(k)}} \mathcal{V}(\pi_{\theta^{(k)}})$ denotes the oracle true policy gradient evaluated at $\theta^{(k)}$ and $\nabla_{\theta^{(k)}} \widehat{\mathcal{V}(\pi_{\theta^{(k)}})}$ represents the estimated one from the min-max estimation procedure (9).

By setting $w_t^{(k)}$ as $\nabla_{\theta_t^{(k)}} \widehat{\mathcal{V}(\pi_{\theta_t^{(k)}})})$ in Proposition 1, we have

$$\mathcal{V}(\pi_{\theta^*}) - \frac{1}{K} \sum_{k=0}^{K-1} \mathcal{V}(\pi_{\theta^{(k)}})$$

$$\leq \varepsilon_{approx} + G \underbrace{\frac{1}{K} \sum_{k=0}^{K-1} \sum_{t=1}^{T} \sqrt{d_{\Theta_t}} \|(w_{*,t}^{(k)}(\theta^{(k)}) - \nabla_{\theta_t^{(k)}} \widehat{\mathcal{V}(\pi_{\theta_t^{(k)}})}))\|_{\ell^2}}_{(a)} + \frac{\beta\eta}{2} \underbrace{\frac{1}{K} \sum_{k=0}^{K-1} \|\nabla_{\theta^{(k)}} \widehat{\mathcal{V}(\pi_{\theta^{(k)}})}\|_{\ell^2}^2}_{(b)}$$

$$+ \underbrace{\frac{1}{K\eta} \sum_{t=1}^{T} \mathbb{E}_{(o_t, h_{t-1}) \sim \mathbb{P}_t^{\pi_{\theta^*}}} \left[ \mathrm{KL}(\pi^*(\cdot \mid o_t, h_{t-1}) \| \pi_{\theta^{(0)}}(\cdot \mid o_t, h_{t-1})) \right]}_{(c)}.$$

$$(128)$$

We first upper bound $(a) = \frac{G}{K} \sum_{k=0}^{K-1} \sum_{t=1}^{T} \sqrt{d_{\Theta_t}} \|(w_{*,t}^{(k)}(\theta^{(k)}) - \nabla_{\theta_t^{(k)}} \widehat{\mathcal{V}(\pi_{\theta_t^{(k)}})}))\|_{\ell^2}$. To this end, we have

$$(a)$$

$$= G \frac{1}{K} \sum_{k=0}^{K-1} \sum_{t=1}^{T} \sqrt{d_{\Theta_t}} \|(w_{*,t}^{(k)}(\theta^{(k)}) - \nabla_{\theta_t^{(k)}} \widehat{\mathcal{V}(\pi_{\theta_t^{(k)}})}))\|_{\ell^2}$$

$$\leq G \frac{\max_{t=1:T} \sqrt{d_{\Theta_t}}}{K} \sum_{k=0}^{K-1} \sum_{t=1}^{T} \|(w_{*,t}^{(k)}(\theta^{(k)}) - \nabla_{\theta_t^{(k)}} \widehat{\mathcal{V}(\pi_{\theta_t^{(k)}})}))\|_{\ell^2}$$

$$\leq G \frac{\max_{t=1:T} \sqrt{d_{\Theta_t}}}{K} \sum_{k=0}^{K-1} \sum_{t=1}^{T} \|\nabla_{\theta_t^{(k)}} \mathcal{V}(\pi_{\theta_t^{(k)}}) - \nabla_{\theta_t^{(k)}} \widehat{\mathcal{V}(\pi_{\theta_t^{(k)}})}))\|_{\ell^2}$$

$$+ G \frac{\max_{t=1:T} \sqrt{d_{\Theta_t}}}{K} \sum_{k=0}^{K-1} \|(w_{*,t}^{(k)}(\theta^{(k)}) - \nabla_{\theta_t^{(k)}} \mathcal{V}(\pi_{\theta_t^{(k)}}))\|_{\ell^2} \tag{129}$$

$$\leq G \frac{\max_{t=1:T} \sqrt{d_{\Theta_t}}}{K} \sqrt{T} \sum_{k=0}^{K-1} \|\nabla_{\theta^{(k)}} \mathcal{V}(\pi_{\theta^{(k)}}) - \widehat{\nabla_{\theta^{(k)}} \mathcal{V}(\pi_{\theta^{(k)}}}))\|_{\ell^2}$$

$$+ G \frac{\max_{t=1:T} \sqrt{d_{\Theta_t}}}{K} \sum_{k=0}^{K-1} \sum_{t=1}^{T} \|(w_{*,t}^{(k)}(\theta^{(k)}) - \nabla_{\theta_t^{(k)}} \mathcal{V}(\pi_{\theta_t^{(k)}}))\|_{\ell^2}$$

$$= \underbrace{G \frac{\max_{t=1:T} \sqrt{d_{\Theta_t}}}{K} \sqrt{T} \sum_{k=0}^{K-1} \|\widehat{E}_k\|_{\ell^2}}_{(a.1)} + \underbrace{G \frac{\max_{t=1:T} \sqrt{d_{\Theta_t}}}{K} \sum_{k=0}^{K-1} \sum_{t=1}^{T} \|(w_{*,t}^{(k)}(\theta^{(k)}) - \nabla_{\theta_t^{(k)}} \mathcal{V}(\pi_{\theta_t^{(k)}}))\|_{\ell^2}}_{(a.2)} .$$

We note that $(a.1)$ is a term caused by estimation error. Now we focus on $(a.2)$. Recall that $w_{*,t}^{(k)}(\theta^{(k)})$ is defined in Definition D.2, which is the minimizer of the loss function related to compatible function approximation. Here we provide an explicit expression for $w_{*,t}^{(k)}(\theta^{(k)})$.

Given a parameter $\theta$, we notice that minimizing with respect to $w = vec(w_1, ..., w_T)$

$$\sum_{t=1}^{T} \left( \mathbb{E}_{(a_t,o_t,s_t,h_{t-1}) \sim \mathbb{P}_t^{\pi_\theta}} \left[ \left( A_t^{\pi_\theta}(a_t, o_t, s_t, h_{t-1}) - w_t^\top \nabla_{\theta_t} \log \pi_{\theta_t}(a_t \mid o_t, h_{t-1}))^2 \right] \right)^{\frac{1}{2}}$$

is equivalent to minimizing

$$\mathbb{E}_{(a_t,o_t,s_t,h_{t-1}) \sim \mathbb{P}_t^{\pi_\theta}} \left[ \left( A_t^{\pi_\theta}(a_t, o_t, s_t, h_{t-1}) - w_t^\top \nabla_{\theta_t} \log \pi_{\theta_t}(a_t \mid o_t, h_{t-1}))^2 \right] \right]$$

for each $t = 1, ..., T$ by concatenating all the $w_{*,t}(\theta)$ into a vector $w_*(\theta) = vec(w_{*,1}(\theta), ..., w_{*,T}(\theta))$ with dimension $d_\Theta$. Since it is a convex quadratic optimization, we can get $w_{*,t}(\theta)$ by directly setting the derivative to be zero. By calculating the derivative of the loss function at $t$ with respect to $w_t$ and set it as zero, we get

$$\mathbb{E}_{(a_t,o_t,s_t,h_{t-1}) \sim \mathbb{P}_t^{\pi_\theta}} \left[ A_t^{\pi_\theta}(a_t, o_t, s_t, h_{t-1}) \nabla_{\theta_t} \log \pi_{\theta_t}(a_t \mid o_t, h_{t-1}) \right]$$
$$= \mathbb{E}_{(a_t,o_t,s_t,h_{t-1}) \sim \mathbb{P}_t^{\pi_\theta}} \left[ \nabla_{\theta_t} \log \pi_{\theta_t}(a_t \mid o_t, h_{t-1}) \nabla_{\theta_t} \log \pi_{\theta_t}(a_t \mid o_t, h_{t-1})^\top \right] w_t$$
$$= \mathbb{E}_{(a_t,o_t,s_t,h_{t-1}) \sim \mathbb{P}_t^{\pi_\theta}} \left[ \nabla_{\theta_t} \log \pi_{\theta_t}(a_t \mid o_t, h_{t-1}) \nabla_{\theta_t} \log \pi_{\theta_t}(a_t \mid o_t, h_{t-1})^\top \right] w_t \tag{130}$$
$$= F_t(\theta) w_t \text{ (by Assumption 5(c))}.$$

In addition, by the expression of policy gradient from Lemma 10, we have

$$\nabla_\theta \mathcal{V}(\pi_\theta) = \sum_{t=1}^{T} \mathbb{E}^{\pi_\theta} \left[ \nabla_\theta \log \pi_{\theta_t}(A_t \mid O_t, H_{t-1}) Q_t^{\pi_\theta}(A_t, O_t, S_t, H_{t-1}) \right]. \tag{131}$$

Since $\sum_{a \in \mathcal{A}} \pi_{\theta_t}(a \mid o_t, h_{t-1}) = 1$, we can also show that

$$\nabla_\theta \mathcal{V}(\pi_\theta) = \sum_{t=1}^{T} \mathbb{E}^{\pi_\theta} \left[ \nabla_\theta \log \pi_{\theta_t}(A_t \mid O_t, H_{t-1}) A_t^{\pi_\theta}(A_t, O_t, S_t, H_{t-1}) \right]. \tag{132}$$

by simple calculation and expressing $A_t = Q_t - V_t$ where $V_t$ does not depend on $a_t$. Therefore, we have

$$\nabla_{\theta_t} \mathcal{V}(\pi_\theta) = \sum_{t=1}^{T} \mathbb{E}^{\pi_\theta} \left[ \nabla_{\theta_t} \log \pi_{\theta_t}(A_t \mid O_t, H_{t-1}) A_t^{\pi_\theta}(A_t, O_t, S_t, H_{t-1}) \right] \tag{133}$$

for each $t = 1, ..., T$.

Plugging equation 133 into (130) and we have

$$\nabla_{\theta_t} \mathcal{V}(\pi_\theta) = F_t(\theta) w_t. \tag{134}$$

Therefore $w_{*,t}^{(k)}(\theta^{(k)}) = F_t^{-1}(\theta^{(k)}) \nabla_{\theta_t^{(k)}} \mathcal{V}(\pi_{\theta^{(k)}})$ for each $t = 1, ..., T$, where the non-singularity of $F_t(\theta^{(k)})$ is implied by Assumption 5(c). This is actually a direction for natural policy gradient (Agarwal et al. (2021)).

Consequently, we have

$$
\begin{aligned}
&\|(w_{*,t}^{(k)}(\theta^{(k)}) - \nabla_{\theta_t^{(k)}} \mathcal{V}(\pi_{\theta_t^{(k)}})\|_{\ell^2} \\
=&\|F_t^{-1}(\theta^{(k)}) \nabla_{\theta_t^{(k)}} \mathcal{V}(\pi_{\theta^{(k)}}) - \nabla_{\theta_t^{(k)}} \mathcal{V}(\pi_{\theta_t^{(k)}})\|_{\ell^2} \\
=&\|(F_t^{-1}(\theta^{(k)}) - I) \nabla_{\theta_t^{(k)}} \mathcal{V}(\pi_{\theta^{(k)}})\|_{\ell^2} \\
\leq&\|(F_t^{-1}(\theta^{(k)}) - I)\|_{2,2} \|\nabla_{\theta_t^{(k)}} \mathcal{V}(\pi_{\theta^{(k)}})\|_{\ell^2} \ (\|\|_{2,2} \text{ denotes matrix 2-norm}) \\
\leq&(\|F_t^{-1}(\theta^{(k)})\|_{2,2} + \|I\|_{2,2}) \|\nabla_{\theta_t^{(k)}} \mathcal{V}(\pi_{\theta^{(k)}})\|_{\ell^2} \\
\leq&(1 + \frac{1}{\mu}) \|\nabla_{\theta_t^{(k)}} \mathcal{V}(\pi_{\theta^{(k)}})\|_{\ell^2} \text{ by Assumption 5(c).}
\end{aligned}
\tag{135}
$$

Plug equation 135 into (129) and we have

$$
\begin{aligned}
&(a) \\
\leq&(a.1) + (a.2) \\
=&G \frac{\max_{t=1:T} \sqrt{d_{\Theta_t}}}{K} \sqrt{T} \sum_{k=0}^{K-1} \|\widehat{E}_k\|_{\ell^2} + G \frac{\max_{t=1:T} \sqrt{d_{\Theta_t}}}{K} \sum_{k=0}^{K-1} \sum_{t=1}^{T} \|(w_{*,t}^{(k)}(\theta^{(k)}) - \nabla_{\theta_t^{(k)}} \mathcal{V}(\pi_{\theta_t^{(k)}})\|_{\ell^2} \\
\leq&G \frac{\max_{t=1:T} \sqrt{d_{\Theta_t}}}{K} \sqrt{T} \sum_{k=0}^{K-1} \|\widehat{E}_k\|_{\ell^2} + G \frac{\max_{t=1:T} \sqrt{d_{\Theta_t}}}{K} \sum_{k=0}^{K-1} \sum_{t=1}^{T} (1 + \frac{1}{\mu}) \|\nabla_{\theta_t^{(k)}} \mathcal{V}(\pi_{\theta^{(k)}})\|_{\ell^2} \\
\leq&G \frac{\max_{t=1:T} \sqrt{d_{\Theta_t}}}{K} \sqrt{T} \sum_{k=0}^{K-1} \|\widehat{E}_k\|_{\ell^2} + G \frac{\max_{t=1:T} \sqrt{d_{\Theta_t}}}{K} (1 + \frac{1}{\mu}) \sum_{k=0}^{K-1} \sum_{t=1}^{T} \|\nabla_{\theta_t^{(k)}} \mathcal{V}(\pi_{\theta^{(k)}})\|_{\ell^2} \\
\leq&G \frac{\max_{t=1:T} \sqrt{d_{\Theta_t}}}{K} \sqrt{T} \sum_{k=0}^{K-1} \|\widehat{E}_k\|_{\ell^2} + G \frac{\max_{t=1:T} \sqrt{d_{\Theta_t}}}{K} (1 + \frac{1}{\mu}) \sqrt{T} \sum_{k=0}^{K-1} \|\nabla_{\theta^{(k)}} \mathcal{V}(\pi_{\theta^{(k)}})\|_{\ell^2}.
\end{aligned}
\tag{136}
$$

Now we need to upper bound $\frac{1}{K} \sum_{k=0}^{K-1} \|\nabla_{\theta^{(k)}} \mathcal{V}(\pi_{\theta^{(k)}})\|_{\ell^2}$. We will relate the gradient with the improvement in each step. Since $\mathcal{V}(\pi_\theta)$ is assumed to be $L$-smooth w.r.t. $\theta$, then according to lemma 7, we have

$$
\begin{aligned}
&\mathcal{V}(\pi_{\theta^{(k+1)}}) - \mathcal{V}(\pi_{\theta^{(k)}}) \\
\geq&\langle \nabla_{\theta^{(k)}} \mathcal{V}(\pi_{\theta^{(k)}}), \theta^{(k+1)} - \theta^{(k)} \rangle - \frac{L}{2} \|\theta^{(k+1)} - \theta^{(k)}\|_2^2 \\
=&\eta \langle \nabla_{\theta^{(k)}} \mathcal{V}(\pi_{\theta^{(k)}}), \nabla_{\theta^{(k)}} \widehat{\mathcal{V}(\pi_{\theta^{(k)}})} \rangle - \frac{\eta^2 L}{2} \|\nabla_{\theta^{(k)}} \widehat{\mathcal{V}(\pi_{\theta^{(k)}})}\|_2^2 \\
=&\eta \langle \nabla_{\theta^{(k)}} \mathcal{V}(\pi_{\theta^{(k)}}), \nabla_{\theta^{(k)}} \mathcal{V}(\pi_{\theta^{(k)}}) \rangle + \eta \langle \nabla_{\theta^{(k)}} \mathcal{V}(\pi_{\theta^{(k)}}), \nabla_{\theta^{(k)}} \widehat{\mathcal{V}(\pi_{\theta^{(k)}})} - \nabla_{\theta^{(k)}} \mathcal{V}(\pi_{\theta^{(k)}}) \rangle \\
&- \frac{\eta^2 L}{2} \|\nabla_{\theta^{(k)}} \widehat{\mathcal{V}(\pi_{\theta^{(k)}})} - \nabla_{\theta^{(k)}} \mathcal{V}(\pi_{\theta^{(k)}}) + \nabla_{\theta^{(k)}} \mathcal{V}(\pi_{\theta^{(k)}})\|_2^2 \\
\geq&\eta \|\nabla_{\theta^{(k)}} \mathcal{V}(\pi_{\theta^{(k)}})\|^2 - \frac{1}{2} \eta (\|\nabla_{\theta^{(k)}} \mathcal{V}(\pi_{\theta^{(k)}})\|^2 + \|\widehat{E}_k\|^2) - \frac{\eta^2 L}{2} (2\|\widehat{E}_k\|^2 + 2\|\nabla_{\theta^{(k)}} \mathcal{V}(\pi_{\theta^{(k)}})\|^2) \\
=&\eta \|\nabla_{\theta^{(k)}} \mathcal{V}(\pi_{\theta^{(k)}})\|^2 - \frac{1}{2} \eta (\|\nabla_{\theta^{(k)}} \mathcal{V}(\pi_{\theta^{(k)}})\|^2 + \|\widehat{E}_k\|^2) - \frac{\eta^2 L}{2} (2\|\widehat{E}_k\|^2 + 2\|\nabla_{\theta^{(k)}} \mathcal{V}(\pi_{\theta^{(k)}})\|^2) \\
=&(\frac{\eta}{2} - L\eta^2) \|\nabla_{\theta^{(k)}} \mathcal{V}(\pi_{\theta^{(k)}})\|^2 - (\frac{\eta}{2} + L\eta^2) \|\widehat{E}_k\|^2.
\end{aligned}
\tag{137}
$$

Therefore we have

$$(\frac{\eta}{2} - L\eta^2)\frac{1}{K} \sum_{k=0}^{K-1} \|\nabla_{\theta^{(k)}} \mathcal{V}(\pi_{\theta^{(k)}})\|^2$$

$$\leq \frac{1}{K} \sum_{k=0}^{K-1} (\mathcal{V}(\pi_{\theta^{(k+1)}}) - \mathcal{V}(\pi_{\theta^{(k)}})) + (\frac{\eta}{2} + L\eta^2)\frac{1}{K} \sum_{k=0}^{K-1} \|\widehat{E}_k\|^2 \tag{138}$$

$$= \frac{1}{K} (\mathcal{V}(\pi_{\theta^{(K)}}) - \mathcal{V}(\pi_{\theta^{(0)}})) + (\frac{\eta}{2} + L\eta^2)\frac{1}{K} \sum_{k=0}^{K-1} \|\widehat{E}_k\|^2.$$

Let $\eta = \frac{1}{4L}$, we have

$$\frac{1}{K} \sum_{k=0}^{K-1} \|\nabla_{\theta^{(k)}} \mathcal{V}(\pi_{\theta^{(k)}})\|^2 \leq \frac{16L}{K} (\mathcal{V}(\pi_{\theta^{(K)}}) - \mathcal{V}(\pi_{\theta^{(0)}})) + \frac{3}{K} \sum_{k=0}^{K-1} \|\widehat{E}_k\|^2. \tag{139}$$

Therefore, the following holds

$$\frac{1}{K} \sum_{k=0}^{K-1} \|\nabla_{\theta^{(k)}} \mathcal{V}(\pi_{\theta^{(k)}})\|$$

$$\leq \frac{1}{K} (\sum_{k=0}^{K-1} \|\nabla_{\theta^{(k)}} \mathcal{V}(\pi_{\theta^{(k)}})\|^2)^{\frac{1}{2}} \sqrt{K} \tag{140}$$

$$\leq \frac{1}{\sqrt{K}} \left( 16L(\mathcal{V}(\pi_{\theta^{(K)}}) - \mathcal{V}(\pi_{\theta^{(0)}})) + 3 \sum_{k=0}^{K-1} \|\widehat{E}_k\|^2 \right).$$

Plug equation 140 into equation 136 and we have

$$(a)$$

$$\leq G\frac{\max_{t=1:T} \sqrt{d_{\Theta_t}}}{K} \sqrt{T} \sum_{k=0}^{K-1} \|\widehat{E}_k\|_{\ell^2} + G\frac{\max_{t=1:T} \sqrt{d_{\Theta_t}}}{K}(1 + \frac{1}{\mu})\sqrt{T} \sum_{k=0}^{K-1} \|\nabla_{\theta^{(k)}} \mathcal{V}(\pi_{\theta^{(k)}})\|_{\ell^2}$$

$$\leq G\frac{\max_{t=1:T} \sqrt{d_{\Theta_t}}}{K} \sqrt{T} \sum_{k=0}^{K-1} \|\widehat{E}_k\|_{\ell^2}$$

$$+ G \max_{t=1:T} \sqrt{d_{\Theta_t}}(1 + \frac{1}{\mu})\sqrt{T} \frac{1}{\sqrt{K}} \left( 16L(\mathcal{V}(\pi_{\theta^{(K)}}) - \mathcal{V}(\pi_{\theta^{(0)}})) + 3 \sum_{k=0}^{K-1} \|\widehat{E}_k\|^2 \right) \tag{141}$$

We also have

$$(b)$$

$$= \frac{\beta\eta}{2} \frac{1}{K} \sum_{k=0}^{K-1} \|\nabla_{\theta^{(k)}} \widehat{\mathcal{V}(\pi_{\theta^{(k)}})}\|^2$$

$$= \frac{\beta\eta}{2} \frac{1}{K} \sum_{k=0}^{K-1} \|\nabla_{\theta^{(k)}} \widehat{\mathcal{V}(\pi_{\theta^{(k)}})} - \nabla_{\theta^{(k)}} \mathcal{V}(\pi_{\theta^{(k)}}) + \nabla_{\theta^{(k)}} \mathcal{V}(\pi_{\theta^{(k)}})\|^2 \tag{142}$$

$$\leq \frac{\beta\eta}{2} \frac{1}{K} \sum_{k=0}^{K-1} (2\|\nabla_{\theta^{(k)}} \widehat{\mathcal{V}(\pi_{\theta^{(k)}})} - \nabla_{\theta^{(k)}} \mathcal{V}(\pi_{\theta^{(k)}})\|^2 + 2\|\nabla_{\theta^{(k)}} \mathcal{V}(\pi_{\theta^{(k)}})\|^2)$$

$$\leq \frac{\beta\eta}{2} \frac{32L}{K} (\mathcal{V}(\pi_{\theta^{(K)}}) - \mathcal{V}(\pi_{\theta^{(0)}})) + \frac{\beta\eta}{2} \frac{8}{K} \sum_{k=0}^{K-1} \|\widehat{E}_k\|^2.$$

where the last inequality is from equation (139).

Combining all the results, we have

$$\mathcal{V}(\pi_{\theta^*}) - \max_{k=0:K-1} \mathcal{V}(\pi_{\theta^{(k)}})$$

$$\leq \mathcal{V}(\pi_{\theta^*}) - \frac{1}{K} \sum_{k=0}^{K-1} \mathcal{V}(\pi_{\theta^{(k)}})$$

$$\leq \varepsilon_{approx} + (a) + (b) + (c)$$

$$= \varepsilon_{approx} + G \frac{1}{K} \sum_{k=0}^{K-1} \sum_{t=1}^{T} \sqrt{d_{\Theta_t}} \|(w_{*,t}^{(k)}(\theta^{(k)}) - \nabla_{\theta_t^{(k)}} \widehat{\mathcal{V}(\pi_{\theta_t^{(k)}})})\|_{\ell^2} + \frac{\beta\eta}{2} \frac{1}{K} \sum_{k=0}^{K-1} \|\nabla_{\theta^{(k)}} \widehat{\mathcal{V}(\pi_{\theta^{(k)}})}\|_{\ell^2}^2$$

$$+ \frac{1}{K\eta} \sum_{t=1}^{T} \mathbb{E}_{(o_t, h_{t-1}) \sim \mathbb{P}_t^{\pi_{\theta^*}}} [\mathrm{KL}(\pi^*(\cdot \mid o_t, h_{t-1}) \| \pi_{\theta^{(0)}}(\cdot \mid o_t, h_{t-1}))]$$

$$\leq \varepsilon_{approx} + \frac{1}{K\eta} \sum_{t=1}^{T} \mathbb{E}_{(o_t, h_{t-1}) \sim \mathbb{P}_t^{\pi_{\theta^*}}} [\mathrm{KL}(\pi^*(\cdot \mid o_t, h_{t-1}) \| \pi_{\theta^{(0)}}(\cdot \mid o_t, h_{t-1}))]$$

$$+ G \frac{\max_{t=1:T} \sqrt{d_{\Theta_t}}}{K} \sqrt{T} \sum_{k=0}^{K-1} \|\widehat{E}_k\|_{\ell^2}$$

$$+ \underbrace{G \max_{t=1:T} \sqrt{d_{\Theta_t}} (1 + \frac{1}{\mu}) \sqrt{T} \frac{1}{\sqrt{K}} \left( 16L(\mathcal{V}(\pi_{\theta^{(K)}}) - \mathcal{V}(\pi_{\theta^{(0)}})) + 3 \sum_{k=0}^{K-1} \|\widehat{E}_k\|^2 \right)}_{(*)} \text{ by (141) for (a)}$$

$$+ \frac{\beta\eta}{2} \left( \frac{32L}{K} (\mathcal{V}(\pi_{\theta^{(K)}}) - \mathcal{V}(\pi_{\theta^{(0)}})) + \frac{8}{K} \sum_{k=0}^{K-1} \|\widehat{E}_k\|^2 \right) \text{ by (142) for (b)} \tag{143}$$

We note that equation 143 provides an upper bound in terms of all key parameters. In addition, $(*)$ is the dominating term. We can now employ Theorem 2 for explicitly quantifying $\|\widehat{E}_k\|$. According to Theorem 2, we have that with probability at least $1 - \zeta$, it holds that

$$\sup_{k=0:K-1} \|\widehat{E}_k\| \lesssim \tau_{\max} C_{\pi^b} T^{\frac{5}{2}} M_{\mathcal{B}} M_{\mathcal{F}} d_{\Theta} \sqrt{\frac{\log(T/\zeta) \gamma(\mathcal{F}, \mathcal{B})}{N}}. \tag{144}$$

For simplicity, we can drop the terms $\gamma(\mathcal{F}, \mathcal{B})$ and $M_{\mathcal{F}}$, and get

$$\sup_{k=0:K-1} \|\widehat{E}_k\| \lesssim \tau_{\max} C_{\pi^b} T^{\frac{5}{2}} M_{\mathcal{B}} d_{\Theta} \sqrt{\frac{\log(T/\zeta)}{N}} \tag{145}$$

with probability at least $1 - \zeta$.

Then, we have that with probability at least $1 - \zeta$

$$\mathcal{V}(\pi_{\theta^*}) - \mathcal{V}(\widehat{\pi})$$

$$\lesssim (*) + \varepsilon_{approx}$$

$$\lesssim \max_{t=1:T} \sqrt{d_{\Theta_t}} (1 + \frac{1}{\mu}) \sqrt{T} \frac{L}{\sqrt{K}} + \max_{t=1:T} \sqrt{d_{\Theta_t}} (1 + \frac{1}{\mu}) \sqrt{T} \frac{1}{\sqrt{K}}$$

$$K \left( \tau_{\max} C_{\pi^b} T^{\frac{5}{2}} M_{\mathcal{B}} d_{\Theta} \sqrt{\frac{\log(T/\zeta)}{N}} \right)^2 + \varepsilon_{approx} \tag{146}$$

$$\lesssim \sqrt{d_{\Theta}} \sqrt{T} \frac{L}{\sqrt{K}} + \sqrt{T} \sqrt{K} \tau_{\max}^2 C_{\pi^b}^2 T^5 M_{\mathcal{B}}^2 d_{\Theta}^{2.5} \frac{\log(T/\zeta)}{N} + \varepsilon_{approx}$$

$$\lesssim \sqrt{d_{\Theta}} \sqrt{T} \frac{L}{\sqrt{K}} + \tau_{\max}^2 C_{\pi^b}^2 T^{5.5} M_{\mathcal{B}}^2 d_{\Theta}^{2.5} \frac{\log(T/\zeta)}{\sqrt{N}} + \varepsilon_{approx}$$

where the last inequality holds if $K \asymp N$.

In particular, we have $L \asymp T^4$ and $M_{\mathcal{B}} \asymp T^2$ according to Appendix B.2, B.3. Consequently, with probability at least $1 - \zeta$, we have

$$\mathcal{V}(\pi_{\theta^*}) - \mathcal{V}(\widehat{\pi}) \lesssim \quad \sqrt{d_\Theta} \frac{T^{4.5}}{\sqrt{K}} + \tau_{\max}^2 C_{\pi^b}^2 T^{9.5} d_\Theta^{2.5} \frac{\log(T/\zeta)}{\sqrt{N}} + \varepsilon_{approx}. \tag{147}$$

Remark: we can consider specific examples such as VC subgraphs and RKHSs for analyzing $\sup_{k=0:K-1} \|\widehat{E}_k\|$. See Appendix C for details.

## H   PROOFS FOR LEMMAS

### H.1   PROOF OF LEMMA 3

Let $\tau_{i:j}$ denote a trajectory from $i$ to $j$, i.e., $\tau_{i:j} := (S_i, A_i, O_i, R_i, ..., S_j, A_j, O_j, R_j)$. Then we have

$$
\begin{aligned}
&\nabla_\theta \mathcal{V}(\pi_\theta) \\
=&\nabla_\theta \mathbb{E}^{\pi_\theta} \left[ \sum_{j=1}^T R_j \right] \\
=&\nabla_\theta \int (\sum_{t=1}^T R_t) dP^{\pi_\theta}(\tau_{1:T}) \\
=&\nabla_\theta \sum_{t=1}^T \int R_t dP^{\pi_\theta}(\tau_{1:T}) \\
=&\nabla_\theta \sum_{t=1}^T \int R_t dP^{\pi_\theta}(\tau_{1:t}) \\
=&\sum_{t=1}^T \int R_t \nabla_\theta dP^{\pi_\theta}(\tau_{1:t}) \\
=&\sum_{t=1}^T \int R_t \nabla_\theta \log P^{\pi_\theta}(\tau_{1:t}) dP^{\pi_\theta}(\tau_{1:t}) \\
=&\sum_{t=1}^T \int R_t \sum_{j=1}^t \nabla_\theta \log \pi_{\theta_j}(A_j \mid O_j, H_{j-1}) dP^{\pi_\theta}(\tau_{1:t}) \text{ by equation 148} \\
=&\mathbb{E}^{\pi_\theta} \left[ \sum_{t=1}^T R_t \sum_{j=1}^t \nabla_\theta \log \pi_{\theta_j}(A_j \mid O_j, H_{j-1}) \right]
\end{aligned}
$$

where

$$
\begin{aligned}
&\nabla_\theta \log p_{\pi_\theta}(\tau_{1:t}) \\
=&\nabla_\theta \log \prod_{i=1}^t p(R_i \mid S_i, A_i) \pi_{\theta_i}(A_i \mid O_i, H_{i-1}) Z(O_i \mid S_i) p(S_i \mid S_{i-1}, A_{i-1}) \\
=&\sum_{i=1}^t \nabla_\theta \log \pi_{\theta_i}(A_i \mid O_i, H_{i-1})
\end{aligned}
\tag{148}
$$

with $p(S_1 \mid S_0, A_0) := \nu_1(S_1)$.

## H.2 Proof of Lemma 5

We provide a sketch of proof here. Readers can refer to Wainwright (2019); Maurer (2016); Foster & Syrgkanis (2019)for more details.

Define the following function indexed by $r \in (0, 1]$:

$$Z_n(r) := \sup_{f \in \mathbb{B}_2(r;\mathcal{F})} \left| \|f\|_{2,2}^2 - \|f\|_{n,2,2}^2 \right|, \quad \text{where } \mathbb{B}_2(r;\mathcal{F}) = \{f \in \mathcal{F} \mid \|f\|_{2,2} \leq r\}.$$

Let $\mathcal{E}$ be the event that the inequality (23) is violated. We also define an auxiliary event $\mathcal{A}(r) := \{Z_n(r) \geq \frac{dr^2}{2}\}$. By a contraction argument from Maurer (2016) and the Talagrand's concentration inequality (Theorem 3.27 of Wainwright (2019)), we get the tail bound for $Z_n(r)$. Specifically, we have $P(Z_n(r) \geq \frac{1}{4}\delta_n^2 d + u) \leq c_1 \exp(-c_2 n u)$ for constants $c_1, c_2$. Finally, according to the proof of Lemma 14.8 in Wainwright (2019), we get that $\mathcal{E}$ is contained in $\mathcal{A}(r)$, which has exponentially small probability.

## H.3 Proof of Lemma 8

Recall that we defined

$$Q_j^{\pi_\theta}(s_j, a_j, o_j, h_{j-1}) = \mathbb{E}^{\pi_\theta}[\sum_{t=j}^T R_t \mid S_j = s_j, A_j = a_j, H_{j-1} = h_{j-1}, O_j = o_j]. \quad (149)$$

and

$$V_j^{\pi_\theta}(s_j, o_j, h_{j-1}) = \mathbb{E}^{\pi_\theta}[\sum_{t=j}^T R_t \mid S_j = s_j, H_{j-1} = h_{j-1}, O_j = o_j]. \quad (150)$$

Then we notice that

$$\begin{aligned} &V_j^{\pi_\theta}(s_j, o_j, h_{j-1}) \\ =&\mathbb{E}^{\pi_\theta}[\sum_{t=j}^T R_t \mid S_j = s_j, H_{j-1} = h_{j-1}, O_j = o_j] \\ =&\mathbb{E}^{\pi_\theta}[\mathbb{E}^{\pi_\theta}[\sum_{t=j}^T R_t \mid S_j = s_j, H_{j-1} = h_{j-1}, O_j = o_j, A_j] \mid S_j = s_j, H_{j-1} = h_{j-1}, O_j = o_j] \\ =&\mathbb{E}_{A_j \sim \pi_{\theta,j}(\cdot|o_j,h_{j-1})}[Q_j^{\pi_\theta}(s_j, A_j, o_j, h_{j-1})] \end{aligned}$$

$$(151)$$

and

$$\begin{aligned} &Q_j^{\pi_\theta}(S_j, A_j, O_j, H_{j-1}) \\ =&\mathbb{E}[R_j \mid S_j, A_j] + \mathbb{E}^{\pi_\theta}[\sum_{t=j+1}^T R_t \mid S_j, A_j, H_{j-1}, O_j] \\ =&\mathbb{E}[R_j \mid S_j, A_j] + \mathbb{E}^{\pi_\theta}[\mathbb{E}^{\pi_\theta}[\sum_{t=j+1}^T R_t \mid S_{j+1}, O_{j+1}, A_{j+1}, S_j, H_j] \mid S_j, A_j, H_{j-1}, O_j] \\ =&\mathbb{E}[R_j \mid S_j, A_j] + \mathbb{E}^{\pi_\theta}[\mathbb{E}^{\pi_\theta}[\sum_{t=j+1}^T R_t \mid S_{j+1}, O_{j+1}, A_{j+1}, H_j] \mid S_j, A_j, H_{j-1}, O_j] \\ =&\mathbb{E}[R_j \mid S_j, A_j] + \mathbb{E}^{\pi_\theta}\left[Q_{j+1}^{\pi_\theta}(S_{j+1}, O_{j+1}, A_{j+1}, H_j) \mid S_j, A_j, O_j, H_{j-1}\right] \\ =&\mathbb{E}[R_j \mid S_j, A_j] + \mathbb{E}[V_{j+1}^{\pi_\theta}(S_{j+1}, O_{j+1}, H_j) \mid S_j, A_j, O_j, H_{j-1}]. \end{aligned} \quad (152)$$

For clarity, we denote $\mathbb{E}[R_j \mid S_j = s, A_j = a]$ as $r_j(s, a)$. Then we have

$$
\begin{aligned}
&V_1^{\pi}(s_1, o_1) - V_1^{\pi'}(s_1, o_1) \\
=&V_1^{\pi}(s_1, o_1) - \mathbb{E}^{\pi}[r_1(s_1, A_1) + \mathbb{E}[V_2^{\pi'}(S_2, O_2, H_1) \mid A_1, s_1, o_1]] \\
&\quad + \mathbb{E}^{\pi}[r_1(s_1, A_1) + \mathbb{E}[V_2^{\pi'}(S_2, O_2, H_1) \mid A_1, s_1, o_1]] - V_1^{\pi'}(s_1, o_1) \\
=&\mathbb{E}^{\pi}[r_1(s_1, A_1) + \mathbb{E}[V_2^{\pi}(S_2, O_2, H_1) \mid A_1, s_1, o_1]] - \mathbb{E}^{\pi}[r_1(s_1, A_1) + \mathbb{E}[V_2^{\pi'}(S_2, O_2, H_1) \mid A_1, s_1, o_1]] \\
&\quad + \mathbb{E}^{\pi}[r_1(s_1, A_1) + \mathbb{E}[V_2^{\pi'}(S_2, O_2, H_1) \mid A_1, s_1, o_1]] - V_1^{\pi'}(s_1, o_1) \text{ by (151)(152)} \\
=&\mathbb{E}^{\pi}[\mathbb{E}[V_2^{\pi}(S_2, O_2, H_1) - V_2^{\pi'}(S_2, O_2, H_1) \mid A_1, s_1, o_1]] \\
&\quad + \mathbb{E}^{\pi}[r_1(s_1, A_1) + \mathbb{E}[V_2^{\pi'}(S_2, O_2, H_1) \mid A_1, s_1, o_1]] - V_1^{\pi'}(s_1, o_1) \\
=&\mathbb{E}^{\pi}[\mathbb{E}[V_2^{\pi}(S_2, O_2, H_1) - V_2^{\pi'}(S_2, O_2, H_1) \mid A_1, s_1, o_1]] + \mathbb{E}^{\pi}[Q_1^{\pi'}(A_1, s_1, o_1)] - V_1^{\pi'}(s_1, o_1) \text{ by (152)} \\
=&\mathbb{E}^{\pi}[\mathbb{E}[V_2^{\pi}(S_2, O_2, H_1) - V_2^{\pi'}(S_2, O_2, H_1) \mid A_1, s_1, o_1]] + \mathbb{E}^{\pi}[Q_1^{\pi'}(A_1, s_1, o_1) - V_1^{\pi'}(s_1, o_1)] \\
=&\mathbb{E}^{\pi}[\mathbb{E}[V_2^{\pi}(S_2, O_2, H_1) - V_2^{\pi'}(S_2, O_2, H_1) \mid A_1, s_1, o_1]] + \mathbb{E}^{\pi}[A_1^{\pi'}(A_1, s_1, o_1)] \text{ by Definition D.7} \\
=&\mathbb{E}^{\pi}[V_2^{\pi}(S_2, O_2, H_1) - V_2^{\pi'}(S_2, O_2, H_1) \mid s_1, o_1] + \mathbb{E}^{\pi}[A_1^{\pi'}(A_1, s_1, o_1)] \text{ by law of total expectation.}
\end{aligned}
\tag{153}
$$

Next, we consider $V_2^{\pi}(S_2, O_2, H_1) - V_2^{\pi'}(S_2, O_2, H_1)$.

$$
\begin{aligned}
&V_2^{\pi}(S_2, O_2, H_1) - V_2^{\pi'}(S_2, O_2, H_1) \\
=&V_2^{\pi}(S_2, O_2, H_1) - \mathbb{E}^{\pi}[r_2(S_2, A_2) + \mathbb{E}[V_3^{\pi'}(S_3, O_3, H_2) \mid A_2, S_2, O_2, H_1] \mid S_2, O_2, H_1] \\
&\quad + \mathbb{E}^{\pi}[r_2(S_2, A_2) + \mathbb{E}[V_3^{\pi'}(S_3, O_3, H_2) \mid A_2, S_2, O_2, H_1] \mid S_2, O_2, H_1] - V_2^{\pi'}(S_2, O_2, H_1) \\
=&\mathbb{E}^{\pi}[\mathbb{E}[V_3^{\pi}(S_3, O_3, H_2) - V_3^{\pi'}(S_3, O_3, H_2) \mid A_2, S_2, O_2, H_1] \mid S_2, O_2, H_1] \\
&\quad + \mathbb{E}^{\pi}[r_2(S_2, A_2) + \mathbb{E}[V_3^{\pi'}(S_3, O_3, H_2) \mid A_2, S_2, O_2, H_1] \mid S_2, O_2, H_1] - V_2^{\pi'}(S_2, O_2, H_1) \text{ by (151)(152)} \\
=&\mathbb{E}^{\pi}[\mathbb{E}[V_3^{\pi}(S_3, O_3, H_2) - V_3^{\pi'}(S_3, O_3, H_2) \mid A_2, S_2, O_2, H_1] \mid S_2, O_2, H_1] \\
&\quad + \mathbb{E}^{\pi}[Q_2^{\pi'}(A_2, S_2, O_2, H_1) \mid S_2, O_2, H_1] - V_2^{\pi'}(S_2, O_2, H_1) \text{ by (152)} \\
=&\mathbb{E}^{\pi}[V_3^{\pi}(S_3, O_3, H_2) - V_3^{\pi'}(S_3, O_3, H_2) \mid S_2, O_2, H_1] + \mathbb{E}^{\pi}[A_2^{\pi'}(A_2, S_2, O_2, H_1) \mid S_2, O_2, H_1]
\end{aligned}
\tag{154}
$$

Plug (154) into (153) and we have

$$
\begin{aligned}
&V_1^{\pi}(s_1, o_1) - V_1^{\pi'}(s_1, o_1) \\
=&\mathbb{E}^{\pi}[V_3^{\pi}(S_3, O_3, H_2) - V_3^{\pi'}(S_3, O_3, H_2) \mid s_1, o_1] + \mathbb{E}^{\pi}[A_2^{\pi'}(A_2, S_2, O_2, H_1) \mid s_1, o_1] + \mathbb{E}^{\pi}[A_1^{\pi'}(A_1, s_1, o_1)]
\end{aligned}
\tag{155}
$$

where we have used the fact that given $(S_2, O_2, H_1)$, $(S_3, O_3, H_2)$ is independent of $S_1, O_1$ and the law of total expectation.

Repeating this procedure and using $(S_{k+1}, O_{k+1}, H_k) \perp\!\!\!\perp (S_1, O_1) \mid (S_k, O_k, H_{k-1})$, we have

$$
V_1^{\pi}(s_1, o_1) - V_1^{\pi'}(s_1, o_1) = \sum_{t=1}^{T} \mathbb{E}^{\pi}[A_t^{\pi'}(A_t, S_t, O_t, H_{t-1}) \mid s_1, o_1].
\tag{156}
$$

By taking expectation with respect to $S_1, O_1$, we have

$$
\mathcal{V}(\pi) - \mathcal{V}(\pi') = \sum_{t=1}^{T} \mathbb{E}^{\pi}[A_t^{\pi'}(A_t, S_t, O_t, H_{t-1})].
\tag{157}
$$

## H.4   PROOF OF LEMMA 10

*Proof.* Let $\tau_{1:T}$ denotes the trajectory $(s_1, o_1, a_1, r_1, ..., s_T, o_T, a_T, r_T)$.

$$
\nabla_\theta \mathcal{V}(\pi_\theta)
$$

$$=\nabla_\theta \mathbb{E}^{\pi_\theta}\left[\sum_{j=1}^T R_j\right]$$

$$=\nabla_\theta \int (\sum_{t=1}^T R_t)dP^{\pi_\theta}(\tau_{1:T})$$

$$=\int (\sum_{t=1}^T R_t)\nabla_\theta \log P^{\pi_\theta}(\tau_{1:T})dP^{\pi_\theta}(\tau_{1:T})$$

$$=\mathbb{E}^{\pi_\theta}\left[\sum_{t=1}^T R_t \sum_{i=1}^t \nabla_\theta \log \pi_{\theta,i}(A_i \mid O_i, H_{i-1})\right] \text{ by Lemma 3}$$

$$=\mathbb{E}^{\pi_\theta}\left[\sum_{i=1}^T \nabla_\theta \log \pi_{\theta,i}(A_i \mid O_i, H_{i-1}) \sum_{t=i}^T R_t\right] \text{ by changing the order of summation for } t \text{ and } i$$

$$=\sum_{i=1}^T \mathbb{E}^{\pi_\theta}\left[\nabla_\theta \log \pi_{\theta,i}(A_i \mid O_i, H_{i-1}) \sum_{t=i}^T R_t\right]$$

$$=\sum_{i=1}^T \mathbb{E}^{\pi_\theta}\left[\mathbb{E}^{\pi_\theta}\left[\nabla_\theta \log \pi_{\theta,i}(A_i \mid O_i, H_{i-1}) \sum_{t=i}^T R_t \mid S_i, A_i, O_i, H_{i-1}\right]\right] \text{ by law of total expectation}$$

$$=\sum_{i=1}^T \mathbb{E}^{\pi_\theta}\left[\nabla_\theta \log \pi_{\theta,i}(A_i \mid O_i, H_{i-1})\mathbb{E}^{\pi_\theta}\left[\sum_{t=i}^T R_t \mid S_i, A_i, O_i, H_{i-1}\right]\right] \text{ by measurability}$$

$$=\sum_{i=1}^T \mathbb{E}^{\pi_\theta}\left[\nabla_\theta \log \pi_{\theta,i}(A_i \mid O_i, H_{i-1})Q_i^{\pi_\theta}(S_i, A_i, O_i, H_{i-1})\right] \text{ by definition of } Q_i^{\pi_\theta} \tag{158}$$

$$\square$$

### H.5 PROOF OF LEMMA 11

*proof of Lemma 11.* We prove it by induction. At the base step $t = T$, we have

$$\mathbb{E}^{\pi_\theta}[R_T \mid S_T, H_{T-1}]$$
$$=\mathbb{E}[\mathbb{E}^{\pi_\theta}[\mathbb{E}[R_T \mid S_T, H_{T-1}, O_T, A_T] \mid S_T, H_{T-1}, O_T] \mid S_T, H_{T-1}] \text{ by law of total expectation}$$

$$=\mathbb{E}^{\pi_\theta}\left[\mathbb{E}[\sum_a \mathbb{E}[R_T \mid S_T, H_{T-1}, O_T, A_T = a]\pi_{\theta_T}(a \mid O_T, H_{T-1}) \mid S_T, H_{T-1}, O_T] \mid S_T, H_{T-1}\right]$$

$$=\mathbb{E}\left[\sum_a \mathbb{E}[R_T \mid S_T, H_{T-1}, O_T, A_T = a]\pi_{\theta_T}(a \mid O_T, H_{T-1}) \mid S_T, H_{T-1}\right]$$

$$=\mathbb{E}\left[\sum_a \mathbb{E}[R_T \mid S_T, H_{T-1}, A_T = a]\pi_{\theta_T}(a \mid O_T, H_{T-1}) \mid S_T, H_{T-1}\right] \text{ by } R_T \perp\!\!\!\perp O_T \mid S_T, A_T, H_{T-1}$$

$$=\sum_a \mathbb{E}[R_T \mid S_T, H_{T-1}, A_T = a]\mathbb{E}[\pi_{\theta_T}(a \mid O_T, H_{T-1}) \mid S_T, H_{T-1}]$$

$$=\sum_a \mathbb{E}[R_T \mid S_T, H_{T-1}, A_T = a]\mathbb{E}[\pi_{\theta_T}(a \mid O_T, H_{T-1}) \mid S_T, H_{T-1}, A_T = a] \text{ by } O_T \perp\!\!\!\perp A_T \mid S_T, H_{T-1}$$

$$=\sum_a \mathbb{E}[R_T\pi_{\theta_T}(a \mid O_T, H_{T-1}) \mid S_T, H_{T-1}, A_T = a] \text{ by } O_T \perp\!\!\!\perp R_T \mid S_T, A_T, H_{T-1}$$

$$=\sum_a \mathbb{E}\left[b_{V,T}^{\pi_\theta}(O_T, H_{T-1}, a) \mid S_T, H_{T-1}, A_T = a\right] \text{ by (33)}$$

$$= \sum_a \mathbb{E}\left[ b_{V,T}^{\pi_\theta}(O_T, H_{T-1}, a) \mid S_T, H_{T-1} \right] \text{ by } O_T \perp\!\!\!\perp A_T \mid S_T, H_{T-1}$$

$$= \mathbb{E}\left[ \sum_a b_{V,T}^{\pi_\theta}(a, O_T, H_{T-1}) \mid S_T, H_{T-1} \right] \tag{159}$$

According to the above derivation, we have shown $\mathbb{E}\left[ \sum_a b_{V,j}^{\pi_\theta}(a, O_j, H_{j-1}) \mid S_j, H_{j-1} \right] = \mathbb{E}^{\pi_\theta}\left[ \sum_{t=j}^T R_t \mid S_j, H_{j-1} \right]$ when $j = T$. We proceed with the derivation by induction. Assume that $\mathbb{E}\left[ \sum_a b_{V,j}^{\pi_\theta}(a, O_j, H_{j-1}) \mid S_j, H_{j-1} \right] = \mathbb{E}^{\pi_\theta}\left[ \sum_{t=j}^T R_t \mid S_j, H_{j-1} \right]$ holds for $j = k+1$, we will show that it also holds for $j = k$.

For $j = k$, we first notice that

$$\mathbb{E}^{\pi_\theta}\left[ \sum_{t=k}^T R_t \mid S_k, H_{k-1} \right] = \mathbb{E}^{\pi_\theta}\left[ R_k \mid S_k, H_{k-1} \right] + \mathbb{E}^{\pi_\theta}\left[ \sum_{t=k+1}^T R_t \mid S_k, H_{k-1} \right].$$

Next, we analyze these two terms separately. Analyzing the first term is the same as $\mathbb{E}^{\pi_\theta}\left[ R_T \mid S_T, H_{T-1} \right]$ by replacing $T$ with $k$.

$$\mathbb{E}^{\pi_\theta}\left[ R_k \mid S_k, H_{k-1} \right]$$
$$= \mathbb{E}\left[ \mathbb{E}^{\pi_\theta}\left[ \mathbb{E}\left[ R_k \mid S_k, H_{k-1}, O_k, A_k \right] \mid S_k, H_{k-1}, O_k \right] \mid S_k, H_{k-1} \right] \text{ by law of total expectation}$$

$$= \mathbb{E}^{\pi_\theta}\left[ \mathbb{E}[\sum_a \mathbb{E}\left[ R_k \mid S_k, H_{k-1}, O_k, A_k = a \right] \pi_{\theta_k}(a \mid O_k, H_{k-1}) \mid S_k, H_{k-1}, O_k] \mid S_k, H_{k-1} \right]$$

$$= \mathbb{E}\left[ \sum_a \mathbb{E}\left[ R_k \mid S_k, H_{k-1}, O_k, A_k = a \right] \pi_{\theta_k}(a \mid O_k, H_{k-1}) \mid S_k, H_{k-1} \right]$$

$$= \mathbb{E}\left[ \sum_a \mathbb{E}\left[ R_k \mid S_k, H_{k-1}, A_k = a \right] \pi_{\theta_k}(a \mid O_k, H_{k-1}) \mid S_k, H_{k-1} \right] \text{ by } R_k \perp\!\!\!\perp O_k \mid S_k, A_k, H_{k-1}$$

$$= \sum_a \mathbb{E}\left[ R_k \mid S_k, H_{k-1}, A_k = a \right] \mathbb{E}\left[ \pi_{\theta_k}(a \mid O_k, H_{k-1}) \mid S_k, H_{k-1} \right]$$

$$= \sum_a \mathbb{E}\left[ R_k \mid S_k, H_{k-1}, A_k = a \right] \mathbb{E}\left[ \pi_{\theta_k}(a \mid O_k, H_{k-1}) \mid S_k, H_{k-1}, A_k = a \right] \text{ by } O_k \perp\!\!\!\perp A_k \mid S_k, H_{k-1}$$

$$= \sum_a \mathbb{E}\left[ R_k \pi_{\theta_k}(a \mid O_k, H_{k-1}) \mid S_k, H_{k-1}, A_k = a \right] \text{ by } O_k \perp\!\!\!\perp R_k \mid S_k, A_k, H_{k-1}$$

$$= \sum_a \mathbb{E}\left[ R_k \pi_{\theta_k}(a \mid O_k, H_{k-1}) \mid S_k, H_{k-1}, A_k = a \right] \tag{160}$$

For the second term, we have

$$\mathbb{E}^{\pi_\theta}\left[ \sum_{t=k+1}^T R_t \mid S_k, H_{k-1} \right]$$

$$= \mathbb{E}^{\pi_\theta}\left[ \mathbb{E}^{\pi_\theta}\left[ \sum_{t=k+1}^T R_t \mid S_{k+1}, H_k, S_k \right] \mid S_k, H_{k-1} \right] \text{ by law of total expectation}$$

$$= \mathbb{E}^{\pi_\theta}\left[ \mathbb{E}^{\pi_\theta}\left[ \sum_{t=k+1}^T R_t \mid S_{k+1}, H_k \right] \mid S_k, H_{k-1} \right] \text{ by } \{R_t\}_{t=k+1}^T \perp\!\!\!\perp_\theta S_k \mid S_{k+1}, H_k$$

$$= \mathbb{E}^{\pi_\theta} \left[ \mathbb{E} \left[ \sum_{a'} b_{V,k+1}^{\pi_\theta}(a', O_{k+1}, H_k) \mid S_{k+1}, H_k \right] \mid S_k, H_{k-1} \right] \quad \text{by assumption in induction}$$

$$= \mathbb{E}^{\pi_\theta} \left[ \mathbb{E} \left[ \sum_{a'} b_{V,k+1}^{\pi_\theta}(a', O_{k+1}, H_k) \mid S_{k+1}, H_k, S_k \right] \mid S_k, H_{k-1} \right] \quad O_{k+1} \perp\!\!\!\perp S_k \mid S_{k+1}, H_k$$

$$= \mathbb{E}^{\pi_\theta} \left[ \sum_{a'} b_{V,k+1}^{\pi_\theta}(a', O_{k+1}, H_k) \mid S_k, H_{k-1} \right] \quad \text{by law of total expectation} \tag{161}$$

$$= \mathbb{E}^{\pi_\theta} \left[ \mathbb{E} \left[ \sum_{a'} b_{V,k+1}^{\pi_\theta}(a', O_{k+1}, H_k) \mid S_k, H_{k-1}, O_k, A_k \right] \mid S_k, H_{k-1} \right] \quad \text{by law of total expectation}$$

$$= \mathbb{E} \left[ \sum_a \mathbb{E} \left[ \sum_{a'} b_{V,k+1}^{\pi_\theta}(a', O_{k+1}, H_k) \mid S_k, H_{k-1}, O_k, A_k = a \right] \pi_{\theta_k}(a \mid O_k, H_{k-1}) \mid S_k, H_{k-1} \right]$$

$$= \mathbb{E} \left[ \sum_a \mathbb{E} \left[ \sum_{a'} b_{V,k+1}^{\pi_\theta}(a', O_{k+1}, H_k) \pi_{\theta_k}(a \mid O_k, H_{k-1}) \mid S_k, H_{k-1}, O_k, A_k = a \right] \mid S_k, H_{k-1} \right]$$

(by measurability)

$$= \sum_a \mathbb{E} \left[ \mathbb{E} \left[ \sum_{a'} b_{V,k+1}^{\pi_\theta}(a', O_{k+1}, H_k) \pi_{\theta_k}(a \mid O_k, H_{k-1}) \mid S_k, H_{k-1}, O_k, A_k = a \right] \mid S_k, H_{k-1} \right]$$

$$= \sum_a \mathbb{E} \left[ \mathbb{E} \left[ \sum_{a'} b_{V,k+1}^{\pi_\theta}(a', O_{k+1}, H_k) \pi_{\theta_k}(a \mid O_k, H_{k-1}) \mid S_k, H_{k-1}, O_k, A_k = a \right] \mid S_k, H_{k-1}, A_k = a \right]$$

(by $O_k \perp\!\!\!\perp A_k \mid S_k$)

$$= \sum_a \mathbb{E} \left[ \sum_{a'} b_{V,k+1}^{\pi_\theta}(a', O_{k+1}, H_k) \pi_{\theta_k}(a \mid O_k, H_{k-1}) \mid S_k, H_{k-1}, A_k = a \right] \quad \text{by law of total expectation}$$

Combine equations (160)(161) and we have

$$\mathbb{E}^{\pi_\theta} \left[ \sum_{t=k}^{T} R_t \mid S_k, H_{k-1} \right]$$

$$= \mathbb{E}^{\pi_\theta} [R_k \mid S_k, H_{k-1}] + \mathbb{E}^{\pi_\theta} \left[ \sum_{t=k+1}^{T} R_t \mid S_k, H_{k-1} \right]$$

$$= \sum_a \mathbb{E} [R_k \pi_{\theta_k}(a \mid O_k, H_{k-1}) \mid S_k, H_{k-1}, A_k = a]$$

$$+ \sum_a \mathbb{E} \left[ \sum_{a'} b_{V,k+1}^{\pi_\theta}(a', O_{k+1}, H_k) \pi_{\theta_k}(a \mid O_k, H_{k-1}) \mid S_k, H_{k-1}, A_k = a \right] \quad \text{by (160)(161)}$$

$$= \sum_a \mathbb{E} \left[ R_k \pi_{\theta_k}(a \mid O_k, H_{k-1}) + \sum_{a'} b_{V,k+1}^{\pi_\theta}(a', O_{k+1}, H_k) \pi_{\theta_k}(a \mid O_k, H_{k-1}) \mid S_k, H_{k-1}, A_k = a \right]$$

$$= \sum_a \mathbb{E} \left[ b_{V,k}^{\pi_\theta}(a, O_k, H_{k-1}) \mid S_k, H_{k-1}, A_k = a \right] \quad \text{by (33)}$$

$$= \sum_a \mathbb{E} \left[ b_{V,k}^{\pi_\theta}(a, O_k, H_{k-1}) \mid S_k, H_{k-1} \right] \quad \text{by } O_k \perp\!\!\!\perp A_k \mid S_k, H_{k-1}$$

$$= \mathbb{E} \left[ \sum_a b_{V,k}^{\pi_\theta}(a, O_k, H_{k-1}) \mid S_k, H_{k-1} \right]$$

Therefore, $\mathbb{E}^{\pi_\theta}\left[\sum_{t=k}^T R_t \mid S_k, H_{k-1}\right] = \mathbb{E}\left[b_{V,k}^{\pi_\theta}(O_k, H_{k-1}) \mid S_k, H_{k-1}\right]$ also holds for $j = k$, if it holds for $j = k+1$. By the induction argument, the proof is done. $\qquad\square$

## H.6 PROOF OF LEMMA 12

*Proof.*

$\nabla_\theta \mathbb{E}^{\pi_\theta}[R_T \mid S_T, H_{T-1}]$

$=\nabla_\theta \mathbb{E}\left[\mathbb{E}^{\pi_\theta}\left[\mathbb{E}[R_T \mid S_T, H_{T-1}, O_T, A_T] \mid S_T, H_{T-1}, O_T\right] \mid S_T, H_{T-1}\right]$ by law of total expectation

$=\nabla_\theta \mathbb{E}^{\pi_\theta}\left[\mathbb{E}[\sum_a \mathbb{E}[R_T \mid S_T, H_{T-1}, O_T, A_T = a]\pi_{\theta_T}(a \mid O_T, H_{T-1}) \mid S_T, H_{T-1}, O_T] \mid S_T, H_{T-1}\right]$

$=\nabla_\theta \mathbb{E}\left[\sum_a \mathbb{E}[R_T \mid S_T, H_{T-1}, O_T, A_T = a]\pi_{\theta_T}(a \mid O_T, H_{T-1}) \mid S_T, H_{T-1}\right]$

$=\nabla_\theta \mathbb{E}\left[\sum_a \mathbb{E}[R_T \mid S_T, H_{T-1}, A_T = a]\pi_{\theta_T}(a \mid O_T, H_{T-1}) \mid S_T, H_{T-1}\right]$ by $R_T \perp\!\!\!\perp O_T \mid S_T, A_T, H_{T-1}$

$=\mathbb{E}\left[\sum_a \mathbb{E}[R_T \mid S_T, H_{T-1}, A_T = a]\nabla_{\theta_T}\pi_{\theta_T}(a \mid O_T, H_{T-1}) \mid S_T, H_{T-1}\right]$

$=\sum_a \mathbb{E}[R_T \mid S_T, H_{T-1}, A_T = a]\mathbb{E}[\nabla_{\theta_T}\pi_{\theta_T}(a \mid O_T, H_{T-1}) \mid S_T, H_{T-1}]$

$=\sum_a \mathbb{E}[R_T \mid S_T, H_{T-1}, A_T = a]\mathbb{E}[\nabla_{\theta_T}\pi_{\theta_T}(a \mid O_T, H_{T-1}) \mid S_T, H_{T-1}, A_T = a]$ by $O_T \perp\!\!\!\perp A_T \mid S_T, H_{T-1}$

$=\sum_a \mathbb{E}[R_T\nabla_{\theta_T}\pi_{\theta_T}(a \mid O_T, H_{T-1}) \mid S_T, H_{T-1}, A_T = a]$ by $O_T \perp\!\!\!\perp R_T \mid S_T, A_T, H_{T-1}$

$=\sum_a \mathbb{E}\left[b_{\nabla V,T}^{\pi_\theta}(O_T, H_{T-1}, a) \mid S_T, H_{T-1}, A_T = a\right]$ by (37)

$=\sum_a \mathbb{E}\left[b_{\nabla V,T}^{\pi_\theta}(O_T, H_{T-1}, a) \mid S_T, H_{T-1}\right]$ by $O_T \perp\!\!\!\perp A_T \mid S_T, H_{T-1}$

$=\mathbb{E}\left[\sum_a b_{\nabla V,T}^{\pi_\theta}(O_T, H_{T-1}, a) \mid S_T, H_{T-1}\right]$

According to the above derivation, we have shown $\mathbb{E}\left[b_{\nabla V,j}^{\pi_\theta}(O_j, H_{j-1}) \mid S_j, H_{j-1}\right] = \nabla_\theta \mathbb{E}^{\pi_\theta}\left[\sum_{t=j}^T R_t \mid S_j, H_{j-1}\right]$ when $j = T$. We proceed with the derivation by induction. Assume that $\mathbb{E}\left[b_{\nabla V,j}^{\pi_\theta}(O_j, H_{j-1}) \mid S_j, H_{j-1}\right] = \nabla_\theta \mathbb{E}^{\pi_\theta}\left[\sum_{t=j}^T R_t \mid S_j, H_{j-1}\right]$ holds for $j = k+1$, we will show that it also holds for $j = k$.

For $j = k$, we first notice that

$$\nabla_\theta \mathbb{E}^{\pi_\theta}\left[\sum_{t=k}^T R_t \mid S_k, H_{k-1}\right] = \nabla_\theta \mathbb{E}^{\pi_\theta}[R_k \mid S_k, H_{k-1}] + \nabla_\theta \mathbb{E}^{\pi_\theta}\left[\sum_{t=k+1}^T R_t \mid S_k, H_{k-1}\right]. \tag{163}$$

Next, we analyze these two terms separately. Analyzing the first term is the same as $\nabla_\theta \mathbb{E}^{\pi_\theta}[R_T \mid S_T, H_{T-1}]$ by replacing $T$ with $k$.

$\nabla_\theta \mathbb{E}^{\pi_\theta}[R_k \mid S_k, H_{k-1}]$

$$=\nabla_\theta \mathbb{E}\left[\mathbb{E}^{\pi_\theta}\left[\mathbb{E}\left[R_k \mid S_k, H_{k-1}, O_k, A_k\right] \mid S_k, H_{k-1}, O_k\right] \mid S_k, H_{k-1}\right] \text{ by law of total expectation}$$

$$=\nabla_\theta \mathbb{E}^{\pi_\theta}\left[\mathbb{E}[\sum_a \mathbb{E}\left[R_k \mid S_k, H_{k-1}, O_k, A_k = a\right]\pi_{\theta_k}(a \mid O_k, H_{k-1}) \mid S_k, H_{k-1}, O_k] \mid S_k, H_{k-1}\right]$$

$$=\nabla_\theta \mathbb{E}\left[\sum_a \mathbb{E}\left[R_k \mid S_k, H_{k-1}, O_k, A_k = a\right]\pi_{\theta_k}(a \mid O_k, H_{k-1}) \mid S_k, H_{k-1}\right]$$

$$=\nabla_\theta \mathbb{E}\left[\sum_a \mathbb{E}\left[R_k \mid S_k, H_{k-1}, A_k = a\right]\pi_{\theta_k}(a \mid O_k, H_{k-1}) \mid S_k, H_{k-1}\right] \text{ by } R_k \perp\!\!\!\perp O_k \mid S_k, A_k, H_{k-1}$$

$$\text{(164)}$$

$$=\mathbb{E}\left[\sum_a \mathbb{E}\left[R_k \mid S_k, H_{k-1}, A_k = a\right]\nabla_\theta\pi_{\theta_k}(a \mid O_k, H_{k-1}) \mid S_k, H_{k-1}\right]$$

$$=\sum_a \mathbb{E}\left[R_k \mid S_k, H_{k-1}, A_k = a\right]\mathbb{E}\left[\nabla_\theta\pi_{\theta_k}(a \mid O_k, H_{k-1}) \mid S_k, H_{k-1}\right]$$

$$=\sum_a \mathbb{E}\left[R_k \mid S_k, H_{k-1}, A_k = a\right]\mathbb{E}\left[\nabla_\theta\pi_{\theta_k}(a \mid O_k, H_{k-1}) \mid S_k, H_{k-1}, A_k = a\right] \text{ by } O_k \perp\!\!\!\perp A_k \mid S_k, H_{k-1}$$

$$=\sum_a \mathbb{E}\left[R_k\nabla_\theta\pi_{\theta_k}(a \mid O_k, H_{k-1}) \mid S_k, H_{k-1}, A_k = a\right] \text{ by } O_k \perp\!\!\!\perp R_k \mid S_k, A_k, H_{k-1}$$

For the second term, we have

$$\nabla_\theta \mathbb{E}^{\pi_\theta}\left[\sum_{t=k+1}^T R_t \mid S_k, H_{k-1}\right]$$

$$=\nabla_\theta \mathbb{E}^{\pi_\theta}\left[\mathbb{E}^{\pi_\theta}\left[\sum_{t=k+1}^T R_t \mid S_{k+1}, H_k, S_k\right] \mid S_k, H_{k-1}\right] \text{ by law of total expectation}$$

$$=\nabla_\theta \mathbb{E}^{\pi_\theta}\left[\mathbb{E}^{\pi_\theta}\left[\sum_{t=k+1}^T R_t \mid S_{k+1}, H_k\right] \mid S_k, H_{k-1}\right] \text{ by } \{R_t\}_{t=k+1}^T \perp\!\!\!\perp_\theta S_k \mid S_{k+1}, H_k$$

$$=\nabla_\theta \mathbb{E}^{\pi_\theta}\left[\mathbb{E}\left[\sum_{a'} b_{V,k+1}^{\pi_\theta}(a', O_{k+1}, H_k) \mid S_{k+1}, H_k\right] \mid S_k, H_{k-1}\right] \text{ by Lemma 11} \qquad \text{(165)}$$

$$=\nabla_\theta \mathbb{E}^{\pi_\theta}\left[\mathbb{E}\left[\sum_{a'} b_{V,k+1}^{\pi_\theta}(a', O_{k+1}, H_k) \mid S_{k+1}, A_k, O_k, H_{k-1}\right] \mid S_k, H_{k-1}\right] \text{ by expanding } H_k$$

$$=\nabla_\theta \int \mathbb{E}\left[\sum_{a'} b_{V,k+1}^{\pi_\theta}(a', O_{k+1}, H_k) \mid s_{k+1}, a_k, o_k, h_{k-1}\right] p_\theta(s_{k+1}, a_k, o_k, h_{k-1} \mid S_k, H_{k-1})ds_{k+1}da_kdo_kdh_k$$

$$=\int \nabla_\theta \mathbb{E}\left[\sum_{a'} b_{V,k+1}^{\pi_\theta}(a', O_{k+1}, H_k) \mid s_{k+1}, a_k, o_k, h_{k-1}\right] p_\theta(s_{k+1}, a_k, o_k, h_{k-1} \mid S_k, H_{k-1})ds_{k+1}da_kdo_kdh_k$$

$$+\int \mathbb{E}\left[\sum_{a'} b_{V,k+1}^{\pi_\theta}(a', O_{k+1}, H_k) \mid s_{k+1}, a_k, o_k, h_{k-1}\right] \nabla_\theta p_\theta(s_{k+1}, a_k, o_k, h_{k-1} \mid S_k, H_{k-1})ds_{k+1}da_kdo_kdh_k$$

$$=I + II$$

where

$$I = \int \nabla_\theta \mathbb{E}\left[\sum_{a'} b_{V,k+1}^{\pi_\theta}(a', O_{k+1}, H_k) \mid s_{k+1}, a_k, o_k, h_{k-1}\right] p_\theta(s_{k+1}, a_k, o_k, h_{k-1} \mid S_k, H_{k-1})ds_{k+1}da_kdo_kdh_k$$

$$= \mathbb{E}^{\pi_\theta} \left[ \nabla_\theta \mathbb{E} \left[ \sum_{a'} b_{V,k+1}^{\pi_\theta}(a', O_{k+1}, H_k) \mid S_{k+1}, A_k, O_k, H_{k-1} \right] \mid S_k, H_{k-1} \right]$$

$$= \mathbb{E}^{\pi_\theta} \left[ \nabla_\theta \mathbb{E} \left[ \sum_{a'} b_{V,k+1}^{\pi_\theta}(a', O_{k+1}, H_k) \mid S_{k+1}, H_k \right] \mid S_k, H_{k-1} \right] \text{ by } H_k = (A_k, O_k, H_{k-1})$$

$$= \mathbb{E}^{\pi_\theta} \left[ \nabla_\theta \mathbb{E}^{\pi_\theta} \left[ \sum_{t=k+1}^{T} R_t \mid S_{k+1}, H_k \right] \mid S_k, H_{k-1} \right] \text{ by Lemma 11}$$

$$= \mathbb{E}^{\pi_\theta} \left[ \mathbb{E} \left[ \sum_{a'} b_{\nabla V,k+1}^{\pi_\theta}(a', O_{k+1}, H_k) \mid S_{k+1}, H_k \right] \mid S_k, H_{k-1} \right] \text{ by assumption in induction}$$

$$= \mathbb{E}^{\pi_\theta} \left[ \mathbb{E} \left[ \sum_{a'} b_{\nabla V,k+1}^{\pi_\theta}(a', O_{k+1}, H_k) \mid S_{k+1}, H_k, S_k \right] \mid S_k, H_{k-1} \right] \quad O_{k+1} \perp\!\!\!\perp S_k \mid S_{k+1}, H_k$$

$$= \mathbb{E}^{\pi_\theta} \left[ \sum_{a'} b_{\nabla V,k+1}^{\pi_\theta}(a', O_{k+1}, H_k) \mid S_k, H_{k-1} \right] \text{ by law of total expectation} \tag{166}$$

$$= \mathbb{E}^{\pi_\theta} \left[ \mathbb{E} \left[ \sum_{a'} b_{\nabla V,k+1}^{\pi_\theta}(a', O_{k+1}, H_k) \mid S_k, H_{k-1}, O_k, A_k \right] \mid S_k, H_{k-1} \right] \text{ by law of total expectation}$$

$$= \mathbb{E} \left[ \sum_{a} \mathbb{E} \left[ \sum_{a'} b_{\nabla V,k+1}^{\pi_\theta}(a', O_{k+1}, H_k) \mid S_k, H_{k-1}, O_k, A_k = a \right] \pi_{\theta_k}(a \mid O_k, H_{k-1}) \mid S_k, H_{k-1} \right]$$

$$= \mathbb{E} \left[ \sum_{a} \mathbb{E} \left[ \sum_{a'} b_{\nabla V,k+1}^{\pi_\theta}(a', O_{k+1}, H_k) \pi_{\theta_k}(a \mid O_k, H_{k-1}) \mid S_k, H_{k-1}, O_k, A_k = a \right] \mid S_k, H_{k-1} \right]$$

$$= \sum_{a} \mathbb{E} \left[ \mathbb{E} \left[ \sum_{a'} b_{\nabla V,k+1}^{\pi_\theta}(a', O_{k+1}, H_k) \pi_{\theta_k}(a \mid O_k, H_{k-1}) \mid S_k, H_{k-1}, O_k, A_k = a \right] \mid S_k, H_{k-1} \right]$$

$$= \sum_{a} \mathbb{E} \left[ \mathbb{E} \left[ \sum_{a'} b_{\nabla V,k+1}^{\pi_\theta}(a', O_{k+1}, H_k) \pi_{\theta_k}(a \mid O_k, H_{k-1}) \mid S_k, H_{k-1}, O_k, A_k = a \right] \mid S_k, H_{k-1}, A_k = a \right]$$
$$\text{by } O_k \perp\!\!\!\perp A_k \mid S_k$$

$$= \sum_{a} \mathbb{E} \left[ \sum_{a'} b_{\nabla V,k+1}^{\pi_\theta}(a', O_{k+1}, H_k) \pi_{\theta_k}(a \mid O_k, H_{k-1}) \mid S_k, H_{k-1}, A_k = a \right] \text{ by law of total expectation}$$

and

$$II$$
$$= \int \mathbb{E} \left[ \sum_{a'} b_{V,k+1}^{\pi_\theta}(a', O_{k+1}, H_k) \mid s_{k+1}, a_k, o_k, h_{k-1} \right] \nabla_\theta p_\theta(s_{k+1}, a_k, o_k, h_{k-1} \mid S_k, H_{k-1}) ds_{k+1} da_k do_k dh_k$$

$$= \int \mathbb{E} \left[ \sum_{a'} b_{V,k+1}^{\pi_\theta}(a', O_{k+1}, H_k) \mid s_{k+1}, a_k, o_k, h_{k-1} \right]$$
$$(\nabla_\theta \log p_\theta \times p_\theta)(s_{k+1}, a_k, o_k, h_{k-1} \mid S_k, H_{k-1}) ds_{k+1} da_k do_k dh_k$$

$$= \mathbb{E}^{\pi_\theta} \left[ \mathbb{E} \left[ \sum_{a'} b_{V,k+1}^{\pi_\theta}(a', O_{k+1}, H_k) \mid S_{k+1}, A_k, O_k, H_{k-1} \right] \right.$$
$$\left. \nabla_\theta \log p_\theta(S_{k+1}, A_k, O_k, H_{k-1} \mid S_k, H_{k-1}) \mid S_k, H_{k-1} \right]$$

$$=\mathbb{E}^{\pi_\theta}\left[\mathbb{E}\left[\sum_{a'}b_{V,k+1}^{\pi_\theta}(a',O_{k+1},H_k)\mid S_{k+1},A_k,O_k,H_{k-1}\right]\mid S_k,H_{k-1}\right]$$
$$\nabla_\theta\log p_\theta(S_{k+1},A_k,O_k,H_{k-1}\mid S_k,H_{k-1})$$

$$=\mathbb{E}^{\pi_\theta}\left[\mathbb{E}\left[\sum_{a'}b_{V,k+1}^{\pi_\theta}(a',O_{k+1},H_k)\mid S_{k+1},H_k\right]\mid S_k,H_{k-1}\right]\nabla_\theta\log p_\theta(S_{k+1},A_k,O_k,H_{k-1}\mid S_k,H_{k-1})$$

$$=\mathbb{E}^{\pi_\theta}\left[\mathbb{E}\left[\sum_{a'}b_{V,k+1}^{\pi_\theta}(a',O_{k+1},H_k)\mid S_{k+1},H_k,S_k\right]\mid S_k,H_{k-1}\right]\nabla_\theta\log p_\theta(S_{k+1},A_k,O_k,H_{k-1}\mid S_k,H_{k-1})$$
$$\text{(by } O_{k+1}\perp\!\!\!\perp S_k\mid S_{k+1},H_k)$$

$$=\mathbb{E}^{\pi_\theta}\left[\sum_{a'}b_{V,k+1}^{\pi_\theta}(a',O_{k+1},H_k)\mid S_k,H_{k-1}\right]\nabla_\theta\log p_\theta(S_{k+1},A_k,O_k,H_{k-1}\mid S_k,H_{k-1})$$

$$=\mathbb{E}^{\pi_\theta}\left[\sum_{a'}b_{V,k+1}^{\pi_\theta}(a',O_{k+1},H_k)\nabla_\theta\log p_\theta(S_{k+1},A_k,O_k,H_{k-1}\mid S_k,H_{k-1})\mid S_k,H_{k-1}\right]$$

$$=\mathbb{E}^{\pi_\theta}\left[\sum_{a'}b_{V,k+1}^{\pi_\theta}(a',O_{k+1},H_k)\nabla_\theta\log\pi_{\theta_k}(A_k\mid O_k,H_{k-1})\mid S_k,H_{k-1}\right]\text{ by (168)}$$

$$=\mathbb{E}^{\pi_\theta}\left[\mathbb{E}\left[\sum_{a'}b_{V,k+1}^{\pi_\theta}(a',O_{k+1},H_k)\nabla_\theta\log\pi_{\theta_k}(A_k\mid O_k,H_{k-1})\mid O_k,A_k,S_k,H_{k-1}\right]\mid S_k,H_{k-1}\right]$$

$$=\mathbb{E}^{\pi_\theta}\left[\mathbb{E}\left[\sum_{a'}b_{V,k+1}^{\pi_\theta}(a',O_{k+1},H_k)\mid O_k,A_k,S_k,H_{k-1}\right]\nabla_\theta\log\pi_{\theta_k}(A_k\mid O_k,H_{k-1})\mid S_k,H_{k-1}\right]$$

$$=\mathbb{E}\left[\sum_{a}\mathbb{E}\left[\sum_{a'}b_{V,k+1}^{\pi_\theta}(a',O_{k+1},H_k)\mid O_k,A_k=a,S_k,H_{k-1}\right]\right.$$
$$\left.\nabla_\theta\log\pi_{\theta_k}(a\mid O_k,H_{k-1})\pi_{\theta_k}(a\mid O_k,H_{k-1})\mid S_k,H_{k-1}\right]$$

$$=\mathbb{E}\left[\sum_{a}\left[\sum_{a'}b_{V,k+1}^{\pi_\theta}(a',O_{k+1},H_k)\mid O_k,A_k=a,S_k,H_{k-1}\right]\nabla_\theta\pi_{\theta_k}(a\mid O_k,H_{k-1})\mid S_k,H_{k-1}\right]$$

$$=\mathbb{E}\left[\sum_{a}\mathbb{E}\left[\sum_{a'}b_{V,k+1}^{\pi_\theta}(a',O_{k+1},H_k)\nabla_\theta\pi_{\theta_k}(a\mid O_k,H_{k-1})\mid O_k,A_k=a,S_k,H_{k-1}\right]\mid S_k,H_{k-1}\right]$$

$$=\sum_{a}\mathbb{E}\left[\mathbb{E}\left[\sum_{a'}b_{V,k+1}^{\pi_\theta}(a',O_{k+1},H_k)\nabla_\theta\pi_{\theta_k}(a\mid O_k,H_{k-1})\mid O_k,A_k=a,S_k,H_{k-1}\right]\mid S_k,H_{k-1}\right]$$

$$=\sum_{a}\mathbb{E}\left[\mathbb{E}\left[\sum_{a'}b_{V,k+1}^{\pi_\theta}(a',O_{k+1},H_k)\nabla_\theta\pi_{\theta_k}(a\mid O_k,H_{k-1})\mid O_k,A_k=a,S_k,H_{k-1}\right]\mid S_k,H_{k-1},A_k=a\right]$$

$$=\sum_{a}\mathbb{E}\left[\sum_{a'}b_{V,k+1}^{\pi_\theta}(a',O_{k+1},H_k)\nabla_\theta\pi_{\theta_k}(a\mid O_k,H_{k-1})\mid S_k,H_{k-1},A_k=a\right] \tag{167}$$

where

$$\nabla_\theta\log p_\theta(s_{k+1},a_k,o_k,h_{k-1}\mid s_k,h_{k-1})$$
$$=\nabla_\theta\log\{p_\theta(s_{k+1}\mid a_k,o_k,s_k,h_{k-1})p_\theta(a_k,o_k\mid s_k,h_{k-1})\}$$
$$=\nabla_\theta\log\{p(s_{k+1}\mid a_k,s_k)p_\theta(a_k,o_k\mid s_k,h_{k-1})\}\text{ by }S_{k+1}\perp\!\!\!\perp(O_k,H_{k-1})\mid S_k,A_k$$
$$=\nabla_\theta\log p_\theta(a_k,o_k\mid s_k,h_{k-1}) \tag{168}$$
$$=\nabla_\theta\log\{p_\theta(a_k\mid o_k,s_k,h_{k-1})p(o_k\mid s_k,h_{k-1})\}$$
$$=\nabla_\theta\log p_\theta(a_k\mid o_k,s_k,h_{k-1})$$
$$=\nabla_\theta\log\pi_{\theta_k}(a_k\mid o_k,h_{k-1})\text{by }A_k\perp\!\!\!\perp S_k\mid O_k,H_{k-1}.$$

Combining equations (164)(165)(166)(167), and we have

$$\nabla_\theta \mathbb{E}^{\pi_\theta} \left[ \sum_{t=k}^{T} R_t \mid S_k, H_{k-1} \right]$$

$$= \nabla_\theta \mathbb{E}^{\pi_\theta} \left[ R_k \mid S_k, H_{k-1} \right] + \nabla_\theta \mathbb{E}^{\pi_\theta} \left[ \sum_{t=k+1}^{T} R_t \mid S_k, H_{k-1} \right]$$

$$= \sum_a \mathbb{E} \left[ R_k \nabla_\theta \pi_{\theta_k}(a \mid O_k, H_{k-1}) \mid S_k, H_{k-1}, A_k = a \right] + I + II \text{ by (164)(165)}$$

$$= \sum_a \mathbb{E} \left[ R_k \nabla_\theta \pi_{\theta_k}(a \mid O_k, H_{k-1}) \mid S_k, H_{k-1}, A_k = a \right]$$

$$+ \sum_a \mathbb{E} \left[ \sum_{a'} b_{\nabla V, k+1}^{\pi_\theta}(a', O_{k+1}, H_k) \pi_{\theta_k}(a \mid O_k, H_{k-1}) \mid S_k, H_{k-1}, A_k = a \right]$$

$$+ \sum_a \mathbb{E} \left[ \sum_{a'} b_{V, k+1}^{\pi_\theta}(a', O_{k+1}, H_k) \nabla_\theta \pi_{\theta_k}(a \mid O_k, H_{k-1}) \mid S_k, H_{k-1}, A_k = a \right] \text{ by (166)(167)}$$

$$= \sum_a \mathbb{E} \left[ b_{\nabla V, k}^{\pi_\theta}(O_k, H_{k-1}, a) \mid S_k, A_k = a, H_{k-1} \right] \text{ by (37).}$$

$$= \sum_a \mathbb{E} \left[ b_{\nabla V, k}^{\pi_\theta}(O_k, H_{k-1}, a) \mid S_k, H_{k-1} \right] \text{ by } O_k \perp\!\!\!\perp A_k \mid S_k, H_{k-1}$$

$$= \mathbb{E} \left[ \sum_a b_{\nabla V, k}^{\pi_\theta}(a, O_k, H_{k-1}) \mid S_k, H_{k-1} \right] \tag{169}$$

Therefore, $\nabla_\theta \mathbb{E}^{\pi_\theta} \left[ \sum_{t=k}^{T} R_t \mid S_k, H_{k-1} \right] = \mathbb{E} \left[ \sum_a b_{\nabla V, k}^{\pi_\theta}(a, O_k, H_{k-1}) \mid S_k, H_{k-1} \right]$ also holds for $j = k$ if it holds for $j = k + 1$. By induction, the proof is done. $\square$

## H.7   PROOF OF LEMMA 13

We need to show

$$\mathbb{E}^{\pi^b} \left[ \frac{p_t^{\pi_\theta}(S_t, H_{t-1})}{p_t^{\pi^b}(S_t, H_{t-1}) \pi_t^b(A_t \mid S_t)} f_t(A_t, O_t, H_{t-1}) \right]$$

$$= \mathbb{E}^{\pi^b} \left[ \frac{p_{t-1}^{\pi_\theta}(S_{t-1}, H_{t-2}) \pi_{\theta_{t-1}}(A_{t-1} \mid O_{t-1}, H_{t-2})}{p_{t-1}^{\pi^b}(S_{t-1}, H_{t-2}) \pi_{t-1}^b(A_{t-1} \mid S_{t-1})} \sum_a f_t(a, O_t, H_{t-1}) \right]. \tag{170}$$

To see this, we calculate these two terms separately.

$$\mathbb{E}^{\pi^b} \left[ \frac{p_t^{\pi_\theta}(S_t, H_{t-1})}{p_t^{\pi^b}(S_t, H_{t-1}) \pi_t^b(A_t \mid S_t)} f_t(A_t, O_t, H_{t-1}) \right]$$

$$= \int \left[ \frac{p_t^{\pi_\theta}(s_t, h_{t-1})}{p_t^{\pi^b}(s_t, h_{t-1}) \pi_t^b(a_t \mid s_t)} f_t(a_t, o_t, h_{t-1}) \right] p_t^{\pi^b}(a_t, o_t, s_t, h_{t-1}) da_t do_t ds_t dh_{t-1}$$

$$= \int \left[ \frac{p_t^{\pi_\theta}(s_t, h_{t-1})}{p_t^{\pi^b}(s_t, h_{t-1}) \pi_t^b(a_t \mid s_t)} f_t(a_t, o_t, h_{t-1}) \right] \pi_t^b(a_t \mid s_t) p(o_t \mid s_t) p_t^{\pi^b}(s_t, h_{t-1}) da_t do_t ds_t dh_{t-1}$$

$$= \int_{o_t, s_t, h_{t-1}} \sum_a \frac{p_t^{\pi_\theta}(s_t, h_{t-1})}{p_t^{\pi^b}(s_t, h_{t-1}) \pi_t^b(a \mid s_t)} f_t(a, o_t, h_{t-1}) \pi_t^b(a \mid s_t) p(o_t \mid s_t) p_t^{\pi^b}(s_t, h_{t-1}) do_t ds_t dh_{t-1}$$

$$= \int_{o_t, s_t, h_{t-1}} \sum_a f_t(a, o_t, h_{t-1}) p(o_t \mid s_t) p_t^{\pi_\theta}(s_t, h_{t-1}) do_t ds_t dh_{t-1}. \tag{171}$$

The first equality is simply from expanding the expectation. The second equality is based on $O_t \perp\!\!\!\perp A_t \mid S_t$. The third equality is by taking expectation with respect to $A_t$ at first. The final equality is by canceling out the same terms.

Now we analyze another term.

$$\mathbb{E}^{\pi^b} \left[ \frac{p^{\pi_\theta}_{t-1}(S_{t-1}, H_{t-2})\pi_{\theta_{t-1}}(A_{t-1} \mid O_{t-1}, H_{t-2})}{p^{\pi^b}_{t-1}(S_{t-1}, H_{t-2})\pi^b_{t-1}(A_{t-1} \mid S_{t-1})} \sum_a f_t(a, O_t, H_{t-1}) \right]$$

$$= \int \left[ \frac{p^{\pi_\theta}_{t-1}(s_{t-1}, h_{t-2})\pi_{\theta_{t-1}}(a_{t-1} \mid o_{t-1}, h_{t-2})}{p^{\pi^b}_{t-1}(s_{t-1}, h_{t-2})\pi^b_{t-1}(a_{t-1} \mid s_{t-1})} \sum_a f_t(a, o_t, h_{t-1}) \right]$$

$$p^{\pi^b}(o_t, a_{t-1}, o_{t-1}, s_{t-1}, h_{t-2}) do_t da_{t-1} do_{t-1} ds_{t-1} dh_{t-2}$$

$$= \int \left[ \frac{p^{\pi_\theta}_{t-1}(s_{t-1}, h_{t-2})\pi_{\theta_{t-1}}(a_{t-1} \mid o_{t-1}, h_{t-2})}{p^{\pi^b}_{t-1}(s_{t-1}, h_{t-2})\pi^b_{t-1}(a_{t-1} \mid s_{t-1})} \sum_a f_t(a, o_t, h_{t-1}) \right]$$

$$\int p^{\pi^b}(o_t, s_t, a_{t-1}, o_{t-1}, s_{t-1}, h_{t-2}) ds_t do_t da_{t-1} do_{t-1} ds_{t-1} dh_{t-2} \quad \text{(marginalization over } S_t)$$

$$= \int \left[ \frac{p^{\pi_\theta}_{t-1}(s_{t-1}, h_{t-2})\pi_{\theta_{t-1}}(a_{t-1} \mid o_{t-1}, h_{t-2})}{p^{\pi^b}_{t-1}(s_{t-1}, h_{t-2})\pi^b_{t-1}(a_{t-1} \mid s_{t-1})} \sum_a f_t(a, o_t, h_{t-1}) \right]$$

$$= \int p^{\pi_\theta}_{t-1}(s_{t-1}, h_{t-2})\pi_{\theta_{t-1}}(a_{t-1} \mid o_{t-1}, h_{t-2}) \sum_a f_t(a, o_t, h_{t-1})$$

$$p(o_t \mid s_t)p(s_t \mid a_{t-1}, s_{t-1})p(o_{t-1} \mid s_{t-1}) do_t ds_t da_{t-1} do_{t-1} ds_{t-1} dh_{t-2}$$

$$= \int_{o_t, s_t, h_{t-1}} \sum_a f_t(a, o_t, h_{t-1})p(o_t \mid s_t)p^{\pi_\theta}_t(s_t, h_{t-1}) do_t ds_t dh_{t-1}. \tag{172}$$

where the third equality is by expanding the joint density under the assumptions on POMDP. The fourth equality is by canceling out all the same terms. The last equality is by noticing that $p^{\pi_\theta}_t(s_t, h_{t-1}) = p(s_t \mid s_{t-1}, a_{t-1})\pi_{\theta_t}(a_{t-1} \mid o_{t-1}, h_{t-2})p(o_{t-1} \mid s_{t-1})p^{\pi_\theta}_{t-1}(s_{t-1}, h_{t-2})$. Then the proof is done by comparing these two terms.

The proof is done by comparing equation 171 and equation 172.

# I  IMPLEMENTATION

## I.1  IMPLEMENTATION DETAILS USING RKHS

We provide implementation details for Algorithm 1 in this section. Specifically, we mainly focus on steps 4-6 for estimating the policy gradient when function classes are assumed to be RKHSs.

For a specific coordinate $j$ at each iteration $t$, $\mathcal{B}^{(t)}_j$ can be set as a RKHS endowed with a reproducing kernel $K_{\mathcal{B}^{(t)}_j}(\cdot, \cdot)$ and a canonical RKHS norm $\| \cdot \|_{\mathcal{B}^{(t)}_j} = \| \cdot \|_{K_{\mathcal{B}^{(t)}_j}}$. Similarly, $\mathcal{F}^{(t)}_j$ can be set as a RKHS endowed with a reproducing kernel $K_{\mathcal{F}^{(t)}_j}(\cdot, \cdot)$ and a canonical RKHS norm $\| \cdot \|_{\mathcal{F}^{(t)}_j} = \| \cdot \|_{K_{\mathcal{F}^{(t)}_j}}$. Then, with these bridge function classes and test function classes, we are able to get the closed-form solution of the min-max optimization problem (9) for each coordinate by adopting Propositions 9, 10 in Appendix E.3 of Dikkala et al. (2020). Specifically, we can first get two empirical kernel matrices based on the offline data:

$$\mathbb{K}_{\mathcal{B}^{(t)}_j, N} := [K_{\mathcal{B}^{(t)}_j}([a^i_t, o^i_t, h^i_{t-1}], [a^j_t, o^j_t, h^j_{t-1}])]^{N,N}_{i=1, j=1}$$

and

$$\mathbb{K}_{\mathcal{F}^{(t)}_j, N} := [K_{\mathcal{F}^{(t)}_j}([a^i_t, o^i_t, h^i_{t-1}], [a^j_t, o^j_t, h^j_{t-1}])]^{N,N}_{i=1, j=1}.$$

Given these two empirical kernel matrices, we define a matrix

$$\mathbb{M}_{t,j,N} := \mathbb{K}_{\mathcal{F}_j^{(t)},N}^{\frac{1}{2}} \big( \frac{\lambda}{N\xi} \mathbb{K}_{\mathcal{F}_j^{(t)},N} + \mathbb{I}_N \big)^{-1} \mathbb{K}_{\mathcal{F}_j^{(t)},N}^{\frac{1}{2}}$$

where $\lambda, \xi$ are tuning parameters and $\mathbb{I}_N$ denotes the $N \times N$ identity matrix.

Next, if $j \leq d_\Theta$, then we consider the empirical "gradient response vector" $\mathbb{Y}_{t,j,N} \in \mathbb{R}^N$ whose $n$-th element $\mathbb{Y}_{t,j,n} := (r_t^n + \sum_{a'} \widehat{b}_{V,t+1}^{\pi_\theta}(a', o_{t+1}^n, h_t^n))[\nabla_\theta \pi_{\theta_t}(a_t^n \mid o_t^n, h_{t-1}^n)]_j + \sum_{a'} [\widehat{b}_{\nabla V,t+1}^{\pi_\theta}(a', o_{t+1}^n, h_t^n)]_j \pi_{\theta_t}(a_t^n \mid o_t^n, h_{t-1}^n)$ where $[\cdot]_j$ denotes the j-th coordinate. If $j = d_\Theta + 1$, then we consider the empirical "value response vector" $\mathbb{Y}_{t,d_\Theta+1,N} \in \mathbb{R}^N$ whose $n$-th element $\mathbb{Y}_{t,d_\Theta+1,n} := (r_t^n + \sum_{a'} \widehat{b}_{V,t+1}^{\pi_\theta}(a', o_{t+1}^n, h_t^n)) \pi_{\theta_t}(a_t^n \mid o_t^n, h_{t-1}^n)$.

We note that the empirical "gradient response vector" and the empirical "value response vector" can be understood as the empirical version of the "response" (i.e. evaluated at the offline dataset) on the right hand sides of equations (3)(4) in Assumption 2. All the involved terms are computable because we know the form of reproducing kernel functions (e.g. Gaussian kernel), the form of policies and their gradients (e.g. log-linear policies), the bridge functions $\widehat{b}_{V,t+1}^{\pi_\theta}, \widehat{b}_{\nabla V,t+1}^{\pi_\theta}$ that have been learned from the previous iteration $t + 1$ and the given offline dataset $\mathcal{D} = \{(o_t^n, a_t^n, r_t^n)_{t=1}^T\}_{n=1}^N$. Finally, by adopting the mentioned results in Dikkala et al. (2020), for each coordinate $j = 1, ..., d_\Theta + 1$, the solution to the min-max optimization problem (9) is given in the following form:

$$\widehat{b}_{t,j}^{\pi_\theta}(\cdot) = \sum_{n=1}^N \alpha_{n,t,j} K_{\mathcal{B}_j^{(t)}}([a_t^n, o_t^n, h_{t-1}^n], \cdot)$$

where $\alpha_{n,t,j}$ is the $n$-th element of the $N$-dimensional vector

$$\alpha_{t,j} := (\mathbb{K}_{\mathcal{B}_j^{(t)},N} \mathbb{M}_{t,j,N} \mathbb{K}_{\mathcal{B}_j^{(t)},N} + 4\xi\mu \mathbb{K}_{\mathcal{B}_j^{(t)},N})^\dagger \mathbb{K}_{\mathcal{B}_j^{(t)},N} \mathbb{M}_{t,j,N} \mathbb{Y}_{t,j,N}.$$

Here $\mu$ is a tuning parameter and $(\cdot)^\dagger$ denotes the Moore-Penrose pseudo-inverse. The hyper-parameters $\xi, \mu, \lambda$ can be chosen by using cross-validation as suggested in Dikkala et al. (2020). Repeating this procedure sequentially for $t = T, ..., 1$ and we get $\widehat{b}_1^{\pi_\theta}(\cdot)$ whose $j$-th coordinate is $\sum_{n=1}^N \alpha_{n,1,j} K_{\mathcal{B}_j^{(1)}}([a_1^n, o_1^n, h_0^n], \cdot)$ with $\alpha_{1,j} = [\alpha_{1,1,j}, \alpha_{2,1,j}, ..., \alpha_{N,1,j}]^\top = (\mathbb{K}_{\mathcal{B}_j^{(1)},N} \mathbb{M}_{1,j,N} \mathbb{K}_{\mathcal{B}_j^{(1)},N} + 4\xi\mu \mathbb{K}_{\mathcal{B}_j^{(1)},N})^\dagger \mathbb{K}_{\mathcal{B}_j^{(1)},N} \mathbb{M}_{1,j,N} \mathbb{Y}_{1,j,N}$. We then use the first $d_\Theta$ elements of $\widehat{b}_1^{\pi_\theta}(\cdot)$ (i.e. $j = 1, ...., d_\Theta$) to estimate the policy gradient, by following the procedure described at steps 7,8 of Algorithm 1. Specifically, we compute $\widehat{b}_{1,j}^{\pi_{\theta^{(k)}}}(a, o_1^i) = \sum_{n=1}^N \alpha_{n,1,j} K_{\mathcal{B}_j^{(1)}}([a_1^n, o_1^n], [a, o_1^i])$ for each $j = 1, ..., d_\Theta$, $a \in \mathcal{A}(|\mathcal{A}| \leq \infty)$, and $i = 1, ..., N$. Lastly we estimate the policy gradient evaluated at $\theta^{(k)}$ by $\frac{1}{N} \sum_{i=1}^N [\sum_{a \in \mathcal{A}} \widehat{b}_{1,1:d_\Theta}^{\pi_{\theta^{(k)}}}(a, o_1^i)]$, and the first stage for estimation is done. In the subsequent stage, we employ the estimated policy gradient to update the parameter using the procedure described in step 9 of Algorithm 1.

## I.2 COMPUTATIONAL COMPLEXITY OF ALGORITHM 1 USING RKHS

We discuss the computational complexity of Algorithm 1 using RKHS here. We first focus on the steps 4-6 regarding the min-max estimation procedure. At each $t$, for a specific $j - th$ coordinate, we follow the implementation details described above.

**(a) Computing $\mathbb{K}_{\mathcal{B}_j^{(t)},N}$ and $\mathbb{K}_{\mathcal{F}_j^{(t)},N}$.** It can be seen that we need to first evaluate the empirical kernel matrix $\mathbb{K}_{\mathcal{B}_j^{(t)},N}$ and $\mathbb{K}_{\mathcal{F}_j^{(t)},N}$. As they are both $N \times N$ matrix, it takes $N^2$ operations to evaluate each element (kernel function values between two vectors with length $t\dim(\mathcal{O})\dim(\mathcal{A})$), which is the dimension of the history space $\mathcal{H}_t$. Here we note that both $\dim(\mathcal{O})$ and $\dim(\mathcal{A})$ are fixed, and we care about the key parameter - the scale with $t$. Throughout this discussion, we simply ignore $\dim(\mathcal{O})$ and $\dim(\mathcal{A})$. The complexity of computing the kernel function value for two vectors with dimension $t$ depends on the specific kernel functions we use. For example, for linear kernels, the computational complexity of evaluating the kernel function is $O(t)$. For more complex kernels like polynomial kernel, this complexity is higher than linear and depends on the degree of

the polynomial. For the Gaussian kernel (RBF) that we used in our numerical experiments, the computational complexity is usually $O(t)$. Consequently, the computational complexity of evaluating two empirical kernel matrices is $O(N^2 t)$.

**(b) Computing** $\mathbb{M}_{t,j,N}$**.** The computation of $\mathbb{M}_{t,j,N}$ involves taking the square root of matrices, matrix multiplications, and the calculation of the inverse of a matrix. The computational time is dominated by the calculation of the inverse of a matrix, which is $O(N^3)$.

**(c) Computing** $\pi_{\theta_t}(a_t \mid o_t, h_{t-1})$ **and** $\nabla_{\theta_t} \pi_{\theta_t}(a_t \mid o_t, h_{t-1})$ **for any fixed** $(a_t, o_t, h_{t-1})$**.** The computational time depends on the specific policy class. For the log-linear policy class with $\pi_{\theta_t}(a_t \mid o_t, h_{t-1}) \propto \exp(\theta_t^\top \phi(o_t, a_t, h_{t-1}))$, we have $\nabla_{\theta_t} \pi_{\theta_t}(a_t \mid o_t, h_{t-1}) = \phi(o_t, a_t, h_{t-1}) - \sum_{a'_t} \pi_{\theta_t}(a'_t \mid o_t, h_{t-1}) \phi(o_t, a'_t, h_{t-1})$. It can be seen directly that both the computational time in evaluating $\pi_{\theta_t}(a_t \mid o_t, h_{t-1})$ and $\nabla_{\theta_t} \pi_{\theta_t}(a_t \mid o_t, h_{t-1})$ is $O(d_{\Theta_t})$

**(d) Computing** $\mathbb{Y}_{t,j,N}$**.** For each $n$, it takes $O(d_{\Theta_t})$ to compute $\pi_{\theta_t}(a_t^n \mid o_t^n, h_{t-1}^n)$ and $\nabla_{\theta_t} \pi_{\theta_t}(a_t^n \mid o_t^n, h_{t-1}^n)$ for log-linear policies according to (c). In addition, it takes $O(N(t+1))$ to compute $[\widehat{b}_{\nabla V,t+1}^{\pi_\theta}]_j(a', o_{t+1}^n, h_t^n)$ and $\widehat{b}_{V,t+1}^{\pi_\theta}(a', o_{t+1}^n, h_t^n)$ according to (f). Since we need to do the same operation for each $n$, the computational time for this step is $O(Nd_{\Theta_t} + N^2(t+1))$

**(e) Computing** $\alpha_{t,j}$**.** Having the results of $\mathbb{K}_{\mathcal{B}_j^{(t)},N}$, $\mathbb{K}_{\mathcal{F}_j^{(t)},N}$, $\mathbb{M}_{t,j,N}$, and $\mathbb{Y}_{t,j,N}$, we are able to calculate the coefficient $\alpha_{t,j}$. The computation of $\alpha_{t,j}$ involves matrix multiplication and solving a linear system. The computational time is $O(N^3)$.

**(f) Computing** $\widehat{b}_{t,j}^{\pi_\theta}(a_t, o_t, h_{t-1})$ **for any fixed** $(a_t, o_t, h_{t-1})$**.** It involves the calculation of $N$ kernel function values and a linear combination of them. For the Gaussian kernel, the computational complexity is $O(t)$ for each $n$. Therefore the computational complexity in this step is $O(Nt)$.

We note that at each $t$, steps (a), (b), (d), (e), (f) should be repeated $d_\Theta + 1$ times, while (c) is only repeated $T$ times. Consequently, the computational time for solving the min-max optimization problem for each $t$ is $O(d_\Theta N^2 t + d_\Theta N^3 + N d_\Theta + N^2 d_\Theta t + d_\Theta N^3) = O(d_\Theta N^2 t + d_\Theta N^3)$. As we repeat the procedure from $t = T$ to $t = 1$, the computational complexity of steps 4-6 in Algorithm 1 is $O(d_\Theta N^2 T^2 + d_\Theta N^3 T)$.

It can be easily seen that the computational time of steps 8,9 is dominated by the one of steps 4-6. Consequently, by considering the for-loop over $k$, the overall computational complexity of Algorithm 1 is
$$O(K d_\Theta N^2 T \max\{T, N\})$$
by assuming history-dependent log-linear policy class and RKHS with Gaussian kernels.

### I.3 MORE DISCUSSIONS ON PRACTICAL IMPLEMENTATIONS

In practical implementations on real RL problems, the exact execution of the proposed algorithm may sometimes computationally costly. For instance, when dealing with large sample sizes or a high number of stages, solving the min-max optimization problem (9) precisely in each iteration may become computationally expensive. Particularly, exact solutions with RKHSs involve calculating the inverse of an $N \times N$ (which takes $O(N^3)$) and evaluating the empirical kernel matrix on the space of history (taking approximately $O(N^2 T)$). Additionally, when assuming other function classes for bridge functions and test functions, such as neural networks, achieving an exact solution for the min-max optimization problem (9) is not feasible.

To enhance computational efficiency and accommodate general function classes, a potential alternative approach is to compute the (stochastic) gradient of the empirical loss function in the min-max optimization problem (9) with respect to $b$ and $f$ respectively. Subsequently, updating $f$ and $b$ alternately using gradient ascent and gradient descent only once in each iteration can be performed. In this case, Algorithm 1 could be potentially expedited, contributing to faster computations.

## J SIMULATION DETAILS

In this section, we provide the simulation details for the numerical results of the conducted experiment as shown in section 7.

Table 2: Comparisons of policy gradient estimations under the proposed method and the naive method at the uniform random policy when $\mathcal{O} = \mathcal{S} = \mathcal{A} = \{+1, -1\}$. Simulated data are generated by assuming $P(S_0 = 1) = \frac{1}{2}$, $\mathbb{P}(O_0 = S_0) = 0.8$, $P(S_1 = S_0) = 0.95$, $\mathbb{P}(O_1 = S_1) = 0.8$, $R_1 = \frac{2}{1+\exp(-4S_1 A_1)} - 1$, $\pi_b(+1 \mid 1) = \pi_b(-1 \mid -1) = 0.3$, $\pi_\theta(a_1 \mid o_1) \propto \exp(\theta_{a_1,o_1})$. $T =$ number of stages; $N =$ number of samples. Means (standard deviations) of $\|\widehat{\nabla V} - \nabla V\| / \|\nabla V\|$ are reported based on 20 replicates.

| SAMPLE SIZE | $T$=1, PROPOSED | $T$=1, NAIVE | $T$=2, PROPOSED | $T$=2, NAIVE |
|---|---|---|---|---|
| $N = 500$ | 0.124 (0.011) | 0.115 (0.0011) | 0.192 (0.0189) | 0.345 (0.011) |
| $N = 2000$ | 0.073 (0.006) | 0.110 (0.005) | 0.133 (0.012) | 0.342 (0.007) |
| $N = 8000$ | 0.031 (0.003) | 0.110 (0.003) | 0.066 (0.004) | 0.352 (0.003) |

We consider $T = 2$ and $N = 10000$. Simulated data are generated by assuming

$$S_0 \sim unif(-2, 2),$$

$$O_0 \sim 0.8\mathcal{N}(S_0, 0.1) + 0.2\mathcal{N}(-S_0, 0.1),$$

$$S_1 \sim \mathcal{N}(S_0, 0.1),$$

$$S_2 \sim \mathcal{N}(S_1 A_1, 0.01),$$

$$\pi_b(+1 \mid S_t > 0) = \pi_b(-1 \mid S_t < 0) = 0.3,$$

$$O_t \sim 0.8\mathcal{N}(S_t, 0.1) + 0.2\mathcal{N}(-S_t, 0.1),$$

$$R_t(S_t, A_t) = \frac{2}{1 + \exp(-4S_t A_t)} - 1,$$

$$\pi_{\theta_t}(A_t \mid O_t) \propto \exp(\theta_t^\top \phi_t(A_t, O_t)),$$

where $\phi_{t,1}(a_t, o_t) := 2o_t\mathbb{I}(a_t > 0, o_t > 0)$, $\phi_{t,2}(a_t, o_t) := 2o_t\mathbb{I}(a_t < 0, o_t > 0)$, $\phi_{t,3}(a_t, o_t) := 2o_t\mathbb{I}(a_t > 0, o_t < 0)$, $\phi_{t,1}(a_t, o_t) := 2o_t\mathbb{I}(a_t < 0, o_t < 0)$ for $t = 1, 2$. The dimension of policy class $d_\Theta = 8$.

## K  ADDITIONAL NUMERICAL RESULTS FOR TABULAR CASES

Table 2 reports the normalized $l_2$ norm of the error for the policy gradient estimator under the proposed method and the naive method. We can see that the proposed method under both single-stage and multi-stage settings is consistent, i.e. the error of the policy gradient estimator approaches to 0 as the sample size $N$ increases. In contrast, the naive estimator has an irreducible bias ($\approx 0.11$) in the single-stage setting, and this bias becomes significantly larger ($\approx 0.352$) in the multi-stage setting, indicating that the bias caused by unmeasured confounding may be more severe when the number of stages increases.

## L  ADDITIONAL NUMERICAL RESULTS FOR CONTINUOUS CASES

**Numerical results of policy gradient estimation in the function approximation settings.** We consider an example with continuous state/observation space and $T = 2$, and implement the proposed policy gradient estimation procedure (9) using RKHS. Simulation results of the proposed method and the naive method for estimating policy gradient are provided in Figure 3. We observe that the naive estimator has an irreducible bias. In contrast, the proposed estimator eliminates the bias and outperforms the naive estimator significantly. Simulation details can be found in Appendix J.

**Computational time.** We summarize the computational time (in seconds) of running Algorithm 1 with varying sample sizes in table 3. For all scenarios, the proposed method only takes minutes to find the optimal policy.

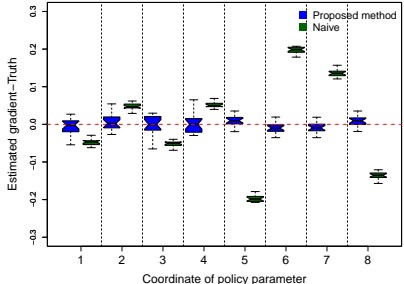

Figure 3: Comparisons of the proposed policy gradient and the naive policy gradient estimators to the truth at the uniform random policy. Results are computed using 20 replicates under $T = 2$, $N = 10000$ with general function approximations.

Table 3: Computational time (in seconds) for $T = 2$, $d_\Theta = 8$, $K = 50$ with varying sample sizes $N$.

| SAMPLE SIZE | TIME |
|---|---|
| $N = 200$ | 9.81563 |
| $N = 1000$ | 28.79569 |
| $N = 5000$ | 118.81537 |
| $N = 10000$ | 227.02766 |

