# OpenReview forum: "A Policy Gradient Method for Confounded POMDPs"
_ICLR.cc/2024/Conference — ICLR 2024 poster_

### Official Review · Reviewer_4mko · 2023-10-29

**Soundness:** 4 excellent
**Presentation:** 3 good
**Contribution:** 3 good
**Rating:** 8
**Confidence:** 2

**Summary:**

This paper theoretically studies the problem of offline policy optimization in POMDPs from a confounded offline data dataset, following the line of previous works on confounded POMDPs and policy gradient in MDPs. The core contribution is a deconfounded policy gradient ascent algorithm (Algorithm 1) based on proximal causal inference with theoretical guarantees. There are also numerical demonstrations to show the effectiveness of the proposed method.

**Strengths:**

1. The confounded offline POMDP setting itself is a well-motivated problem, and based on existing works it is natural to ask how to learn the optimal history dependent policy with computational efficiency. The authors answer this question by looking into the policy gradient style method which has not been explored in this literature.
2. The idea of using bridge functions (originated from proximal causal inference) to identify not only the policy value $\mathcal{V}(\pi_{\theta})$ but also the policy gradient $\nabla_{\theta}\mathcal{V}(\pi_{\theta})$ is new.
3. The theoretical derivations of the policy gradient identification is novel and is of independent interest to future research in confounded offline RL area.
4. The theoretical results are self-content and sound.

**Weaknesses:**

1. From my viewpoint, by looking into the previous line of works, the main theory (Sections 6.1 & 6.2) of this work is mostly based on **(i)** the theoretical understanding of statistical analysis for using minimax estimator to solve bridge functions in confounded offline POMDP settings, e.g., [1, 2, 3]; **(ii)** the analysis of global convergence of policy gradient ascent methods in standard MDP settings, e.g., [4, 5]. So the technical contributions of the main theory part are somehow weakened given these prior works.
2. As stated in 1., a consequence is that the theoretical assumptions regarding the policy gradient analysis (Section 6.2) are mostly adapted directly from those for MDP setups. How to understand these assumptions in POMDP settings with a history dependent policy class is less discussed.

**References:**

[1] Bennett, Andrew, and Nathan Kallus. "Proximal Reinforcement Learning: Efficient Off-policy Evaluation in Partially Observed Markov Decision Processes." *Operations Research* (2023).

[2] Shi, Chengchun, et al. "A Minimax Learning Approach to Off-policy Evaluation in Confounded Partially Observable Markov Decision Processes." *International Conference on Machine Learning*. PMLR, 2022.

[3] Lu, Miao, et al. "Pessimism in the Face of Confounders: Provably Efficient Offline Reinforcement Learning in Partially Observable Markov Decision Processes." *The Eleventh International Conference on Learning Representations*, 2023.

[4] Agarwal, Alekh, et al. "On the Theory of Policy Gradient Methods: Optimality, Approximation, and Distribution Shift." *The Journal of Machine Learning Research* 22.1 (2021): 4431-4506.

[5] Liu, Yanli, et al. "An Improved Analysis of (Variance-reduced) Policy Gradient and Natural Policy Gradient Methods." *Advances in Neural Information Processing Systems* 33 (2020): 7624-7636.

**Questions:**

1. Continued from Weakness 1., I would appreciate it if the authors could highlight more on the technical contributions behind the main theory, especially when compared with the previous line of works listed.
2. Continued from Weakness 2., it seems that the paper does not contain any discussion of a concrete policy class example. I think this is important since we are now dealing with history-dependent policy class which is different from previous MDP policy gradient problems. How does the dependence on history change (or not change) the difficulty of doing policy gradient and why? It would be great if such discussions can be included.

---

> ### Author Response · Authors · 2023-11-20
>
> We are grateful for your valuable feedback and the recognition you've given to our work. We would like to address your concerns in this response.
>
> > Weakness 1 and question 1 regarding technical contributions in Sections 6.1 & 6.2
>
> Thank you for acknowledging **one of our main theoretical contributions in the derivation of the identification results** for the policy gradient.  In addition to this, we would like to highlight some crucial technical contributions in estimation and optimization presented in Sections 6.1 and 6.2. Below we provide a detailed summary.
>
> In the theoretical results related to the estimation of the policy gradient (section 6.1), our main
> technical contributions are as follows.
>
> - **Novel Error Decomposition.** We derived a novel decomposition of error for policy gradient estimation, outlined in Appendix F.1 or equations (63)(64)(65). Existing works such as [1, 2, 3] typically consider the decomposition of the estimation error of *policy value* into multiple one-step errors caused by min-max estimation for the conditional operator of the *value* bridge function. In contrast, we study the decomposition of the estimation error of *policy gradient*, which turns out to involve one-step errors caused by min-max estimation for *both* the conditional operator of the *value* bridge function and the *gradient* bridge functions. This unique decomposition arises from our method of policy gradient identification, incorporating policy value identification results as assistance. Importantly, the derivation of this decomposition is not a trivial extension from existing works. Such error decomposition enhances the understanding of the source of all involved statistical errors, and can be potentially used to design more efficient or robust estimators for policy gradient in confounded POMDPs. For example, a direct idea is to simply consider more efficient/robust off-policy evaluation methods in confounded POMDPs and see if their incorporation can improve the efficiency/robustness of the policy gradient estimators by looking at the proposed error decomposition of the policy gradient.
>
> - **Focus on Policy Gradient Estimation with Dimension $>1$.** In contrast to existing works [1, 2, 3], which primarily focused on policy value estimation with dimension equal to 1, our work uniquely concentrates on policy gradient estimation with dimensions greater than 1. This emphasis, driven by the significance of history-dependent policies, introduces an additional layer of complexity to the analysis. Specifically, the dimensionality of the policy parameter space, denoted as $d_\Theta$, becomes a crucial factor in our theoretical results. Therefore, our derivation of upper bounds with explicit dependence on $d_\Theta$ facilitates understanding of the complexity in higher-dimensional min-max estimation.
>
> - **Upper Bounds with Explicit Dependence on Key Parameters (in the Nonparametric Settings).** In contrast to [1,2,3], our work provides upper bounds in terms of all key parameters when function classes are assumed to be  RKHSs (see details in Appendix B). This allows for non-parametric estimation. Existing works often provide examples only under linear settings or with finite VC dimensions, making our results more versatile and applicable to a broader range of scenarios.
>
> In the theoretical results related to the global convergence of the proposed algorithm (section 6.2), our main technical contributions are as follows.
>
> - **Extension from MDPs to POMDPs.**  Building upon the theoretical results of [4, 5], we extend their results from MDPs to POMDPs. This extension necessitates the incorporation of specific lemmas tailored for POMDPs, including the policy gradient lemma and the performance difference lemma (Lemmas 3, 8, 10). Our work thus contributes to advancing theoretical understanding in the context of POMDPs.
>
> - **Finite-horizon v.s. Infinite-horizon**. While [4,5] focus on the infinite-horizon discounted MDPs, we study the finite-horizon settings. Therefore, our work requires us to study the upper bound with explicit dependence on the length of horizon $T$, especially when history is included in our POMDP settings.
>
> References:
>
> [1] Bennett, Andrew, and Nathan Kallus. "Proximal Reinforcement Learning: Efficient Off-policy Evaluation in Partially Observed Markov Decision Processes."
>
> [2] Shi, Chengchun, et al. "A Minimax Learning Approach to Off-policy Evaluation in Confounded Partially Observable Markov Decision Processes."
>
> [3] Lu, Miao, et al. "Pessimism in the Face of Confounders: Provably Efficient Offline Reinforcement Learning in Partially Observable Markov Decision Processes."
>
> [4] Agarwal, Alekh, et al. "On the Theory of Policy Gradient Methods: Optimality, Approximation, and Distribution Shift."
>
> [5] Liu, Yanli, et al. "An Improved Analysis of (Variance-reduced) Policy Gradient and Natural Policy Gradient Methods."

---

> > ### Author Response · Authors · 2023-11-20
> >
> > > Weakness 2 and question 2 regarding discussions on assumptions (made in Section 6.2) in POMDP settings with a history dependent policy class and how the dependence on history changes (or does not change) the difficulty of doing policy gradient
> >
> > Thank you for your suggestions. We have included such a discussion in Appendix B.7 of the revised manuscript. To address your concern, below we discuss how the inclusion of history impacts the complexity of policy gradient ascent for confounded POMDPs under the offline settings across statistical, optimization, and computational aspects. Specifically, we illustrate this with the example of a log-linear policy:
> > $$\pi_{\theta_t}(a_t\mid o_{t},h_{t-1}) \propto \exp(\theta_t^\top \phi_t(a_t,o_t,h_{t-1})).$$
> >
> > - **Statistical Aspect.** In terms of statistical estimation, we examine the upper bound presented in Theorem 2 for the policy gradient estimation error. First, the dimension of the policy space $d_\Theta$ implicitly depends on $T$. At each step $t$, $\phi_t$ is a feature map from a space of dimension $t|\mathcal{A}|dim(\mathcal{O})$ to a space of dimension $d_{\Theta_t}$. To preserve information adequately, it's reasonable to assume that $d_{\Theta_t}$ grows with $t$. In contrast, in MDP settings, a fixed $d_{\Theta_t}$ assumption may suffice for all $t$. Second, the function classes for the bridge functions and test functions should also be rich enough because they are functions of histories that scale with $t$ at each stage $t$. Therefore the complexity of the function classes $\gamma(\mathcal{F},\mathcal{B})$ also grows when the number of stages $T$ increases. These factors collectively contribute to the complexity of estimation when dealing with history-dependent policies.
> >
> > - **Optimization Aspect.** We discuss the assumptions in section 6.2 in response to your concern. Firstly, assumption 5(a) (optimal policy coverage) can be dropped with the introduction of a notion we term ``transferred compatible approximation error." We respectfully direct you to our response regarding "Regarding Optimal Policy Coverage (Assumption 5(a))" to Reviewer KDFK for more details. We then focus on the revised manuscript in the rest. Regarding Assumption 5(a) $\beta$-smoothness, when $\phi _t(a _t,o _t,h _{t-1})$ is in a compact region with $\\|\phi _t(a _t,o _t,h _{t-1})\\| _{\ell _2}\le B$, it is straightforward to show that $\left\\|\nabla _{\theta _t} \log \pi _{\theta _t}\left(a _t \mid o _t, h _{t-1}\right)-\nabla _{\theta _t} \log \pi _{\theta _t^{\prime}}\left(a _t \mid o _t, h _{t-1}\right)\right\\| _{\ell^2} \leq B^2\left\\|\theta _t-\theta _t^{\prime}\right\\| _{\ell^2}$. This assures that the Lipschitz constant remains unaffected by the historical dependence. Assumption 5(b) (Positive definiteness of Fisher information matrix)) requires the positive definiteness of Fisher information matrix, where the constant $\mu$ implicitly depends on the number of stage $T$. Intuitively, obtaining a large $\mu$ becomes more challenging in the context of history-dependent policies due to the high dimensionality of the history space. A potential approach to mitigate this challenge involves mapping the history to a lower-dimensional space that retains sufficient information. For Assumption 5(c) (L-smoothness), the scale of the constant $L$ increases when considering history, compared to the standard MDP settings. See Appendix B.3 for more details. It's evident that the historical dependence amplifies the complexity through Assumptions 5(b) and 5(c). Furthermore, the dimension of the parameter space $d _\Theta$, implicitly depending on $T$, heightens the challenge of gradient ascent for the same reasons elucidated in the statistical aspect.
> >
> > -  **Computational Aspect.** We have added the analysis of the computational complexity of Algorithm 1 in Appendix I.2 of the revised manuscript, yielding a time complexity of $O(Kd _{\Theta}N^2 T\max\\{T,N\\})$. Compared to the standard MDP settings, the introduction of history-dependence primarily increases the computational complexities in two steps: the evaluation of the empirical kernel matrix to derive the closed-form solution for the min-max optimization problem (Equation 9) and the update of the policy parameter. For the empirical kernel matrix evaluation, kernel functions must be computed in the history space, a task that scales with $T$. Furthermore, we need $d _{\Theta}$ operations to update each coordinate, where $d _\Theta$ implicitly depends on $T$.
> >
> > We hope that we have addressed your concerns, and are open to any further questions or feedback. If our revised manuscript and rebuttal more closely meet your expectations for the paper, we respectfully ask you to reconsider your initial rating.

---

> > > ### Comment · Reviewer_4mko · 2023-11-22
> > >
> > > Thank you very much for your detailed answer to my questions and addressing my concerns, and thanks for your detailed discussion of my **Q2** in the revised version. From the authors' response, I acknowledge the additional contributions compared to the previous works. Most of my concerns have been addressed, and I am happy to increase my rate to 8.

---

> > > > ### Author Response · Authors · 2023-11-23
> > > >
> > > > Thanks for increasing the score. We greatly appreciate the time and effort you've taken to review our paper and provide insightful feedback!

---

### Official Review · Reviewer_KDFK · 2023-10-29

**Soundness:** 3 good
**Presentation:** 2 fair
**Contribution:** 3 good
**Rating:** 8
**Confidence:** 3

**Summary:**

This paper introduces a policy gradient method tailored for confounded POMDPs with continuous state and observation spaces in the offline learning context. The authors present a novel method for non-parametrically estimating any history-dependent policy gradient in POMDPs using offline data. They employ a min-max learning procedure with general function approximation to estimate the policy gradient through solving a sequence of conditional moment restrictions.

The authors provide a finite-sample, non-asymptotic bound for the gradient estimation. Using the proposed gradient estimation method within a gradient ascent algorithm, the paper demonstrates the global convergence of the algorithm.

**Strengths:**

- The paper addresses a challenging problem on policy gradient methods for POMDPs in the offline setting with continuous state and observation spaces. This is a novel contribution to the best of my knowledge; most existing work on policy gradient methods has been centered around fully observable environments.

- The paper’s contributions seem to be significant with the global convergence result of the algorithm to find the history-dependent optimal policy. Additionally, the identification result for non-parametrically estimating any history-dependent policy gradient under POMDPs using offline data is a unique contribution.

**Weaknesses:**

- The paper introduces a significant number of symbols and notations, which might overwhelm readers, especially those who are less familiar with the topic. To improve clarity, the authors could consider providing a table or appendix that lists all the symbols used along with their definitions. Additionally, they might simplify the notation where possible and ensure that each symbol is clearly defined upon its first use. For example, the notation seems to be quite heavy at the end of Page 5 where $\mathcal{Z_t}$ and $\mathcal{W_t}$ are introduced.

- The full coverage assumptions stated in Assumption 4(a) and Assumption 5(a) are indeed common in offline RL literature, but they bring about challenges and potential limitations to the proposed method in the paper.

1. **Full Coverage Assumption (Assumption 4(a))**: This assumption, which requires that the offline distribution $P_{\pi_b}$ can calibrate the distribution $P_{\pi_\theta}$ induced by $\pi_\theta$ for all $\theta$, is strong and might not always be satisfied in practical scenarios. In real-world applications, especially in domains like healthcare or finance, obtaining an offline dataset that sufficiently covers all possible actions and states can be impractical due to ethical, logistical, or financial constraints. The paper could improve by discussing the potential implications of this assumption, providing guidance on how to assess whether this assumption is reasonable in a given setting, or suggesting alternative approaches if the assumption is not met.

2. **Optimal Policy Coverage (Assumption 5(a))**: This assumption requires that the optimal policy $\pi^*$ is covered by all the policies in the class. While this condition ensures that the policy class is rich enough to contain the optimal policy, it might be too restrictive in practice.

**Questions:**

How might the proposed method be adapted or extended to accommodate partial coverage assumptions, and what would be the technical challenges associated with such an adaptation? Could you provide insights or discuss potential ways for relaxing the full coverage assumptions while maintaining the theoretical guarantees of the method?


================================================

**After Rebuttal:**

Thank you for the detailed response. I feel like my concerns have been addressed by the authors' response, and I would like to raise my score to 8.

---

> ### Author Response · Authors · 2023-11-20
>
> We sincerely appreciate your constructive feedback and your acknowledgment of our work.
>
> >Regarding notations
>
>  Thank you for the valuable suggestion. We acknowledge the importance of clarity in notation. In response to your feedback, we have included a table in Appendix A that lists all the main notations used along with their definitions. Additionally, we have simplified the notations where feasible and ensured that each symbol is explicitly defined upon its initial use in the manuscript. See Section 5 in the revised version.
>
> > Regarding Full Coverage Assumption (Assumption 4(a))
>
> Thank you for your suggestions. In the following, we discuss on practical scenarios in which this assumption is either met or not, and propose a potential way to address this concern. We have also added this discussion in Appendix B.4 of the revised manuscript.
>
> In certain real-world applications, such as the sequential multiple assignment randomized trials (SMART) designed to build optimal adaptive interventions, the assumption of full coverage is usually satisfied. This is because data collection in these trials is typically randomized, ensuring a comprehensive representation by the behavior policy. However, we acknowledge that in domains such as electronic medical records, meeting the full coverage assumption may pose challenges due to ethical or logistical constraints.
>
> To address scenarios where the full coverage assumption might not hold, we could incorporate the principle of pessimism into our approach. This involves penalizing state-action pairs that are rarely visited under the offline distribution. The idea of incorporating pessimism has been widely used in the offline RL literature for MDPs. For example, a practical implementation of this idea can be adapted from [Zhan et al., 2022], where a regularity term is added to the objective function to measure the discrepancy between the policy of interest and the behavior policy. By identifying and estimating the gradient of this modified objective function, we could potentially provide an upper bound on suboptimality and maintain a similar theoretical result by only assuming partial coverage, i.e.
> $$C_{\pi^b}^{\pi_{\theta^*}}:=\max_{t=1,..,T}(\mathbb E^{\pi^b}[(\frac{p_t^{\pi_{\theta^*}}(S_t,H_{t-1})}{p_t^{\pi^b}(S_t,H_{t-1})\pi^b_t(A_t\mid S_t)})^2])^{\frac{1}{2}}<\infty.$$
> This partial coverage assumption only needs that the offline distribution $\mathbb{P}^{\pi^b}$ can calibrate the distribution $\mathbb{P}^{\pi_{\theta^*}}$ induced by the in-class optimal policy, which is a milder condition compared to the full coverage assumption.
>
> >Regarding Optimal Policy Coverage (Assumption 5(a))
>
> Thank you for your comment on this assumption. Inspired by [Liu et al.][Masiha et al.], we can relax this assumption by introducing a transferred compatible function approximation error in the following way:
>
> $$\varepsilon _{approx}:=\sup  _{\theta \in \Theta}  \sum _{t=1}^T\left(\mathbb{E}^{\pi _{\theta^*}}\left[\left(\mathbb{A} _t^{\pi _\theta}\left(A _t, O _t, S _t, H _{t-1}\right)-w _t^*(\theta)^{\top} \nabla _{\theta _t} \log \pi _{\theta _t}\left(A _t \mid O _t, H _{t-1}\right)\right)^2\right]\right)^{\frac{1}{2}},$$
> where $w _t^*(\theta)$ denotes the optimal solution of the following optimization problem for each $\theta$:
> $$\arg\min _{w _t}  \sum _{t=1}^T\left(\mathbb{E}^{\pi _{\theta}}\left[\left(\mathbb{A} _t^{\pi _\theta}\left(A _t, O _t, S _t, H _{t-1}\right)-w _t^{\top} \nabla _{\theta _t} \log \pi _{\theta _t}\left(A _t \mid O _t, H _{t-1}\right)\right)^2\right]\right)^{\frac{1}{2}}.$$
>
> Note that this notion is similar to the definition (Def. C.2 in Appendix C) in our original paper, which also quantifies the richness of the policy class. It tends to be small when the policy class is sufficiently rich. In such cases, the requirement for optimal policy coverage (Assumption 5(a)) becomes unnecessary. In the revised manuscript, we have introduced this transferred compatible approximation error in Definition D.2, deleted the original optimal policy coverage assumption, and updated the results \& proof of Theorem 3.
>
> We respectfully hope that we have addressed your concerns. If you have any further questions or require more information to raise your initial score, please feel free to let us know.
>
>
> Reference:
>
> 1. Zhan W, Huang B, Huang A, et al. Offline reinforcement learning with realizability and single-policy concentrability[C]//Conference on Learning Theory. PMLR, 2022: 2730-2775.
>
> 2. Liu Y, Zhang K, Basar T, et al. An improved analysis of (variance-reduced) policy gradient and natural policy gradient methods[J]. Advances in Neural Information Processing Systems, 2020, 33: 7624-7636.
>
> 3. Masiha S, Salehkaleybar S, He N, et al. Stochastic second-order methods improve best-known sample complexity of sgd for gradient-dominated functions[J]. Advances in Neural Information Processing Systems, 2022, 35: 10862-10875.

---

> ### Author Response · Authors · 2023-11-23
>
> Thanks for your positive feedback and the upgraded score. We are again grateful for your time and efforts to review our paper and give constructive comments!

---

### Official Review · Reviewer_YGPH · 2023-10-31

**Soundness:** 3 good
**Presentation:** 3 good
**Contribution:** 3 good
**Rating:** 8
**Confidence:** 3

**Summary:**

This paper proposes a policy gradient method for confounded partially observable Markov decision processes (POMDPs) in the offline setting with novel gradient identification and estimation. Also, a theoretical analysis of the suboptimality of the proposed method is provided. Finally, numerical experiments are conducted to evaluate the performance of the algorithm.

**Strengths:**

The gradient identification proposed is new to confounded POMDPs, and the policy gradient based on that is complimented by strong theoretical guarantees. The statistical error of the gradient estimation and the suboptimal of the obtained policy are both discussed with a comprehensive analysis. Also, The theoretical results are also complimented by experimental results.

**Weaknesses:**

While I typically do not complain about the empirical results of a theory paper, I do expect that the authors could show how to implement such estimation and algorithm on real practical RL problems.

**Questions:**

1. In the third paragraph of Section 3, does the definition of history $\mathcal{H}_{t}$ lack the subscript on $\mathcal{O}$ and $\mathcal{A}$?
2. The notation $Z_{t}$ has already been defined in Section 3; however, another $\mathcal{z}$ is used in Section 5 as additional notation. Are they the same things?

---

> ### Author Response · Authors · 2023-11-20
>
> We are grateful for your insightful comment and would like to address your concerns in this response.
>
> >While I typically do not complain about the empirical results of a theory paper, I do expect that the authors could show how to implement such estimation and algorithm on real practical RL problems.
>
>  We appreciate your feedback and understand the importance of practical implementation in reinforcing the applicability of our proposed estimation and algorithm. In response to your suggestion, we have added discussion about the practical implementation of the proposed algorithm on real-world RL problems in Appendix I.3 of the revised manuscript.
>
> First of all, an exact implementation method has been  provided in Appendix I.1 of the manuscript. Specifically, assuming all function classes are reproducing kernel Hilbert spaces (RKHSs), the min-max optimization problem (9) comprises a quadratic concave inner maximization problem and a quadratic convex outer minimization problem. Consequently, a closed-form solution can be computed, as demonstrated in Appendix I.1.
>
> In practical implementations on real RL problems, the exact execution of the proposed algorithm may sometimes computationally costly. For instance, when dealing with large sample sizes or a high number of stages, solving the min-max optimization problem (9) precisely in each iteration may become computationally expensive. Particularly, exact solutions with RKHSs involve calculating the inverse of an $N\times N$ (which takes $O(N^3)$) and evaluating the empirical kernel matrix on the space of history (taking approximately $O(N^2 T)$).  Additionally, when assuming other function classes for bridge functions and test functions, such as neural networks, achieving an exact solution for the min-max optimization problem (9) is not feasible.
>
> To enhance computational efficiency and accommodate general function classes, an alternative approach is to compute the (stochastic) gradient of the empirical loss function in the min-max optimization problem (9) with respect to $b$ and $f$ respectively. Subsequently, updating $f$ and $b$
> alternately using gradient ascent and gradient descent only once in each iteration can be performed. In this case, gradient estimation is expedited, contributing to faster computations. We acknowledge the need for further exploration to assess its efficacy in real-world applications, and we consider this an avenue for future research.
>
> > In the third paragraph of Section 3, does the definition of history $\mathcal{H}_t$
>  lack the subscript on $\mathcal{O}$ and $\mathcal{A}$?
>
> Thank you for your question. We assume the observation space $\mathcal{O}$ and the action space $\mathcal{A}$ are the same for each stage $t=1,2,...,T$, eliminating the need for subscripts on
> $\mathcal{O}$ and $\mathcal{A}$. The space of history
> $\mathcal{H}_t$ can be conceptualized as a product space. One can generalize the setting by allowing $\mathcal{O}$ and $\mathcal{A}$ to be time-dependent.
>
> > The notation $Z_t$
>  has already been defined in Section 3; however, another $z$
>  is used in Section 5 as additional notation. Are they the same things?
>
> Thank you for pointing this out. They are not the same thing, and we have corrected the notations in the revised manuscript.  We now use $\mathcal E_t$ rather than $Z_t$ to denote the emission kernel in Section 3.
>
> We hope that we have addressed your concerns, and welcome any questions or feedback you may have. If the rebuttal reflects a better alignment with your expectations for the paper, we kindly ask you to consider raising your rating.

---

### Official Review · Reviewer_5Pwh · 2023-11-02

**Soundness:** 4 excellent
**Presentation:** 3 good
**Contribution:** 4 excellent
**Rating:** 8
**Confidence:** 1

**Summary:**

This paper introduces a novel policy gradient method for confounded partially observable Markov decision processes (POMDPs), and proves it converges to global optimal under certain assumptions. Their method estimates the policy gradient using bridge functions calculated from offline data, which is adopted from min-max estimator of [Dikkala et al. (2020)]. The paper provides a finite-sample non-asymptotic bound for estimating the gradient uniformly over a pre-specified policy class. Additionally, the authors show the global convergence of the proposed algorithm in finding the history-dependent optimal policy under certain technical conditions. This work is claimed to be the first to study the policy gradient method for POMDPs under the offline setting.

The paper also discusses the challenges in studying policy gradient methods in confounded POMDPs under the offline setting, such as the bias in estimation due to unobserved state variables, the need for function approximation in continuous spaces, and the challenge in achieving global convergence for finding the optimal policy.

The authors contribute by proposing a policy gradient method with both statistical and computational guarantees, establishing a non-asymptotic error bound for estimating the policy gradient, and providing a solution for global convergence in POMDPs.

**Strengths:**

**Dsiclaimer:** I should first note that the results of this paper are super technical. A proper review of this article needs a full working week of my time, which clearly I couldn't put. I tried going through Appendix A and B, but even then I cannot say I understood completely.

Despite these challenges, it is evident that the results derived in this paper solid. Establishing any form of convergence for the Policy Gradient (PG) algorithms within the context of POMDPs is immensely valuable to the community. Furthermore, their method for gradient estimation in this study presents itself as a potentially advantageous tool in its own right.

**Weaknesses:**

Given the technical nature of this paper, I am compelled to express my reservations about the suitability of the ICLR conference as the platform for its publication. I believe that this work might find a more fitting home in a scholarly journal, where reviewers are afforded ample time to thoroughly validate the results presented. Additionally, the attempt to condense the material into a 9-page format has significantly hindered its readability. In particular, the contents of Appendix A are critical enough that they warrant inclusion (or partial inclusion) in the main body of the text.

**Questions:**

[Vlassis et. al., 2012] have proved that finding a stochastic controller of polynomial size that achieves a certain target, is an NP-hard problem. Optimizing policy for confounded POMDPs is even more challenging. While [Vlassis et al., 2012] does not directly contradict this paper, it does raise questions about the real-world applicability of the assumptions made herein. Is there a possibility to provide examples that provide lower bounds? [Agrawal et. al.] provided an example demonstrating the necessity of distribution mismatch coefficient (which I presume it isn't very difficult to have a similar one for POMDPs). With this work, I also love to see more about computational complexity (and polynomial / NP-hardness / ...) of specific classes of POMDPs.

I must note that I am unable to specifically identify any of the assumptions as unreasonable, but I guess Assumption 2 is the most critical assumption.

References:
- Alekh Agarwal, Sham M Kakade, Jason D Lee, and Gaurav Mahajan. On the theory of policy
gradient methods: Optimality, approximation, and distribution shift. The Journal of Machine Learning Research, 22(1):4431–4506, 2021.
- Vlassis, Nikos, Michael L. Littman, and David Barber. "On the computational complexity of stochastic controller optimization in POMDPs." ACM Transactions on Computation Theory (TOCT) 4.4 (2012): 1-8.

---

> ### Author Response · Authors · 2023-11-20
>
> Thank you for your constructive feedback. We appreciate the opportunity to address each of your concerns.
>
> > [Vlassis et. al., 2012] have proved that finding a stochastic controller of polynomial size that achieves a certain target, is an NP-hard problem. Optimizing policy for confounded POMDPs is even more challenging. While [Vlassis et al., 2012] does not directly contradict this paper, it does raise questions about the real-world applicability of the assumptions made herein.
>
> Thank you for bringing attention to [Vlassis et al., 2012], which addresses the computational complexity of finding a stochastic controller of polynomial size in POMDPs. We appreciate your acknowledgement of the challenges associated with policy learning in POMDPs, as demonstrated by the NP-hardness results presented in [Burago et al., 1996] and [Vlassis et al., 2012]. It is important to note that our proposed method is designed as an approximate solution, specifically tailored to address a subclass of confounded POMDPs based on the assumptions outlined in our paper. By doing so, we navigate away from the NP-hard complexity, focusing on a more manageable problem class. In the following, we discuss the sources of "approximation" and the underlying assumptions.
>
> - **Policy parameterization.** In our work, the policy is indexed by a finite-dimensional parameter, introducing an approximation by restricting the policy class. The transferred compatible function approximation error $\varepsilon_{approx}$, presented in Theorem 3, quantifies the richness of the policy space.
>
> - **Gradient ascent as an iterative method.** Convergence to the optimal solution is not guaranteed due to the non-concave nature of the policy value function with respect to the policy parameter. To address this, we leverage the commonly-considered gradient dominance property in the field of non-convex/non-concave optimization. This property arises from a notion of function approximation for the advantage function $\varepsilon_{approx}$ and the positive definiteness of the Fisher information matrix (as per Assumption 5(b)). Up to a finite number of iterations, the output policy value can approximate the optimal policy value. The related assumptions are commonly used in the literature on policy gradient methods in MDPs.
>
> - **Policy gradient identification and estimation.** The approximation in our method stems from two main components: identification and estimation. 1) In the identification part, we restrict to a subclass of POMDPs that satisfies Assumptions 1, 2, and 3 in Section 4. These assumptions ensure that historical information and observations carry sufficient information about the latent state at each stage (completeness from Assumptions 1, 3) and that we can effectively use this information (Assumptions 1, 2) to derive a solution from observational data. In the tabular setting, this subclass of POMDPs is akin to (actually more general than) the under-completeness POMDPs introduced in [Jin et al. 2020]. 2) In the estimation part, we utilize offline samples to estimate the policy gradient instead of exact computation. We introduce function classes with controlled complexity to allow the use of a polynomial number of samples for policy gradient estimation. This relies on Assumption 4 in Section 6.1. Given our consideration of non-parametric models, the associated assumptions are weak.
>
> We plan to study the policy gradient methods under looser assumptions in future research. Nevertheless, the
> current paper is a first try to study policy gradient for confounded POMDPs under offline settings.
>
> > Is there a possibility to provide examples that provide lower bounds? [Agrawal et. al.] provided an example demonstrating the necessity of distribution mismatch coefficient (which I presume it isn't very difficult to have a similar one for POMDPs).
>
> Thank you for your suggestion to provide examples illustrating lower bounds. We believe a similar sparse-reward example can be developed to illustrate the necessity of the distribution mismatch coefficient in our problem. However, since we consider history-dependent policy in the confounded POMDP, the derivation of the policy gradient is much more involved. Because of the limited time for the rebuttal, achieving this for POMDPs is a more intricate task. Nevertheless, we assure you that we will strive to include such an example in the future revision.
>
> 1. Burago D, De Rougemont M, Slissenko A. On the complexity of partially observed Markov decision processes[J]. Theoretical Computer Science, 1996, 157(2): 161-183.
>
> 2. Vlassis N, Littman M L, Barber D. On the computational complexity of stochastic controller optimization in POMDPs[J]. ACM Transactions on Computation Theory (TOCT), 2012, 4(4): 1-8.
>
> 3. Jin C, Kakade S, Krishnamurthy A, et al. Sample-efficient reinforcement learning of undercomplete pomdps[J]. Advances in Neural Information Processing Systems, 2020, 33: 18530-18539.

---

> > ### Author Response · Authors · 2023-11-20
> >
> > > With this work, I also love to see more about computational complexity (and polynomial / NP-hardness / ...) of specific classes of POMDPs.
> >
> > Thank you for your valuable suggestions. For POMDPs satisfying Assumptions 1-5, the computational time for Algorithm 1 is polynomial in key parameters. We have included a detailed analysis of the computational complexity in Appendix I.2 of the revised manuscript. Specifically, the complexity is $O(Kd_ {\Theta}N^2 T\max\\{T,N\\})$ when utilizing RKHSs as function classes of interest. Here, $K$ is the number of iterations, $d_\Theta$ is the dimension of the parameter space, $N$ is the offline sample size, and $T$ is the number of stages.
> >
> > The dependence on $K$ arises from the iterative nature of the algorithm, while the dependence on $d_\Theta$ is related to manipulating the policy gradient and updating each coordinate of the policy parameter. The dependence on $N$ stems from matrix operations, including matrix inversions, multiplications, and the evaluation of the gram matrix in RKHSs. The dependence on $T$ comes from the iterative estimation of the bridge functions from $T$ to $1$ (steps 4-6), and the evaluation of kernel functions of histories, which scales with $T$.
> >
> > > I must note that I am unable to specifically identify any of the assumptions as unreasonable, but I guess Assumption 2 is the most critical assumption.
> >
> > Thank you for bringing this up. Assumption 2 indeed plays an important role in our paper, enabling the identification of the policy gradient through available information. This assumption is satisfied under some regularity conditions imposed on the linear integral operators, akin to the infinite-dimensional version of a linear system, as detailed in Appendix B.1. In the tabular case, the required assumptions lead to a subclass of POMDPs, that is more general than undercomplete POMDPs.
> >
> > We welcome any further questions or feedback you may have, and thank you again for taking the time to review our paper.

---

> > > ### Comment · Reviewer_5Pwh · 2023-11-22
> > > **Response**
> > >
> > > Thank you for your comprehensive response. I particularly value your commitment to including examples of the lower bound, as I believe they would significantly enhance this article. While I recognize that the article is already quite extensive, my queries were primarily directed towards identifying future research opportunities. For instance, I am aware you have calculated the sample complexity of your algorithm. In the following comment:
> > > >  With this work, I also love to see more about computational complexity (and polynomial / NP-hardness / ...) of specific classes of POMDPs.
> > >
> > > I meant to say I love to see for which sub-class of POMDPs the NP-hardness barrier breaks (More as a TCS perspective in contrast to learning theory one)

---

> > > > ### Author Response · Authors · 2023-11-23
> > > >
> > > > Thank you for your insightful comments and suggestions. Your emphasis on exploring the computational complexity of specific subclasses of Partially Observable Markov Decision Processes (POMDPs) from a theoretical computer science (TCS) perspective is well-received. Though we are not deeply familiar with the TCS perspective, we will explore it in the future based on some recent references such as [Kurniawati et al., 2021][Liu et al., 2022][Uehara et al., 2023].
> > > >
> > > > We greatly value your comments and suggestions, which are helpful in enhancing the quality of our manuscript.
> > > >
> > > > References:
> > > >
> > > > [1] Kurniawati H. Partially observable markov decision processes (pomdps) and robotics[J]. arXiv preprint arXiv:2107.07599, 2021.
> > > >
> > > > [2] Liu Q, Chung A, Szepesvári C, et al. When is partially observable reinforcement learning not scary?[C]//Conference on Learning Theory. PMLR, 2022: 5175-5220.
> > > >
> > > > [3] Uehara M, Sekhari A, Lee J D, et al. Computationally efficient pac rl in pomdps with latent determinism and conditional embeddings[C]//International Conference on Machine Learning. PMLR, 2023: 34615-34641.

---

### Meta-Review · Area_Chair_Qx8u · 2023-12-04

**Metareview:**

This paper presents a substantial theoretical result that tackles offline PG for confounded POMDPs, an open problem in offline RL. Through a novel identification result, the paper proposed a minimax approach to solve for a policy gradient estimator using offline data. A non-asymptotic sample complexity analysis is then provided using the aforementioned PG estimation subroutine.

All reviewers unanimously appreciate the theoretical contribution of this work. However, the paper did not put forward a strong case supporting the empirical applicability of the proposed method. In particular, the paper described how the proposed method can be implemented using kernel-based methods, but no experiments are provided.

Nevertheless, the theoretical contribution alone is sufficient to recommend acceptance for this manuscript.

**Justification For Why Not Higher Score:**

The results is primarily theoretical and on a very specific problem of offline PG estimation.

**Justification For Why Not Lower Score:**

NA

---

### Decision · Program_Chairs · 2024-01-16

Accept (poster)